# Gapless Phases in (2+1)d with Non-Invertible Symmetries

Lakshya Bhardwaj, Yuhan Gai[1], Sheng-Jie Huang[1], Kansei Inamura[1,2]

Sakura Schäfer-Nameki[1], Apoorv Tiwari[3,4], Alison Warman[1]

[1] *Mathematical Institute, University of Oxford,*
*Andrew Wiles Building, Woodstock Road, Oxford, OX2 6GG, UK*
[2] *Rudolf Peierls Centre for Theoretical Physics, University of Oxford,*
*Parks Road, Oxford, OX1 3PU, UK*
[3] *Center for Quantum Mathematics, University of Southern Denmark*
[4] *Danish Institute of Advanced Study, University of Southern Denmark*

The study of gapless phases with categorical (or so-called non-invertible) symmetries is a formidable task, in particular in higher than two space-time dimensions. In this paper we build on previous works [1,2] on gapped phases in (2+1)d and provide a systematic framework to study phase transitions with categorical symmetries. The Symmetry Topological Field Theory (SymTFT) is, as often in these matters, the central tool. Applied to gapless theories, we need to consider the extension of the SymTFT to interfaces between topological orders, so-called "club sandwiches", which realize generalizations of so-called Kennedy-Tasaki (KT) transformations. This requires an input phase transition for a smaller symmetry, such as the Ising transition for $\mathbb{Z}_2$, and the SymTFT constructs a transformation to a gapless phase with a larger categorical symmetry. We carry this out for categorical symmetries whose SymTFT is a (3+1)d Dijkgraaf-Witten (DW) theory for a finite group $G$ with twist – so-called all bosonic fusion 2-categories. We classify such interfaces using a physically motivated picture of generalized gauging, as well as with a complementary analysis using (bi-)module 2-categories. This is exemplified in numerous abelian and non-abelian DW theories, giving rise to interesting gapless phases such as intrinsically gapless symmetry protected phases (igSPTs) and spontaneous symmetry breaking phases (igSSBs) from abelian, $S_3$, and $D_8$ DW theories.

## A   Partial Ordering on Minimal Interfaces     114

## B   Gapless Phases from Abelian DW Theories     115

# 1   Introduction and Summary

It is by now well-documented that gapped phases with categorical or non-invertible symmetries go far beyond the standard classification of phases using symmetry groups, as initiated by Landau. Categorical phases are qualitatively distinct to the ones with group symmetries, in that e.g. order parameters are not only genuine operators, but can be non-genuine or disorder operators. This can be made precise and studied systematically, using the so-called Symmetry Topological Field theory [3–10] approach on general grounds and in concrete continuum and lattice models in (1+1)d and (2+1)d implementing this *Categorical Landau Paradigm* [11], applied in several setups [8, 9, 12–27].

**Phase Transitions from Interfaces between TOs.** Second order phase transitions in the presence of categorical symmetries are substantially more difficult to study in general. However given two gapped phases, that have a SymTFT realization, it is possible to systematically construct an associated phase transition by generalizing the concept of so-called Kennedy-Tasaki (KT) transformations to categorical symmetries. The framework of the SymTFT is applicable in any dimension and for any fusion $(d-1)$-category symmetry. The analysis is general, and we will first outline the procedure and then specialize subsequently to (2+1)-dimensional phase transitions. Concrete known implementations focus on (1+1)d [12, 15–17, 21, 22, 24]. The SymTFT realization of phase transitions relies on the concept called the "club sandwich": Starting with a known phase transition for a symmetry $\mathcal{S}_0$ between two gapped phases $\mathcal{P}'_A$ and $\mathcal{P}'_B$

$$\mathcal{P}'_A \quad \longleftarrow \quad \mathrm{CFT}' \quad \longrightarrow \quad \mathcal{P}'_B \,, \tag{1.1}$$

the club sandwich is a map from $\mathcal{S}_0$-symmetric theories to $\mathcal{S}$-symmetric theories that gives rise to a phase transition for a larger symmetry $\mathcal{S}$ such that $\mathcal{P}_i$ are $\mathcal{S}$-symmetric gapped phases

$$\mathcal{P}_A \quad \longleftarrow \quad \text{CFT} \quad \longrightarrow \quad \mathcal{P}_B\,. \tag{1.2}$$

The canonical example is that of $\mathcal{S}_0$ a $\mathbb{Z}_2$ 0-form symmetry and the associated transition is the $\mathbb{Z}_2$ trivial to $\mathbb{Z}_2$ spontaneous symmetry breaking (SSB) phase, that is given by the Ising CFT:

$$\mathbb{Z}_2\ \text{SSB} \quad \longleftarrow \quad \text{Ising} \quad \longrightarrow \quad \mathbb{Z}_2\ \text{Trivial}\,. \tag{1.3}$$

Inputing this into the construction we e.g. obtain phase transitions as detailed in tables 1, 3, and 2 for categorical symmetries that can be constructed from the $D_8$ topological order in 3+1d, including $\mathsf{Rep}(D_8)$ 1-form symmetries and 2-representations of 2-groups.

In (1+1)d for finite abelian group symmetries in particular, there are well-known phase transitions, which can be used in such KT-transformations. We will see that in (2+1)d the situation is less developed. The construction that we advocate in this paper, does rely on the input phase transitions, but we will see that already with $\mathbb{Z}_2$ there are many new second order transitions, that are categorically symmetric.

Technically the main task is to understand which symmetries $\mathcal{S}_0$ can arise, given a symmetry $\mathcal{S}$. More precisely, given the SymTFT for $\mathcal{S}$, and its topological defects, the Drinfeld center $\mathcal{Z}(\mathcal{S})$, one would like to determine the possible interfaces to reduced TOs, given by SymTFTs for $\mathcal{S}_0$, with center $\mathcal{Z}(\mathcal{S}_0)$. For fusion categories, these are determined by the so-called condensable algebras and can be characterized systematically for group-theoretical fusion categories (i.e. symmetries that are Morita equivalent to $\mathsf{Vec}_G^\omega$) [28].

Interfaces between the two topological orders (TOs) can be equivalently studied in terms of gapped boundary conditions (BCs) for the folded TO, i.e. in this case $\overline{\mathcal{Z}(\mathcal{S})} \boxtimes \mathcal{Z}(\mathcal{S}_0)$. Thus the classification of gapped BCs for general TOs can be used to study also interfaces. However in generic dimensions, it is a challenge to determine what the possible reduced TOs are.

**Phase Transitions in (2+1)d with Categorical Symmetries.** Luckily in (2+1)d at least for so-called all bosonic fusion 2-category symmetries [29], the SymTFTs are simply Dijkgraaf-Witten (DW) theories [30] for finite groups $G$, possibly with a twist $\varpi$. For these, we can provide a fully systematic classification of gapped boundary conditions [1, 2, 31, 32]. Physically this means we can characterize all symmetry categories that can arise for a given SymTFT, and secondly, applied to the folded TOs, we can characterize all interfaces. In turn this implies a *classification* of all possible interfaces, and the associated reduced TOs. Thus, for

| $D_8$ gapped phase $\mathcal{P}_A$ $(H_A, N_A)$ | $D_8$ CFT $(H_C, N_C)$ | $D_8$ gapped phase $\mathcal{P}_B$ $(H_B, N_B)$ |
|---|---|---|
| $\mathbb{Z}_2^r \times \mathbb{Z}_2^x$ SSB $(\mathbb{Z}_2^{r^2}, \mathbb{Z}_2^{r^2})$ | Ising $\boxplus$ Ising $\boxplus$ Ising $\boxplus$ Ising $(\mathbb{Z}_2^{r^2}, 1)$ | $D_8$ SSB $(1,1)$ |
| $\mathbb{Z}_4^r$ SSB $(\mathbb{Z}_2^x, \mathbb{Z}_2^x)$ | Ising $\boxplus$ Ising $\boxplus$ Ising $\boxplus$ Ising $(\mathbb{Z}_2^x, 1)$ | $D_8$ SSB $(1,1)$ |
| $\mathbb{Z}_4^r$ SSB $(\mathbb{Z}_2^{xr}, \mathbb{Z}_2^{xr})$ | Ising $\boxplus$ Ising $\boxplus$ Ising $\boxplus$ Ising $(\mathbb{Z}_2^{xr}, 1)$ | $D_8$ SSB $(1,1)$ |
| $\mathbb{Z}_2^x$ SSB $(\mathbb{Z}_2^{r^2} \times \mathbb{Z}_2^{xr}, \mathbb{Z}_2^{r^2} \times \mathbb{Z}_2^{xr})$ | Ising $\boxplus$ Ising $(\mathbb{Z}_2^{r^2} \times \mathbb{Z}_2^{xr}, \mathbb{Z}_2^{r^2})$ | $\mathbb{Z}_2^r \times \mathbb{Z}_2^x$ SSB $(\mathbb{Z}_2^{r^2}, \mathbb{Z}_2^{r^2})$ |
| $\mathbb{Z}_2^x$ SSB $(\mathbb{Z}_2^{r^2} \times \mathbb{Z}_2^{xr}, \mathbb{Z}_2^{r^2} \times \mathbb{Z}_2^{xr})$ | Ising $\boxplus$ Ising $(\mathbb{Z}_2^{r^2} \times \mathbb{Z}_2^{xr}, \mathbb{Z}_2^{xr})$ | $\mathbb{Z}_4^r$ SSB $(\mathbb{Z}_2^{xr}, \mathbb{Z}_2^{xr})$ |
| $\mathbb{Z}_2^r$ SSB $(\mathbb{Z}_2^{r^2} \times \mathbb{Z}_2^x, \mathbb{Z}_2^{r^2} \times \mathbb{Z}_2^x)$ | Ising $\boxplus$ Ising $(\mathbb{Z}_2^{r^2} \times \mathbb{Z}_2^x, \mathbb{Z}_2^{r^2})$ | $\mathbb{Z}_2^r \times \mathbb{Z}_2^x$ SSB $(\mathbb{Z}_2^{r^2}, \mathbb{Z}_2^{r^2})$ |
| $\mathbb{Z}_2^r$ SSB $(\mathbb{Z}_2^{r^2} \times \mathbb{Z}_2^x, \mathbb{Z}_2^{r^2} \times \mathbb{Z}_2^x)$ | Ising $\boxplus$ Ising $(\mathbb{Z}_2^{r^2} \times \mathbb{Z}_2^x, \mathbb{Z}_2^x)$ | $\mathbb{Z}_4^r$ SSB $(\mathbb{Z}_2^x, \mathbb{Z}_2^x)$ |
| $\mathbb{Z}_2^x$ SSB $(\mathbb{Z}_4^r, \mathbb{Z}_4^r)$ | Ising $\boxplus$ Ising $(\mathbb{Z}_4^r, \mathbb{Z}_2^{r^2})$ | $\mathbb{Z}_2^r \times \mathbb{Z}_2^x$ SSB $(\mathbb{Z}_2^{r^2}, \mathbb{Z}_2^{r^2})$ |
| $D_8$ SPT $(D_8, D_8)$ | Ising $(D_8, \mathbb{Z}_4^r)$ | $\mathbb{Z}_2^x$ SSB $(\mathbb{Z}_4^r, \mathbb{Z}_4^r)$ |
| $D_8$ SPT $(D_8, D_8)$ | Ising $(D_8, \mathbb{Z}_2^{r^2} \times \mathbb{Z}_2^{xr})$ | $\mathbb{Z}_2^x$ SSB $(\mathbb{Z}_2^{r^2} \times \mathbb{Z}_2^{xr}, \mathbb{Z}_2^{r^2} \times \mathbb{Z}_2^{xr})$ |
| $D_8$ SPT $(D_8, D_8)$ | Ising $(D_8, \mathbb{Z}_2^{r^2} \times \mathbb{Z}_2^x)$ | $\mathbb{Z}_2^r$ SSB $(\mathbb{Z}_2^{r^2} \times \mathbb{Z}_2^x, \mathbb{Z}_2^{r^2} \times \mathbb{Z}_2^x)$ |

Table 1: Phase transitions between two gapped phases for $D_8$ 0-form symmetry. Columns one and three display the type of phase and subgroup data $(H, N)$ that determine the gapped phases $\mathcal{P}_A$ and $\mathcal{P}_B$, while the middle column shows the $D_8$-symmetric CFT obtained after the KT transformation with input transition given by the 3d Ising CFT. Phases with a single copy of Ising are **gSPTs**, while those with multiple copies are **gSSBs**. The gapped phases were discussed in [2]. Here we have taken all 3-cocyles to be trivial.

all-bosonic fusion 2-categories, we can systematically determine the symmetric second-order phase transitions between gapped phases. This is the main result of this paper.

Although the formalism is comprehensive in (2+1)d, the problem has an intrinsic complexity in that for any SymTFT there is an *infinite number* of gapped boundary conditions, and thus (applied to the folded TO), an infinite number of interfaces between two topological orders, and thus an infinite number of phase transitions.

Let us recall the origin of this abundance [1, 2]: given a SymTFT $\mathcal{Z}(2\mathsf{Vec}_G)$ there is a canonical gapped boundary condition $\mathfrak{B}_{\text{Dir}}$ (Dirichlet), on the boundary of which the $2\mathsf{Vec}_G$ symmetry category is realized. We can generate an infinite number of gapped boundary conditions from this by stacking the Dirichlet boundary condition of the folded theory with an $H$ (where $H < G$) symmetric topological order $\mathfrak{T}_H$ to obtain

$$\mathfrak{B}_{\text{Dir}}^{\mathfrak{T}_H} = \mathfrak{B}_{\text{Dir}} \boxtimes \mathfrak{T}_H \,, \tag{1.4}$$

and then gauging a diagonal $H$. The category of lines in $\mathfrak{T}_H$ form a modular tensor category (MTC) $\mathcal{M}$, with the $H$ action giving rise to an $H$-crossed braided extension of $\mathcal{M}$, $\mathcal{M}_H^\times$. If the TQFT is an SPT (or trivial phase) we call these **minimal boundary conditions**, otherwise **non-minimal**. Applied to the folded topological order this gives rise to **minimal and non-minimal interfaces**, respectively.

**Classification of Interfaces.** One of the findings of this paper is that a large class of interfaces, which we call (non-chiral)[1] interfaces, can be classified as follows. Starting with the TO, given by the DW theory for $G$ with twist $\varpi$,

$$\mathcal{Z}(2\mathsf{Vec}_G^\varpi) \,, \tag{1.5}$$

the interfaces are fully characterized as

$$\mathcal{I}(H, N, \pi, \mathcal{A}) \,, \tag{1.6}$$

where

1. $H < G$ is a subgroup (up to conjugation),

2. $N$ is a normal subgroup of $H$ (up to $H$-preserving conjugation),

3. $[\pi] \in H^4(H/N, U(1))$,

---

[1] We will assume throughout that we have a non-chiral TQFT associated to $\mathcal{A}$.

4. A $(p^*\pi - \varpi|_H)$-twisted $H$-graded fusion category $\mathcal{A}$, which is faithfully graded by $H$. Here, $p : H \to H/N$ is the canonical projection.

The distinct choices are characterized by the choice of $H$ up to conjugation, and $N$ up to conjugation that preserves $H$. The reduced topological order is

$$\mathcal{Z}(2\mathsf{Vec}_{H/N}^\pi), \tag{1.7}$$

i.e. the reduced TO is the SymTFT, which is the DW theory for $H/N$ with twist $\pi$.

For minimal interfaces $\mathcal{A} = \mathsf{Vec}_H^\omega$, where $\omega \in H^3(H, U(1))$ characterizes the associator for the lines on the interface, that arise as the intersection with the surfaces that are labeled by $[h]$ conjugacy classes with $h \in H$. The minimal interfaces can exist only when $[p^*\pi - \varpi|_H] = 0 \in H^4(H, \mathrm{U}(1))$. By non-chiral interfaces, we mean interfaces that admit gapped interfaces to the Dirichlet interface $\mathcal{I}_{\mathrm{Dir}}$ of $\mathcal{Z}(2\mathsf{Vec}_G^\varpi)$ and $\mathcal{Z}(2\mathsf{Vec}_{H/N}^\pi)$. Here, $\mathcal{I}_{\mathrm{Dir}}$ is the topological interface obtained by unfolding the Dirichlet boundary of the folded TO $\overline{\mathcal{Z}(2\mathsf{Vec}_G^\varpi)} \boxtimes \mathcal{Z}(2\mathsf{Vec}_{H/N}^\pi)$.

**Interfaces from Folded TO.** Physically, the interface $\mathcal{I}_{(H,N,\pi,\mathcal{A})}$ is obtained as follows. First, we start from the Dirichlet boundary $\mathfrak{B}_{\mathrm{Dir}}$ of the folded TO

$$\overline{\mathcal{Z}(2\mathsf{Vec}_G^\varpi)} \boxtimes \mathcal{Z}(2\mathsf{Vec}_{H/N}^\pi). \tag{1.8}$$

We then stack $\mathfrak{B}_{\mathrm{Dir}}$ with some non-chiral anomalous $H$-symmetric TFT $\mathfrak{T}_{\mathcal{A}}^H$. Finally, we gauge the non-anomalous diagonal $H$ subgroup, where the $H$ symmetry on $\mathfrak{B}_{\mathrm{Dir}}$ is given by $H^{\mathrm{diag}} = \{(h, p(h)) \mid h \in H\}$. Upon unfolding, this procedure gives us the interface $\mathcal{I}_{(H,N,\pi,\mathcal{A})}$.

More generally, we can do the same construction of the interface using a chiral 3d TFT instead of a non-chiral one $\mathfrak{T}_{\mathcal{A}}^H$. Applied to chiral TFTs, the same procedure gives us chiral interfaces between the original TO $\mathcal{Z}(2\mathsf{Vec}_G^\varpi)$ and the reduced TO $\mathcal{Z}(2\mathsf{Vec}_{H/N}^\pi)$. We will not consider the chiral interfaces in this paper.

Note that for most of the SymTFT discussion we will simplify the analysis to start with $\mathcal{Z}(2\mathsf{Vec}_G^\varpi)$ with trivial $\varpi$. But this has an extension, which in particular will be discussed in the perspective from module categories.

**Club Sandwiches and KT-Transformations.** The interfaces can be used to obtain new phase transitions from old, (1.3) and (1.2). We will mostly focus on the input transition that is the Ising transition for the 0-form symmetry $\mathbb{Z}_2$ (or the gauged version, 1-form symmetry). The setup we have is shown in figure 1. The left boundary condition $\mathfrak{B}_{\mathrm{sym}}$ is gapped and determines the symmetry $\mathcal{S}$, so that $\mathcal{Z}(\mathcal{S}) = \mathcal{Z}(2\mathsf{Vec}_G^\varpi) \equiv \mathcal{Z}(G, \varpi)$. The interface $\mathcal{I}_{(H,N,\pi,\mathcal{A})}$ gives a map to $\mathcal{Z}(H/N, \pi) = \mathcal{Z}(2\mathsf{Vec}_{H/N}^\pi)$. The smaller symmetry $\mathcal{S}_0$ has center given by this

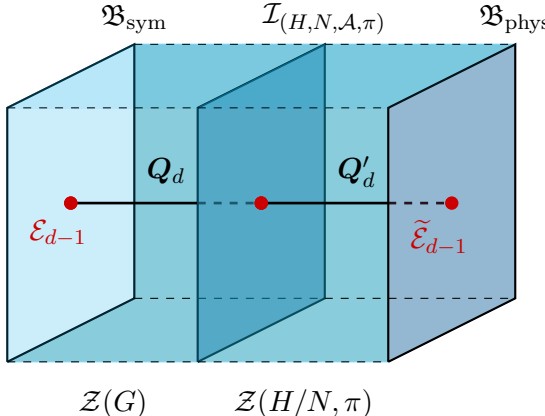

Figure 1: Club Sandwich Picture that generalizes the KT transitions to categorical symmetries. This is applicable in any dimension. Here we have drawn the concrete case relevant for $(2+1)$d fusion 2-category symmetries. $\boldsymbol{Q}_d$ are the topological defects of the SymTFT, which will provide order parameters and symmetry generators.

smaller TO: $\mathcal{Z}(\mathcal{S}_0) = \mathcal{Z}(H/N, \pi)$. The right boundary condition is the physical boundary, which is where we input the phase transition for $\mathcal{S}_0$. Inputting the gapped phases for $\mathcal{S}_0$ results in gapped phases for $\mathcal{S}$, which of course fall into the known classification [1, 2]. Inputting the gapless phase transition for $\mathcal{S}'$ results in a gapless theory with $\mathcal{S}$ symmetry.

**Module Category Picture.** The interface $\mathcal{I}_{(H,N,\pi,\mathcal{A})}$ is characterized by the property that the topological interfaces between $\mathcal{I}_{(H,N,\pi,\mathcal{A})}$ and the Dirichlet interface $\mathcal{I}_{\mathrm{Dir}}$ form a $(2\mathsf{Vec}_G^\varpi, 2\mathsf{Vec}_{H/N}^\pi)$-bimodule 2-category $_{\mathcal{A}}(2\mathsf{Vec}_{G \times H/N}^{\overline{\varpi}\pi})$, which is the 2-category of left $\mathcal{A}$-modules in $2\mathsf{Vec}_{G \times H/N}^{\overline{\varpi}\pi}$. The cocycle $\overline{\varpi}\pi \in Z^4(G \times H/N, \mathrm{U}(1))$ is defined by (3.34). The above (bi)module 2-category description allows us to show that no topological defects in the reduced TO $\mathcal{Z}(2\mathsf{Vec}_{H/N}^\pi)$ can end on the interface. This suggests that the reduced TO is obtained by the condensation in the original TO $\mathcal{Z}(2\mathsf{Vec}_G^\varpi)$.

**Symmetry Protected Criticality: igSPTs and igSSBs.** Particularly interesting gapless phases are those whose criticality is symmetry protected. Gapless phases which generalize SPT or SSB phases have been studied in [12, 15, 33–41]. A more stringent set of conditions arise for intrinsically gapless SPT or SSB phases (igSPT and igSSB). These are gapless phases, which either preserve the full symmetry or spontaneously break it to a subsymmetry, but which cannot be gapped without further breaking the symmetry, i.e. the number of genuine local operators (for 0-form symmetries) or genuine line operators (for 1-form symmetries) that can be resolved by linking with the symmetry generators strictly increases. Examples of

| 2Rep($\mathbb{G}$) gapped phase $\mathcal{P}_A$ $(H_A, N_A)$ | 2Rep($\mathbb{G}$) CFT $(H_C, N_C)$ | 2Rep($\mathbb{G}$) gapped phase $\mathcal{P}_B$ $(H_B, N_B)$ |
|---|---|---|
| 2Rep($\mathbb{G}$)/($\mathbb{Z}_2^{(1)} \times \mathbb{Z}_2^{(0)}$) SSB $(\mathbb{Z}_2^{r^2}, \mathbb{Z}_2^{r^2})$ | Ising $\boxplus$ Ising | 2Rep($\mathbb{G}$)/$\mathbb{Z}_2^{(1)}$ SSB $(1,1)$ |
| 2Rep($\mathbb{G}$) SSB $(\mathbb{Z}_2^x, \mathbb{Z}_2^x)$ | $\frac{\text{Ising}}{\mathbb{Z}_2^{(0)}} \boxplus \frac{\text{Ising}}{\mathbb{Z}_2^{(0)}} \boxplus$ Ising $(\mathbb{Z}_2^x, 1)$ | 2Rep($\mathbb{G}$)/$\mathbb{Z}_2^{(1)}$ SSB $(1,1)$ |
| $\mathbb{Z}_2^{(0)}$ SSB with mixed SPT $(\mathbb{Z}_2^{xr}, \mathbb{Z}_2^{xr})$ | Ising $\boxplus$ Ising $(\mathbb{Z}_2^{xr}, 1)$ | 2Rep($\mathbb{G}$)/$\mathbb{Z}_2^{(1)}$ SSB $(1,1)$ |
| 2Rep($\mathbb{G}$) Mixed SPT $(\mathbb{Z}_2^{r^2} \times \mathbb{Z}_2^{xr}, \mathbb{Z}_2^{r^2} \times \mathbb{Z}_2^{xr})$ | Ising $(\mathbb{Z}_2^{r^2} \times \mathbb{Z}_2^{xr}, \mathbb{Z}_2^{r^2})$ | 2Rep($\mathbb{G}$)/($\mathbb{Z}_2^{(1)} \times \mathbb{Z}_2^{(0)}$) SSB $(\mathbb{Z}_2^{r^2}, \mathbb{Z}_2^{r^2})$ |
| 2Rep($\mathbb{G}$) Mixed SPT $(\mathbb{Z}_2^{r^2} \times \mathbb{Z}_2^{xr}, \mathbb{Z}_2^{r^2} \times \mathbb{Z}_2^{xr})$ | Ising $(\mathbb{Z}_2^{r^2} \times \mathbb{Z}_2^{xr}, \mathbb{Z}_2^{xr})$ | $\mathbb{Z}_2^{(0)}$ SSB with mixed SPT $(\mathbb{Z}_2^{xr}, \mathbb{Z}_2^{xr})$ |
| 2Rep($\mathbb{G}$)/$\mathbb{Z}_2^{(0)}$ SSB $(\mathbb{Z}_2^{r^2} \times \mathbb{Z}_2^x, \mathbb{Z}_2^{r^2} \times \mathbb{Z}_2^x)$ | $\frac{\text{Ising}}{\mathbb{Z}_2^{(0)}} \boxplus \frac{\text{Ising}}{\mathbb{Z}_2^{(0)}}$ $(\mathbb{Z}_2^{r^2} \times \mathbb{Z}_2^x, \mathbb{Z}_2^{r^2})$ | 2Rep($\mathbb{G}$)/($\mathbb{Z}_2^{(1)} \times \mathbb{Z}_2^{(0)}$) SSB $(\mathbb{Z}_2^{r^2}, \mathbb{Z}_2^{r^2})$ |
| 2Rep($\mathbb{G}$)/$\mathbb{Z}_2^{(0)}$ SSB $(\mathbb{Z}_2^{r^2} \times \mathbb{Z}_2^x, \mathbb{Z}_2^{r^2} \times \mathbb{Z}_2^x)$ | (DW($\mathbb{Z}_2$) $\boxtimes$ Ising) $\boxplus \frac{\text{Ising}}{\mathbb{Z}_2^{(0)}}$ $(\mathbb{Z}_2^{r^2} \times \mathbb{Z}_2^x, \mathbb{Z}_2^x)$ | 2Rep($\mathbb{G}$) SSB $(\mathbb{Z}_2^x, \mathbb{Z}_2^x)$ |
| 2Rep($\mathbb{G}$) SPT $(\mathbb{Z}_4^r, \mathbb{Z}_4^r)$ | Ising $(\mathbb{Z}_4^r, \mathbb{Z}_2^{r^2})$ | 2Rep($\mathbb{G}$)/($\mathbb{Z}_2^{(1)} \times \mathbb{Z}_2^{(0)}$) SSB $(\mathbb{Z}_2^{r^2}, \mathbb{Z}_2^{r^2})$ |
| $\mathbb{Z}_2^{(1)}$ SSB $(D_8, D_8)$ | $\frac{\text{Ising}}{\mathbb{Z}_2^{(0)}}$ $(D_8, \mathbb{Z}_4^r)$ | 2Rep($\mathbb{G}$) SPT $(\mathbb{Z}_4^r, \mathbb{Z}_4^r)$ |
| $\mathbb{Z}_2^{(1)}$ SSB $(D_8, D_8)$ | $\frac{\text{Ising}}{\mathbb{Z}_2^{(0)}}$ $(D_8, \mathbb{Z}_2^{r^2} \times \mathbb{Z}_2^{xr})$ | 2Rep($\mathbb{G}$) Mixed SPT $(\mathbb{Z}_2^{r^2} \times \mathbb{Z}_2^{xr}, \mathbb{Z}_2^{r^2} \times \mathbb{Z}_2^{xr})$ |
| $\mathbb{Z}_2^{(1)}$ SSB $(D_8, D_8)$ | DW($\mathbb{Z}_2$) $\boxtimes$ Ising $(D_8, \mathbb{Z}_2^{r^2} \times \mathbb{Z}_2^x)$ | 2Rep($\mathbb{G}$)/$\mathbb{Z}_2^{(0)}$ SSB $(\mathbb{Z}_2^{r^2} \times \mathbb{Z}_2^x, \mathbb{Z}_2^{r^2} \times \mathbb{Z}_2^x)$ |

Table 2: Phase transitions between two gapped phases for 2Rep($\mathbb{G}$) symmetry where $\mathbb{G} \equiv \mathbb{G}^{(2)} = \mathbb{Z}_4^{(1)} \rtimes \mathbb{Z}_2^{(0)}$. This 2-representation of 2-group symmetry can be obtained from $D_8$ 0-form symmetry by gauging the non-normal $\mathbb{Z}_2^x$. Columns one and three display the type of phase and subgroup data for the gapped phases $\mathcal{P}_A$ and $\mathcal{P}_B$, while the middle column shows the 2Rep($\mathbb{G}^{(2)}$) CFT obtained after the KT transformation with input transition given by the 3d Ising CFT. The transition between the 2Rep($\mathbb{G}$)/$\mathbb{Z}_2^0$ SSB and the 2Rep($\mathbb{G}$) SSB phase is an **igSSB** for the 1-form symmetry: the number of 2Rep($\mathbb{G}$) charged lines (obtained from bulk SymTFT surfaces ending on both $\mathfrak{B}_{\text{sym}}$ and $\mathfrak{B}_{\text{phys}}$) strictly increases when we gap the theory, as can be seen from the algebras in tables 8 and 7. The gapped phases were discussed in [2]. Here we have taken all 3-cocyles to be trivial.

this were constructed in various dimensions for groups [12, 35, 42, 43], and for non-invertible symmetries [17, 42]. Here we will give concrete criteria for when they can occur in 2+1d and determine how the associated interfaces are characterized. We give concrete examples for gapless theories coming from $D_8$ DW theory. The main bottleneck for the construction in 2+1d of these phases is however the input phase transition, which requires gapless phases e.g. with anomalous group-like symmetries.

**Relation to Classification of Étale Algebras in Mathematics.** The characterization of the interfaces in a topological order $\mathcal{Z}(2\mathsf{Vec}_G^\varpi)$ in the folded picture matches up with the classification of (some special types of) condensable algebras[2] (also known as connected étale algebras) in $\mathcal{Z}(2\mathsf{Vec}_G^\varpi)$ in the mathematical literature [32]. We will in this paper not use this as an input, but motivate the construction and classification from very natural physical intuition. The data that determine these étale algebras seem to match that description, but we find it less ad hoc and this supports the conjecture that this corresponds to the correct physical concept as well.

**Summary of Gapless Phases and Phase Transitions.** We provide two distinct perspectives on the general theory of gapless phases and phase-transitions with fusion 2-category symmetries: **SymTFT sandwich** on the one hand and **module 2-category** on the other. These perspectives agree and give a classification first of interfaces and then phase transitions between symmetric gapped phases.

In terms of concrete examples we consider $\mathbb{Z}_4$, general abelian $\mathbb{A}$, $S_3$ and $D_8$ DW theories. From the Abelian DW theories, we can construct already interesting intrinsically gapless phases. In the tables 4 and 5 we summarize the type of gapless phases we get for the concrete examples $\mathbb{Z}_4$ and $S_3$ DW theory, respectively. Here, the input transitions left generic as $\mathfrak{T}_{\mathcal{S}'}$. For $\mathcal{S}' = \mathbb{Z}_2$ this can be taken to be the Ising CFT. We furthermore provide the action of the categorical symmetry. In the main text for each transition we also detail the order parameters. An important point to stress is that, although the reduced topological order is usually $\mathcal{Z}(2\mathsf{Vec}_{\mathbb{Z}_n})$, the actual symmetry category on the symmetry boundary of the reduced TO, obtained after collapsing the $\mathcal{Z}(\mathcal{S})$ part of the club-sandwich, is usually a multifusion category.

Finally in tables 1, 2 and 3 we give a summary of interesting gapless phases arising from $D_8$ DW theory, including igSSB phases. This includes the symmetries $D_8$ 0-form symmetry (table

---

[2]The interface corresponding to a condensable algebra $A$ in a topological order is the (2-)category of modules of $A$ in the topological order [44].

| 2Rep($D_8$) gapped phase $\mathcal{P}_A$ $(H_A, N_A)$ | 2Rep($D_8$) CFT $(H_C, N_C)$ | 2Rep($D_8$) gapped phase $\mathcal{P}_B$ $(H_B, N_B)$ |
|---|---|---|
| 2Rep($D_8$)/2Rep($\mathbb{Z}_2 \times \mathbb{Z}_2$) SSB $(\mathbb{Z}_2^{r^2}, \mathbb{Z}_2^{r^2})$ | $\frac{\mathsf{Ising}}{\mathbb{Z}_2^{(0)}}$ $(\mathbb{Z}_2^{r^2}, 1)$ | 2Rep($D_8$) Triv $(1,1)$ |
| 2Rep($\mathbb{Z}_2$) $(\mathbb{Z}_2^x, \mathbb{Z}_2^x)$ | $\frac{\mathsf{Ising}}{\mathbb{Z}_2^{(0)}}$ $(\mathbb{Z}_2^x, 1)$ | 2Rep($D_8$) Triv $(1,1)$ |
| 2Rep($\mathbb{Z}_2$) $(\mathbb{Z}_2^{xr}, \mathbb{Z}_2^{xr})$ | $\frac{\mathsf{Ising}}{\mathbb{Z}_2^{(0)}}$ $(\mathbb{Z}_2^{xr}, 1)$ | 2Rep($D_8$) Triv $(1,1)$ |
| 2Rep($\mathbb{Z}_2 \times \mathbb{Z}_2$) SSB $(\mathbb{Z}_2^{r^2} \times \mathbb{Z}_2^{xr}, \mathbb{Z}_2^{r^2} \times \mathbb{Z}_2^{xr})$ | DW($\mathbb{Z}_2$) $\boxtimes \frac{\mathsf{Ising}}{\mathbb{Z}_2^{(0)}}$ $(\mathbb{Z}_2^{r^2} \times \mathbb{Z}_2^{xr}, \mathbb{Z}_2^{r^2})$ | 2Rep($D_8$)/2Rep($\mathbb{Z}_2 \times \mathbb{Z}_2$) SSB $(\mathbb{Z}_2^{r^2}, \mathbb{Z}_2^{r^2})$ |
| 2Rep($\mathbb{Z}_2 \times \mathbb{Z}_2$) SSB $(\mathbb{Z}_2^{r^2} \times \mathbb{Z}_2^{xr}, \mathbb{Z}_2^{r^2} \times \mathbb{Z}_2^{xr})$ | DW($\mathbb{Z}_2$) $\boxtimes \frac{\mathsf{Ising}}{\mathbb{Z}_2^{(0)}}$ $(\mathbb{Z}_2^{r^2} \times \mathbb{Z}_2^{xr}, \mathbb{Z}_2^{xr})$ | 2Rep($\mathbb{Z}_2$) $(\mathbb{Z}_2^{xr}, \mathbb{Z}_2^{xr})$ |
| 2Rep($\mathbb{Z}_2 \times \mathbb{Z}_2$) SSB $(\mathbb{Z}_2^{r^2} \times \mathbb{Z}_2^x, \mathbb{Z}_2^{r^2} \times \mathbb{Z}_2^x)$ | DW($\mathbb{Z}_2$) $\boxtimes \frac{\mathsf{Ising}}{\mathbb{Z}_2^{(0)}}$ $(\mathbb{Z}_2^{r^2} \times \mathbb{Z}_2^x, \mathbb{Z}_2^{r^2})$ | 2Rep($D_8$)/2Rep($\mathbb{Z}_2 \times \mathbb{Z}_2$) SSB $(\mathbb{Z}_2^{r^2}, \mathbb{Z}_2^{r^2})$ |
| 2Rep($\mathbb{Z}_2 \times \mathbb{Z}_2$) SSB $(\mathbb{Z}_2^{r^2} \times \mathbb{Z}_2^x, \mathbb{Z}_2^{r^2} \times \mathbb{Z}_2^x)$ | DW($\mathbb{Z}_2$) $\boxtimes \frac{\mathsf{Ising}}{\mathbb{Z}_2^{(0)}}$ $(\mathbb{Z}_2^{r^2} \times \mathbb{Z}_2^x, \mathbb{Z}_2^x)$ | 2Rep($\mathbb{Z}_2$) $(\mathbb{Z}_2^x, \mathbb{Z}_2^x)$ |
| 2Rep($\mathbb{Z}_2 \times \mathbb{Z}_2$) SSB $(\mathbb{Z}_4^r, \mathbb{Z}_4^r)$ | DW($\mathbb{Z}_2$) $\boxtimes \frac{\mathsf{Ising}}{\mathbb{Z}_2^{(0)}}$ $(\mathbb{Z}_4^r, \mathbb{Z}_2^{r^2})$ | 2Rep($D_8$)/2Rep($\mathbb{Z}_2 \times \mathbb{Z}_2$) SSB $(\mathbb{Z}_2^{r^2}, \mathbb{Z}_2^{r^2})$ |
| 2Rep($D_8$) SSB $(D_8, D_8)$ | $\frac{\text{DW}(\mathbb{Z}_4) \boxtimes \mathsf{Ising}}{\mathbb{Z}_2^{(0),\text{diag}}}$ $(D_8, \mathbb{Z}_4^r)$ | 2Rep($\mathbb{Z}_2 \times \mathbb{Z}_2$) SSB $(\mathbb{Z}_4^r, \mathbb{Z}_4^r)$ |
| 2Rep($D_8$) SSB $(D_8, D_8)$ | $\frac{\text{DW}(\mathbb{Z}_2 \times \mathbb{Z}_2) \boxtimes \mathsf{Ising}}{\mathbb{Z}_2^{(0),\text{diag}}}$ $(D_8, \mathbb{Z}_2^{r^2} \times \mathbb{Z}_2^{xr})$ | 2Rep($\mathbb{Z}_2 \times \mathbb{Z}_2$) SSB $(\mathbb{Z}_2^{r^2} \times \mathbb{Z}_2^{xr}, \mathbb{Z}_2^{r^2} \times \mathbb{Z}_2^{xr})$ |
| 2Rep($D_8$) SSB $(D_8, D_8)$ | $\frac{\text{DW}(\mathbb{Z}_2 \times \mathbb{Z}_2) \boxtimes \mathsf{Ising}}{\mathbb{Z}_2^{(0),\text{diag}}}$ $(D_8, \mathbb{Z}_2^{r^2} \times \mathbb{Z}_2^x)$ | 2Rep($\mathbb{Z}_2 \times \mathbb{Z}_2$) SSB $(\mathbb{Z}_2^{r^2} \times \mathbb{Z}_2^x, \mathbb{Z}_2^{r^2} \times \mathbb{Z}_2^x)$ |

Table 3: Phase transitions between two gapped phases for 2Rep($D_8$), i.e. Rep($D_8$) 1-form symmetry. Columns one and three display the type of phase and subgroup data for the gapped phases $\mathcal{P}_A$ and $\mathcal{P}_B$, while the middle column shows the 2Rep($D_8$) CFT obtained after the KT transformation with input transition given by the 3d Ising CFT. The bottom three phases, with gauged $\mathbb{Z}_2^{(0),\text{diag}}$, are **igSSBs** for the 1-form symmetry: the number of 2Rep($D_8$) charged lines (obtained from bulk SymTFT surfaces ending on both $\mathfrak{B}_\text{sym}$ and $\mathfrak{B}_\text{phys}$) strictly increases when we gap the theory, as can be seen from the algebras in tables 8 and 7. The gapped phases were discussed in [2]. Here we have taken all 3-cocyles to be trivial.

1) and $\mathsf{Rep}(D_8)$ 1-form symmetry (table 3) and the 2-representation of 2-group symmetry $2\mathsf{Rep}(\mathbb{Z}_4^{(1)} \ltimes \mathbb{Z}_2^{(0)})$ (table 2). In these tables we have specified as the $\mathbb{Z}_2$ symmetric input phase transition the Ising transition in 2+1d. For $D_8$ there are also igSPTs, as discussed in section 6.4.1. However in this case the input phase transition requires a $\mathbb{Z}_2 \times \mathbb{Z}_2$ symmetric gapless phase with specific cocycle. It would be interesting to determine this and construct using our results the $D_8$-symmetric igSPT.

**Plan of the Paper.** The paper lays out the theory of phase transitions in the presence of (all bosonic) fusion 2-categories, whose SymTFTs are DW theories for finite groups $G$, and illustrates this with a detailed analysis of examples.

The reader interested in the main theoretical results is invited to focus on sections 2 for a SymTFT derivation of the gapless phases, and section 3 for a module category description. These are equivalent, but one provides a derivation using the club sandwich construction in the SymTFT, studying interfaces of DW theories (possibly with twist). The other generalizes the module category picture known for fusion categories, that corresponds to generalized gauging, to fusion 2-categories. In both descriptions we characterize minimal and non-minimal interfaces.

The simplest non-trivial example is $G = \mathbb{Z}_4$ DW theory, which we discuss in section 4. This has a natural generalization to DW for any abelian group $\mathbb{A}$, which is carried out in appendix B. These gapless phases based on abelian TOs are already interesting and realize igSPT and igSSB phases.

The framework can be applied to any $G$-DW theory, i.e. any bosonic fusion 2-category symmetry, but the case of non-abelian groups is substantially more involved. We present two examples: $S_3$ and $D_8$ and develop the full SymTFT picture including a recap of the gapped BCs, a complete analysis of the interfaces to reduced TOs, and the resulting KT transformations and second order phase transitions. This analysis can be found in section 5 for $S_3$ and for $D_8$ in section 6. These gives rise to igSSB ($S_3$ and $D_8$) and igSPTs ($D_8$).

## 2 Gapless Phases from the SymTFT

The goal of this paper is to determine systematically properties of critical models that arise as second order phase transitions in (2+1)d in theories with fusion 2-categorical symmetries. Such gapless phases can be constructed using the SymTFT framework, which in turn provides a systematic exploration of Kennedy-Tasaki (KT) transformations. The key idea is to start with a known phase transition for a small symmetry category $\mathcal{S}_0$ and to KT it to a larger

symmetry category $\mathcal{S}$. Concretely we will in fact start with $\mathcal{S}$ and determine all possible $\mathcal{S}_0$, which then enables us to construct $\mathcal{S}$ symmetric gapless phases from $\mathcal{S}_0$-symmetric ones. In our discussion in this section we will focus on the initial TO given by $\mathcal{Z}(\mathcal{S}) = \mathcal{Z}(2\mathsf{Vec}_G)$, though it can be generalized to $\mathcal{Z}(2\mathsf{Vec}_G^\varpi)$, as will be done in section 3, using module categories.

## 2.1 Categorical Symmetries in (2+1)d and their SymTFTs

Recently, fusion 2-categories have been classified [29] in terms of braided fusion categories and certain group cohomological data into two broad classes, dubbed *All Bosonic* (AB) type and *Emergent Fermionic* (EF) type. We stress that EF-type fusion 2-categories are still bosonic fusion 2-categories describing symmetries of bosonic 3d theories, and should not be confused with the notion of fermionic fusion 2-categories that would describe symmetries of fermionic 3d theories.

We consider AB fusion 2-category symmetries, which are related to ordinary 0-form symmetries for a finite group $G$, possibly with an anomaly $\varpi \in H^4(G, U(1))$ (denoted as $2\mathsf{Vec}_G^\varpi$), by generalized gauging operations. This property makes such symmetries amenable to a systematic SymTFT based study. The SymTFT associated to such symmetry types is simply the Dijkgraaf-Witten (DW) theory for a finite group $G$ with a topological action $\varpi \in H^4(G, U(1))$. Then the different fusion 2-category symmetries in the gauging web [45, 46] of $2\mathsf{Vec}_G^\varpi$ are nothing but the 2-categories of symmetry defects on the various gapped boundaries of the $2\mathsf{Vec}_G^\varpi$ SymTFT. The gapped boundary conditions (BCs) of (3+1)d DW theories have been classified [1, 2, 31, 32]. In contrast to 1+1d gapped boundaries of (2+1)d SymTFTs, in the (2+1)d setting there are infinitely many gapped boundary conditions for any given (3+1)d $G$-DW theory. This is due to the fact that there are infinitely many generalized gauging operators in (2+1)d.

**Symmetry Boundaries.** The canonical BC of the (3+1)d $G$-DW theory is the Dirichlet one $\mathfrak{B}_{\mathrm{Dir}}$ which realizes the symmetry category $2\mathsf{Vec}_G$ (i.e. 0-form symmetry $G^{(0)}$). Any other (of the infinitely many) gapped BCs is obtained by stacking with a $G$-symmetric TQFT and gauging an anomaly free diagonal subgroup of $G$. A gapped boundary condition is described in terms of which bulk topological defects can end on (or are condensed on) the given boundary. We will refer to an open neighborhood of a symmetry boundary $\mathfrak{B}_{\mathrm{sym}}$ with a certain collection

of bulk defects ending topologically as a **symmetry quiche**, which is depicted as:

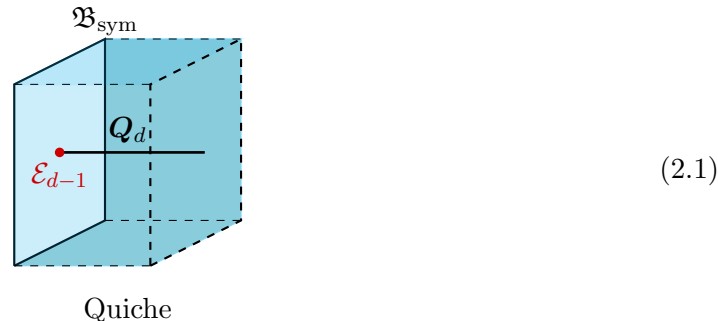

$$\text{(2.1)}$$

Quiche

The topological defects of the SymTFT, which form the Drinfeld center $\mathcal{Z}(2\mathsf{Vec}_G)$ can be of dimension $d = 0, 1, 2, 3$, however the 3d defects are purely condensation defects. From general considerations [47], we know that the topological defects of co-dimension 2 and higher can be organized as

$$\mathcal{Z}(2\mathsf{Vec}_G) = \boxplus_{[g]} \, 2\mathsf{Rep}(H_g) \,, \tag{2.2}$$

where we sum over all conjugacy classes $[g]$ of $G$, $H_g$ denotes the centralizer of $g \in [g]$ in $G$, and in each component $2\mathsf{Rep}(H_g)$ there is one non-trivial (up to condensation) surface defect labeled by

$$\boldsymbol{Q}_2^{[g]} \,, \tag{2.3}$$

with the identity surface operator given by $[g] = [\mathrm{id}]$. The line operators are denoted by

$$\boldsymbol{Q}_1^R \,, \qquad R \in \mathsf{Rep}(G) \,. \tag{2.4}$$

Furthermore there are line operators on the surfaces $\boldsymbol{Q}_2^{[g]}$, which form $\mathsf{Rep}(H_g)$.

**SymTFT Sandwiches and Generalized Charges.** Let $\mathcal{S}$ be a symmetry that can be constructed from a gapped boundary for $\mathcal{Z}(2\mathsf{Vec}_G)$, as in (2.1). We can couple any theory with $\mathcal{S}$ symmetry as the physical boundary and obtain the sandwich compactification of the SymTFT as follows:

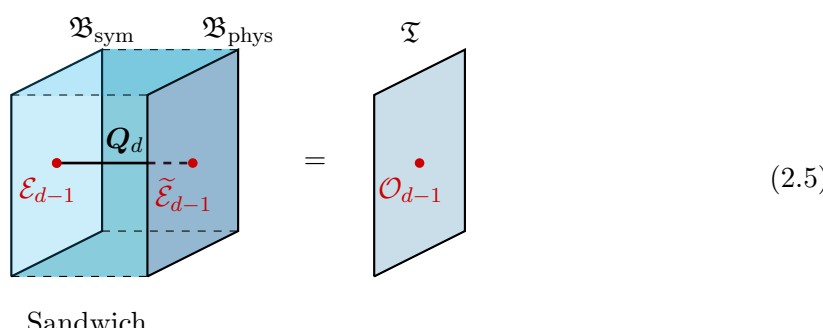

$$\text{(2.5)}$$

Sandwich

Since the symmetry defects are on the left boundary $\mathfrak{B}_{\text{sym}}$, while the dynamical non-symmetry defects are on the right boundary $\mathfrak{B}_{\text{phys}}$, the only way an operator can talk to (be charged under) $\mathcal{S}$ is by being attached to a bulk SymTFT defect that extends between the two boundaries. This provides a heuristic argument as to why the generalized $\mathcal{S}$ charges are classified by topological defects in the $\mathcal{S}$ SymTFT. Generalized charges of fusion 2-categories have recently been studied from equivalent perspectives of tube algebras [48,49] and SymTFT defects [50,51].

The (2+1)d theory $\mathfrak{T}$ obtained in (2.5) can be gapped or gapless. For gapped theories the physical boundary $\mathfrak{B}_{\text{phys}}$ is a gapped BC of the SymTFT, as we have studied in [1, 2]. This can be achieved systematically, including order parameters, symmetry action on the phase as gapped BC can be extremely well understood. We now describe the physical content of gapped boundaries and gapped phases in terms of generalized charges, that leads to a natural generalization to gapless phases which is the focus of this work.

## 2.2 Phases and Generalized Charges

The SymTFT $\mathfrak{Z}(\mathcal{S})$ for a fusion 2-categorical symmetry $\mathcal{S}$ can be used to systematically study the structure of gapped and gapless phases with symmetry $\mathcal{S}$ as well as the structure of certain transitions between gapless phases.

This is an extension of Bosonic [11, 13, 16, 17] and Fermionic [23, 24] fusion categorical symmetries in (1+1)d to AB-type fusion 2-categorical symmetries in (2+1)d [1, 2, 31].

A **phase** with $\mathcal{S}$ symmetry corresponds to a class of universal IR phenomena that are compatible with the symmetry $\mathcal{S}$. The universal symmetry properties of these phases can be understood in terms of condensed, confined and deconfined charges:

- **Condensed Charges:** At the most basic level an $\mathcal{S}$-symmetric phase $\mathcal{P}$ is characterized by the set of $\mathcal{S}$ generalized charges that are condensed in $\mathcal{P}$. Let us denote the set as $\mathcal{Q}_\mathcal{P}$ which generically contains generalized charges corresponding to both line and surface defects in $\mathfrak{Z}(\mathcal{S})$. The set of condensed charges acquire vacuum or ground state expectation values in the phase $\mathcal{P}$ and therefore act within the manifold of ground states. These charges appear in the IR theory. The set of generalized charges in $\mathcal{Q}_\mathcal{P}$ need to be mutually compatible. Firstly they need to be Bosonic such that the IR vacuum is Lorentz invariant. Secondly they need to be mutually local. These conditions equivalently describe a condensable algebra in $\mathcal{Z}(2\mathsf{Vec}_G)$. As we will see that condensable algebra also define co-dimension-1 topological defects (boundaries and interfaces) in the SymTFT and will be the central objects in our study of phases.

- **Confined Charges:** Any charge that is mutually non-local with at least one element

in $\mathcal{Q}_\mathcal{P}$ does not appear in the IR theory. In other words, in the UV theory acting with an operator carrying this generalized charge takes us out of the low energy subspace and is therefore projected out under the renormalization group flow. This phenomenon is completely analogous to the Meissner effect familiar from superconductivity.

- **Deconfined Charges:** Finally there may be generalized defects that are neither condensed nor confined, i.e., they do not appear in $\mathcal{Q}_\mathcal{P}$ but are mutually local with every element in $\mathcal{Q}_\mathcal{P}$. The set of deconfined charges will be denoted as $\mathcal{Q}_\mathcal{D}$. Mathematically the deconfined charges correspond to local modules of the condensable algebra defined via $\mathcal{Q}_\mathcal{P}$ [44,52]. These generalized charges potentially act within the low energy subspace and contribute to the dynamics of the phase. Operators carrying generalized charges in $\mathcal{Q}_\mathcal{D}$ potentially carry gapless excitations.

Let us contextualize the condensed, confined and deconfined charges within the SymTFT. This is summarized in figure 2. Firstly, as mentioned previously, the set of condensed charges $\mathcal{Q}_\mathcal{P}$ represents a condensable algebra in the SymTFT, which in turn defines a codimension-1 topological defect, which we denote as $\mathcal{I}_\mathcal{P}$. Elements in $\mathcal{Q}_\mathcal{P}$ are precisely the defects that can end (on untwisted operators) on $\mathcal{I}_\mathcal{P}$. The confined operators, as the name suggests are confined on the interface as they are non-local with respect to (at least one element in) $\mathcal{Q}_\mathcal{P}$. Finally, the deconfined defects $\mathcal{Q}_\mathcal{D}$ can pass through and the interface $\mathcal{I}_\mathcal{P}$ and become defects in a smaller reduced topological order (TO).

A comment is in order. Everything described thus far is modulo symmetry preserving deformations. For instance, there might be local uncharged operators that create gapless excitations which can clearly be gapped out while preserving the symmetry. The situation gets more interesting when one allows for topological (or condensed) line operators that are uncharged under the symmetry. This is precisely the scenario of stacking an $\mathcal{S}$-symmetric theory with a topological order that does not interact with the symmetry but does however influence the universal characteristics of the phase. The corresponding condensable algebras are called non-minimal and will be treated in detail in later sections.

**Gapped Phases from the SymTFT Sandwich.** As described above, a gapped phase corresponds to a set of condensed charges which are maximal, in the sense that all the charges that are not condensed are confined, i.e., $\mathcal{Q}_\mathcal{D} = \emptyset$. The interface corresponding to such a condensable algebra is indeed a boundary condition for the SymTFT. Mathematically, the 2-category of local modules over such a condensable algebra is trivial, i.e., the reduced TO is the vacuum. Then by taking the SymTFT sandwich, with a fixed symmetry boundary

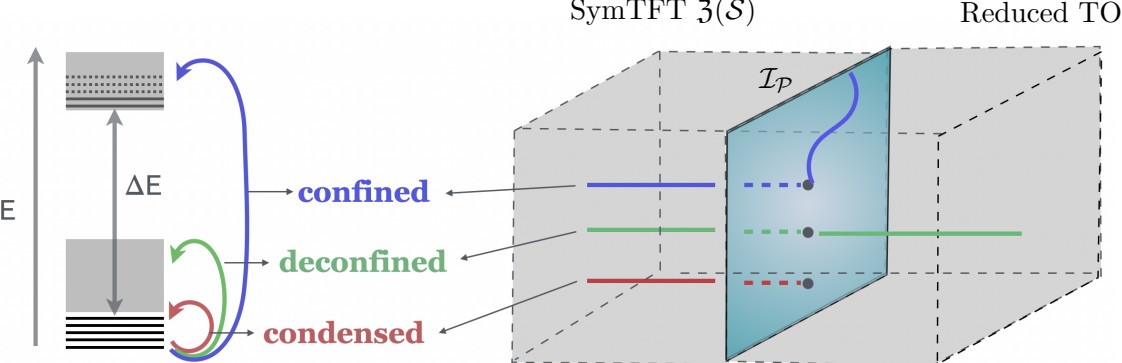

Figure 2: An $\mathcal{S}$-symmetric phase $\mathcal{P}$ is characterized by the set of condensed, confined and deconfined symmetry charges. Within the SymTFT, such a phase is modeled via a condensable algebra that defines a topological interface $\mathcal{I}_\mathcal{P}$ to a reduced topological order (TO). When $\mathcal{P}$ is a gapped phase, the reduced TO is trivial and $\mathcal{I}_\mathcal{P}$ is a gapped boundary.

corresponding to a fusion 2-category $\mathcal{S}$ and a gapped boundary condition for the physical boundary, one can construct and systematically characterize all $\mathcal{S}$-symmetric gapped phases [1, 2].

**Gapless Phases, KT Transformations and Club Sandwiches.** We may try to generalize the SymTFT approach for gapped phases to gapless phases. The most naive approach would be to simply input a gapless physical boundary and study the corresponding SymTFT sandwich. However this approach does not give one much mileage since gapless BCs of 4d TQFTs (as opposed to gapped BCs) are not any easier to study than gapless 3d quantum field theories. Instead we study gapless phases via KT transformations which are obtained from topological interfaces in the SymTFT. In this construction one obtains a reduced TO across the interface and therefore the gapless dynamics are formulated in terms of fewer generalized charges. Using an interface and an input gapless theory with a smaller symmetry structure, one can construct a theory carrying the larger symmetry:

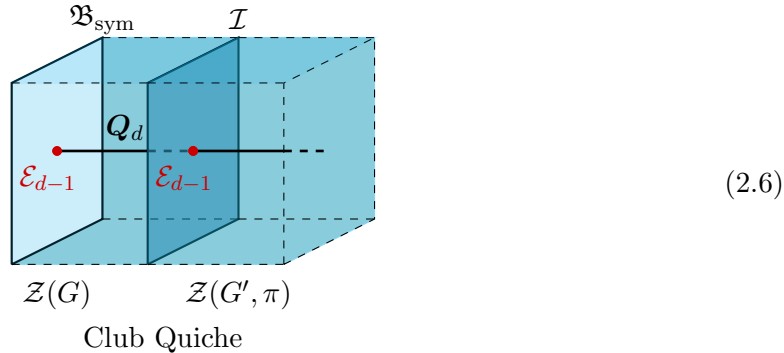

(2.6)

In this work we classify all such interfaces as well as provide their physical understanding in terms of topological defects of the SymTFT and the reduced TO. Starting from the club quiche, one can compactify the interval occupied by $\mathcal{Z}(G)$ such that we obtain a boundary condition $\mathfrak{B}'$ for $\mathcal{Z}(G', \pi)$ where

$$\mathfrak{B}' = \mathfrak{B}_{\text{sym}} \otimes_{\mathcal{Z}(G)} \mathcal{I}. \tag{2.7}$$

This is depicted as

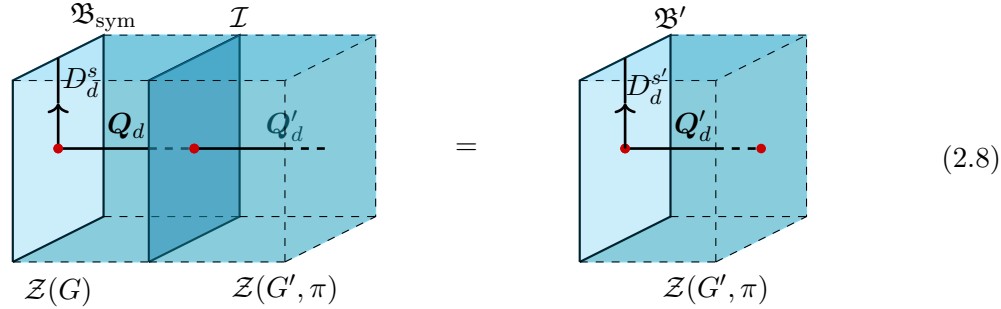

The symmetry on $\mathfrak{B}'$ will be denoted as $\mathcal{S}'$. We emphasize that this is distinct from $\mathcal{S}_0$. In (2.8) a bulk symmetry defect $\boldsymbol{Q}_d$ ends on a topological operator in the twisted sector of a symmetry defect $D_d^s$ on $\mathfrak{B}_{\text{sym}}$. At the interface $\boldsymbol{Q}_d$ is attached to $\boldsymbol{Q}_d'$ in the reduced TO. Therefore after compactifying the interval occupied by $\mathcal{Z}(G)$, one obtains a configuration where $\boldsymbol{Q}_d'$ ends on $\mathfrak{B}'$ on some topological local operator attached to a symmetry defect $D_d^{s'} \in \mathcal{S}'$. This construction furnishes a functor

$$\mathcal{F} : \mathcal{S} \longrightarrow \mathcal{S}', \tag{2.9}$$

such that $D_d^{s'} \in \mathcal{F}(D_d^s)$. The number of topological local operators in $\mathfrak{B}'$ are

$$\mathsf{N} = \sum_R \mathsf{N}_{\text{sym}}^R \mathsf{N}_{\mathcal{I}}^R, \tag{2.10}$$

where $\mathsf{N}_{\text{sym}}^R$ and $\mathsf{N}_{\mathcal{I}}^R$ are the number of topological (untwisted) ends of $\boldsymbol{Q}_1^R$ for $R \in \mathsf{Rep}(G)$ on $\mathfrak{B}_{\text{sym}}$ and $\mathcal{I}$ respectively. Correspondingly the symmetry category $\mathcal{S}'$ is a multifusion 2-category consisting of $\mathsf{N}$ indecomposable fusion 2-categories $\mathcal{S}_{ii}'$ and $(\mathcal{S}_{ii}', \mathcal{S}_{jj}')$-bimodule 2-categories $\mathcal{S}_{ij}'$, where $i, j = 1, 2, \cdots, \mathsf{N}$. This maybe viewed as a 3-category, with objects labeling the components which occupy the 3-strata on the 3-manifold where $\mathfrak{B}'$ is located. Morphisms being topological surfaces that occupy 2-strata and so on.

The club quiche can be used to then construct gapless phases for $\mathcal{S}$ symmetric theories by

starting with those for $G'$-symmetric theories:

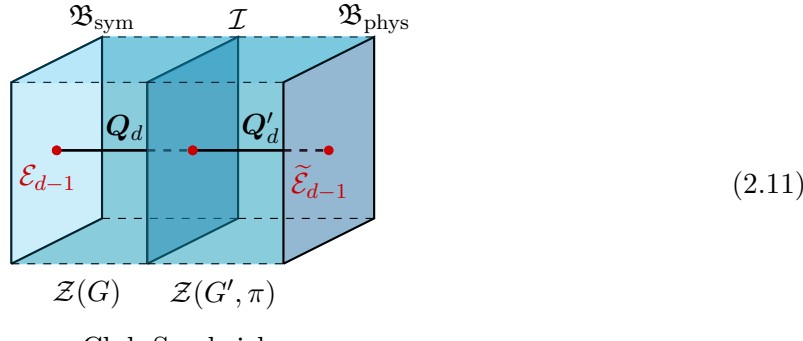

$$\text{(2.11)}$$

Club Sandwich

In order to obtain $\mathcal{S}$ symmetric gapless phase (where $\mathcal{S}$ is the fusion 2-category on $\mathfrak{B}_{\text{sym}}$ on the boundary of $\mathcal{Z}(G)$), using the club quiche with the interface $\mathcal{I}$, we insert a physical boundary corresponding to a gapless phase with $2\mathsf{Vec}_{G'}^\pi$ symmetry. The physical boundary for a gapless phase is required to satisfy certain physically motivated requirements:

- **Maximal Gaplessness:** None of the topological defects in the reduced TO can end on $\mathfrak{B}_{\text{phys}}$ on untwisted topological operators. This ensures that there are no condensed charges in addition to the ones already accounted for in the condensable algebra defining the interface $\mathcal{I}$.

- **Completeness of the Spectrum:** Every topological defect is attached to at least one non-topological operator on $\mathfrak{B}_{\text{phys}}$, i.e., $\mathfrak{B}_{\text{phys}}$ has a complete spectrum of (reduced) generalized charges.

- **Irreducibility:** There are no topological local operators unattached to any bulk or boundary lines on $\mathfrak{B}_{\text{phys}}$. This implies that the input gapless theory is irreducible, in the sense that it does not decompose into a direct sum of dynamically disconnected universes.

A special kind of a gapless theory is one that realizes a phase transition. Consider a phase transition between gapped phases for $G'$: $\mathfrak{T}'_1$ and $\mathfrak{T}'_2$, and denote the gapless phase by $\mathfrak{T}'_{12}$. Then the club sandwich provides a transformation from this phase transition to an $\mathcal{S}$-symmetry phase transition, by considering the partial compactification of the club sandwich. The key is then to classify all possible interfaces $\mathcal{I}$, which we will identify with the problem of classifying condensable algebras of the Drinfeld center $\mathcal{Z}(G)$, and the possible reduced topological orders. The interfaces define map between these two topological orders. Equivalently we can think about this interface in terms of a gapped BC of the folded TO

$$\overline{\mathcal{Z}(G)} \boxtimes \mathcal{Z}(G', \pi) \,. \tag{2.12}$$

**Interfaces from Condensable Algebras and Folded TOs.** Our starting point is the topological order $\mathcal{Z}(G) = \mathcal{Z}(2\mathsf{Vec}_G)$, where $G$ is a finite group. We would like to determine the complete set of interfaces $\mathcal{I}$ between this TO and a reduced topological order. The interfaces need to satisfy some conditions in order to give rise to KT transformations. Intuitively these correspond to the fact that any non-trivial topological defect $\boldsymbol{Q}_2^{\alpha'}$ in the reduced order $\mathcal{Z}(G', \pi)$, needs to come from a non-trivial one in the original TO $\mathcal{Z}(G)$, see (2.11). We can think of this in terms of the folded TO (2.12), where the interface corresponds to a gapped BC, which has to be such that the surfaces that end

$$\sum_{\alpha, \alpha'} n_{\alpha, \alpha'} \overline{\boldsymbol{Q}_2^{\alpha}} \boxtimes \boldsymbol{Q}_2^{\alpha'} \tag{2.13}$$

satisfy that for all $\boldsymbol{Q}_2^{\alpha'}$ in $\mathcal{Z}(G', \pi)$, there exists $\alpha$ such that $n_{\alpha, \alpha'} \neq 0$. Mathematically we require there to be a map

$$\mathcal{F}: \quad \mathcal{Z}(G', \pi) \to \mathcal{Z}(G), \tag{2.14}$$

such that the kernel $\ker \mathcal{F}$ is trivial. In the main text we will focus on the characterization of such interfaces coming from the folded picture.

**Minimal and Non-Minimal Interfaces** For $(2+1)$d, the $(3+1)$d SymTFT has an infinite number of gapped BCs. Likewise, there is an infinite number of such interfaces to reduced TOs, which is easy to see once we consider the interface as coming from a gapped BC of the folded TO (2.12). As with gapped BCs, we will distinguish again minimal and non-minimal such interfaces:

- Minimal Interfaces: All the topological line operators on a minimal interface are obtainable as the interface projection of a line operator in the SymTFT. These are constructed as minimal gapped BCs of the folded TO (2.12), and thus obtained by stacking with SPTs and gauging a subgroup.

- Non-minimal Interfaces: Conversely, not all the topological line operators on a non-minimal interface are obtainable as the interface projection of a line operator in the SymTFT. Some lines are intrinsic to the interface. These correspond to non-minimal gapped BCs of the folded TO (2.12), and thus obtained by stacking with a non-trivial $G \times G'$ TQFT and subsequently gauging a diagonal subsymmetry.

**General Criteria for igSPT and igSSB.** There are various interesting symmetry protected gapless theories.

- igSPT for a symmetry $\mathcal{S}$ is a gapless phase, which cannot be gapped to a fully $\mathcal{S}$ symmetric phase.

- igSSB for a symmetry $\mathcal{S}$ is a gapless phase that breaks $\mathcal{S}$ spontaneously, but cannot be gapped without further breaking the symmetry.

Concretely, this means that applied to a 0-form symmetry the number of charged local operators increases if we gap the theory, therefore the 0-form symmetry gets (further) spontaneously broken along the RG flow. For a 1-form symmetry the number of charged line operators increases if we gap the theory, therefore the 1-form symmetry gets (further) spontaneously broken along the RG flow. So in both instances of igSPTs and igSSBs the degree of symmetry breaking increases when we gap the theory. The difference is whether the gapless theory fully preserves the symmetry $\mathcal{S}$ or is breaking it spontaneously.

**Partial Order on Phases and Hasse Diagrams.** Starting from a certain phase $\mathcal{P}$ defined via a set of condensed charges $\mathcal{Q}_\mathcal{P}$, a further deconfined charge, i.e., an element of $\mathcal{Q}_\mathcal{D}$ can be condensed without un-condensing any of the condensed charges. Such an additional condensation produces a less gapless phase $\mathcal{P}'$. Specifically we mean that the number of dynamical or deconfined generalized charges in $\mathcal{P}'$ is fewer than in $\mathcal{P}$. A gapped phase is one where there are no deconfined charges. Therefore one obtains a partial ordering on phases, given by the inclusion of algebras. The complete set of condensable algebras along with the partial ordering can be conveniently organized into a Hasse diagram. We will carry this out concretely for the minimal interfaces.

## 2.3 Minimal Interfaces

In this section we will start with the characterization of minimal interfaces (MI) from the 4d $G$ DW with a trivial topological action to a reduced TO. These topological interfaces are specified by the following data:

1. A subgroup $H$ of $G$ (up to conjugation).

2. A normal subgroup $N$ of $H$ (up to $H$-preserving conjugation).

3. A cohomology class $[\omega] \in H^3(H, U(1))$.

4. A cohomology class $[\pi] \in H^4(H/N, U(1))$ such that $[p^*\pi] = 0 \in H^4(H, U(1))$, where $p : H \to H/N$ is the canonical projection.

We will refer to this interface as

$$\mathcal{I}_{(H,N,\pi,\omega)}. \tag{2.15}$$

and we will drop some of these if they are redundant. Note that $\mathcal{I}_{(H,N,\pi,\omega)}$ provides an interface to the reduced TO $\mathcal{Z}(H/N,\pi)$:

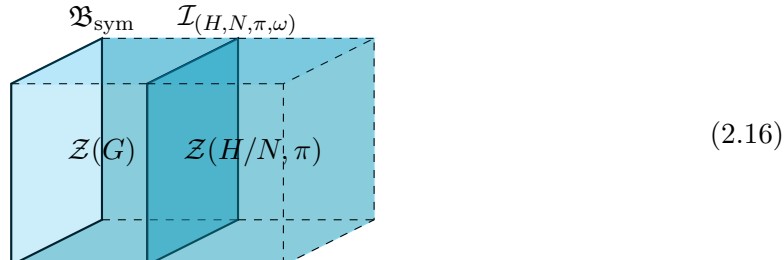

$$\tag{2.16}$$

This implies in particular that interfaces with $H = N$ correspond to gapped BCs. Let's unpack the meaning of each datum, and what is specified in terms of the interfaces:

- **The subgroup $H < G$:** Corresponds to the following information: There is a functor

$$\kappa : \mathsf{Rep}(G) \longrightarrow \mathsf{Rep}(H), \tag{2.17}$$

which decomposes $G$-irreps into $H$-irreps. The category of genuine lines on the interface forms $\mathsf{Rep}(H)$ which is obtained by a projection of bulk lines labeled by $R \in \mathsf{Rep}(G)$ to $\kappa(R) \in \mathsf{Rep}(H)$. Consequently all the representation lines $\boldsymbol{Q}_1^R$ with $R \in \ker(\kappa)$ can end on the interface[3]:

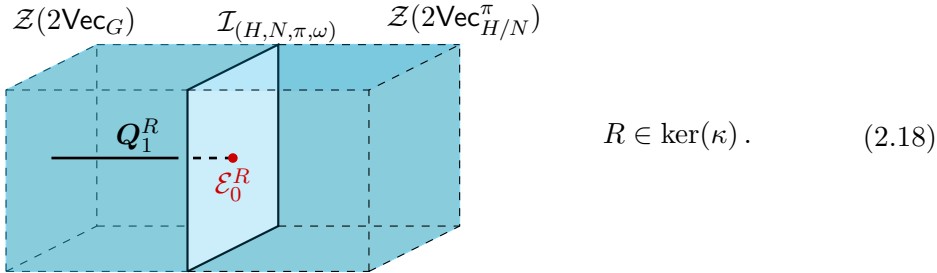

$$R \in \ker(\kappa). \tag{2.18}$$

- **$N$ a normal subgroup of $H$:** The SymTFT surfaces $\boldsymbol{Q}_2^{[n]}$ for $n \in N$ can have genuine (or untwisted) ends on the interface, i.e. they end in lines $\mathcal{E}_1^{[n]}$ on the interface $\mathcal{I}_{(H,N,\pi,\omega)}$:

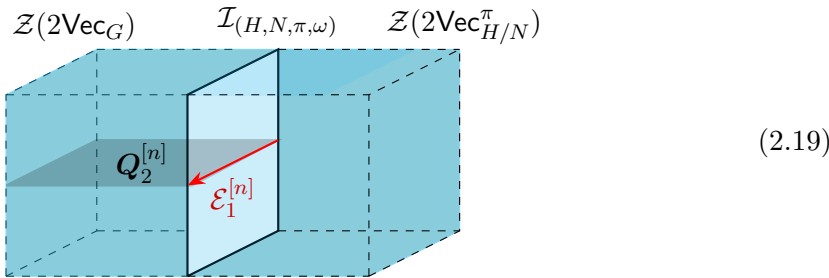

$$\tag{2.19}$$

---

[3]Here, we say that $R \in \mathsf{Rep}(G)$ is in the kernel of $\kappa$ if and only if $\kappa(R)$ contains the unit object of $\mathsf{Rep}(H)$.

Furthermore the surfaces that can pass through the interface are $Q_2^{[h]}$ where $h \in H$. These surfaces are attached to $Q_2^{[p(h)]}$ in the reduced TO, where we have used the fact that $p(h) \in H/N$ and the surfaces in the reduced TO are labeled by conjugacy classes in $H/N$.

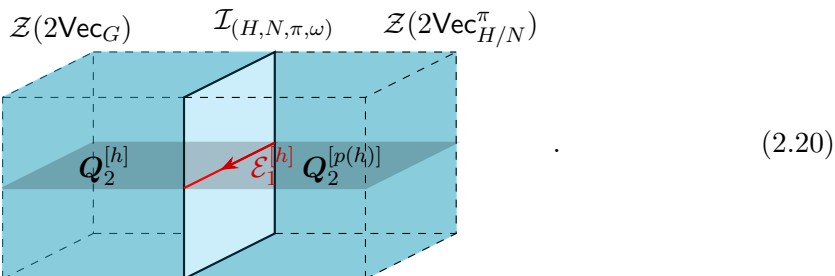

$$(2.20)$$

- **The cohomology class $\omega \in H^3(H, U(1))$:** The associativity properties of the lines $\mathcal{E}_1^{[h]}$ are controlled by $\omega$. More precisely, the category of genuine and non-genuine lines on the interface $\mathcal{I}_{(H,N,\pi,\omega)}$ form the 3d $H$ Dijkgraaf-Witten with topological action $\omega \in H^3(H, U(1))$. The genuine lines are precisely the (charge) Wilson lines that are labeled by representations of $H$, while the non-genuine lines are all the remaining lines.

- **$\pi \in H^4(H/N, U(1))$:** This cohomology class pulls back trivially to $H$. It determines the detailed structure of the topological defects on the interface. Specifically the topological defects on the interface form a category related to $2\mathsf{Vec}_G \boxtimes 2\mathsf{Vec}_{H/N}^\pi$ by a gauging of $H^{\mathrm{diag}}$ defined by (2.24).[4]

**Interfaces from Folding and Gauging.** Equivalently, the interface $\mathcal{I}_{(H,N,\omega,\pi)}$ can also be constructed via the "folding and gauging" approach depicted as

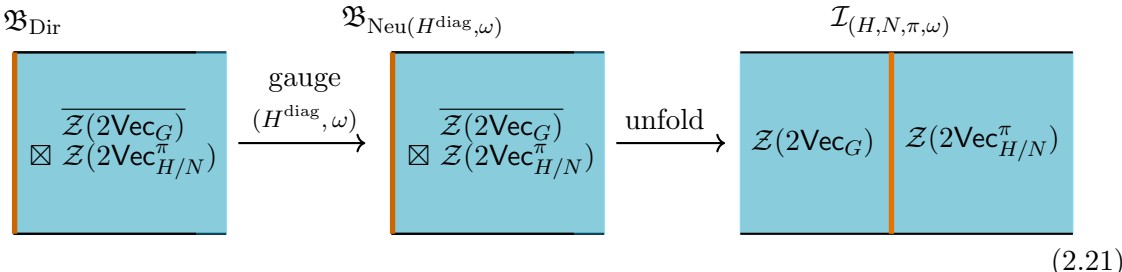

$$(2.21)$$

in the following way. We consider the folded theory

$$\overline{\mathcal{Z}(2\mathsf{Vec}_G)} \boxtimes \mathcal{Z}(2\mathsf{Vec}_{H/N}^\pi), \qquad (2.22)$$

---

[4]The choice of discrete torsion $\omega \in H^3(H, U(1))$ does not modify the category of genuine topological defects on the interface.

with the Dirichlet boundary condition. The Dirichlet boundary condition corresponds to an impenetrable interface upon unfolding, i.e., one where all lines from the topological orders from either side end. Instead by gauging appropriately on the Dirichlet and then unfolding, we can naturally recover all possible interfaces between $\mathcal{Z}(2\mathsf{Vec}_G)$ and $\mathcal{Z}(2\mathsf{Vec}^{\pi}_{H/N})$. The category of topological defects on $\mathfrak{B}_{\mathrm{Dir}}$ corresponds to the $G \times H/N$ 0-form symmetry, with $\pi$ anomaly on the second factor. In order to recover the interface $\mathcal{I}_{(H,N,\pi,\omega)}$ we can gauge a diagonal $H^{\mathrm{diag}} \subset G \times H/N$ with a choice of discrete torsion $\omega \in H^3(H^{\mathrm{diag}}, U(1))$ on $\mathfrak{B}_{\mathrm{Dir}}$ and unfold the SymTFT. We have the exact sequence

$$1 \longrightarrow N \longrightarrow H \overset{p}{\longrightarrow} H/N \longrightarrow 1, \tag{2.23}$$

The subgroup $H$ can be written as

$$H \cong H^{\mathrm{diag}} := \{(h, p(h))\} < G \times H/N. \tag{2.24}$$

The surfaces in the folded theory can be denoted by

$$\boldsymbol{Q}_2^{[g_L],[g_R]} \equiv \boldsymbol{Q}_2^{[g_L]} \times \boldsymbol{Q}_2^{[g_R]}, \tag{2.25}$$

where $\boldsymbol{Q}_2^{[g_L]}$ and $\boldsymbol{Q}_2^{[g_R]}$ are surfaces in $\mathcal{Z}(2\mathsf{Vec}_G)$ and $\mathcal{Z}(2\mathsf{Vec}^{\pi}_{H/N})$, respectively. After gauging $H^{\mathrm{diag}}$, we obtain a gapped boundary condition

$$\mathfrak{B}_{\mathrm{Neu}(H^{\mathrm{diag}}),\omega} = \frac{\mathfrak{B}_{\mathrm{Dir}}}{(H^{\mathrm{diag}}, \omega)}, \tag{2.26}$$

on which any surface labeled by a conjugacy class in $H^{\mathrm{diag}}$ can end. In particular, $H^{\mathrm{diag}}$ does not have any element of the form $(1, g_R) \in G \times H/N$ for $g_R \neq 1$. This implies that no surface can end on the corresponding interface from the reduced TO after unfolding. In general there are two kinds of surfaces that have untwisted ends on $\mathfrak{B}_{\mathrm{Neu}(H^{\mathrm{diag}}),\omega}$. The first are those that belong entirely to $\mathcal{Z}(2\mathsf{Vec}_G)$, i.e., for which $h \in \ker(p) \cong N$. The second are those for which $h \notin N$. The untwisted ends of these surfaces on $\mathfrak{B}_{\mathrm{Neu}(H^{\mathrm{diag}}),\omega}$ are (see (2.19) and (2.20) for the picture after unfolding)

$$\begin{aligned} \boldsymbol{Q}_2^{[n],[\mathrm{id}]} \bigg|_{\mathrm{Neu}(H^{\mathrm{diag}}),\omega} &= \mathcal{E}_1^{[n]}, \\ \boldsymbol{Q}_2^{[h],[p(h)]} \bigg|_{\mathrm{Neu}(H^{\mathrm{diag}}),\omega} &= \mathcal{E}_1^{[h],[p(h)]}, \end{aligned} \tag{2.27}$$

The choice of discrete torsion determines the associators of surfaces that can end on the boundary. After unfolding back, we obtain an interface $\mathcal{I}_{(H,N,\pi,\omega)}$ (see (5.38)). The surfaces

$\boldsymbol{Q}_2^{[h]}$ become the surfaces $\boldsymbol{Q}_2^{[p(h)]}$ on the other side of the interface. The topological line located at the locus where $\boldsymbol{Q}_2^{[h]}$ from $\mathcal{Z}(2\mathsf{Vec}_G)$ intersects the interface to become $\boldsymbol{Q}_2^{[p(h)]}$ from $\mathcal{Z}(2\mathsf{Vec}_{H/N}^\pi)$ corresponds to $\mathcal{E}_1^{[h],[p(h)]}$. This line has F-symbols that are determined by the 3-cocycle $\omega \in H^3(H, U(1))$. The folding construction will be particularly useful in constructing non-minimal interfaces in the SymTFT which would in turn provide an understanding of either gapless phases with "non-minimal symmetries" or "non-minimal gapless phases" with minimal symmetries.

**Hasse Diagram.** The partial order on condensable algebras and thus phases can be concretely characterized in terms of the data that defines the interfaces as follows: The partial order is such that

$$\mathcal{I}(H_1, N_1, \pi_1, \omega_1) < \mathcal{I}(H_2, N_2, \pi_2, \omega_2), \tag{2.28}$$

if $H_1 > H_2$, $N_1 < N_2$ and $\omega_1|_{H_2} = \omega_2$ and $\pi_1|_{H2/N1} = p^*\pi_2$ where $p : H_2/N_1 \to H_2/N_2$. A proof of this is in appendix A. We can then arrange the algebras and associated gapless/gapped phases in a Hasse diagram.

**Criteria for igSPT and igSSB Phases.** In terms of topological defects of the SymTFT, a phase is characterized by the number of topological local operators

$$\mathsf{N}_0 = \sum_R \mathsf{N}_{\text{sym}}^R \mathsf{N}_{\text{phys}}^R, \tag{2.29}$$

where $\mathsf{N}_{\text{sym/phys}}^R$ is the number of topological (untwisted) ends of $\boldsymbol{Q}_1^R$ for $R \in \mathsf{Rep}(G)$ on the SymTFT symmetry or physical boundary, and by the number $\mathsf{N}_1$ of linearly independent charged topological line operators.

A gapless phase with $\mathsf{N}_0^{\text{gapless}}$ topological local operators and $\mathsf{N}_1^{\text{gapless}}$ topological charged line operators a given set of topological line operators is intrinsically gapless if all the gapped phases it can be deformed to have:

$$\mathsf{N}_0^{\text{gapped}} > \mathsf{N}_0^{\text{gapless}}, \quad \text{and/or} \quad \mathsf{N}_1^{\text{gapped}} > N_1^{\text{gapless}}. \tag{2.30}$$

The fact that the number of linearity independent charged local and/or charged line operators increases implies that the symmetry of the gapless theory gets (further) spontaneously broken when we gap it, i.e. the gaplessness of the theory is protected by this symmetry. Examples are provided for gapless theories arising from abelian group DW theories in section B.5, and for $D_8$ DW theory in section 6.4.

**Intrinsically gapless 0-form symmetric phases from twist in reduced topological order.** Let us now return to the possible 4-cocycle twist $\pi \in H^4(H/N, U(1))$. Recall that when $N \neq H$ and $\pi$ is non-trivial, the partial order in section A implies that the gapless phase determined by the interface $\mathcal{I}_{(H,N,\pi,\omega)}$ can only flow to gapped phases labeled by $H' < H$ and such that the 4-cocycle trivializes on $H'/N$, i.e., $\pi|_{H'/N} = 1$. Therefore, if we fix a symmetry boundary preserving $H$ 0-form symmetry, the interface $\mathcal{I}_{(H,N,\pi,\omega)}$ will necessarily be intrinsically gapless, since only a subgroup $H' < H$ can be preserved in the gapped phases obtainable from deformations of the gapless phase preserving $H$. For example, if we fix $\mathfrak{B}_{\text{sym}} = \mathfrak{B}_{\text{Dir}}$ and $H = G$ then the gapless phase is an igSPT for $2\text{Vec}_G$, while if $H < G$ the gapless phase is an igSSB.

Note that if we write $\pi$ as a cup product between $\eta \in H^2(H/N, \widehat{N})$ and $e \in H^2(H/N, N)$

$$\pi(h_1, h_2, h_3, h_4) = (\eta \cup e)(h_1, h_2, h_3, h_4) = \eta\big(h_1, h_2 | e(h_3, h_4)\big), \tag{2.31}$$

then its pullback to $H$ is trivial (this condition is necessary if $H$ had no 4-cocycle twist).[5] Here, $\eta(\cdot, \cdot | n)$ defines a 2-cocycle on $H/N$ for each $n \in N$ and for each $h_1 N, h_2 H \in H/N$, $\eta(h_1 N, h_2 N | \cdot)$ defines a one-dimensional representation of $N$, while $e \in H^2(H/N, N)$ denotes the extension class

$$1 \to N \hookrightarrow H \xrightarrow{p} H/N \to 1 \tag{2.32}$$

and can be computed via any section $s : H/N \to H$ to the canonical projection $p : H \to H/N$ as:

$$e(h_1 H, h_2 N) = s(h_1 N)\, s(h_2 N)\, s(h_1 N h_2 N)^{-1}. \tag{2.33}$$

Therefore, from $\eta$ and $e$ one can construct $\pi = \eta \cup e \in H^4(H/N, U(1))$ which is guaranteed to be trivial when pulled-back to $H$. If $\pi$ is then a representative of a non-trivial cohomology class in $H^4(H/N, U(1))$, after fixing a symmetry boundary preserving $H$ 0-form symmetry, the interface $\mathcal{I}_{(H,N,\pi,\omega)}$ will give rise to an intrinsically gapless phase.

## 2.4 Non-Minimal Interfaces

A non-minimal interface has the defining property that it contains some topological line operators that are native to it, i.e., they are not obtainable by the boundary projection of a line in the ambient 4d TFT. Equivalently there is no topological local operator that can be used to lift such a line off the 3d interface. Any such line is remotely detectable, i.e., braids

---

[5]This is because $\pi = \eta \cup e \in H^4(H/N, U(1))$ is in the image of the homomorphism $d_2' : H^2\big(H/N, \widehat{N}\big) \to H^4(H/N, U(1))$, corresponding to the $d_2$ differential of the Lyndon-Hochschild-Serre spectral sequence on the $E_2$-page [53], which is hence pulled back via the canonical projection $p : H \twoheadrightarrow H/N$ to the trivial cohomology class in $H^4(H, U(1))$ i.e., $p^*(\eta \cup e) = d\nu$ for some 3-cochain $\nu$ on $H$.

non-trivially with at least one other topological line. Non-chiral non-minimal interfaces are classified by the following data:[6]

1. A subgroup $H$ of $G$ (up to conjugation).

2. A normal subgroup $N$ of $H$ (up to $H$-preserving conjugation).

3. A cohomology class $[\pi] \in H^4(H/N, U(1))$.

4. A $p^*\pi$-twisted $H$-graded fusion category $\mathcal{A}$.

We will denote them by

$$\mathcal{I}_{(H,N,\pi,\mathcal{A})} \,. \tag{2.34}$$

The case of minimal interfaces can be recovered by fixing $\mathcal{A}$ to be $\mathsf{Vec}_H^\omega$ (viewed as an $H$ graded fusion category). Then the properties of this algebra are completely captured in $\omega \in H^3(H, U(1))$. Let us now provide a physical characterization of the different input data in the definition of a non-minimal interface in terms of topological defects of the SymTFT and the reduced TO.

The first three pieces of input data, i.e., $N$, $H$ and $\pi$ play exactly the same role as for minimal interfaces. The difference lies in the fourth input datum $\mathcal{A}$. Specifically, genuine and non-genuine lines on the interface $\mathcal{I}_{(H,N,\pi,\mathcal{A})}$ form the category

$$\mathcal{Z}(\mathcal{A}) \,, \tag{2.35}$$

which, when $[p^*\pi] = 0$, becomes a Modular Tensor Category (MTC) and agrees with the Drinfeld center of $\mathcal{A}$. In particular, $\mathcal{Z}(\mathcal{A})$ always has a (maximal) condensable sub-category of $\mathsf{Rep}(H)$ lines. These can be used to endow $\mathcal{Z}(\mathcal{A})$ with a grading into $H$ conjugacy classes. Specifically we decompose

$$\mathcal{Z}(\mathcal{A}) = \bigoplus_{[h]} \widetilde{\mathcal{M}}^{[h]} \,. \tag{2.36}$$

Let us denote the lines generating $\mathsf{Rep}(H)$ as $D_1^R$. Then any line in $\widetilde{\mathcal{M}}^{[h]}$ has a braiding (Hopf-linking) phase with $D_1^R$ given by the character $\chi_R([h])$. Physically the difference to minimal interfaces, where $\mathcal{E}_1^{[h]}$ was a simple line (up to fusion by $\mathsf{Rep}(H)$ lines) is that for non-minimal ones, it is a grade in a braided fusion category, i.e. we consider the replacement

$$\mathcal{E}_1^{[h]} \longrightarrow \widetilde{\mathcal{M}}^{[h]} \,. \tag{2.37}$$

---

[6]Here, non-chiral interfaces are those that have gapped interfaces to the Dirichlet interface, see section 1.

This is depicted as follows:

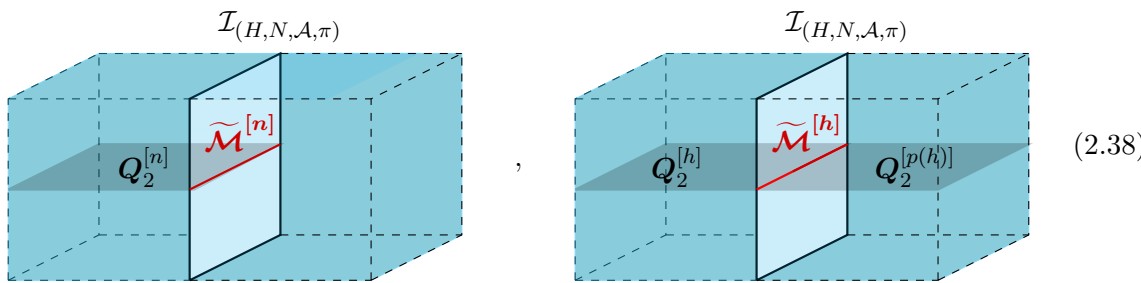

$$\tag{2.38}$$

**Non-Minimal Interfaces from Folding and Gauging.** As with minimal interfaces, all non-minimal interfaces can be constructed by starting from a Dirichlet (impenetrable) interface or equivalently the canonical Dirichlet boundary and performing a non-minimal gauging. Specifically the non-minimal gauging involves first stacking with the MTC

$$\mathcal{M} = \mathcal{Z}(\mathcal{A}^{\mathrm{id}}) \,, \tag{2.39}$$

where $\mathcal{A}^{\mathrm{id}}$ is the identity grade in the $H$-graded $p^*\pi$-twisted fusion category $\mathcal{A}$. $\mathcal{Z}(\mathcal{A}^{\mathrm{id}})$ has a natural action of $H$ symmetry that can also be formulated in terms of $\mathcal{A}$ [54]. Specifically, the category of lines on the 2d boundary of $\mathcal{Z}(\mathcal{A}^{\mathrm{id}})$ form $\mathcal{A}$, where the lines in some grade $\mathcal{A}^h$ are attached to the bulk condensation defect in $\mathcal{Z}(\mathcal{A}^{\mathrm{id}})$ implementing the $h \in H$ symmetry. Given this $H$ action, one can gauge the diagonal $H$ symmetry between the stacked topological order and

$$H = \{(h\,, p(h))\} < G \times H/N \,. \tag{2.40}$$

Upon unfolding the gauged boundary condition, we obtain the interface $\mathcal{I}_{(H,N,\pi,\mathcal{A})}$.

## 2.5 Input Phase Transitions

A key ingredient for the generalized KT transformation is the input phase transition for the smaller symmetry $\mathcal{S}'$, whose SymTFT is the DW theory for $H/N$. In (2+1)d the main example of a well-established symmetric phase transition is $H/N = \mathbb{Z}_2$.

$\mathbb{Z}_2$ **Phase Transition.** The first one is a $\mathbb{Z}_2$ SSB transition given by the 3d Ising CFT, see figure 3. The phase transition between the $\mathbb{Z}_2$ SPT and the $\mathbb{Z}_2$ SSB phase is given by the stacking of the 3d Ising CFT with the $\mathbb{Z}_2$ SPT state, which we denote as Ising$\boxtimes\mathbb{Z}_2$-SPT. This is an example of the symmetry enriched quantum critical point, where the twisted sectors of the CFT carry the non-trivial topological invariants of the SPT state [55]. The transition between the $\mathbb{Z}_2$ trivial and the $\mathbb{Z}_2$ SPT phases was found to be described by a first order transition [56]. We summarize the phase diagram in figure 3.

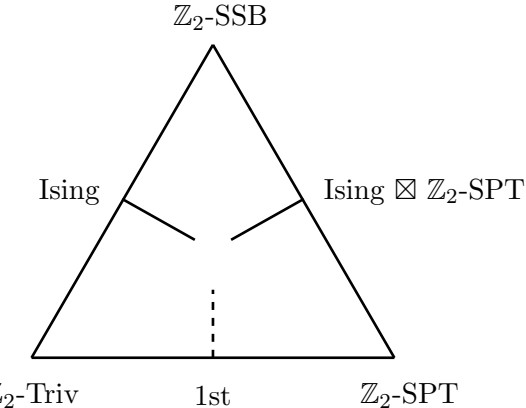

Figure 3: Schematic phase diagram for phases with $\mathbb{Z}_2$ symmetry. The $\mathbb{Z}_2$ SSB transition is described by the 3d Ising CFT, which is denoted as Ising. The phase transition between the $\mathbb{Z}_2$ SPT and the $\mathbb{Z}_2$ SSB phase is given by the stacking of the 3d Ising CFT with the $\mathbb{Z}_2$ SPT state, which we denote as Ising $\boxtimes$ $\mathbb{Z}_2$-SPT. The transition between the $\mathbb{Z}_2$ trivial and the $\mathbb{Z}_2$ SPT phases was found to be described by a first order transition.

The situation near the center of the phase diagram is less clear. One proposal is that it is a multicritical point described by the $SO(5)$ deconfined quantum critical point. A more detailed discussion of the phase diagram and the numerical studies can be found in Ref. [56].

**Gauged $\mathbb{Z}_2$ Phase Transition.** We can gauge the $\mathbb{Z}_2$ phase diagram and obtain a phase diagram for phase transitions with the $\mathbb{Z}_2^{(1)}$ 1-form symmetry. A schematic phase diagram is shown in figure 4. After gauging the $\mathbb{Z}_2$ symmetry, the $\mathbb{Z}_2$ SSB phase becomes the $\mathbb{Z}_2^{(1)}$ trivial phase. The $\mathbb{Z}_2$ trivial phase becomes the $\mathbb{Z}_2^{(1)}$ SSB phase, which can be recognized as the $\mathbb{Z}_2$ DW theory without twist, or equivalently as the Toric Code topological order [57]. We denote this phase as $\mathrm{DW}(\mathbb{Z}_2)_0$. The $\mathbb{Z}_2$ SPT phase becomes the other $\mathbb{Z}_2^{(1)}$ SSB phase, which is the $\mathbb{Z}_2$ DW theory with a non-trivial twist, or equivalently the double semion topological order. We denote this phase as $\mathrm{DW}(\mathbb{Z}_2)_1$. The phase transition between the $\mathbb{Z}_2^{(1)}$ trivial and the $\mathrm{DW}(\mathbb{Z}_2)_0$ phases is described by the CFT that results from gauging the $\mathbb{Z}_2$ symmetry in the 3d Ising CFT, which we denote as Ising$/\mathbb{Z}_2$. The numerical study of this theory can be found in Ref. [58]. The phase transition between the $\mathbb{Z}_2^{(1)}$ trivial and the $\mathrm{DW}(\mathbb{Z}_2)_1$ phases is described by gauging the $\mathbb{Z}_2$ symmetry in Ising $\boxtimes$ $\mathbb{Z}_2$-SPT, which we denote as (Ising $\boxtimes$ $\mathbb{Z}_2$-SPT)$/\mathbb{Z}_2$. The phase transition between $\mathrm{DW}(\mathbb{Z}_2)_0$ and $\mathrm{DW}(\mathbb{Z}_2)_1$ is expected to be a gauged first order transition.

**$\mathbb{Z}_q$ Phase Transition.** The phase transitions with $\mathbb{Z}_q$ symmetry for $q > 2$ are less explored. Based on the Monte Carlo simulation of the classical $\mathbb{Z}_q$ clock model [59–63], the $\mathbb{Z}_3$ SSB tran-

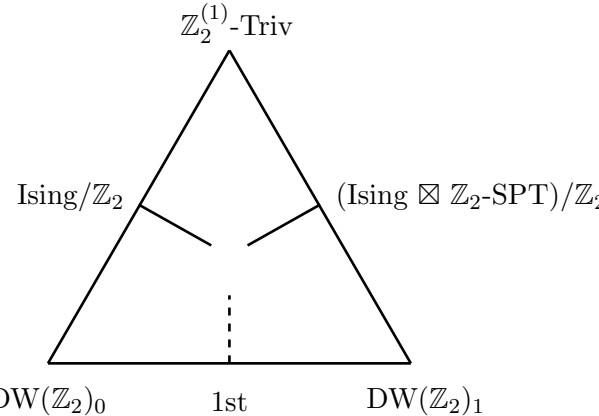

Figure 4: Schematic phase diagram for phases with $\mathbb{Z}_2^{(1)}$ 1-form symmetry. We denote the trivial gapped phase with $\mathbb{Z}_2^{(1)}$ symmetry as $\mathbb{Z}_2^{(1)}$-Triv. $DW(\mathbb{Z}_2)_0$ denotes the DW theory without twist, or equivalently the Toric Code topological order. $DW(\mathbb{Z}_2)_1$ denotes the DW theory with the non-trivail twist, or equivalently the double semion topological order. The transition between the $\mathbb{Z}_2^{(1)}$ trivial and the $DW(\mathbb{Z}_2)_0$ phases is described by the gauged Ising CFT, which is denoted as Ising/$\mathbb{Z}_2$. The transition between the $\mathbb{Z}_2^{(1)}$ trivial and the $DW(\mathbb{Z}_2)_0$ phases is described by the CFT obtained from gauging the $\mathbb{Z}_2$ symmetry in Ising $\boxtimes$ $\mathbb{Z}_2$-SPT, which we denote as (Ising $\boxtimes$ $\mathbb{Z}_2$-SPT)/$\mathbb{Z}_2$. The phase transition between $DW(\mathbb{Z}_2)_0$ and $DW(\mathbb{Z}_2)_1$ is expected to be a gauged first order transition.

sition was found to be described by a first order transition. Regarding the $\mathbb{Z}_4$ SSB transition, while it's possible to fine tune the transition to the Ising$^2$ CFT, a generic $\mathbb{Z}_4$ SSB transition is described by the $O(2)$ CFT. For $q > 4$, the critical behavior is generally described by the $O(2)$ CFT [60, 64–66].

# 3 Module Category Description of Interfaces

In this section, we describe the topological interfaces $\mathcal{I}_{(H,N,\pi,\mathcal{A})}$ between $\mathcal{Z}(2\mathsf{Vec}_G^\varpi)$ and $\mathcal{Z}(2\mathsf{Vec}_{H/N}^\pi)$ in terms of (bi)module 2-categories. This analysis generalizes the previous one in the SymTFT to include the nontrivial twist $\varpi$. The cocycles will be denoted multiplicatively.

## 3.1 Gapped Boundaries from Module 2-Categories

**Module 2-categories over $2\mathsf{Vec}_K^\lambda$.** Let us start with a brief review of module 2-categories over $2\mathsf{Vec}_K^\lambda$, where $K$ is a finite group and $\lambda \in Z^4(K, U(1))$ is a normalized 4-cocycle, i.e., $\lambda(k_1, k_2, k_3, k_4) = 1$ if $k_i = e$ for some $i$.

Due to [67], any right module 2-category over $2\mathsf{Vec}_K^\lambda$ is equivalent to the 2-category $_\mathcal{A}(2\mathsf{Vec}_K^\lambda)$ of left $\mathcal{A}$-modules in $2\mathsf{Vec}_K^\lambda$, where $\mathcal{A}$ is a separable algebra in $2\mathsf{Vec}_K^\lambda$. Objects,

1-morphisms, and 2-morphisms of $_{\mathcal{A}}(2\mathsf{Vec}_K^\lambda)$ are left $\mathcal{A}$-modules, left $\mathcal{A}$-module 1-morphisms, and left $\mathcal{A}$-module 2-morphisms in $2\mathsf{Vec}_K^\lambda$, see [68, Section 2.3] for more detailed definitions. The right module action $\triangleleft\colon {}_{\mathcal{A}}(2\mathsf{Vec}_K^\lambda) \times 2\mathsf{Vec}_K^\lambda \to {}_{\mathcal{A}}(2\mathsf{Vec}_K^\lambda)$ is given by the tensor product of $2\mathsf{Vec}_K^\lambda$.

A separable algebra $\mathcal{A} \in 2\mathsf{Vec}_K^\lambda$ is a $K$-graded $\lambda$-twisted multifusion 1-category [68, 69], i.e., it is a $K$-graded semisimple 1-category $\mathcal{A} = \bigoplus_{k \in K} \mathcal{A}_k$ equipped with a tensor product $\otimes\colon \mathcal{A} \times \mathcal{A} \to \mathcal{A}$ whose associator $\alpha_{a,b,c}\colon (a \otimes b) \otimes c \to a \otimes (b \otimes c)$ satisfies the following commutaive diagram:

$$
\begin{array}{ccccc}
((a \otimes b) \otimes c) \otimes d & \xrightarrow{\alpha_{a\otimes b,c,d}} & (a \otimes b) \otimes (c \otimes d) & \xrightarrow{\alpha_{a,b,c\otimes d}} & a \otimes (b \otimes (c \otimes d)) \\
{\scriptstyle \lambda(k,l,m,n)} \downarrow & & & & \uparrow {\scriptstyle \mathrm{id}_a \otimes \alpha_{b,c,d}} \\
((a \otimes b) \otimes c) \otimes d & \xrightarrow[\alpha_{a,b,c}\otimes\mathrm{id}_d]{} & (a \otimes (b \otimes c)) \otimes d & \xrightarrow[\alpha_{a,b\otimes c,d}]{} & a \otimes ((b \otimes c) \otimes d)
\end{array}
\tag{3.1}
$$

where $a \in \mathcal{A}_k$, $b \in \mathcal{A}_l$, $c \in \mathcal{A}_m$, and $d \in \mathcal{A}_n$. We call (3.1) the twisted pentagon equation. We note that the trivially graded component $\mathcal{A}_e$ is an ordinary multifusion category because $\lambda$ is normalized. In what follows, we suppose that $\mathcal{A}_e$ is fusion, i.e., its unit object is simple.

As an object of $2\mathsf{Vec}_K^\lambda$, $\mathcal{A}$ can be decomposed as

$$
\mathcal{A} = \boxplus_{k \in K} (D_2^k)^{\boxplus \mathrm{rank}(\mathcal{A}_k)},
\tag{3.2}
$$

where $D_2^k$ is a simple object of $2\mathsf{Vec}_K^\lambda$ and $\mathrm{rank}(\mathcal{A}_k)$ is the number of (isomorphism classes of) simple objects of $\mathcal{A}_k$. The multiplication 1- and 2-morphisms are given by the tensor product of $\mathcal{A}$ and the associator of $\mathcal{A}$, respectively.

When $\mathcal{A}$ is faithfully graded by $L \subset K$, the connected components of $_{\mathcal{A}}(2\mathsf{Vec}_K^\lambda)$ are labeled by right $L$-cosets in $K$.[7] The representatives of these connected components can be chosen to be $\{\mathcal{A} \otimes D_2^{k_i} \mid i = 1, 2, \cdots, |L\backslash K|\}$, where $k_i$ is the representative of a right $L$-coset in $K$. For later use, we denote the representative of each connected component by $M_i := \mathcal{A} \otimes D_2^{k_i}$.

**Topological Boundaries of $\mathcal{Z}(2\mathsf{Vec}_K^\lambda)$.** Among all topological boundaries of $\mathcal{Z}(2\mathsf{Vec}_K^\lambda)$, there is a large class of topological boundaries that are labeled by module 2-categories over $2\mathsf{Vec}_K^\lambda$. Physically, these topological boundaries are obtained by stacking (possibly anomalous) non-chiral TFTs on the Dirichlet boundary and gauging a non-anomalous subgroup.[8] Below, we describe such topological boundary conditions.

---

[7]Two objects $X$ and $Y$ are said to be connected if and only if there exists a non-zero 1-morphism between them. The set of connected objects is called a connected component.

[8]We do not even allow stacking chiral topological orders with chiral central charge $c_- \in 8\mathbb{Z}$.

The topological boundary $\mathfrak{B}_\mathcal{A}$ labeled by a module 2-category $_\mathcal{A}(2\mathsf{Vec}_K^\lambda)$ is defined as a topological boundary such that the topological interfaces between $\mathfrak{B}_\mathcal{A}$ and $\mathfrak{B}_{\mathrm{Dir}}$ form $_\mathcal{A}(2\mathsf{Vec}_K^\lambda)$. Specifically, 2d interfaces between $\mathfrak{B}_\mathcal{A}$ and $\mathfrak{B}_{\mathrm{Dir}}$ are labeled by objects of $_\mathcal{A}(2\mathsf{Vec}_K^\lambda)$, 1d interfaces between 2d interfaces are labeled by 1-morphisms of $_\mathcal{A}(2\mathsf{Vec}_K^\lambda)$, and 0d interfaces between 1d interfaces are labeled by 2-morphisms of $_\mathcal{A}(2\mathsf{Vec}_K^\lambda)$. The number of topological interfaces up to condensation is equal to $|L\backslash K| = |K|/|L|$ because the connected components of $_\mathcal{A}(2\mathsf{Vec}_K^\lambda)$ are in one-to-one correspondence with right $L$-cosets in $K$. We note that the boundary $\mathfrak{B}_\mathcal{A}$ is indecomposable because $_\mathcal{A}(2\mathsf{Vec}_K^\lambda)$ is indecomposable as a module 2-category when $\mathcal{A}_e$ is fusion.

The topological lines on the 2d interface $M_i \in {}_\mathcal{A}(2\mathsf{Vec}_K^\lambda)$ between $\mathfrak{B}_\mathcal{A}$ and $\mathfrak{B}_{\mathrm{Dir}}$ form a fusion 1-category

$$\mathrm{Hom}_{\mathcal{A}(2\mathsf{Vec}_K^\lambda)}(M_i, M_i) \cong \mathcal{A}_e^{\mathrm{rev}}. \tag{3.3}$$

Here, $\mathcal{A}^{\mathrm{rev}}$ is the reverse category of $\mathcal{A}$, i.e., the fusion category obtained by reversing all morphisms of $\mathcal{A}$. In particular, the associator of $\mathcal{A}^{\mathrm{rev}}$ is given by $\alpha^{\mathrm{rev}}_{a,b,c} := \alpha^{-1}_{a,b,c}$. We note that $\mathcal{A}^{\mathrm{rev}}$ is a $K$-graded $\lambda^{-1}$-twisted fusion category.[9] Similarly, the topological lines at the intersection of the 2d interfaces $M_i \lhd D_2^k$ and $M_j$ form an $(\mathcal{A}_e^{\mathrm{rev}}, \mathcal{A}_e^{\mathrm{rev}})$-bimodule category

$$\mathrm{Hom}_{\mathcal{A}(2\mathsf{Vec}_K^\lambda)}(M_j, M_i \lhd D_2^k) \cong \mathcal{A}^{\mathrm{rev}}_{k_i k k_j^{-1}}. \tag{3.4}$$

The objects of this category represent non-genuine lines on the interface of $\mathfrak{B}_\mathcal{A}$ and $\mathfrak{B}_{\mathrm{Dir}}$. Drawing the 2+1d boundary only looks as follows:

$$\tag{3.5}$$

The black arrows specify the coorientations of the interface and the surface. The above picture shows the (2+1)d boundary of the (3+1)d TFT. From the 4d perspective, the non-genuine line $a \in \mathcal{A}^{\mathrm{rev}}_{k_i k k_j^{-1}}$ looks like the following:

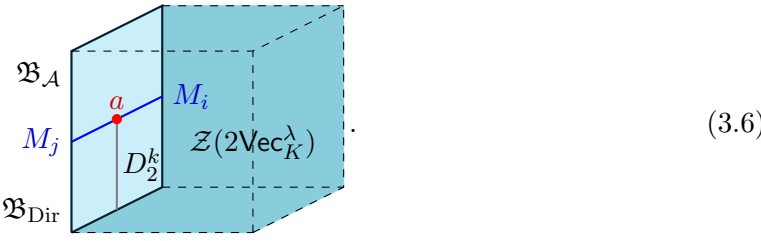

$$\tag{3.6}$$

<hr>

[9]The category $\mathcal{A}_e^{\mathrm{rev}}$ is the trivially graded component of the $K$-graded category $\mathcal{A}^{\mathrm{rev}}$. Equivalently, one can think of $\mathcal{A}_e^{\mathrm{rev}}$ as the reverse category of $\mathcal{A}_e$.

The fusion of these non-genuine lines is described by the tensor product of $\mathcal{A}^{\text{rev}}$.

**Symmetry 2-category on $\mathfrak{B}_\mathcal{A}$.** The topological defects on $\mathfrak{B}_\mathcal{A}$ form the dual 2-category

$$(2\text{Vec}_K^\lambda)^*_{\mathcal{A}(2\text{Vec}_K^\lambda)} := \text{Fun}_{2\text{Vec}_K^\lambda}(_\mathcal{A}(2\text{Vec}_K^\lambda), {}_\mathcal{A}(2\text{Vec}_K^\lambda)), \tag{3.7}$$

which is the 2-category of right $2\text{Vec}_K^\lambda$-module endofunctors of $_\mathcal{A}(2\text{Vec}_K^\lambda)$. The dual 2-category is monoidally equivalent to $_\mathcal{A}(2\text{Vec}_K^\lambda)_\mathcal{A}$, the 2-category of $(\mathcal{A}, \mathcal{A})$-bimodules in $2\text{Vec}_K^\lambda$ [70]. Thus, topological surfaces, topological lines, and topological point defects on $\mathfrak{B}_\mathcal{A}$ are labeled by objects, 1-morphisms, and 2-morphisms of $_\mathcal{A}(2\text{Vec}_K^\lambda)_\mathcal{A}$. In particular, the identity surface on $\mathfrak{B}_\mathcal{A}$ is labeled by the regular bimodule $\mathcal{A}$. We note that the symmetry 2-category $_\mathcal{A}(2\text{Vec}_K^\lambda)_\mathcal{A}$ is fusion because the boundary $\mathfrak{B}_\mathcal{A}$ is indecomposable.

More generally, the symmetry 2-category on a decomposable boundary $\boxplus_i \mathfrak{B}_{\mathcal{A}_i}$ becomes a multifusion 2-category

$$(2\text{Vec}_K^\lambda)^*_{\boxplus_i\,\mathcal{A}_i(2\text{Vec}_K^\lambda)} \cong \boxplus_{i,j}\,{}_{\mathcal{A}_i}(2\text{Vec}_K^\lambda)_{\mathcal{A}_j}. \tag{3.8}$$

The diagonal component $\mathcal{C}_{ii} := {}_{\mathcal{A}_i}(2\text{Vec}_K^\lambda)_{\mathcal{A}_i}$ is the 2-category of topological defects on $\mathfrak{B}_{\mathcal{A}_i}$, while the off-diagonal component $\mathcal{C}_{ij} := {}_{\mathcal{A}_i}(2\text{Vec}_K^\lambda)_{\mathcal{A}_j}$ is the 2-category of topological defects between $\mathfrak{B}_{\mathcal{A}_i}$ and $\mathfrak{B}_{\mathcal{A}_j}$. The monoidal structure on the symmetry 2-category (3.8) is given by the $(\mathcal{C}_{ii}, \mathcal{C}_{jj})$-bimodule structure on $\mathcal{C}_{ij}$.

**Stacking and gauging.** The topological boundary $\mathfrak{B}_\mathcal{A}$ is obtained by stacking a 3d $L$-symmetric TFT $\mathfrak{T}_\mathcal{A}^L$ on $\mathfrak{B}_{\text{Dir}}$ and gauging a non-anomalous diagonal subgroup $L^{\text{diag}} \subset K \times L$. That is, we have

$$\mathfrak{B}_\mathcal{A} = (\mathfrak{B}_{\text{Dir}} \boxtimes \mathfrak{T}_\mathcal{A}^L)/L^{\text{diag}}. \tag{3.9}$$

We note that $\mathfrak{T}_\mathcal{A}^L$ must have an anomaly $\lambda^{-1}|_L$ so that $L^{\text{diag}}$ becomes non-anomalous. In what follows, we define $\mathfrak{T}_\mathcal{A}^L$ and argue that (3.9) holds.

The TFT $\mathfrak{T}_\mathcal{A}^L$ is a 3d non-chiral TFT with an anomalous symmetry $L$. The underlying TFT of $\mathfrak{T}_\mathcal{A}^L$ is described by the Drinfeld center $\mathcal{Z}(\mathcal{A}_e)$ of the trivially graded component $\mathcal{A}_e \subset \mathcal{A}$. The $L$ symmetry of $\mathfrak{T}_\mathcal{A}^L$ is implemented by invertible topological surfaces $\{D_2^l \mid l \in L\}$. These topological surfaces can end on the non-genuine lines on the topological boundaries of $\mathcal{Z}(\mathcal{A}_e)$. On the canonical boundary labeled by the regular $\mathcal{A}_e$-module, the non-genuine lines attached

to $D_2^l$ are labeled by objects of $\mathcal{A}_l^{\mathrm{rev}}$ (again showing the 2+1d boundary):

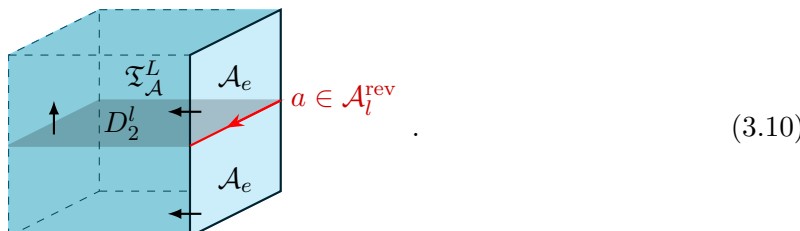

$$(3.10)$$

The fusion of these non-genuine lines is described by the tensor product of $\mathcal{A}^{\mathrm{rev}}$. The $L$ symmetry generated by $\{D_2^l \mid l \in L\}$ has an anomaly $\lambda^{-1}|_L$ beucase of the twisted pentagon equation for $\mathcal{A}^{\mathrm{rev}}$. The 3d TFT $\mathfrak{T}_{\mathcal{A}}^L$ should be completely determined by the category of boundary lines (3.10) via the bulk-boundary correspondence. In particular, the bulk topological lines at the end of $D_2^l$ are given by the boundary non-genuine lines that admit half-braiding with the genuine lines on the boundary. The category of these bulk lines is known as the relative center $\mathcal{Z}_{\mathcal{A}_e}(\mathcal{A})$ [10,71]. When $L$ is non-anomalous, $\mathcal{Z}_{\mathcal{A}_e}(\mathcal{A})$ has the structure of an $L$-crossed non-degenerate braided fusion category, whose $L$-equivariantization is braided equivalent to $\mathcal{Z}(\mathcal{A})$ [71].[10]

Now, let us consider stacking $\mathfrak{T}_{\mathcal{A}}^L$ on $\mathfrak{B}_{\mathrm{Dir}}$ and gauging the diagonal subgroup $L^{\mathrm{diag}}$. We first consider the following configuration:

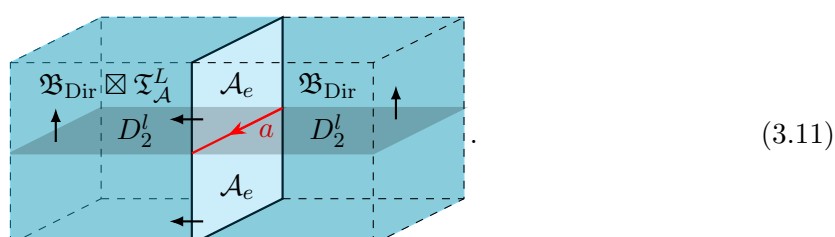

$$(3.11)$$

Here, the interface between $\mathfrak{B}_{\mathrm{Dir}} \boxtimes \mathfrak{T}_{\mathcal{A}}^L$ and $\mathfrak{B}_{\mathrm{Dir}}$ is given by the stacking of the identity surface of $\mathfrak{B}_{\mathrm{Dir}}$ and the canonical boundary of $\mathfrak{T}_{\mathcal{A}}^L$. The topological line at the intersection is a non-genuine line on the canonical boundary of $\mathfrak{T}_{\mathcal{A}}^L$, which is labeled by an object of $\mathcal{A}_l^{\mathrm{rev}}$ due to (3.10). If we gauge the $L^{\mathrm{diag}}$ symmetry on the left side, we obtain an interface between $(\mathfrak{B}_{\mathrm{Dir}} \boxtimes \mathfrak{T}_{\mathcal{A}}^L)/L^{\mathrm{diag}}$ and $\mathfrak{B}_{\mathrm{Dir}}$. We denote this interface by $\mathfrak{I}_0$. Since the $L^{\mathrm{diag}}$ surfaces become invisible after the gauging, the configuration (3.11) gives rise to a non-genuine line on $\mathfrak{I}_0$ attached to $D_2^l$ on $\mathfrak{B}_{\mathrm{Dir}}$. This shows that the surface $D_2^l$ for $l \in L$ can end on $\mathfrak{I}_0$, and the non-genuine lines attached to $D_2^l$ are labeled by objects of $\mathcal{A}_l^{\mathrm{rev}}$.

On the other hand, the surface $D_2^k$ for $k \in K$ cannot end on $\mathfrak{I}_0$ if $k$ is not in $L$. This means that the interface $\mathfrak{I}_0 \lhd D_2^k$, which is the fusion of $\mathfrak{I}_0$ and $D_2^k$, is not connected to $\mathfrak{I}_0$ if $k$ is

---

[10] The lattice models for $\mathfrak{T}_{\mathcal{A}}^L$ are discussed in [72,73].

not in $L$.[11] The interfaces $\mathfrak{I}_0 \lhd D_2^k$ and $\mathfrak{I}_0 \lhd D_2^{k'}$ are connected if and only if $k$ and $k'$ are in the same right $L$-coset in $K$. Therefore, the connected components of the interfaces between $(\mathfrak{B}_{\mathrm{Dir}} \boxtimes \mathfrak{T}_{\mathcal{A}}^L)/L^{\mathrm{diag}}$ and $\mathfrak{B}_{\mathrm{Dir}}$ are labeled by right $L$-cosets in $K$. We choose $\mathfrak{I}_i := \mathfrak{I}_0 \lhd D_2^{k_i}$ as the representative of each connected component, where $k_i$ is the representative of a right $L$-coset. Then, by construction, the topological lines at the junction of $\mathfrak{I}_i$, $\mathfrak{I}_j$, and $D_2^k$ are labeled by objects of $\mathcal{A}_{k_i k k_j^{-1}}^{\mathrm{rev}}$. This agrees with the defining property (3.4) of the topological boundary $\mathfrak{B}_{\mathcal{A}}$. Thus, we conclude that $(\mathfrak{B}_{\mathrm{Dir}} \boxtimes \mathfrak{T}_{\mathcal{A}}^L)/L^{\mathrm{diag}}$ agrees with $\mathfrak{B}_{\mathcal{A}}$, i.e., (3.9) holds.

## 3.2   Genuine and Non-Genuine Lines on the Boundary

In this subsection, we discuss the category of non-genuine lines on $\mathfrak{B}_{\mathcal{A}}$ attached to the bulk surface $\boldsymbol{Q}_2^{[k]} \in \mathcal{Z}(2\mathsf{Vec}_K^\lambda)$.[12] Genuine lines can be regarded as a special case of non-genuine lines where $\boldsymbol{Q}_2^{[k]}$ is trivial. As in section 3.1, we suppose that $\mathcal{A}$ is faithfully graded by $L \subset K$.

The non-genuine lines attached to $\boldsymbol{Q}_2^{[k]}$ are labeled by 1-morphisms between the identity surface on $\mathfrak{B}_{\mathcal{A}}$ and $F(\boldsymbol{Q}_2^{[k]})$, where $F : \mathcal{Z}(2\mathsf{Vec}_K^\lambda) \to {}_{\mathcal{A}}(2\mathsf{Vec}_K^\lambda)_{\mathcal{A}}$ is the bulk-to-boundary functor for $\mathfrak{B}_{\mathcal{A}}$. Namely, the non-genuine lines form a 1-category

$$\mathrm{Hom}_{{}_{\mathcal{A}}(2\mathsf{Vec}_K^\lambda)_{\mathcal{A}}}(F(\boldsymbol{Q}_2^{[k]}), \mathcal{A}). \tag{3.12}$$

The bulk-to-boundary functor $F$ is given by

$$F(\boldsymbol{Q}_2^{[k]}) = \mathrm{Forg}(\boldsymbol{Q}_2^{[k]}) \otimes \mathcal{A} = \bigoplus_{k' \in [k]} n_{k'} D_2^{k'} \otimes \mathcal{A}. \tag{3.13}$$

Here, Forg denotes the forgetful functor from $\mathcal{Z}(2\mathsf{Vec}_K^\lambda)$ to $2\mathsf{Vec}_K^\lambda$, and $n_{k'}$ is some non-negative integer. The right $\mathcal{A}$-module structure on $F(\boldsymbol{Q}_2^{[k]})$ is given by the right multiplication of $\mathcal{A}$, while the left $\mathcal{A}$-module structure is given by using the half-braiding of $\boldsymbol{Q}_2^{[k]}$ as follows:

$$\mathcal{A} \otimes F(\boldsymbol{Q}_2^{[k]}) = \mathcal{A} \otimes \mathrm{Forg}(\boldsymbol{Q}_2^{[k]}) \otimes \mathcal{A} \xrightarrow{\text{half-braiding of } \boldsymbol{Q}_2^{[k]}} \mathrm{Forg}(\boldsymbol{Q}_2^{[k]}) \otimes \mathcal{A} \otimes \mathcal{A}$$
$$\xrightarrow{\text{tensor product of } \mathcal{A}} \mathrm{Forg}(\boldsymbol{Q}_2^{[k]}) \otimes \mathcal{A} = F(\boldsymbol{Q}_2^{[k]}). \tag{3.14}$$

The category (3.12) is non-vanishing if and only if $[k] \cap L$ is not empty. To see this, we consider the following equivalence of semisimple 1-categories:[13]

$$\mathrm{Hom}_{{}_{\mathcal{A}}(2\mathsf{Vec}_K^\lambda)_{\mathcal{A}}}(\mathcal{A} \otimes \mathcal{A}, F(\boldsymbol{Q}_2^{[k]})) \cong \mathrm{Hom}_{2\mathsf{Vec}_K^\lambda}(D_2^e, F(\boldsymbol{Q}_2^{[k]})) \cong \boxplus_{l \in [k] \cap L} \mathsf{Vec}^{\boxplus n_l \mathrm{rank}(\mathcal{A}_{l-1})}. \tag{3.15}$$

---

[11]Two interfaces are said to be connected if and only if there exists a genuine line between them.

[12]The Drinfeld center $\mathcal{Z}(2\mathsf{Vec}_K^\lambda)$ is described in detail in [47].

[13]In general, there is an equivalence of semisimple 1-categories $\mathrm{Hom}_{{}_{\mathcal{A}}(2\mathsf{Vec}_K^\lambda)_{\mathcal{A}}}(\mathcal{A} \otimes X \otimes \mathcal{A}, M) \cong \mathrm{Hom}_{2\mathsf{Vec}_K^\lambda}(X, M)$ for any $X \in 2\mathsf{Vec}_K^\lambda$ and any $M \in {}_{\mathcal{A}}(2\mathsf{Vec}_K^\lambda)_{\mathcal{A}}$ [68].

The second equivalence follows from eqs. (3.2) and (3.13). The above equation implies that $F(\boldsymbol{Q}_2^{[k]})$ contains a simple direct summand connected to $\mathcal{A} \otimes \mathcal{A}$ if and only if $[k] \cap L$ is not empty. Here, we note that $\mathcal{A} \otimes \mathcal{A}$ is a simple object of $_{\mathcal{A}}(2\mathsf{Vec}_K^\lambda)_{\mathcal{A}}$ that is connected to the unit object $\mathcal{A}$.[14] Therefore, (3.15) also implies that $F(\boldsymbol{Q}_2^{[k]})$ contains a simple direct summand connected to $\mathcal{A}$ if and only if $[k] \cap L$ is not empty. Hence, the category (3.12) is non-vanishing if and only if $[k] \cap L$ is not empty.

The above result shows that the bulk surface $\boldsymbol{Q}_2^{[k]}$ can end on the boundary $\mathfrak{B}_{\mathcal{A}}$ if and only if $[k]$ contains an element of $L$. In other words, only the surfaces $\boldsymbol{Q}_2^{[l]}$ for some $l \in L$ can end on $\mathfrak{B}_{\mathcal{A}}$. This is consistent with the physical interpretation (3.9) that $\mathfrak{B}_{\mathcal{A}}$ is obtained by gauging the diagonal $L$ symmetry of $\mathfrak{B}_{\mathrm{Dir}} \boxtimes \mathfrak{T}_{\mathcal{A}}^L$.

**The case of non-anomalous abelian $L$.** Let us describe the category of non-genuine lines in more detail. For simplicity, we suppose that $\lambda$ is trivial, in which case we have $\mathrm{Forg}(\boldsymbol{Q}_2^{[k]}) = \bigoplus_{k' \in [k]} D_2^{k'}$. In addition, we also suppose that the subgroup $L \subset K$ is abelian.

We first notice that the category of non-genuine lines can be decomposed as

$$\mathrm{Hom}_{_{\mathcal{A}}(2\mathsf{Vec}_K)_{\mathcal{A}}}(F(\boldsymbol{Q}_2^{[k]}), \mathcal{A}) \cong \mathrm{Hom}_{_{\mathcal{A}}(2\mathsf{Vec}_L)_{\mathcal{A}}}(\boxplus_{l_i \in [k] \cap L} D_2^{l_i} \otimes \mathcal{A}, \mathcal{A})$$

$$\cong \boxplus_{l_i \in [k] \cap L} \mathrm{Hom}_{_{\mathcal{A}}(2\mathsf{Vec}_L)_{\mathcal{A}}}(D_2^{l_i} \otimes \mathcal{A}, \mathcal{A}). \tag{3.16}$$

The first equivalence follows from the fact that there is no non-zero 1-morphism between $\mathcal{A}$ and $D_2^{k'} \otimes \mathcal{A}$ if $k'$ is not in $L$. The second equivalence follows from the fact that $D_2^l \otimes \mathcal{A}$ for $l \in L$ is by itself an $(\mathcal{A}, \mathcal{A})$-bimodule in $2\mathsf{Vec}_L \subset 2\mathsf{Vec}_K$. If we forget the $L$-grading, $D_2^{l_i} \otimes \mathcal{A}$ is equivalent to $\mathcal{A}$ as an $(\mathcal{A}, \mathcal{A})$-bimodule. Therefore, the direct summand $\mathrm{Hom}_{_{\mathcal{A}}(2\mathsf{Vec}_L)_{\mathcal{A}}}(D_2^{l_i} \otimes \mathcal{A}, \mathcal{A})$ is a subcategory of the category $\mathrm{End}_{\mathsf{Bimod}(\mathcal{A})}(\mathcal{A})$ of $(\mathcal{A}, \mathcal{A})$-bimodule endofunctors of $\mathcal{A}$. More specifically, $\mathrm{Hom}_{_{\mathcal{A}}(2\mathsf{Vec}_L)_{\mathcal{A}}}(D_2^{l_i} \otimes \mathcal{A}, \mathcal{A})$ is the category of $(\mathcal{A}, \mathcal{A})$-bimodule functors that shift the grading by $l_i$. Recalling that $\mathrm{End}_{\mathsf{Bimod}(\mathcal{A})}(\mathcal{A})$ is equivalent to the Drinfeld center of $\mathcal{A}$ [74], we find

$$\mathrm{Hom}_{_{\mathcal{A}}(2\mathsf{Vec}_L)_{\mathcal{A}}}(D_2^{l_i} \otimes \mathcal{A}, \mathcal{A}) \subset \mathrm{Hom}_{\mathsf{Bimod}(\mathcal{A})}(\mathcal{A}) \cong \mathcal{Z}(\mathcal{A}). \tag{3.17}$$

Under the equivalence $\mathrm{Hom}_{\mathsf{Bimod}(\mathcal{A})}(\mathcal{A}) \cong \mathcal{Z}(\mathcal{A})$, objects of $\mathrm{Hom}_{_{\mathcal{A}}(2\mathsf{Vec}_L)_{\mathcal{A}}}(D_2^{l_i} \otimes \mathcal{A}, \mathcal{A})$ correspond to objects of $\mathcal{Z}(\mathcal{A})$ whose underlying objects in $\mathcal{A}$ are graded by $l_i$. Thus, we have an equivalence

$$\mathrm{Hom}_{_{\mathcal{A}}(2\mathsf{Vec}_L)_{\mathcal{A}}}(D_2^{l_i} \otimes \mathcal{A}, \mathcal{A}) \cong \mathcal{Z}(\mathcal{A})_{l_i}, \tag{3.18}$$

---

[14]It is shown in [68] that for an algebra object of the form $\mathcal{A} = \mathsf{Vec}_H^\nu \in 2\mathsf{Vec}_K^\lambda$, the object $\mathcal{A} \otimes \mathcal{A}$ is simple in $_{\mathcal{A}}(2\mathsf{Vec}_K^\lambda)_{\mathcal{A}}$. The same proof applies to more general algebra object $\mathcal{A}$ as long as $\mathcal{A}_e$ is fusion as we assume in the main text.

where $\mathcal{Z}(\mathcal{A})_{l_i}$ denotes the full subcategory of $\mathcal{Z}(\mathcal{A})$ consisting of objects whose underlying objects in $\mathcal{A}$ have the grading $l_i$.[15] Plugging the above equation into (3.16) leads us to

$$\text{Hom}_{\mathcal{A}(2\text{Vec}_K^\lambda)_{\mathcal{A}}}(F(\boldsymbol{Q}_2^{[k]}), \mathcal{A}) \cong \bigboxplus_{l_i \in [k] \cap L} \mathcal{Z}(\mathcal{A})_{l_i}. \tag{3.19}$$

This shows that the non-genuine lines attached to $\boldsymbol{Q}_2^{[k]}$ are labeled by objects of $\mathcal{Z}(\mathcal{A})_{l_i}$ for $l_i \in [k] \cap L$.

## 3.3 Non-Genuine Point Defects on the Boundary

In this subsection, we consider the non-genuine point-like defects on $\mathfrak{B}_{\mathcal{A}}$ attached to the bulk genuine line $\boldsymbol{Q}_1^R$, where $R \in \text{Rep}(K)$ is a representation of $K$. In particular, we show that $\boldsymbol{Q}_1^R$ can end on $\mathfrak{B}_{\mathcal{A}}$ if and only if $\kappa(R) \in \text{Rep}(L)$ contains the trivial representation. Here, $\kappa : \text{Rep}(K) \to \text{Rep}(L)$ is the forgetful functor, that is, $\kappa(R)$ for $R \in \text{Rep}(K)$ is $R$ viewed as a representation of $L$.

We first recall the definition of $\boldsymbol{Q}_1^R \in \text{End}_{\mathcal{Z}(2\text{Vec}_K^\lambda)}(\boldsymbol{Q}_2^{[e]})$ [47]. The underlying 1-morphism of $\boldsymbol{Q}_1^R$ is given by

$$\text{Forg}(\boldsymbol{Q}_1^R) = \text{id}_{D_2^e}^{\oplus \dim R} \in \text{End}_{2\text{Vec}_K^\lambda}(D_2^e). \tag{3.20}$$

The half-braiding of $\boldsymbol{Q}_1^R$ and $D_2^k \in 2\text{Vec}_K^\lambda$ is given by

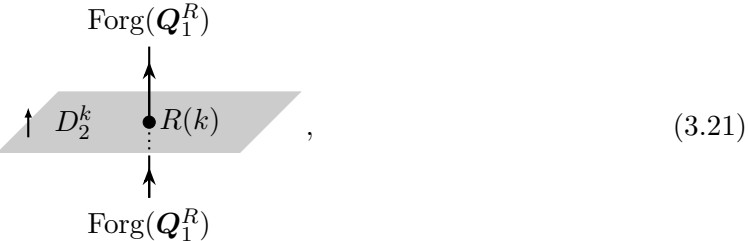

$$\tag{3.21}$$

where $R(k) \in \text{End}_{2\text{Vec}_K^\lambda}(\text{id}_{D_2^k} \otimes \text{Forg}(\boldsymbol{Q}_1^R), \text{Forg}(\boldsymbol{Q}_1^R) \otimes \text{id}_{D_2^k}) \cong \text{End}(\mathbb{C}^{\dim R})$ is the representation matrix of $R$.

If we push $\boldsymbol{Q}_1^R$ onto the boudnary $\mathfrak{B}_{\mathcal{A}}$, it becomes a boundary line $F(\boldsymbol{Q}_1^R)$, where $F$ is the bulk-to-boundary functor:

$$F(\boldsymbol{Q}_1^R) = \text{Forg}(\boldsymbol{Q}_1^R) \otimes \text{id}_{\mathcal{A}} \in \text{End}_{\mathcal{A}(2\text{Vec}_K^\lambda)_{\mathcal{A}}}(\text{Forg}(\boldsymbol{Q}_2^{[e]}) \otimes \mathcal{A}). \tag{3.22}$$

The line $F(\boldsymbol{Q}_1^R)$ is an $(\mathcal{A}, \mathcal{A})$-bimodule 1-endomorphism of $\text{Forg}(\boldsymbol{Q}_2^{[e]}) \otimes \mathcal{A} \cong \mathcal{A}$. The right $\mathcal{A}$-module 1-morphism structure on $F(\boldsymbol{Q}_1^R)$ is trivial, while the left $\mathcal{A}$-module 1-morphism

---

[15]When $\mathcal{A}$ is graded by an abelian group $L$, the Drinfeld center $\mathcal{Z}(\mathcal{A})$ is also graded by $L$.

structure is given by using the half-braiding (3.22) as follows:

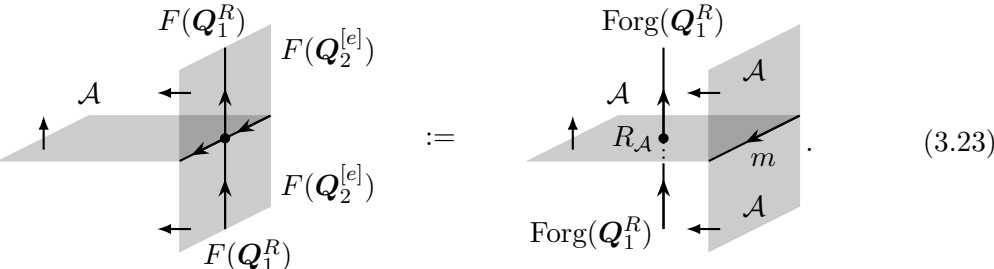

$$(3.23)$$

Here, $m$ is the tensor product of $\mathcal{A}$, and $R_{\mathcal{A}}$ is the half-braiding of $\boldsymbol{Q}_1^R$ and $\mathcal{A}$, which is defined component-wise as follows:

$$(3.24)$$

Now, we consider the point-like defects between $F(\boldsymbol{Q}_1^R)$ and $F(\boldsymbol{Q}_1^{R'})$ on $\mathfrak{B}_{\mathcal{A}}$. These point-like defects form a vector space

$$\mathrm{Hom}_{\mathcal{A}(2\mathsf{Vec}_K^\lambda)_{\mathcal{A}}}(F(\boldsymbol{Q}_1^R), F(\boldsymbol{Q}_1^{R'})). \tag{3.25}$$

Any element of this vector space is of the form

$$f \otimes \mathrm{id}_{\mathrm{id}_{\mathcal{A}}} : \mathrm{Forg}(\boldsymbol{Q}_1^R) \otimes \mathrm{id}_{\mathcal{A}} \to \mathrm{Forg}(\boldsymbol{Q}_1^{R'}) \otimes \mathrm{id}_{\mathcal{A}}, \tag{3.26}$$

where $f$ is a 2-morphism from $\mathrm{Forg}(\boldsymbol{Q}_1^R)$ to $\mathrm{Forg}(\boldsymbol{Q}_1^{R'})$ in $2\mathsf{Vec}_K^\lambda$. The 2-morphism $f$ must satisfy

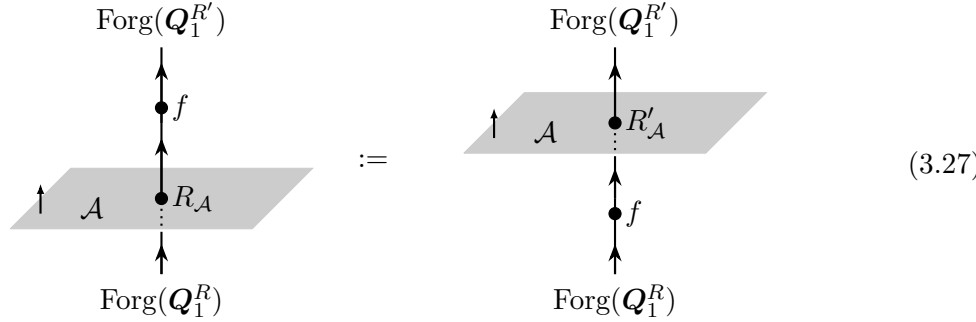

$$(3.27)$$

so that $f \otimes \mathrm{id}_{\mathrm{id}_{\mathcal{A}}}$ is an $(\mathcal{A}, \mathcal{A})$-bimodule 2-morphism. Since the half-braiding $R_{\mathcal{A}}$ is given by (3.24), the above equation reduces to

$$f \circ R(l) = R'(l) \circ f, \qquad \forall l \in L. \tag{3.28}$$

Namely, $f$ is an intertwiner between $R$ and $R'$ viewed as representations of $L$. Therefore, we find

$$\operatorname{Hom}_{\mathcal{A}(2\mathsf{Vec}_K^\lambda)_{\mathcal{A}}}(F(\boldsymbol{Q}_1^R), F(\boldsymbol{Q}_1^{R'})) \cong \operatorname{Hom}_{\mathsf{Rep}(L)}(\kappa(R), \kappa(R')). \tag{3.29}$$

In particular, when $R' = \operatorname{triv}^K$ is the trivial representation of $K$, we have

$$\operatorname{Hom}_{\mathcal{A}(2\mathsf{Vec}_K^\lambda)_{\mathcal{A}}}(F(\boldsymbol{Q}_1^R), \operatorname{id}_{\mathcal{A}}) \cong \operatorname{Hom}_{\mathsf{Rep}(L)}(\kappa(R), \operatorname{triv}^L). \tag{3.30}$$

This implies that $\boldsymbol{Q}_1^R$ can end on the boundary $\mathfrak{B}_{\mathcal{A}}$ if and only if $\kappa(R)$ contains the trivial representation of $L$. This result is consistent with the physical interpretation of $\mathfrak{B}_{\mathcal{A}}$ in terms of stacking and gauging.

## 3.4 Interfaces

Let us describe the interface $\mathcal{I}_{(H,N,\pi,\mathcal{A})}$ between $\mathcal{Z}(2\mathsf{Vec}_G^\varpi)$ and $\mathcal{Z}(2\mathsf{Vec}_{H/N}^\pi)$ in terms of (bi)module 2-categories. We first consider the topological boundary $\mathcal{I}_{(H,N,\pi,\mathcal{A})}^{\mathrm{fold}}$ obtained by folding the bulk TFT. By construction, we have

$$\mathcal{I}_{(H,N,\pi,\mathcal{A})}^{\mathrm{fold}} = (\mathfrak{B}_{\mathrm{Dir}} \boxtimes \mathfrak{T}_{\mathcal{A}}^{H^{\mathrm{diag}}})/(H^{\mathrm{diag}})^{\mathrm{diag}}. \tag{3.31}$$

The notations used in the above equation are as follows.

- $\mathfrak{B}_{\mathrm{Dir}}$ is the Dirichlet boundary of the folded TFT

$$\overline{\mathcal{Z}(2\mathsf{Vec}_G^\varpi)} \boxtimes \mathcal{Z}(2\mathsf{Vec}_{H/N}^\pi) \cong \mathcal{Z}(2\mathsf{Vec}_{G \times H/N}^{\overline{\varpi}\pi}), \tag{3.32}$$

  where $\overline{\varpi}\pi$ is defined by

$$\overline{\varpi}\pi((g_1, x_1), (g_2, x_2), (g_3, x_3), (g_4, x_4)) := \varpi(g_1, g_2, g_3, g_4)^{-1} \pi(x_1, x_2, x_3, x_4) \tag{3.33}$$

  for all $g_1, g_2, g_3, g_4 \in G$ and $x_1, x_2, x_3, x_4 \in H/N$.

- $H^{\mathrm{diag}} = \{(h, p(h)) \mid h \in H\}$ is a subgroup of $G \times H/N$, where $p : H \to H/N$ is the canonical projection.

- $\mathcal{A}$ is a $G \times H/N$-graded $\overline{\varpi}\pi$-twisted fusion category, which is faithfully graded by $H^{\mathrm{diag}}$.

- $\mathfrak{T}_{\mathcal{A}}^{H^{\mathrm{diag}}}$ is an anomalous $H^{\mathrm{diag}}$-symmetric TFT with an anomaly $(\overline{\varpi}\pi)|_{H^{\mathrm{diag}}}^{-1}$. We note that $\overline{\varpi}\pi$ restricted to $H^{\mathrm{diag}}$ can be written as

$$\overline{\varpi}\pi((h_1, p(h_1)), (h_2, p(h_2)), (h_3, p(h_3)), (h_4, p(h_4))) = (\varpi^{-1} p^* \pi)(h_1, h_2, h_3, h_4), \tag{3.34}$$

  which means that $\mathfrak{T}_{\mathcal{A}}^{H^{\mathrm{diag}}}$ can be regarded as an $H$-symmetric TFT with anomaly $\varpi p^* \pi^{-1}$.

- $(H^{\mathrm{diag}})^{\mathrm{diag}}$ is the diagonal $H^{\mathrm{diag}}$ subgroup of $(G \times H/N)|_{\mathfrak{B}_{\mathrm{Dir}}} \times H^{\mathrm{diag}}|_{\mathfrak{T}_{\mathcal{A}}^{H^{\mathrm{diag}}}}$.

Due to the results in section 3.1, the boundary $\mathcal{I}^{\mathrm{fold}}_{(H,N,\pi,\mathcal{A})}$ defined by (3.31) is labeled by a $(2\mathsf{Vec}^{\overline{\varpi}\pi}_{G \times H/N})$-module 2-category

$$\mathcal{A}(2\mathsf{Vec}^{\overline{\varpi}\pi}_{G \times H/N}). \tag{3.35}$$

By unfolding the TFT $\mathcal{Z}(2\mathsf{Vec}^{\overline{\varpi}\pi}_{G \times H/N})$, we obtain the corresponding interface $\mathcal{I}_{(H,N,\pi,\mathcal{A})}$ between $\mathcal{Z}(2\mathsf{Vec}^{\varpi}_G)$ and $\mathcal{Z}(2\mathsf{Vec}^{\pi}_{H/N})$. From the unfolded picture, $\mathcal{I}_{(H,N,\pi,\mathcal{A})}$ is labeled by a $(2\mathsf{Vec}^{\varpi}_G, 2\mathsf{Vec}^{\pi}_{H/N})$-bimodule 2-category $\mathcal{A}(2\mathsf{Vec}^{\overline{\varpi}\pi}_{G \times H/N})$. Here, the left $2\mathsf{Vec}^{\varpi}_G$-action $\rhd: 2\mathsf{Vec}^{\varpi}_G \times \mathcal{A}(2\mathsf{Vec}^{\overline{\varpi}\pi}_{G \times H/N}) \to \mathcal{A}(2\mathsf{Vec}^{\overline{\varpi}\pi}_{G \times H/N})$ is given by

$$D_2^g \rhd M := M \lhd D_2^{g^{-1}}, \qquad \forall M \in \mathcal{A}(2\mathsf{Vec}^{\overline{\varpi}\pi}_{G \times H/N}), \tag{3.36}$$

where $\lhd$ on the right-hand side is the right action of $2\mathsf{Vec}^{\overline{\varpi}}_G$ on $\mathcal{A}(2\mathsf{Vec}^{\overline{\varpi}\pi}_{G \times H/N})$. On the other hand, the right $2\mathsf{Vec}^{\pi}_{H/N}$-action on $\mathcal{A}(2\mathsf{Vec}^{\overline{\varpi}\pi}_{G \times H/N})$ is simply given by restricting the right $2\mathsf{Vec}^{\overline{\varpi}\pi}_{G \times H/N}$-action to $2\mathsf{Vec}^{\pi}_{H/N}$.

From section 3.2, the bulk surface $\boldsymbol{Q}_2^{[(g,x)]}$ of the folded TFT $\mathcal{Z}(2\mathsf{Vec}^{\overline{\varpi}\pi}_{G \times H/N})$ can end on $\mathcal{I}_{(H,N,\pi,\mathcal{A})}$ if and only if $\boldsymbol{Q}_2^{[(g,x)]} = \boldsymbol{Q}_2^{[(h,p(h))]}$ for some $h \in H$. After the unfolding procedure, a non-genuine line attached to $\boldsymbol{Q}_2^{[(h,p(h))]}$ becomes an intersection of a bulk surface and $\mathcal{I}_{(H,N,\pi,\mathcal{A})}$. The surface on the left side of the interface is $\boldsymbol{Q}_2^{[h]}$, while that on the right side is $\boldsymbol{Q}_2^{[p(h)]}$:

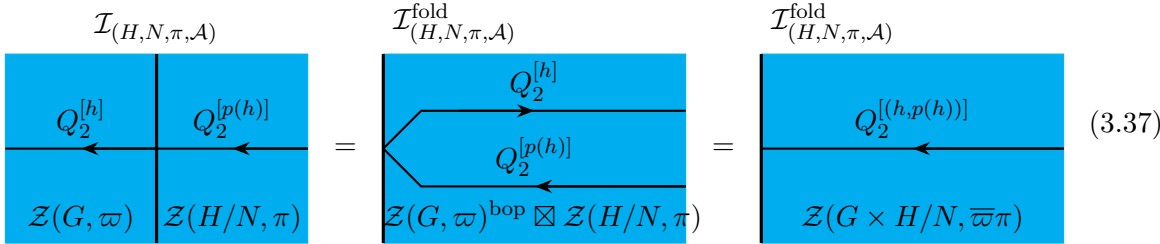

$$\tag{3.37}$$

Here, $\mathcal{Z}(G,\varpi)$ is the abbreviation of $\mathcal{Z}(2\mathsf{Vec}^{\varpi}_G)$ and $\mathcal{Z}(G,\varpi)^{\mathrm{bop}}$ is $\mathcal{Z}(G,\varpi)$ with the opposite braiding. The second equality of (3.37) follows from the braided equivalence

$$\mathcal{Z}(2\mathsf{Vec}^{\varpi}_G)^{\mathrm{bop}} \xrightarrow{\cong} \mathcal{Z}(2\mathsf{Vec}^{\overline{\varpi}}_G) : \boldsymbol{Q}_2^{[h]} \mapsto \overline{\boldsymbol{Q}_2^{[h]}} \tag{3.38}$$

where $\overline{\boldsymbol{Q}_2^{[h]}}$ is the dual of $\boldsymbol{Q}_2^{[h]}$. Equation (3.37) implies that $\boldsymbol{Q}_2^{[n]}$ for $n \in N$ can end on $\mathcal{I}_{(H,N,\pi,\mathcal{A})}$ from the left, while no surfaces can end from the right.

Furthermore, due to section 3.3, the bulk lines $\boldsymbol{Q}_1^{R_G \otimes R_{H/N}}$ of the folded TFT $\mathcal{Z}(2\mathsf{Vec}^{\overline{\varpi}\pi}_{G \times H/N})$ can end on $\mathcal{I}^{\mathrm{fold}}_{(H,N,\pi,\mathcal{A})}$ if and only if $\kappa(R_G \otimes R_{H/N}) \in \mathsf{Rep}(H^{\mathrm{diag}})$ contains the trivial representation. The orthogonality relation of characters implies that $\kappa(R_G \otimes R_{H/N})$ contains the trivial representation of $H^{\mathrm{diag}}$ if and only if the following condition holds:

$$\sum_{h \in H} \mathrm{tr}[R_G(h)]\,\mathrm{tr}[R_{H/N}(p(h))] \neq 0. \tag{3.39}$$

When $R_{H/N}$ is trivial, the above condition reduces to

$$\sum_{h \in H} \mathrm{tr}[R_G(h)] \neq 0. \tag{3.40}$$

This condition is satisfied if and only if $\kappa(R_G) \in \mathsf{Rep}(H)$ contains the trivial representation of $H$. On the other hand, when $R_G$ is trivial, the above condition reduces to

$$\sum_{h \in H} \mathrm{tr}[R_{H/N}(p(h))] = |N| \sum_{x \in H/N} \mathrm{tr}[R_{H/N}(x)] \neq 0. \tag{3.41}$$

This condition is satisfied if and only if $R_{H/N}$ is trivial. The above result shows that after the unfolding, $\boldsymbol{Q}_1^{R_G}$ can end on $\mathcal{I}_{(H,N,\pi,\mathcal{A})}$ from the left if and only if $\kappa(R_G) \in \mathsf{Rep}(H)$ contains the trivial representation, while no lines can end from the right. We emphasize that neither surfaces nor lines can end on $\mathcal{I}_{(H,N,\pi,\mathcal{A})}$ from the right.[16]

## 3.5 Club Quiches

Now, we perform the club quiche construction using the non-minimal interface described in the previous subsection. The club quiche (shown on the left hand side below) can be collapsed to give rise to a BC for $\mathcal{Z}(2\mathsf{Vec}_{H/N}^{\pi})$:

$$\tag{3.42}$$

Here, the boundary on the left-hand side is labeled by a right $2\mathsf{Vec}_G^{\varpi}$-module 2-category $\mathcal{M}$. Collapsing the region occupied by $\mathcal{Z}(2\mathsf{Vec}_G^{\varpi})$ gives us a topological boundary of $\mathcal{Z}(2\mathsf{Vec}_{H/N}^{\pi})$. This boundary is labeled by a right $2\mathsf{Vec}_{H/N}^{\pi}$-module 2-category $\mathcal{M} \boxtimes_{2\mathsf{Vec}_G^{\varpi}} \mathcal{A}(2\mathsf{Vec}_{G \times H/N}^{\overline{\varpi}\pi})$.

For concreteness, we consider the case where the original boundary of $\mathcal{Z}(2\mathsf{Vec}_G^{\varpi})$ is the Dirichlet boundary. The corresponding module 2-category $\mathcal{M}$ is the regular module

$$\mathcal{M} = 2\mathsf{Vec}_G^{\varpi}. \tag{3.43}$$

In this case, the boundary obtained by the club quiche construction (3.42) is labeled by a right $2\mathsf{Vec}_{H/N}^{\pi}$-module 2-category

$$2\mathsf{Vec}_G^{\varpi} \boxtimes_{2\mathsf{Vec}_G^{\varpi}} \mathcal{A}(2\mathsf{Vec}_{G \times H/N}^{\overline{\varpi}\pi}) \cong \mathcal{A}(2\mathsf{Vec}_{G \times H/N}^{\overline{\varpi}\pi}). \tag{3.44}$$

---

[16]More precisely, only the condensation defects can end from the right.

In general, this 2-category is decomposable as a module 2-category over $2\mathsf{Vec}_{H/N}^{\pi}$.

Let us decompose $_{\mathcal{A}}(2\mathsf{Vec}_{G\times H/N}^{\overline{\overline{\omega}}\pi})$ into the direct sum of indecomposable $2\mathsf{Vec}_{H/N}^{\pi}$-module 2-categories. To this end, we first notice that $_{\mathcal{A}}(2\mathsf{Vec}_{G\times H/N}^{\overline{\overline{\omega}}\pi})$ can be decomposed as a semisimple 2-category as follows:

$$_{\mathcal{A}}(2\mathsf{Vec}_{G\times H/N}^{\overline{\overline{\omega}}\pi}) = \boxplus_{w\in H\backslash G}\ \boxplus_{x\in H/N}\ _{\mathcal{A}}(2\mathsf{Vec}_{G\times H/N}^{\overline{\overline{\omega}}\pi})_{(w,x)}. \tag{3.45}$$

Here, $_{\mathcal{A}}(2\mathsf{Vec}_{G\times H/N}^{\overline{\overline{\omega}}\pi})_{(w,x)}$ is a connected sub-2-category that contains $\mathcal{A}\otimes D_2^{(g,x)}$ for $g\in w$ and $x\in H/N$. The right $2\mathsf{Vec}_{H/N}^{\pi}$-action on $\mathcal{A}\otimes D_2^{(g,x)}\in{}_{\mathcal{A}}(2\mathsf{Vec}_{G\times H/N}^{\overline{\overline{\omega}}\pi})$ is given by

$$(\mathcal{A}\otimes D_2^{(g,x)})\lhd D_2^{x'} = \mathcal{A}\otimes D_2^{(g,xx')}, \qquad \forall x'\in H/N. \tag{3.46}$$

In particular, the action of $2\mathsf{Vec}_{H/N}^{\pi}$ does not change $w\in H\backslash G$, while it acts transitively on $x\in H/N$. This implies that $_{\mathcal{A}}(2\mathsf{Vec}_{G\times H/N}^{\overline{\overline{\omega}}\pi})_{(w,x)}$ and $_{\mathcal{A}}(2\mathsf{Vec}_{G\times H/N}^{\overline{\overline{\omega}}\pi})_{(w',x')}$ are sub-2-categories of the same indecomposable $2\mathsf{Vec}_{H/N}^{\pi}$-module 2-category if and only if $w=w'$. Therefore, we obtain the following decomposition as a right $2\mathsf{Vec}_{H/N}^{\pi}$-module 2-category

$$_{\mathcal{A}}(2\mathsf{Vec}_{G\times H/N}^{\overline{\overline{\omega}}\pi}) = \boxplus_{w\in H\backslash G}\ _{\mathcal{A}}(2\mathsf{Vec}_{G\times H/N}^{\overline{\overline{\omega}}\pi})_w, \tag{3.47}$$

where we defined

$$_{\mathcal{A}}(2\mathsf{Vec}_{G\times H/N}^{\overline{\overline{\omega}}\pi})_w := \boxplus_{x\in H/N}\ _{\mathcal{A}}(2\mathsf{Vec}_{G\times H/N}^{\overline{\overline{\omega}}\pi})_{(w,x)}. \tag{3.48}$$

Since $_{\mathcal{A}}(2\mathsf{Vec}_{G\times H/N}^{\overline{\overline{\omega}}\pi})_w$ is an indecomposable module 2-category over $2\mathsf{Vec}_{H/N}^{\pi}$, there exists an $H/N$-graded $\pi$-twisted fusion category $\tilde{\mathcal{A}}$ such that

$$_{\mathcal{A}}(2\mathsf{Vec}_{G\times H/N}^{\overline{\overline{\omega}}\pi})_w \cong {}_{\tilde{\mathcal{A}}}(2\mathsf{Vec}_{H/N}^{\pi}) \tag{3.49}$$

as a $2\mathsf{Vec}_{H/N}^{\pi}$-module 2-category. We note that $\tilde{\mathcal{A}}$ is faithfully graded by the trivial group $\{e\}$ because $_{\mathcal{A}}(2\mathsf{Vec}_{G\times H/N}^{\overline{\overline{\omega}}\pi})_w$ has as many connected components as $|H/N|$.[17] Therefore, we have $\tilde{\mathcal{A}}=\tilde{\mathcal{A}}_e$. To find $\tilde{\mathcal{A}}$, we consider the endomorphism 1-category of a simple object

$$\mathrm{End}_{_{\mathcal{A}}(2\mathsf{Vec}_{G\times H/N}^{\overline{\overline{\omega}}\pi})_w}(\mathcal{A}\otimes D_2^{(g,x)}) \cong \mathcal{A}_e^{\mathrm{rev}}, \qquad \mathrm{End}_{_{\tilde{\mathcal{A}}}(2\mathsf{Vec}_{H/N}^{\pi})}(\tilde{\mathcal{A}}\otimes D_2^x) \cong \tilde{\mathcal{A}}_e^{\mathrm{rev}}. \tag{3.50}$$

The above equation implies that $\tilde{\mathcal{A}}_e$ must be Morita equivalent to $\mathcal{A}_e$. In particular, we can choose $\tilde{\mathcal{A}}_e=\mathcal{A}_e$.[18] Thus, (3.47) leads to the following decomposition of a $2\mathsf{Vec}_{H/N}^{\pi}$-module 2-category:

$$_{\mathcal{A}}(2\mathsf{Vec}_{G\times H/N}^{\overline{\overline{\omega}}\pi}) \cong {}_{\mathcal{A}_e}(2\mathsf{Vec}_{H/N}^{\pi})^{\boxplus|H\backslash G|}. \tag{3.51}$$

---

[17]In general, the number of connected components of $_{\mathcal{A}}(2\mathsf{Vec}_K^{\lambda})$ is $|K|/|L|$ if $\mathcal{A}$ is faithfully graded by $L\subset K$.

[18]The choice of $\tilde{\mathcal{A}}$ is not unique because $_{\tilde{\mathcal{A}}}(2\mathsf{Vec}_{H/N}^{\pi})$ depends only on the Morita equivalence class of $\tilde{\mathcal{A}}$.

We note that $_{\mathcal{A}_e}(2\mathsf{Vec}^\pi_{H/N})$ is equivalent to $2\mathsf{Vec}^\pi_{H/N} \boxtimes \mathsf{LMod}(\mathcal{A}_e)$ as a $2\mathsf{Vec}^\pi_{H/N}$-module 2-category. Here, $2\mathsf{Vec}^\pi_{H/N}$ does not act on $\mathsf{LMod}(\mathcal{A}_e)$, which is the 2-category of left $\mathcal{A}_e$-module categories. Thus, the above equation can also be written as

$$_{\mathcal{A}}(2\mathsf{Vec}^{\overline{\varpi}\pi}_{G\times H/N}) \cong (2\mathsf{Vec}^\pi_{H/N} \boxtimes \mathsf{LMod}(\mathcal{A}_e))^{\boxplus|H\backslash G|}. \tag{3.52}$$

For later use, we denote the $i$th component of the right-hand side as $(2\mathsf{Vec}^\pi_{H/N} \boxtimes \mathsf{LMod}(\mathcal{A}_e))_i$. The equivalence (3.52) maps objects as

$$\mathcal{A} \otimes D_2^{(hg_i,p(h)x)} \mapsto (D_2^x \boxtimes \mathcal{A}_{(h,p(h))^{-1}})_i, \tag{3.53}$$

where $g_i$ is the representative of a right $H$-coset in $G$.

Physically, the topological boundary labeled by $2\mathsf{Vec}^\pi_{H/N} \boxtimes \mathsf{LMod}(\mathcal{A}_e)$ is the stacking of the Dirichelt boundary of $\mathcal{Z}(2\mathsf{Vec}^\pi_{H/N})$ and the 3d TFT $\mathcal{Z}(\mathcal{A}_e)$, cf. (3.9). Therefore, (3.52) implies

$$\mathfrak{B}^{G,\varpi}_{\mathrm{Dir}} \mathcal{I}_{(H,N,\pi,\mathcal{A})} = (\mathfrak{B}^{H/N,\pi}_{\mathrm{Dir}} \boxtimes \mathcal{Z}(\mathcal{A}_e))^{\boxplus|H\backslash G|}, \tag{3.54}$$

where $\mathfrak{B}^{G,\varpi}_{\mathrm{Dir}}$ and $\mathfrak{B}^{H/N,\pi}_{\mathrm{Dir}}$ denote the Dirichlet boundaries of $\mathcal{Z}(2\mathsf{Vec}^\varpi_G)$ and $\mathcal{Z}(2\mathsf{Vec}^\pi_{H/N})$.

**Implementation of the symmetry.** Let us consider the symmetry on the boundary obtained by the club quiche construction. We will mainly focus on the case of the Dirichlet boundary, and briefly comment on the case of more general boundaries.

As we discussed above, if we start from the Dirichlet boundary of $\mathcal{Z}(2\mathsf{Vec}^\varpi_G)$, the club quiche construction gives us the direct sum (3.54) of the Dirichlet boundaries of $\mathcal{Z}(2\mathsf{Vec}^\pi_{H/N})$, stacked with the 3d TFT $\mathcal{Z}(\mathcal{A}_e)$. The symmetry on this boundary is described by the multifusion 2-category

$$\begin{aligned}
(2\mathsf{Vec}^\pi_{H/N})^*_{\mathcal{A}(2\mathsf{Vec}^{\overline{\varpi}\pi}_{G\times H/N})} &= \mathsf{Fun}_{2\mathsf{Vec}^\pi_{H/N}}(_{\mathcal{A}}(2\mathsf{Vec}^{\overline{\varpi}\pi}_{G\times H/N}), _{\mathcal{A}}(2\mathsf{Vec}^{\overline{\varpi}\pi}_{G\times H/N})) \\
&\cong \mathrm{Mat}_{|H\backslash G|}(2\mathsf{Vec}^\pi_{H/N} \boxtimes \Sigma\mathcal{Z}(\mathcal{A}_e)).
\end{aligned} \tag{3.55}$$

Here, $\Sigma\mathcal{Z}(\mathcal{A}_e)$ denotes the condensation completion of $\mathcal{Z}(\mathcal{A}_e)$ [75, 76], which is monoidally equivalent to $\mathsf{Mod}(\mathcal{Z}(\mathcal{A}_e))$ [75], the 2-category of $(\mathcal{Z}(\mathcal{A}_e), \mathcal{Z}(\mathcal{A}_e))$-bimodule categories. On the other hand, by construction, the boundary also has the symmetry described by $2\mathsf{Vec}^\varpi_G$, which is implemented by a monoidal 2-functor

$$\Phi : 2\mathsf{Vec}^\varpi_G \to \mathrm{Mat}_{|H\backslash G|}(2\mathsf{Vec}^\pi_{H/N} \boxtimes \Sigma\mathcal{Z}(\mathcal{A}_e)). \tag{3.56}$$

In what follows, we determine this monoidal 2-functor at the level of objects.

Formally, the 2-functor $\Phi$ is given by

$$\Phi(D_2^g) = \Psi(- \lhd D_2^{g^{-1}}), \tag{3.57}$$

where $- \lhd D_2^{g^{-1}}$ is a $2\mathsf{Vec}_{H/N}^\pi$-module 2-functor from $_\mathcal{A}(2\mathsf{Vec}_{G\times H/N}^{\overline{\omega}\pi})$ to itself, and $\Psi$ denotes the monoidal equivalence in (3.55). The equivalence $\Psi$ is given by the composition of the following equivalences:

$$\mathsf{Fun}_{2\mathsf{Vec}_{H/N}^\pi}(_\mathcal{A}(2\mathsf{Vec}_{G\times H/N}^{\overline{\omega}\pi}), {}_\mathcal{A}(2\mathsf{Vec}_{G\times H/N}^{\overline{\omega}\pi}))$$

$$\overset{\psi_1}{\cong} \mathsf{Fun}_{2\mathsf{Vec}_{H/N}^\pi}\left(\bigboxplus_i(2\mathsf{Vec}_{H/N}^\pi \boxtimes \mathsf{LMod}(\mathcal{A}_e))_i, \ \bigboxplus_j(2\mathsf{Vec}_{H/N}^\pi \boxtimes \mathsf{LMod}(\mathcal{A}_e))_j\right)$$

$$\overset{\psi_2}{\cong} \mathrm{Mat}_{|H\backslash G|}\left(\mathsf{Fun}_{2\mathsf{Vec}_{H/N}^\pi}(2\mathsf{Vec}_{H/N}^\pi \boxtimes \mathsf{LMod}(\mathcal{A}_e), \ 2\mathsf{Vec}_{H/N}^\pi \boxtimes \mathsf{LMod}(\mathcal{A}_e))\right)$$

$$\overset{\psi_3}{\cong} \mathrm{Mat}_{|H\backslash G|}(2\mathsf{Vec}_{H/N}^\pi \boxtimes \mathsf{Bimod}(\mathcal{A}_e)). \tag{3.58}$$

The first equivalence $\psi_1$ follows from (3.52), the second equivalence $\psi_2$ follows in an obvious way, and the last equivalence $\psi_3$ follows from

$$2\mathsf{Vec}_{H/N}^\pi \boxtimes \mathsf{Bimod}(\mathcal{A}_e) \overset{\cong}{\to} \mathsf{Fun}_{2\mathsf{Vec}_{H/N}^\pi}(2\mathsf{Vec}_{H/N}^\pi \boxtimes \mathsf{LMod}(\mathcal{A}_e), \ 2\mathsf{Vec}_{H/N}^\pi \boxtimes \mathsf{LMod}(\mathcal{A}_e))$$

$$X \boxtimes M \quad \mapsto \quad (- \otimes X) \boxtimes (- \boxtimes_{\mathcal{A}_e} M). \tag{3.59}$$

The fusion 2-category $\mathsf{Bimod}(\mathcal{A}_e)$ is monoidally equivalent to $\Sigma\mathcal{Z}(\mathcal{A}_e)$ [77,78].

Let us now compute $\Psi(- \lhd D_2^{g^{-1}})$ by using (3.58). First, we recall that the functor $- \lhd D_2^{g^{-1}}$ maps objects as

$$\mathcal{A} \otimes D_2^{(hg_i,p(h)x)} \overset{-\lhd D_2^{g^{-1}}}{\longmapsto} \mathcal{A} \otimes D_2^{(hg_ig^{-1},p(h)x)} = \mathcal{A} \otimes D_2^{(hh_{ij}^{-1}g_j, \ p(h)x)}. \tag{3.60}$$

Here, we used the decomposition $g = g_j^{-1}h_{ij}g_i$, where $g_i$ and $g_j$ are the representatives of right $H$-cosets in $G$ and $h_{ij} \in H$. We note that the decomposition of $g$ is unique for any $g_i$. The above equation combined with (3.53) implies that the functor $\psi_1(- \lhd D_2^{g^{-1}})$ maps objects as

$$(D_2^x \boxtimes \mathcal{A}_{(h,p(h))^{-1}})_i \overset{\psi_1(-\lhd D_2^{g^{-1}})}{\longmapsto} (D_2^{p(h_{ij})x} \boxtimes \mathcal{A}_{(h_{ij},p(h_{ij}))(h,p(h))^{-1}})_j, \tag{3.61}$$

Furthermore, the equivalence $\psi_2$ maps this functor to a matrix whose components take values in the functor category $\mathsf{Fun}_{2\mathsf{Vec}_{H/N}^\pi}(2\mathsf{Vec}_{H/N}^\pi \boxtimes \mathsf{LMod}(\mathcal{A}_e), 2\mathsf{Vec}_{H/N}^\pi \boxtimes \mathsf{LMod}(\mathcal{A}_e))$. The $(i,j)$-component of the matrix $\psi_2(\psi_1(- \lhd D_2^{g^{-1}}))$ is given by

$$D_2^x \boxtimes \mathcal{A}_{(h,p(h))^{-1}} \overset{\psi_2(\psi_1(-\lhd D_2^{g^{-1}}))_{ij}}{\longmapsto} D_2^{p(h_{ij})x} \boxtimes \mathcal{A}_{(h_{ij},p(h_{ij}))(h,p(h))^{-1}}. \tag{3.62}$$

We note that each column of this matrix has only one non-zero entry because $(h_{ij}, g_j)$ is uniquely determined by $(g, g_i)$. Finally, the equivalence $\psi_3$ maps the above matrix to another matrix whose components take values in $2\mathsf{Vec}_{H/N}^\pi \boxtimes \mathsf{Bimod}(\mathcal{A}_e)$. Due to (3.59), the $(i,j)$-component of the new matrix is given by

$$\psi_3(\psi_2(\psi_1(- \lhd D_2^{g^{-1}})))_{ij} = D_2^{p(h_{ij})} \boxtimes \mathcal{A}_{(h_{ij},p(h_{ij}))}. \tag{3.63}$$

Via the monoidal equivalence $\mathsf{Bimod}(\mathcal{A}_e) \cong \Sigma\mathcal{Z}(\mathcal{A}_e)$ [77,78], the $(\mathcal{A}_e, \mathcal{A}_e)$-bimodule $\mathcal{A}_{(h_{ij}, p(h_{ij}))}$ corresponds to the invertible object $D_2^{h_{ij}} \in \Sigma\mathcal{Z}(\mathcal{A}_e)$, which implements the (possibly anomalous) $H$ symmetry of the stacked TFT $\mathcal{Z}(\mathcal{A}_e)$. Thus, we find

$$\Phi(D_2^g)_{ij} = D_2^{p(h_{ij})} \boxtimes D_2^{h_{ij}}. \tag{3.64}$$

Before concluding this section, we mention that the boundary conditions other than the Dirichlet boundary can be obtained by gauging a separable algebra $\mathcal{B} \in 2\mathsf{Vec}_G^{\varpi}$ on the Dirichlet boundary of $\mathcal{Z}(2\mathsf{Vec}_G^{\varpi})$. Equivalently, these boundaries can be obtained by gauging the separable algebra $\Phi(\mathcal{B}) \in 2\mathsf{Vec}_{H/N}^\pi \boxtimes \Sigma\mathcal{Z}(\mathcal{A}_e)$ after squashing the region occupied by $\mathcal{Z}(2\mathsf{Vec}_G^{\varpi})$ in the club quiche. Such boundaries will be studied later sections of this paper for various concrete examples.

# 4 Gapless Phases from $\mathbb{Z}_4$ DW Theory

To illustrate the general framework developed so far, consider $G = \mathbb{Z}_4$ which is a simple, yet still non-trivial example. We will determine the gapless interfaces, both minimal and non-minimal, from the DW for $\mathbb{Z}_4$. A summary of the transitions from minimal interfaces is shown in table 4. For the reader interested in the general abelian groups, see appendix B

The subgroups are $H = 1, \mathbb{Z}_2, \mathbb{Z}_4$ and we furthermore have

$$H^3(\mathbb{Z}_n, U(1)) = \mathbb{Z}_n\,, \quad H^4(\mathbb{Z}_n, U(1)) = 0\,, \tag{4.1}$$

so the reduced topological orders are all without twist, i.e. $\pi = 0$. The gapped phases have $H = N$ and are given as follows, using the notation that $\mathrm{Neu}(A)$ corresponds to gauging $A$ on the Dirichlet BC:

$$
\begin{aligned}
H = N = 1 : & \quad \mathfrak{B}_{\mathrm{Dir}} \\
H = N = \mathbb{Z}_2,\ \omega \in \mathbb{Z}_2 : & \quad \mathfrak{B}_{\mathrm{Neu}(\mathbb{Z}_2, \omega)} \\
H = N = \mathbb{Z}_4,\ \omega \in \mathbb{Z}_4 : & \quad \mathfrak{B}_{\mathrm{Neu}(\mathbb{Z}_4, \omega)}\,.
\end{aligned} \tag{4.2}
$$

The interfaces to non-trivial TOs are as follows:

$$
\begin{array}{llll}
H = \mathbb{Z}_2\,, & N = 1\,, & \omega \in \mathbb{Z}_2 : & \mathcal{Z}(2\mathsf{Vec}_{\mathbb{Z}_2}) \\
H = \mathbb{Z}_4\,, & N = \mathbb{Z}_2, & \omega \in \mathbb{Z}_4 : & \mathcal{Z}(2\mathsf{Vec}_{\mathbb{Z}_2}) \\
H = \mathbb{Z}_4\,, & N = 1, & \omega \in \mathbb{Z}_4 : & \mathcal{Z}(2\mathsf{Vec}_{\mathbb{Z}_4})\,.
\end{array} \tag{4.3}
$$

The various choices are distinguished in terms of the cocyle $\omega$ that governs the associators of lines on the interface. The case of $H = G$ and $N = 1$ yields the trivial interface, which is not useful for the considerations of KT transformations.

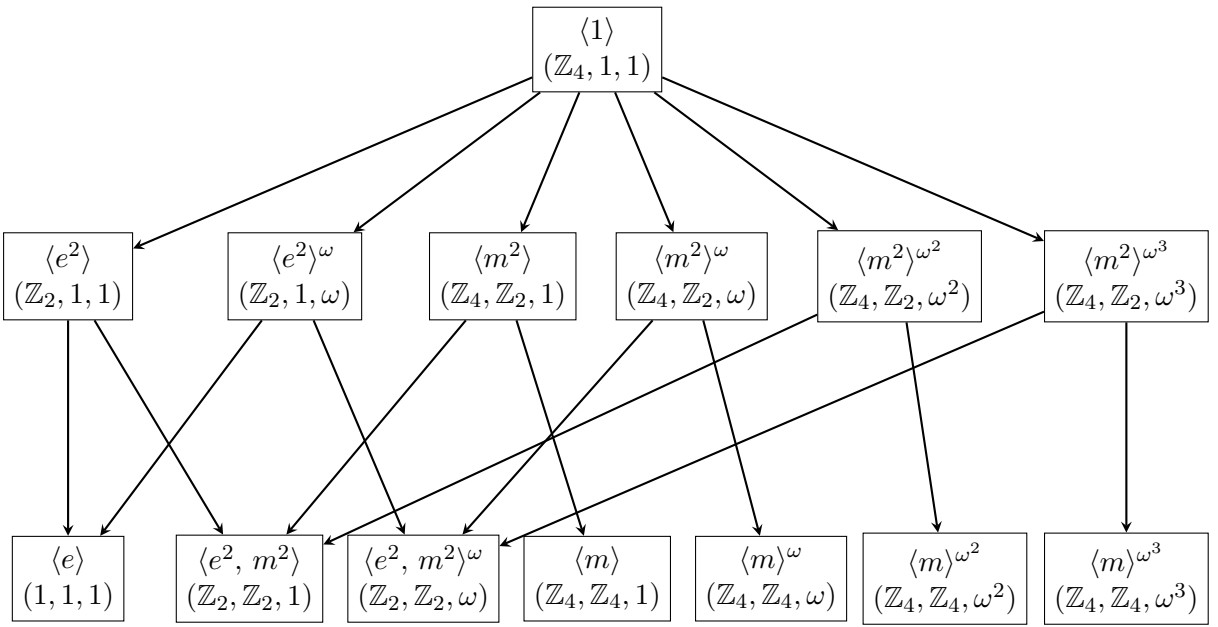

Figure 5: Hasse diagram for condensable algebras in $\mathcal{Z}(2\mathsf{Vec}_{\mathbb{Z}_4})$. Each box contains a condensable algebra, for which we provide generating objects and the group-theoretical data. We list $(H, N, \omega)$, and for the maximal algebras (bottom row) that correspond to gapped BCs, $H = N$. There can be a possible $N$-SPT: $\omega$ is the generator of $H^3(\mathbb{Z}_4, U(1)) = \mathbb{Z}_4$, whose restriction to $\mathbb{Z}_2$ is the generator of $H^3(\mathbb{Z}_2, U(1)) = \mathbb{Z}_2$.

These minimal condensable algebras/interfaces can be put into a partial order, that is depicted by a Hasse diagram as shown in figure 5. We will denote the simple objects in $\mathcal{Z}(2\mathsf{Vec}_{\mathbb{Z}_4})$ as

$$e^p \equiv \boldsymbol{Q}_1^{e^p}, \qquad\qquad m^p \equiv \boldsymbol{Q}_2^{m^p}, \qquad\qquad p \in \{0, 1, 2, 3\}. \qquad (4.4)$$

The condensable algebras are shown in figure 5, where arrows denote inclusion. The maximal algebras determined gapped boundary conditions of the SymTFT and were discussed in [1].

## 4.1 Club Sandwiches

The minimal condensable algebras in $\mathcal{Z}(2\mathsf{Vec}_{\mathbb{Z}_4})$, shown in the middle layer in figure 5, all have reduced TO $\mathcal{Z}(2\mathsf{Vec}_{\mathbb{Z}_2})$. Let us label the group elements of $\mathbb{Z}_4 \times \mathbb{Z}_2$ as $m^p M^q$, where $p \in \{0, 1, 2, 3\}$ and $q \in \{0, 1\}$.

- For the algebra $\langle e^2 \rangle$, for which $H = \mathbb{Z}_2$ and $N = 1$, the group $H^{\text{diag}}$, defined in equation (2.24), is

$$H^{\text{diag}} = \{1, m^2 M\}, \qquad (4.5)$$

therefore the generalized charges of the SymTFT transform as follows when passing through the condensation interface:[19]

$$
\begin{array}{c}
\mathcal{I}_{(\mathbb{Z}_2,1,\omega)=\langle e^2\rangle^\omega} \\
\boxed{\begin{array}{ccc}
\boldsymbol{Q}_1^e & \mathcal{E}_0^{eE} & \boldsymbol{Q}_1^E \\
\boldsymbol{Q}_1^{e^2} & \mathcal{E}_0^{e^2} & \\
\boldsymbol{Q}_2^{m^2} & \mathcal{E}_1^{m^2 M} & \boldsymbol{Q}_2^M
\end{array}} \\
\mathcal{Z}(2\mathsf{Vec}_{\mathbb{Z}_4}) \qquad \mathcal{Z}(2\mathsf{Vec}_{\mathbb{Z}_2})
\end{array}
\tag{4.6}
$$

- For the algebra $\langle m^2\rangle$, for which $H = \mathbb{Z}_4$ and $N = \mathbb{Z}_2$, equation (2.24), gives

$$
H^{\mathrm{diag}} = \{1, m^2, mM, m^3 M\},
\tag{4.7}
$$

which translates to the following transformation of generalized charges when passing through the condensation interface:

$$
\begin{array}{c}
\mathcal{I}_{(\mathbb{Z}_4,\mathbb{Z}_2,\omega)=\langle m^2\rangle^\omega} \\
\boxed{\begin{array}{ccc}
\boldsymbol{Q}_1^{e^2} & \mathcal{E}_0^{e^2 E} & \boldsymbol{Q}_1^E \\
\boldsymbol{Q}_2^{m^2} & \mathcal{E}_1^{m^2} & \\
\boldsymbol{Q}_2^m & \mathcal{E}_1^{mM} & \boldsymbol{Q}_2^M
\end{array}} \\
\mathcal{Z}(2\mathsf{Vec}_{\mathbb{Z}_4}) \qquad \mathcal{Z}(2\mathsf{Vec}_{\mathbb{Z}_2})
\end{array}
\tag{4.8}
$$

- Finally, when condensing the algebras with non-trivial SPT phases, the maps of generalized charges are the same as the previous cases, but now the lines $\boldsymbol{Q}_1^E$ in the reduced TO have non-trivial associator given by the SPT 3-cocycle, which we indicated as $\omega$.

We have summarized our findings in the next few subsections in table 4.

## 4.2  $\mathbb{Z}_4^{(0)}$-Symmetric Phase Transitions

We now provide some examples of $\mathbb{Z}_4$ symmetric phase transitions. We choose the symmetry boundary $\mathfrak{B}_{\mathrm{sym}} = \mathfrak{B}_{\mathrm{Dir}}$, i.e. the group-symmetry $\mathbb{Z}_4$, which mainly serves the purpose to

---

[19]In the projected figures, we indicate bulk lines by dashed lines and bulk surfaces by solid lines.

| $\mathcal{S}$ | $(H, N, \omega)$ | Reduced TO: $\mathcal{Z}(\mathcal{S}_0)$ | $\mathcal{S}$-symmetric CFT $\mathfrak{T}_C^{\mathcal{S}}$ |
|---|---|---|---|
| $\mathbb{Z}_4^{(0)}$ | $(\mathbb{Z}_2, 1, \omega)$ | $\mathcal{Z}(2\mathsf{Vec}_{\mathbb{Z}_2})$ | $\mathbb{Z}_4$ $\quad$ $\mathbb{Z}_2 \curvearrowright \mathfrak{T}_{\mathbb{Z}_2} \boxplus \mathfrak{T}_{\mathbb{Z}_2} \curvearrowleft \mathbb{Z}_2$ $\quad$ $\mathbb{Z}_4$ |
| $\mathbb{Z}_4^{(0)}$ | $(\mathbb{Z}_4, \mathbb{Z}_2, \omega)$ | $\mathcal{Z}(2\mathsf{Vec}_{\mathbb{Z}_2})$ | $\mathfrak{T}_{\mathbb{Z}_2} \circlearrowleft \mathbb{Z}_2 \longleftarrow \mathbb{Z}_4.$ |
| $\mathbb{Z}_4^{(1)}$ | $(\mathbb{Z}_2, 1, \omega)$ | $\mathcal{Z}(2\mathsf{Vec}_{\mathbb{Z}_2})$ | $\frac{\mathfrak{T}_{\mathbb{Z}_2}}{(\mathbb{Z}_2,\omega)} \circlearrowleft \mathbb{Z}_2^{(1)} \longleftarrow \mathbb{Z}_4^{(1)}$ |
| $\mathbb{Z}_4^{(1)}$ | $(\mathbb{Z}_4, \mathbb{Z}_2, \omega)$ | $\mathcal{Z}(2\mathsf{Vec}_{\mathbb{Z}_2})$ | $\frac{\mathfrak{T}_{\mathbb{Z}_2} \boxtimes \mathrm{DW}(\mathbb{Z}_2)_{\omega|_N}}{(\mathbb{Z}_2,\omega)} \circlearrowleft \mathbb{Z}_2^{(1)} \longleftarrow \mathbb{Z}_4^{(1)}$ |
| $\left(\mathbb{Z}_2^{(0)} \times \mathbb{Z}_2^{(1)}\right)^{\beta}$ | $(\mathbb{Z}_2, 1, \omega)$ | $\mathcal{Z}(2\mathsf{Vec}_{\mathbb{Z}_2})$ | $\mathbb{Z}_2^{(1)} \circlearrowright \quad \frac{\mathfrak{T}_{\mathbb{Z}_2}}{\mathbb{Z}_2} \boxplus \frac{\mathfrak{T}_{\mathbb{Z}_2}}{\mathbb{Z}_2} \quad \circlearrowleft \mathbb{Z}_2^{(1)}$ $\quad$ $\mathbb{Z}_2$ |
| $\left(\mathbb{Z}_2^{(0)} \times \mathbb{Z}_2^{(1)}\right)^{\beta}$ | $(\mathbb{Z}_2, 1, \omega)$ | $\mathcal{Z}(2\mathsf{Vec}_{\mathbb{Z}_2})$ | $\mathfrak{T}_{\mathbb{Z}_2} \boxtimes \mathrm{DW}(\mathbb{Z}_2)_{\omega|_N} \circlearrowleft \mathbb{Z}_2$ |

Table 4: Summary table of (minimal) phase transitions from $\mathbb{Z}_4$ DW theory: The symmetries are listed under $\mathcal{S}$ and correspond to the minimal gapped BCs. The data labeling the algebra is $(H, N, \omega)$, and the reduced TO is characterized as the center of $\mathbb{Z}_2$, though the symmetry itself on the boundary may be more complicated (but Morita equivalent to $\mathbb{Z}_2$). $\mathfrak{T}_{\mathbb{Z}_2}$ is a $\mathbb{Z}_2$ symmetric theory and in order to construct KT transformations for gapless theories is chosen to be the Ising CFT.

illustrate the general framework in this very well-known situation. The club quiche is:

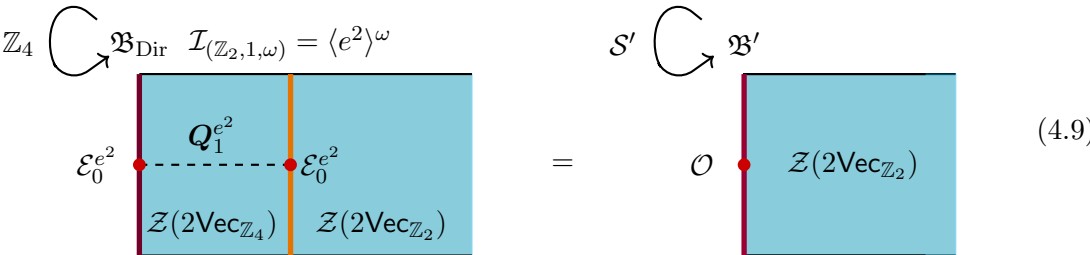

$$\text{(4.9)}$$

All the $e$ lines can end on $\mathfrak{B}_{\text{Dir}}$. The $m^p$ surfaces becomes $D_2^{m^p}$ surfaces on $\mathfrak{B}_{\text{Dir}}$, which generate the $\mathbb{Z}_4^{(0)}$. Non-trival $\omega$ changes the associator of the $\mathbb{Z}_4^{(0)}$ symmetry. Now we consider some possible choices of minimal interfaces and discuss some example of the phase transitions by applying the KT transformation.

- $\langle e^2 \rangle$ **Condensed Interface:** We compactify the left interval as (4.9), which results in a non-trivial local operator. For the reduced TO, this is a reducible BC

$$\mathfrak{B}' = (\mathfrak{B}_e)_0 \oplus (\mathfrak{B}_e)_1, \tag{4.10}$$

  where $\mathfrak{B}_e$ is the $e$-condensed (Dirichlet) boundary condition of the $\mathbb{Z}_2$ SymTFT.

  The KT transformation takes a theory with $\mathbb{Z}_2$ symmetry and output a theory with $\mathbb{Z}_4$ symmetry. We first discuss the KT transformation of the gapped phases with trivial $\omega$. The KT transformation of the $\mathbb{Z}_2$ SSB phase is the $\mathbb{Z}_4$ SSB phase. The KT transformation of the $\mathbb{Z}_2$ trivial is the $\mathbb{Z}_2$ SSB phase, where the $\mathbb{Z}_4$ symmetry is partially broken down to $\mathbb{Z}_2$. Finally, the KT transformation of the SPT phases becomes the stacking of the $\mathbb{Z}_2$ SSB and the $\mathbb{Z}_2$ SPT phase, which we denote as $\mathbb{Z}_2$-SSB $\boxtimes$ $\mathbb{Z}_2$-SPT.

  Now we choose $\mathfrak{B}_{\text{phys}}$ to be a $\mathbb{Z}_2$ symmetric theory $\mathfrak{T}_{\mathbb{Z}_2}$. The KT transformed phase transition is given by

$$\mathbb{Z}_2 \circlearrowright \underset{\mathbb{Z}_4}{\overset{\mathbb{Z}_4}{\mathfrak{T}_{\mathbb{Z}_2} \boxplus \mathfrak{T}_{\mathbb{Z}_2}}} \circlearrowleft \mathbb{Z}_2 \tag{4.11}$$

  The $\mathbb{Z}_4$ symmetry acts by exchanging the two vacua. If $\mathfrak{T}_{\mathbb{Z}_2} = $ Ising, this theory describes the phase transition between the $\mathbb{Z}_4$ SSB and the $\mathbb{Z}_2$ SSB phases. If $\mathfrak{T}_{\mathbb{Z}_2} = $ Ising$\boxtimes\mathbb{Z}_2$-SPT, this theory describes the phase transition between the $\mathbb{Z}_4$ SSB and the $\mathbb{Z}_2$-SSB $\boxtimes$ $\mathbb{Z}_2$-SPT phases.

- $\langle m^2 \rangle$ **Condensed Interface:** In this case the club quiche results in an irreducible BC $\mathfrak{B}_e$ for the reduced TO, as there are no local operators (the $\boldsymbol{Q}_1^e$ lines cannot end on the interface). For trivial $\omega$, the KT transformation of the $\mathbb{Z}_2$ SSB phase is the $\mathbb{Z}_2$ SSB phase, where the $\mathbb{Z}_4$ symmetry is partially broken down to $\mathbb{Z}_2$. The KT transformation of the $\mathbb{Z}_2$ trivial is the $\mathbb{Z}_4$ trivial phase. Finally, the KT transformation of the SPT phases becomes the $\mathbb{Z}_4$ SPT state with $\omega^2$, which we denote as $\mathbb{Z}_4$-SPT$_{\omega^2}$.

For $\mathfrak{B}_{\mathrm{phys}} = \mathfrak{T}_{\mathbb{Z}_2}$, the KT transformed phase transition is given by

$$\mathfrak{T}_{\mathbb{Z}_2} \; \overset{\curvearrowleft}{\bigcirc} \; \mathbb{Z}_2 \longleftarrow \mathbb{Z}_4. \tag{4.12}$$

If $\mathfrak{T}_{\mathbb{Z}_2} = $ Ising, this theory describes the phase transition between the $\mathbb{Z}_2$ SSB and the $\mathbb{Z}_4$ trivial phases. If $\mathfrak{T}_{\mathbb{Z}_2} = $ Ising $\boxtimes \mathbb{Z}_2$-SPT, this theory describes the transition between the $\mathbb{Z}_2$ SSB and the $\mathbb{Z}_4$-SPT$_{\omega^2}$ phases.

## 4.3 $\mathbb{Z}_4^{(1)}$-Symmetric Phase Transitions

We can change the symmetry boundary so that the symmetry is $\mathbb{Z}_4^{(1)}$:

$$\mathfrak{B}_{\mathrm{sym}} = \mathfrak{B}_{\mathrm{Neu}}: \qquad \mathcal{S} = \mathbb{Z}_4^{(1)}. \tag{4.13}$$

Here, all $\boldsymbol{Q}_2^{m^i}$ can end on the symmetry boundary, whereas the $\boldsymbol{Q}_1^{e^p}$ become the 1-form symmetry generators $D_1^{e^p}$.

$\langle e^2 \rangle$ **Condensed Interface.** In this case none of the surfaces that can end on the symmetry boundary have endpoints/lines on the interface.

To apply the KT transformation, we compactify the interval with $\mathcal{Z}(2\mathsf{Vec}_{\mathbb{Z}_4})$. The $Q_2^M$ surface in $\mathcal{Z}(2\mathsf{Vec}_{\mathbb{Z}_2})$ can be connected to $Q_2^{m^2}$ in $\mathcal{Z}(2\mathsf{Vec}_{\mathbb{Z}_4})$ across the interface, and can thus end on $\mathfrak{B}_{\mathrm{sym}}$. The $Q_1^E$ line in $\mathcal{Z}(2\mathsf{Vec}_{\mathbb{Z}_2})$ becomes $Q_1^e$ line in $\mathcal{Z}(2\mathsf{Vec}_{\mathbb{Z}_4})$, which generate the 1-form symmetry on $\mathfrak{B}_{\mathrm{sym}}$. However, the $D_1^{e^2}$ line that generate the $\mathbb{Z}_2^{(1)} \subset \mathbb{Z}_4^{(1)}$ 1-form symmetry is isomorphic to a trivial line since it can end on a local operator in the IR. Therefore, we obtain the $\mathfrak{B}_{m,\omega}$ boundary for the $\mathcal{Z}(2\mathsf{Vec}_{\mathbb{Z}_2})$ reduced TO after compactifying the $\mathcal{Z}(2\mathsf{Vec}_{\mathbb{Z}_4})$ interval.

The KT transformation of the $\mathbb{Z}_2^{(1)}$ trivial phase is the $\mathbb{Z}_4^{(1)}$ trivial phase. The KT transformation of the DW$(\mathbb{Z}_2)_0$ phase is a $\mathbb{Z}_2^{(1)}$ SSB phase, which is equivalent to the Toric Code topological order. There is an unbroken $\mathbb{Z}_2^{(1)}$ symmetry that comes from compactfying the $e^2$ lines and acts trivially on the anyons. We simply denote this gapped phase as DW$(\mathbb{Z}_2)_0$. The KT transformation of the DW$(\mathbb{Z}_2)_1$ phase is another $\mathbb{Z}_2^{(1)}$ SSB phase, which is equivalent

to the double semion topological order. There is also an unbroken $\mathbb{Z}_2^{(1)}$ symmetry that acts trivially in the IR. We denote this gapped phase as $\mathrm{DW}(\mathbb{Z}_2)_1$.

Now we input $\mathfrak{B}_{\mathrm{phys}}$ to be the $\mathfrak{T}_{\mathbb{Z}_2}/\mathbb{Z}_2$, the KT transformed theory is given by

$$\frac{\mathfrak{T}_{\mathbb{Z}_2}}{(\mathbb{Z}_2,\omega)} \circlearrowleft \mathbb{Z}_2^{(1)} \longleftarrow \mathbb{Z}_4^{(1)} \tag{4.14}$$

For $\omega = 0$ (denoting $H^3(\mathbb{Z}_2, U(1))$ additively) and $\mathfrak{T}_{\mathbb{Z}_2} = \mathrm{Ising}$, this theory describes the transition between the $\mathbb{Z}_4^{(1)}$ trivial and $\mathrm{DW}(\mathbb{Z}_2)_0$ phases. If $\omega = 0$ and $\mathfrak{T}_{\mathbb{Z}_2} = \mathrm{Ising} \boxtimes \mathbb{Z}_2\text{-SPT}$, this describes the transition between the $\mathbb{Z}_4^{(1)}$ trivial and $\mathrm{DW}(\mathbb{Z}_2)_1$ phases.

$\langle m^2 \rangle$ **Condensed Interface.** The club quiche for this case is interesting in the case of the $m^2$ interface:

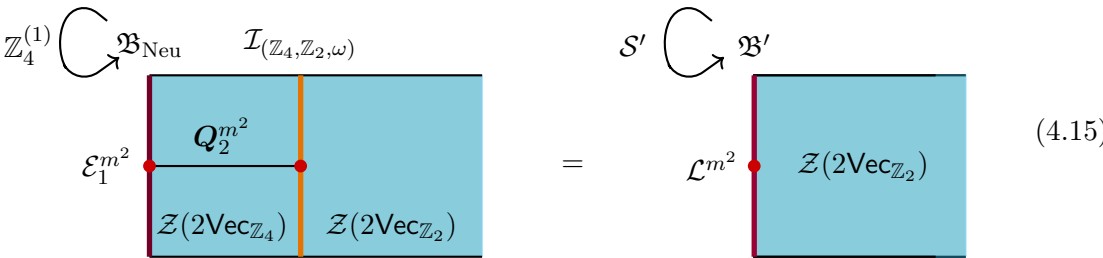

$$\tag{4.15}$$

Here $\mathcal{L}^{m^2}$ is a non-trivial line. We first discuss the boundary of $\mathcal{Z}(2\mathsf{Vec}_{\mathbb{Z}_2})$ obtained by compactifying the interval occupied by $\mathcal{Z}(2\mathsf{Vec}_{\mathbb{Z}_4})$. The $Q_2^M$ surface in $\mathcal{Z}(2\mathsf{Vec}_{\mathbb{Z}_2})$ can be connected to $Q_2^m$ or $Q_2^{m^3}$ in $\mathcal{Z}(2\mathsf{Vec}_{\mathbb{Z}_4})$ across the interface, and can thus end on $\mathfrak{B}_{\mathrm{sym}}$. The $Q_1^E$ line in $\mathcal{Z}(2\mathsf{Vec}_{\mathbb{Z}_2})$ becomes $Q_1^{e^2}$ line in $\mathcal{Z}(2\mathsf{Vec}_{\mathbb{Z}_4})$, and generates the $\mathbb{Z}_2^{(1)} \subset \mathbb{Z}_4^{(1)}$ symmetry on $\mathfrak{B}_{\mathrm{sym}}$. Additionally, the $Q_2^{m^2}$ surface can end on both the symmetry boundary and the interface, and it becomes a line $\mathcal{L}^{m^2}$ with non-trivial associators coming from $\omega$. The $\mathcal{L}^{m^2}$ line is charged under the $\mathbb{Z}_4^{(1)}$ symmetry on $\mathfrak{B}_{\mathrm{sym}}$ generated by $D_1^e$ line. In summary, the boundary of $\mathcal{Z}(2\mathsf{Vec}_{\mathbb{Z}_2})$ is given by

$$\mathfrak{B}' = \frac{\mathfrak{B}_e \boxtimes \mathrm{DW}(\mathbb{Z}_2)_{\omega|_N}}{(\mathbb{Z}_2, \omega)}, \tag{4.16}$$

where $\mathbb{Z}_2$ symmetry fractionalizes on the bosonic anyon in $\mathrm{DW}(\mathbb{Z}_2)_{\omega|_N}$. We note that the boundary of the reduced TO can also be written as

$$\mathfrak{B}' = \frac{\mathfrak{B}_m \boxtimes \mathrm{DW}(\mathbb{Z}_{Z_4})_\omega}{\mathbb{Z}_2^{(1)}}, \tag{4.17}$$

which matches the expression for general abelian groups in Eq. (B.37). In this expression, gauging the diagonal $\mathbb{Z}_2^{(1)}$ symmetry removes extra lines in $\mathrm{DW}(\mathbb{Z}_{Z_4})_\omega$. The confined lines, such as $\mathcal{L}^m$ and $\mathcal{L}^{m^3}$, are then attached to the end of $Q_2^M$ surface in the reduced TO.

We now discuss the KT transformation to the $\mathbb{Z}_4^{(1)}$ gapped phases for $\omega = 0$. The KT transformation of the $\mathbb{Z}_2^{(1)}$ trivial phase is the untwisted $\mathbb{Z}_2^{(1)}$ SSB phase, which we denote as $\mathrm{DW}(\mathbb{Z}_2)_0$. Note that there is an unbroken $\mathbb{Z}_2^{(1)}$ symmetry generated by the $D_1^e$ line. The KT transformation of the $\mathrm{DW}(\mathbb{Z}_2)_0$ phase is the untwisted $\mathbb{Z}_4^{(1)}$ SSB phase, which we denote as $\mathrm{DW}(\mathbb{Z}_4)_0$. The KT transformation of the $\mathrm{DW}(\mathbb{Z}_2)_1$ phase is the $\mathbb{Z}_4^{(1)}$ SSB phase twisted by $2\omega \in H^3(\mathbb{Z}_4, U(1))$, which we denote as $\mathrm{DW}(\mathbb{Z}_4)_2$.

If we input $\mathfrak{T}_{\mathbb{Z}_2}/\mathbb{Z}_2$ on $\mathfrak{B}_{\mathrm{phys}}$, the KT transformed theory is given by

$$\frac{\mathfrak{T}_{\mathbb{Z}_2} \boxtimes \mathrm{DW}(\mathbb{Z}_2)_{\omega|_N}}{(\mathbb{Z}_2, \omega)} \circlearrowleft \quad \mathbb{Z}_2^{(1)} \longleftarrow \mathbb{Z}_4^{(1)} \tag{4.18}$$

For $\omega = 0$ and $\mathfrak{T}_{\mathbb{Z}_2} = \mathrm{Ising}$, this theory describes the phase transition between the $\mathrm{DW}(\mathbb{Z}_2)_0$ and $\mathrm{DW}(\mathbb{Z}_4)_0$ phases. For $\omega = 0$ and $\mathfrak{T}_{\mathbb{Z}_2} = \mathrm{Ising} \boxtimes \mathbb{Z}_2\text{-SPT}$, this theory describes the transition between the $\mathrm{DW}(\mathbb{Z}_2)_0$ and $\mathrm{DW}(\mathbb{Z}_4)_2$ phases.

## 4.4 (Mixed) Anomalous $\mathbb{Z}_2^{(0)} \times \mathbb{Z}_2^{(1)}$-Symmetric Phase Transitions

Finally, consider the symmetry $\mathfrak{B}_{\mathrm{sym}} = \mathfrak{B}_{\mathrm{Neu}(\mathbb{Z}_2)}$, which is obtained by $\mathbb{Z}_2$ gauging (with an SPT) the Dirichlet BC, and results in the 2-group $\mathbb{G}^{(2)} = (Z_2^{(0)} \times \mathbb{Z}_2^{(1)})^\beta$ with mixed anomaly $\beta \in H^4(\mathbb{Z}_2^{(0)} \times \mathbb{Z}_2^{(1)}, U(1))$, which arises from symmetry fractionalization. Both $\boldsymbol{Q}_2^{m^2}$ and $\boldsymbol{Q}_1^{e^2}$ can end on the symmetry boundary. The $\boldsymbol{Q}_2^m$ become the 0-form symmetry generators $D_2^m$, and two $D_2^m$ surfaces is isomorphic to the identity surface $D_2^{\mathrm{id}}$ and the isomorphism is implemented by the non-trivial junction which braids non-trivially with the 1-form symmetry generator $D_1^{e}$[20].

$\langle e^2 \rangle$ **Condensed Interface.** The club quiche is

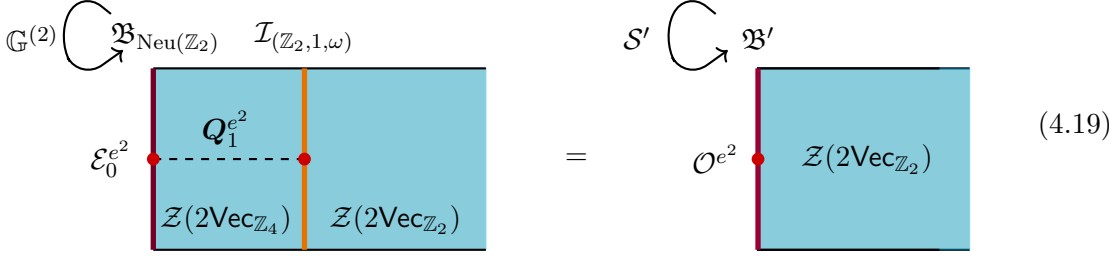

$$\tag{4.19}$$

As $\boldsymbol{Q}_2^M$ maps to $\boldsymbol{Q}_2^{m^2}$, it can end on $\mathfrak{B}_{\mathrm{sym}}$. Likewise $Q_1^E$ becomes $Q_1^e$ which generates the $\mathbb{Z}_2^{(1)}$ 1-form symmetry on $\mathfrak{B}_{\mathrm{sym}}$. There is a local operators $\mathcal{O}$, which come from compactifying

---

[20] This arises in the $\mathcal{Z}(2\mathsf{Vec}_{\mathbb{Z}_4})$ bulk through the non-trivial braiding between $Q_2^{m^2}$ and $Q_1^e$.

the $e^2$ line extended between $\mathfrak{B}_{\text{sym}}$ and the interface. Therefore, we find that boundary of $\mathcal{Z}(2\mathsf{Vec}_{\mathbb{Z}_2})$ reduced TO is given by

$$\mathfrak{B}' = (\mathfrak{B}_m)_0 \oplus (\mathfrak{B}_m)_1. \tag{4.20}$$

The KT transformation takes a theory with $\mathbb{Z}_2$ symmetry and output a theory with $\left(\mathbb{Z}_2^{(0)} \times \mathbb{Z}_2^{(1)}\right)^\beta$ symmetry. We now discuss the KT transformation of the gapped phases with $\omega = 0$. The KT transformation of the $\mathbb{Z}_2$ SSB phase is the $\mathbb{Z}_2$ SSB phase, in which there is a unbroken $\mathbb{Z}_2^{(1)}$ symmetry. The KT transformation of the $\mathbb{Z}_2$ trivial phase is the untwisted $\left(\mathbb{Z}_2^{(0)} \times \mathbb{Z}_2^{(1)}\right)^\beta$ SSB phase, which we denote as $\left(\mathbb{Z}_2^{(0)} \times \mathbb{Z}_2^{(1)}\right)^\beta \text{SSB}_0$. There are 2 vacua exchanged by $\mathbb{Z}_2$ symmetry, and each vacuum realizes the untwisted $\text{DW}(\mathbb{Z}_2)_0$ theory. Moreover, the $\mathbb{Z}_2$ symmetry fractionalized on the $e$ anyon due to the mixed anomaly. The KT transformation of the $\mathbb{Z}_2$ SPT phase is the twisted $\left(\mathbb{Z}_2^{(0)} \times \mathbb{Z}_2^{(1)}\right)^\beta$ SSB phase, which we denote as $\left(\mathbb{Z}_2^{(0)} \times \mathbb{Z}_2^{(1)}\right)^\beta \text{SSB}_1$. There are 2 vacua exchanged by $\mathbb{Z}_2$ symmetry, and each vacuum realizes the twisted $\text{DW}(\mathbb{Z}_2)_1$ theory, which is equivalent to the double semion topological order. Moreover, the $\mathbb{Z}_2$ symmetry fractionalized on the bosonic $s\bar{s}$ anyon due to the mixed anomaly.

If we input $\mathfrak{T}_{\mathbb{Z}_2}/\mathbb{Z}_2$ on $\mathfrak{B}_{\text{phys}}$, the KT transformed theory is given by

$$\mathbb{Z}_2^{(1)} \;\;\curvearrowright\;\; \frac{\mathfrak{T}_{\mathbb{Z}_2}}{\mathbb{Z}_2} \;\boxplus\; \frac{\mathfrak{T}_{\mathbb{Z}_2}}{\mathbb{Z}_2} \;\;\curvearrowleft\;\; \mathbb{Z}_2^{(1)} \atop \mathbb{Z}_2 \tag{4.21}$$

The $\mathbb{Z}_2$ symmetry acts by exchanging the two gauged Ising sectors. If $\omega = 0$ and $\mathfrak{T}_{\mathbb{Z}_2} = \text{Ising}$, this theory describes the phase transition between the $\mathbb{Z}_2$ SSB and the $\left(\mathbb{Z}_2^{(0)} \times \mathbb{Z}_2^{(1)}\right)^\beta \text{SSB}_0$ phases. If $\omega = 0$ and $\mathfrak{T}_{\mathbb{Z}_2} = \text{Ising} \boxtimes \mathbb{Z}_2\text{-SPT}$, this theory describes the transition between the $\mathbb{Z}_2$ SSB and the $\left(\mathbb{Z}_2^{(0)} \times \mathbb{Z}_2^{(1)}\right)^\beta \text{SSB}_1$ phases.

$\langle m^2 \rangle$ **Condensed Interface.** The $Q_1^E$ maps to $Q_1^{e^2}$ and ends on $\mathfrak{B}_{\text{sym}}$. The $Q_2^M$ becomes $D_2^m$ on $\mathfrak{B}_{\text{sym}}$. In addition, the $Q_2^{m^2}$ surface can end on both the symmetry boundary and the interface, and it becomes a line $\mathcal{L}^{m^2}$, which braids non-trivially with the 1-form symmetry generator $D_1^e$. We thus find that the boundary of the $\mathcal{Z}(2\mathsf{Vec}_{\mathbb{Z}_2})$ reduced TO is given by

$$\mathfrak{B}_e \boxtimes \text{DW}(\mathbb{Z}_2)_{\omega|_N} . \tag{4.22}$$

We now discuss the KT transformation of the gapped phases with $\omega = 0$. The KT transformation of the $\mathbb{Z}_2$ SSB phase is the $\left(\mathbb{Z}_2^{(0)} \times \mathbb{Z}_2^{(1)}\right)^\beta \text{SSB}_0$ phase. The KT transformation of the $\mathbb{Z}_2$ trivial phase is the untwisted symmetry enriched $\mathbb{Z}_2^{(1)}$ SSB phase, in which the $\mathbb{Z}_2$ symmetry

fractionalized on the $e$ anyon due to the mixed anomaly. We simply denote this phase as $\mathrm{DW}(\mathbb{Z}_2)_0$. The KT transformation of the $\mathbb{Z}_2$ trivial phase is the stacking of the $\mathbb{Z}_2$ SPT and the untwisted symmetry enriched $\mathbb{Z}_2^{(1)}$ SSB phase. Similarly, the $\mathbb{Z}_2$ symmetry fractionalized on the $e$ anyon. We denote this phase as $\mathrm{DW}(\mathbb{Z}_2)_0 \boxtimes \mathbb{Z}_2$-SPT.

We now input $\mathfrak{T}_{\mathbb{Z}_2}$ on $\mathfrak{B}_{\mathrm{phys}}$, the KT transformed theory is given by

$$\mathfrak{T}_{\mathbb{Z}_2} \boxtimes \mathrm{DW}(\mathbb{Z}_2)_{\omega|_N} \overset{\curvearrowleft}{\phantom{x}} \mathbb{Z}_2 \tag{4.23}$$

The $\mathbb{Z}_2^{(1)}$ symmetry is spontaneously broken in this gapless phase. If $\omega = 0$ and $\mathfrak{T}_{\mathbb{Z}_2} = \mathrm{Ising}$, this theory describes the phase transition between $\mathrm{DW}(\mathbb{Z}_2)_0$ and the $\left(\mathbb{Z}_2^{(0)} \times \mathbb{Z}_2^{(1)}\right)^{\beta}$ SSB$_0$ phases. If $\omega = 0$ and $\mathfrak{T}_{\mathbb{Z}_2} = \mathrm{Ising} \boxtimes \mathbb{Z}_2$-SPT, this theory describes the transition between $\mathrm{DW}(\mathbb{Z}_2)_0 \boxtimes \mathbb{Z}_2$-SPT and the $\left(\mathbb{Z}_2^{(0)} \times \mathbb{Z}_2^{(1)}\right)^{\beta}$ SSB$_0$ phases.

## 4.5  Non-Minimal Phase Transitions

We can also consider non-minimal interfaces and the associated club sandwiches and phase transitions.

**Non-minimal $\langle e^2 \rangle$ Condensed Interface.** We now describe the non-minimal interface on which the $e^2$ line can end. This is described by $H = \mathbb{Z}_2$, $N = 1$ and $\mathcal{A} = \mathcal{A}^{\mathrm{id}} \oplus \mathcal{A}^{m^2}$, a $\mathbb{Z}_2$ graded fusion category. The category of lines on this interface forms the MTC

$$\mathcal{Z}(\mathcal{A}) = \widetilde{\mathcal{M}}^{\mathrm{id}} \oplus \widetilde{\mathcal{M}}^{m^2}, \tag{4.24}$$

where the grading is provided by the $\mathsf{Rep}(\mathbb{Z}_2)$ line $D_1^e$ obtained from the projection of $Q_1^e$ on the interface, and $\widetilde{\mathcal{M}}^{\mathrm{id}} = \mathcal{Z}(\mathcal{A}^{\mathrm{id}})$. The properties of this interface are depicted as follows

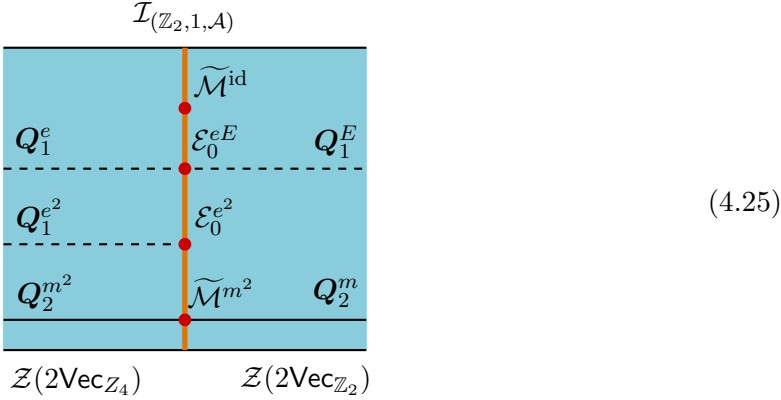

$$\tag{4.25}$$

**Non-minimal $\langle m^2 \rangle$ Condensed Interface.** Here we describe the non-minimal interface on which the $m^2$ surface condensed. This is described by $H = \mathbb{Z}_4$, $N = \mathbb{Z}_2$ and $\mathcal{A}$, a $\mathbb{Z}_4$ graded fusion category. The category of lines on this interface forms the MTC

$$\mathcal{Z}(\mathcal{A}) = \bigoplus_{h \in \mathbb{Z}_4} \widetilde{\mathcal{M}}^h \,, \tag{4.26}$$

where the grading is provided by the $\mathsf{Rep}(\mathbb{Z}_4)$ line $D_1^e$ obtained from the projection of $Q_1^e$ on the interface. The properties of this interface are depicted as follows

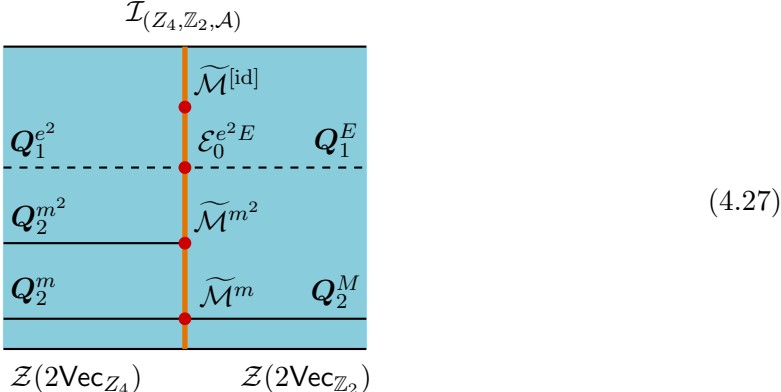

$$\tag{4.27}$$

### 4.5.1 $\mathbb{Z}_4$ Phase Transitions from Non-Minimal Interfaces

We first discuss the $\mathbb{Z}_4$ symmetric phase transitions with $\mathfrak{B}_{\mathrm{sym}} = \mathfrak{B}_{\mathrm{Dir}}$ and non-minimal interfaces.

**Phase Transitions from Non-minimal $\langle e^2 \rangle$.** To apply the KT transformation, we compactify the interval occupied by $\mathcal{Z}(2\mathsf{Vec}_{\mathbb{Z}_4})$. We obtain the following non-minimal boundary of the reduced TO

$$\mathfrak{B}' = \left( \frac{\mathfrak{B}_e / \mathbb{Z}_2 \boxtimes \mathcal{Z}(\mathcal{A})}{\mathbb{Z}_2^{(1)}} \right)_0 \oplus \left( \frac{\mathfrak{B}_e / \mathbb{Z}_2 \boxtimes \mathcal{Z}(\mathcal{A})}{\mathbb{Z}_2^{(1)}} \right)_1 \,, \tag{4.28}$$

Now we choose $\mathfrak{B}_{\mathrm{phys}}$ to be a $\mathbb{Z}_2$ symmetric theory $\mathfrak{T}_{\mathbb{Z}_2}$. The KT transformed phase transition is given by

$$
\mathbb{Z}_2 \hookrightarrow \overset{\displaystyle \mathbb{Z}_4}{\overset{\frown}{\frac{\mathfrak{T}_{\mathbb{Z}_2}/\mathbb{Z}_2 \boxtimes \mathcal{Z}(\mathcal{A})}{\mathbb{Z}_2^{(1)}}}} \boxplus \underset{\displaystyle \mathbb{Z}_4}{\underset{\smile}{\frac{\mathfrak{T}_{\mathbb{Z}_2}/\mathbb{Z}_2 \boxtimes \mathcal{Z}(\mathcal{A})}{\mathbb{Z}_2^{(1)}}}} \circlearrowleft \mathbb{Z}_2 \tag{4.29}
$$

The $\mathbb{Z}_4$ symmetry has been spontaneously broken down to $\mathbb{Z}_2$ and each vacua support a topological order described by the MTC $\mathcal{Z}(\mathcal{A}^{\mathrm{id}})$. This gapless theory describes the $\mathbb{Z}_2$ SSB transition between the $\mathbb{Z}_2$ symmetric MTC and the MTC in which the $\mathbb{Z}_4$ symmetry is completely broken.

A concrete example is obtained where $\mathcal{Z}(\mathcal{A}^{\mathrm{id}})$ is the Toric Code topological order. In this case, $\mathcal{A}$ is the Tambara-Yamagami category of $\mathbb{Z}_2$, which is a $\mathbb{Z}_2$ graded fusion category. The trivial component is $\mathsf{Vec}_{\mathbb{Z}_2}$ and the non-trivial grading component consists of a non-invertible object $\{\sigma\}$. The $\mathbb{Z}_4$ symmetry acts non-faithfully on the Toric Code, in which the $\mathbb{Z}_2$ subgroup acts as the automorphism that exchanges $e$ and $m$ anyons. Therefore, $\mathcal{Z}(\mathcal{A})$ is the double Ising topological order generated by the anyons $\{1, \sigma, \psi\} \times \{1, \bar{\sigma}, \bar{\psi}\}$. The gapless theory (4.29) describes the phase transition from the $\mathbb{Z}_2$ symmetry enriched Toric Code, in which the $\mathbb{Z}_4$ symmetry has been partially broken down to $\mathbb{Z}_2$, to the Toric Code with no 0-form symmetry. If we choose $\mathfrak{T}_{\mathbb{Z}_2} = \mathrm{Ising}$, each sector of the transition in (4.29) is described by the 3d Ising CFT in which the twisted sector line from the $\mathbb{Z}_2$ symmetry defect is decorated by the non-trivial anyons in $\widetilde{\mathcal{M}}^{m^2} = \{\sigma, \bar{\sigma}, \sigma\bar{\psi}, \psi\bar{\sigma}\}$.

**Phase Transitions from Non-minimal $\langle m^2 \rangle$.** In this case the club quiche results in an non-minimal BC

$$\mathfrak{B}' = \frac{\mathfrak{B}_e/\mathbb{Z}_2 \boxtimes \mathcal{Z}(\mathcal{A})}{\mathbb{Z}_2^{(1)}} \tag{4.30}$$

for the reduced TO. If we now input $\mathfrak{B}_{\mathrm{phys}} = \mathfrak{T}_{\mathbb{Z}_2}$, the KT transformed theory is given by

$$\frac{\mathfrak{T}_{\mathbb{Z}_2}/\mathbb{Z}_2 \boxtimes \mathcal{Z}(\mathcal{A})}{\mathbb{Z}_2^{(1)}} \quad \circlearrowleft \quad \mathbb{Z}_2 \longleftarrow \mathbb{Z}_4 \tag{4.31}$$

This gapless theory describes the $\mathbb{Z}_2$ SSB transition from a $\mathbb{Z}_4$ symmetric MTC $\mathcal{Z}(\mathcal{A}^{\mathrm{id}})$ to a $\mathbb{Z}_2$ symmetric MTC.

### 4.5.2 $\mathbb{Z}_4^{(1)}$ Phase Transitions from Non-Minimal Interfaces

We then discuss the $\mathbb{Z}_4^{(1)}$ symmetric phase transitions by choosing $\mathfrak{B}_{\mathrm{sym}} = \mathfrak{B}_{\mathrm{Neu}}$ with non-minimal interfaces.

**Non-minimal $\langle e^2 \rangle$ Condensed Interface.** After compactifying the left interval with $\mathcal{Z}(2\mathsf{Vec}_{\mathbb{Z}_4})$, we obtain the following non-minimal boundary of the reduced TO

$$\mathfrak{B}' = \frac{\mathfrak{B}_e \boxtimes \mathcal{Z}(\mathcal{A}^{\mathrm{id}})}{\mathbb{Z}_2}. \tag{4.32}$$

Now we input $\mathfrak{B}_{\mathrm{phys}} = \mathfrak{T}_{\mathbb{Z}_2}$, the KT transformed theory is given by

$$\frac{\mathfrak{T}_{\mathbb{Z}_2} \boxtimes \mathcal{Z}(\mathcal{A}^{\mathrm{id}})}{\mathbb{Z}_2} \circlearrowleft \quad \mathbb{Z}_2^{(1)} \longleftarrow \mathbb{Z}_4^{(1)} \tag{4.33}$$

This gapless theory describes the $\mathbb{Z}_2^{(1)}$ SSB transition between a non-minimal $\mathbb{Z}_4^{(1)}$ symmetric phase and a non-minimal $\mathbb{Z}_2^{(1)}$ SSB phase.

**Non-minimal $\langle m^2 \rangle$ Condensed Interface.** The club quiche in this case is more interesting as there are additional line $\mathcal{L}^{m^2}$ obtained from the $Q_2^{m^2}$ surface:

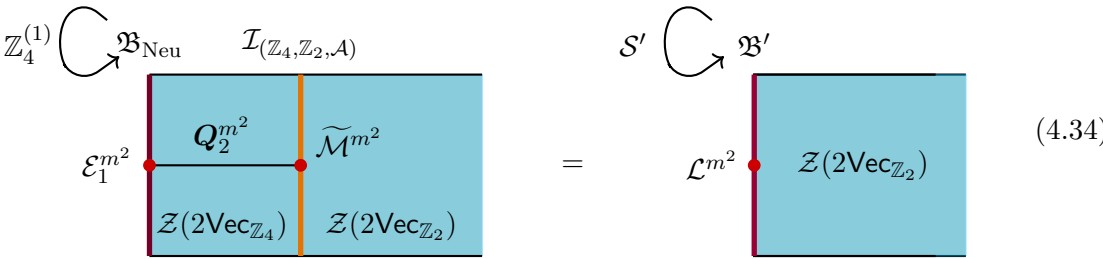

The genuine lines are described by $\widetilde{\mathcal{M}}^{\mathrm{id}} \oplus \widetilde{\mathcal{M}}^{m^2}$, which is a degenerate MTC since the $D_1^{e^2}$ line in $\widetilde{\mathcal{M}}^{\mathrm{id}}$ braids trivial with all other lines. Moreover, lines in $\widetilde{\mathcal{M}}^m \oplus \widetilde{\mathcal{M}}^{m^3}$ are attached to $Q_2^M$ surface after compactification. The non-minimal BC of the reduced TO can be obtained as follows. First, we gauge the $\mathbb{Z}_2^{(1)}$ symmetry generated by the $D_1^{e^2}$ in $\mathcal{Z}(\mathcal{A})$:

$$\widetilde{\mathcal{N}} = \mathcal{Z}(\mathcal{A})/Z_2^{(1)}, \tag{4.35}$$

so that the the lines in $\widetilde{\mathcal{M}}^m$ and $\widetilde{\mathcal{M}}^{m^3}$ are confined. With $D_1^{e^2}$ condensed, $\widetilde{\mathcal{N}}$ is then a $\mathbb{Z}_2$ symmetric MTC. Now the non-minimal BC of the reduced TO is given by

$$\mathfrak{B}' = \frac{\mathfrak{B}_e \boxtimes \widetilde{\mathcal{N}}}{\mathbb{Z}_2}. \tag{4.36}$$

Input $\mathfrak{B}_{\mathrm{phys}} = \mathfrak{T}_{\mathbb{Z}_2}$, the KT transformed theory is given by

$$\frac{\mathfrak{T}_{\mathbb{Z}_2} \boxtimes \widetilde{\mathcal{N}}}{\mathbb{Z}_2} \circlearrowleft \quad \mathbb{Z}_2^{(1)} \longleftarrow \mathbb{Z}_4^{(1)} \tag{4.37}$$

This theory describes the $\mathbb{Z}_2^{(1)}$ SSB between a non-minimal $\mathbb{Z}_2^{(1)}$ SSB and a non-minimal $\mathbb{Z}_4^{(1)}$ SSB phase.

### 4.5.3 $\mathbb{G}^{(2)} = \left( \mathbb{Z}_2^{(0)} \times \mathbb{Z}_2^{(1)} \right)^\beta$ Phase Transitions from Non-Minimal Interfaces

Finally we discuss the $\mathbb{G}^{(2)} = \left( \mathbb{Z}_2^{(0)} \times \mathbb{Z}_2^{(1)} \right)^\beta$ symmetric phase transitions by choosing $\mathfrak{B}_{\mathrm{sym}} = \mathfrak{B}_{\mathrm{Neu}(\mathbb{Z}_2)}$ with non-minimal interfaces.

**Non-minimal $\langle e^2 \rangle$ Condensed Interface.** In this case the club quiche results in an non-minimal BC of the reduced TO

$$\mathfrak{B}' = \left( \frac{\mathfrak{B}_e \boxtimes \mathcal{Z}(\mathcal{A}^{\mathrm{id}})}{\mathbb{Z}_2} \right)_0 \oplus \left( \frac{\mathfrak{B}_e \boxtimes \mathcal{Z}(\mathcal{A}^{\mathrm{id}})}{\mathbb{Z}_2} \right)_1 . \tag{4.38}$$

If we input $\mathfrak{B}_{\mathrm{phys}} = \mathfrak{T}_{\mathbb{Z}_2}$, the KT transformed theory is given by

$$\mathbb{Z}_2^{(1)} \stackrel{\curvearrowright}{} \frac{\mathfrak{T}_{\mathbb{Z}_2} \boxtimes \mathcal{Z}(\mathcal{A}^{\mathrm{id}})}{\mathbb{Z}_2} \boxplus \frac{\mathfrak{T}_{\mathbb{Z}_2} \boxtimes \mathcal{Z}(\mathcal{A}^{\mathrm{id}})}{\mathbb{Z}_2} \stackrel{\curvearrowleft}{} \mathbb{Z}_2^{(1)}$$
$$\mathbb{Z}_2 \tag{4.39}$$

There are two vacua since the $\mathbb{Z}_2$ symmetry is spontaneously broken in theory. This gapless theory describes the $\mathbb{Z}_2^{(1)}$ SSB transition between a non-minimal $\mathbb{Z}_2$ SSB phase and a non-minimal phase in which the $\mathbb{G}^{(2)}$ symmetry is completely broken.

**Non-minimal $\langle m^2 \rangle$ Condensed Interface.** After compactifying the interval with $\mathcal{Z}(2\mathsf{Vec}_{\mathbb{Z}_4})$, we find the genuine lines are described by $\widetilde{\mathcal{N}}^{\mathrm{id}} \oplus \widetilde{\mathcal{N}}^{m^2}$, where $\widetilde{\mathcal{N}}$ is defined in (4.35). Moreover, lines in $\widetilde{\mathcal{M}}^m \oplus \widetilde{\mathcal{M}}^{m^3}$ are attached to $Q_2^M$ surface. This non-minimal BC of the reduced TO is given by

$$\mathfrak{B}' = \frac{\mathfrak{B}_e/\mathbb{Z}_2 \boxtimes \widetilde{\mathcal{N}}/\mathbb{Z}_2}{\mathbb{Z}_2^{(1)}} . \tag{4.40}$$

If we input $\mathfrak{B}_{\mathrm{phys}} = \mathfrak{T}_{\mathbb{Z}_2}$, the KT transformed theory is given by

$$\frac{\mathfrak{T}_{\mathbb{Z}_2}/\mathbb{Z}_2 \boxtimes \widetilde{\mathcal{N}}/\mathbb{Z}_2}{\mathbb{Z}_2^{(1)}} \stackrel{\curvearrowleft}{} \mathbb{Z}_2 \tag{4.41}$$

The $\mathbb{Z}_2^{(1)}$ symmetry has been broken spontaneously. This theory describes the $\mathbb{Z}_2$ SSB transition of a $\mathbb{Z}_2$ symmetric MTC $\widetilde{\mathcal{N}}$.

# 5 Gapless Phases from $S_3$ DW Theory

In this section, we study gapless phases realized in theories with symmetries that are obtainable from $S_3$ 0-form symmetry via generalized gaugings. These symmetries are realized as the fusion 2-categories of topological defects on gapped boundaries of the (3+1)d $S_3$ Dijkgraaf-Witten theory. For the impatient reader, we have summarized the type of phase transitions we find in table 5.

| $\mathcal{S}$ | $\mathcal{Z}(\mathcal{S}_0) \cong$ | $\mathcal{S}$-symmetric CFT $\mathfrak{T}_C^{\mathcal{S}}$ |
|---|---|---|
| $2\mathsf{Vec}_{S_3}$ | $\mathcal{Z}(2\mathsf{Vec}_{\mathbb{Z}_2})$ | $\mathfrak{T}_{\mathbb{Z}_2} \circlearrowleft \quad 2\mathsf{Vec}_{\mathbb{Z}_2} \longleftarrow 2\mathsf{Vec}_{S_3}$ . |
| $2\mathsf{Vec}_{S_3}$ | $\mathcal{Z}(2\mathsf{Vec}_{\mathbb{Z}_2})$ | $\mathbb{Z}_2^b \circlearrowright \mathfrak{T}_{\mathbb{Z}_2^b} \boxplus \mathfrak{T}_{\mathbb{Z}_2^{ab}} \boxplus \mathfrak{T}_{\mathbb{Z}_2^{a^2 b}} \circlearrowleft \mathbb{Z}_2^{a^2 b}$, with $\mathbb{Z}_2^{ab}$ self-loop and $a$-arrows permuting, $a$ |
| $2\mathsf{Vec}_{S_3}$ | $\mathcal{Z}(2\mathsf{Vec}_{\mathbb{Z}_3})$ | $\mathbb{Z}_3^{a^2} \circlearrowright \mathfrak{T}_{\mathbb{Z}_3} \boxplus \mathfrak{T}_{\mathbb{Z}_3} \circlearrowleft \mathbb{Z}_3^{a}$, with $b$ |
| $\mathbb{Z}_3^{(1)} \rtimes \mathbb{Z}_2^{(0)}$ | $\mathcal{Z}(2\mathsf{Vec}_{\mathbb{Z}_2})$ | $\mathfrak{T}_{\mathbb{Z}_2} \boxtimes \mathrm{DW}(\mathbb{Z}_3)_{p-p'} \circlearrowleft \quad 2\mathsf{Vec}_{\mathbb{Z}_2}$ . |
| $\mathbb{Z}_3^{(1)} \rtimes \mathbb{Z}_2^{(0)}$ | $\mathcal{Z}(2\mathsf{Vec}_{\mathbb{Z}_2})$ | $\mathfrak{T}_{\mathbb{Z}_2} \circlearrowleft \quad 2\mathsf{Vec}_{\mathbb{Z}_2} \longleftarrow 2\mathsf{Vec}_{\mathbb{Z}_3^{(1)} \rtimes \mathbb{Z}_2^{(0)}}$ . |
| $\mathbb{Z}_3^{(1)} \rtimes \mathbb{Z}_2^{(0)}$ | $\mathcal{Z}(2\mathsf{Vec}_{\mathbb{Z}_3})$ | $\mathbb{Z}_3^{(1)} = \langle D_1^{\omega^2}\rangle \circlearrowright \mathfrak{T}_{\mathbb{Z}_3}/\mathbb{Z}_3^{(0)} \boxplus \mathfrak{T}_{\mathbb{Z}_3}/\mathbb{Z}_3^{(0)} \circlearrowleft \mathbb{Z}_3^{(1)} = \langle D_1^{\omega}\rangle$, with $b$ |
| $2\mathsf{Rep}(\mathbb{Z}_3^{(1)} \rtimes \mathbb{Z}_2^{(0)})$ | $\mathcal{Z}(2\mathsf{Vec}_{\mathbb{Z}_2})$ | $\mathfrak{T}_{\mathbb{Z}_2}/\mathbb{Z}_2^{(0)} \circlearrowleft \quad 2\mathsf{Rep}\mathbb{Z}_2 \longleftarrow 2\mathsf{Rep}(\mathbb{Z}_3^{(1)} \rtimes \mathbb{Z}_2^{(0)})$ . |
| $2\mathsf{Rep}(\mathbb{Z}_3^{(1)} \rtimes \mathbb{Z}_2^{(0)})$ | $\mathcal{Z}(2\mathsf{Vec}_{\mathbb{Z}_2})$ | $\mathbb{Z}_2^{(1)} \circlearrowright \dfrac{(\mathfrak{T}_{\mathbb{Z}_2})_0}{\mathbb{Z}_2^{(0)}} \boxplus \left(\dfrac{(\mathfrak{T}_{\mathbb{Z}_2})_1 \boxplus (\mathfrak{T}_{\mathbb{Z}_2})_2}{\mathbb{Z}_2^{(0)}}\right) \circlearrowleft \mathbb{Z}_2^{(1)} \ \mathrm{Triv}\ (\mathrm{D}_1^{\hat{b}} \cong \mathrm{D}_1^{\mathrm{id}})$, with $D_2^A$ |
| $2\mathsf{Rep}(\mathbb{Z}_3^{(1)} \rtimes \mathbb{Z}_2^{(0)})$ | $\mathcal{Z}(2\mathsf{Vec}_{\mathbb{Z}_3})$ | $\mathbb{Z}_2^{(1)} \circlearrowright \dfrac{(\mathfrak{T}_{\mathbb{Z}_2})_0}{\mathbb{Z}_2^{(0)}} \boxplus \left(\dfrac{(\mathfrak{T}_{\mathbb{Z}_2})_1 \boxplus (\mathfrak{T}_{\mathbb{Z}_2})_2}{\mathbb{Z}_2^{(0)}}\right) \circlearrowleft \mathbb{Z}_2^{(1)} \ \mathrm{Triv}\ (\mathrm{D}_1^{\hat{b}} \cong \mathrm{D}_1^{\mathrm{id}})$, with $D_2^A$ |
| $2\mathsf{Rep}(S_3)$ | $\mathcal{Z}(2\mathsf{Vec}_{\mathbb{Z}_2})$ | $\dfrac{\mathrm{DW}(\mathbb{Z}_3)_{p-p'} \boxtimes \mathfrak{T}_{\mathbb{Z}_2}}{(\mathbb{Z}_2^{(0),\mathrm{diag}}, t-t')} \quad \mathrm{igSSB}$ |
| $2\mathsf{Rep}(S_3)$ | $\mathcal{Z}(2\mathsf{Vec}_{\mathbb{Z}_2})$ | $\dfrac{\mathfrak{T}_{\mathbb{Z}_2}}{(\mathbb{Z}_2^{(0)}, t-t')} \circlearrowleft \quad 2\mathsf{Rep}\mathbb{Z}_2 \longleftarrow 2\mathsf{Rep}(S_3)$ . |
| $2\mathsf{Rep}(S_3)$ | $\mathcal{Z}(2\mathsf{Vec}_{\mathbb{Z}_3})$ | $\dfrac{\mathfrak{T}_{\mathbb{Z}_2}}{(\mathbb{Z}_3^{(0)}, p-p')} \circlearrowleft \quad 2\mathsf{Rep}\mathbb{Z}_3 \longleftarrow 2\mathsf{Rep}(S_3)$ . |

Table 5: Summary table of phase transitions from $\mathcal{Z}(S_3)$: $\mathfrak{T}_{\mathbb{Z}_2}$ can be chosen as the Ising CFT.

## 5.1 The SymTFT

We begin by very briefly reviewing the $S_3$ Dijkgraaf-Witten theory. We present the group $S_3$ as

$$S_3 = \langle a, b \mid a^3 = b^2 = 1, bab = a^2 \rangle. \tag{5.1}$$

The SymTFT which plays a central role in our analysis of symmetries and phases is the $(3+1)$d $S_3$ Dijkgraaf-Witten theory [30]. Let us briefly recap the topological defects in the SymTFT. For details see [2]. In general, the (co-dimension-2 and higher) topological defects of the $(3+1)$d $G$ Dijkgraaf-Witten theory are organized as [47]

$$\mathcal{Z}(2\mathsf{Vec}_G) = \boxplus_{[g]} 2\mathsf{Rep}(H_g), \tag{5.2}$$

where $[g]$ are conjugacy classes in $G$ and $H_g$ is the centralizer of a representative element in $[g]$. Since, $S_3$ has three conjugacy classes $[\mathrm{id}], [a] = \{a, a^2\}$ and $[b] = \{b, ab, a^2b\}$ with $H_{\mathrm{id}} = S_3, H_a = \mathbb{Z}_3$ and $H_b = \mathbb{Z}_2$, we obtain

$$\mathcal{Z}(2\mathsf{Vec}_{S_3}) = 2\mathsf{Rep}(S_3) \boxplus 2\mathsf{Rep}(\mathbb{Z}_3) \boxplus 2\mathsf{Rep}(\mathbb{Z}_2). \tag{5.3}$$

Up to condensations, each 2-category $2\mathsf{Rep}(H_g)$ has a single simple (non-condensation) object, i.e. topological 2d surface defect. We denote these simple objects by

$$\boldsymbol{Q}_2^{[\mathrm{id}]}, \ \boldsymbol{Q}_2^{[a]}, \ \boldsymbol{Q}_2^{[b]}, \tag{5.4}$$

labeled by the conjugacy classes of $S_3$. The topological surface $\boldsymbol{Q}_2^{[\mathrm{id}]}$ is the transparent or identity surface. The geniune topological lines of the SymTFT form the category $\mathsf{Rep}(S_3)$

$$1\mathrm{End}(\boldsymbol{Q}_2^{[\mathrm{id}]}) = \left\{\boldsymbol{Q}_1^1, \boldsymbol{Q}_1^P, \boldsymbol{Q}_1^E\right\} \cong \mathsf{Rep}(S_3). \tag{5.5}$$

Here $P$ denotes the 1-dimensional irreducible representation of $S_3$ that transforms with the sign $(-1)^j$ under $a^i b^j$ and $E$ denotes the 2-dimensional irreducible representation whose representation space is spanned by vectors $v_1$ and $v_2$ that transform as

$$a : v_i \longmapsto \omega^i v_i, \qquad b : v_1 \longleftrightarrow v_2, \tag{5.6}$$

with $\omega = \exp(2\pi i/3)$. Meanwhile the lines on $\boldsymbol{Q}_2^{[a]}$ and $\boldsymbol{Q}_2^{[b]}$ form

$$\begin{aligned}
1\mathrm{End}(\boldsymbol{Q}_2^{[a]}) &= \left\{\boldsymbol{Q}_1^{[a],1}, \boldsymbol{Q}_1^{[a],\omega}, \boldsymbol{Q}_1^{[a],\omega^2}\right\} \cong \mathsf{Rep}(\mathbb{Z}_3), \\
1\mathrm{End}(\boldsymbol{Q}_2^{[b]}) &= \left\{\boldsymbol{Q}_1^{[b],+}, \boldsymbol{Q}_1^{[b],-}\right\} \cong \mathsf{Rep}(\mathbb{Z}_2).
\end{aligned} \tag{5.7}$$

The phase between obtained by unlinking a genuine line $\boldsymbol{Q}_1^R$ wrapping a circle $S_1$ and a surface $\boldsymbol{Q}_2^{[g]}$ wrapping a sphere $S_2$ is

$$\chi_R([g]). \tag{5.8}$$

## 5.2 Symmetries and Gapped Boundary Conditions

The topological boundary conditions (BCs) of such a (3+1)d SymTFT can be divided into two classes: minimal and non-minimal. These were studied in detail in [2]. Here we summarize them for completeness. The minimal BCs are those for which every topological line defect of the boundary arises by projecting a topological line defect of the bulk. Equivalently, this means that there exist topological local operators on the boundary that connect each boundary line to some bulk line. The non-minimal BCs are those which have additional boundary lines not satisfying this property. On a non-minimal BC, any line that isn't obtainable by the boundary projection of a bulk line, is necessarily remotely detectable [21].

**Minimal boundary Conditions.** A convenient way to obtain any minimal boundary condition is to start with the Dirichlet boundary condition, and gauge a sub-group $H$ of the $S_3$ 0-form symmetry on it. There is an additional choice of discrete torsion $\omega \in H^3(H, U(1))$ in gauging $H$, which can be understood as stacking an $H$-SPT and then gauging [79]. The boundary condition obtained by gauging $H$ with discrete torsion $\omega$ is the $H$-Neumann boundary condition denoted as

$$\mathfrak{B}_{\text{Neu}(H),\omega} = \frac{\mathfrak{B}_{\text{Dir}}}{(H,\omega)} . \tag{5.9}$$

Let us describe the structure of these boundary conditions in terms of how the bulk topological defects of the SymTFT end on them, as this will be important for later purposes.

- **Dirichlet Boundary Condition.** Each SymTFT line $\boldsymbol{Q}_1^R$ for $R \in \mathsf{Rep}(S_3)$ has $\dim(R)$ ends on $\mathfrak{B}_{\text{Dir}}$ which we denote as $\mathcal{E}_0^{R,i}$ with $i = 1, \ldots, \dim(R)$. We denote this as

$$\boldsymbol{Q}_1^R \Big|_{\text{Dir}} = \left\{ \mathcal{E}_0^{R,i} \mid i = 1, \ldots, \dim(R) \right\} . \tag{5.10}$$

  Each bulk surface $\boldsymbol{Q}_2^{[g]}$ splits into a direct sum of surfaces $D_2^g$ for $g \in [g]$ on $\mathfrak{B}_{\text{Dir}}$, which fuse according to the group multiplication in $S_3$. These are the generators of the $S_3$ 0-form symmetry on this boundary. More specifically, the bulk surface $\boldsymbol{Q}_2^{[g]}$ has twisted sector ends $\mathcal{E}_1^g$ for all $g \in [g]$, where the line $\mathcal{E}_1^g$ is in the twisted sector of the $D_2^g$ defect on $\mathfrak{B}_{\text{Dir}}$. We denote this as

$$\boldsymbol{Q}_2^{[g]} \Big|_{\text{Dir}} = \left\{ (D_2^g, \mathcal{E}_1^g) \mid g \in [g] \right\} . \tag{5.11}$$

  The topological operators $\{\mathcal{E}_0^{R,i}\}$ transform as an $R$-multiplet under the $S_3$ symmetry. We summarize the $\mathfrak{B}_{\text{Dir}}$ boundary condition as (only the bulk defects with untwisted

---

[21]This statement is a corollary of the fact that non-modular braided fusion categories are boundary conditions for non-invertible Crane-Yetter-Kauffman TQFTs.

ends are shown)

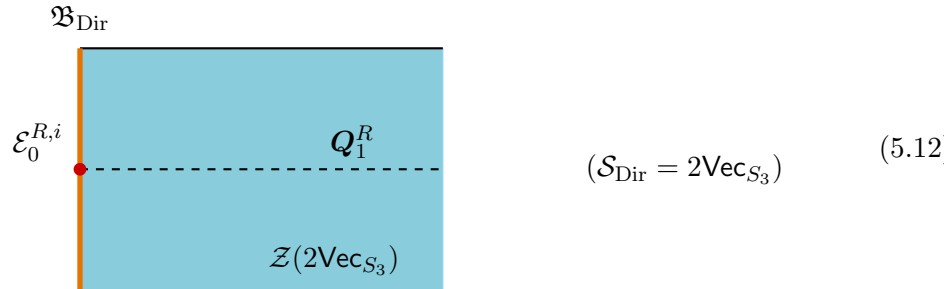

$$(\mathcal{S}_{\mathrm{Dir}} = 2\mathsf{Vec}_{S_3}) \tag{5.12}$$

- **Neumann $\mathbb{Z}_3$ Boundary Conditions.** These are obtained by gauging the $\mathbb{Z}_3$ symmetry generated by $D_2^a$ on $\mathfrak{B}_{\mathrm{Dir}}$. The local operator $\mathcal{E}_0^P$ at the end of $\boldsymbol{Q}_1^P$, being uncharged under $\mathbb{Z}_3$, remains unaltered upon such a gauging. In contrast the ends $\mathcal{E}_0^{E,j}$ which carried an $\omega^j$ (where $\omega = \exp\{2\pi i/3\}$) charge under $D_2^a$, become attached to the dual $\mathbb{Z}_3$ 1-form symmetry generator, which we denote as $\mathsf{D}_1^{\omega^j}$.

$$\boldsymbol{Q}_1^E\Big|_{\mathrm{Neu}(\mathbb{Z}_3),\omega} = \left\{ (D_1^\omega, \mathcal{E}_0^{E,1}), (D_1^{\omega^2}, \mathcal{E}_0^{E,2}) \right\}. \tag{5.13}$$

The $\mathbb{Z}_2$ 0-form symmetry $D_2^b$ exchanges $\mathcal{E}_0^{E,1}$ and $\mathcal{E}_0^{E,2}$ and consequently acts as

$$D_2^b : (D_1^\omega, \mathcal{E}_0^{E,1}) \longleftrightarrow (D_1^{\omega^2}, \mathcal{E}_0^{E,2}), \tag{5.14}$$

on these twisted sector operators. The lines $\mathcal{E}_1^a$ and $\mathcal{E}_1^{a^2}$, at the ends of $\boldsymbol{Q}_2^{[a]}$ become untwisted lines, that are charged under the dual $\mathbb{Z}_3$ 1-form symmetry.

$$\boldsymbol{Q}_2^{[a]}\Big|_{\mathrm{Neu}(\mathbb{Z}_3),\omega} = \left\{ \mathcal{E}_1^a, \mathcal{E}_1^{a^2} \right\}, \tag{5.15}$$

and exchanged under the $\mathbb{Z}_2^{(0)}$ action. These lines have F-symbols that are given by $\omega \in H^3(\mathbb{Z}_3, U(1))$. Finally the $\boldsymbol{Q}_2^{[b]}$ surface becomes a condensation defect on $\mathfrak{B}_{\mathrm{Dir}}$ on which the $\mathbb{Z}_3$ 1-form symmetry lines are condensed.

$$\boldsymbol{Q}_2^{[b]}\Big|_{\mathrm{Neu}(\mathbb{Z}_3),\omega} = \left( \frac{D_2^b}{D_1^{\mathrm{id}} \oplus D_1^\omega \oplus D_1^{\omega^2}}, \mathcal{E}_1^{[b]} \right). \tag{5.16}$$

In summary, this boundary carries a 2-group symmetry with the 1-form and 0-form symmetries generated by $D_1^\omega$ and $D_2^b$ respectively. The 0-form symmetry on the 1-form symmetry as (5.14). This boundary condition is depicted as (only the bulk defects with

untwisted ends are shown)

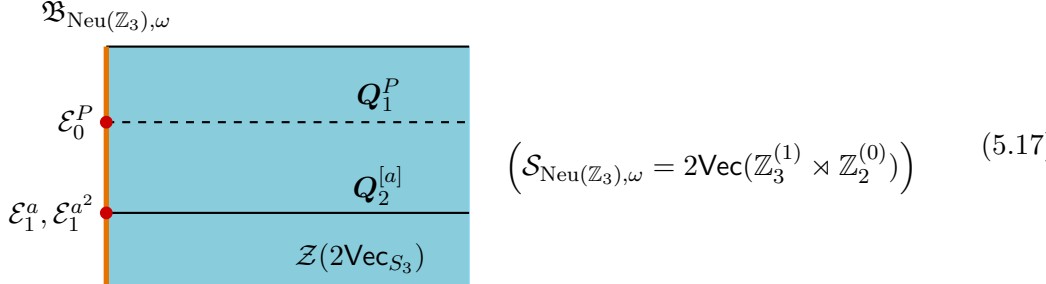

$$\left( \mathcal{S}_{\text{Neu}(\mathbb{Z}_3),\omega} = 2\text{Vec}(\mathbb{Z}_3^{(1)} \rtimes \mathbb{Z}_2^{(0)}) \right) \tag{5.17}$$

- **Neumann $\mathbb{Z}_2$ Boundary Conditions.** These boundary conditions are obtained by gauging $\mathbb{Z}_2^b$ on $\mathfrak{B}_{\text{Dir}}$ with discrete torsion $\omega \in H^3(\mathbb{Z}_2, U(1))$. Upon such a gauging, $\mathcal{E}_0^P$ which was charged under $\mathbb{Z}_2^b$ on $\mathfrak{B}_{\text{Dir}}$, goes to the twisted sector of a dual $\mathbb{Z}_2$ 1-form symmetry, which we denote as $D_1^{\widehat{b}}$.

$$\left. Q_1^P \right|_{\text{Neu}(\mathbb{Z}_2),\omega} = \left( D_1^{\widehat{b}}, \mathcal{E}_0^P \right). \tag{5.18}$$

The ends $\mathcal{E}_0^{E,1}$ and $\mathcal{E}_0^{E,2}$ of $Q_1^E$ on $\mathfrak{B}_{\text{Dir}}$, can be decomposed into linear combinations $\mathcal{E}_0^{E,\pm} = \mathcal{E}_0^{E,1} \pm \mathcal{E}_0^{E,2}$ with $\mathcal{E}_0^{E,+}$ and $\mathcal{E}_0^{E,-}$ being uncharged and charged under $D_2^b$. Therefore upon gauging $\mathbb{Z}_2^b$, one obtains

$$\left. Q_1^E \right|_{\text{Neu}(\mathbb{Z}_2),\omega} = \left\{ \mathcal{E}_0^{E,+}, (D_1^{\widehat{b}}, \mathcal{E}_0^{E,-}) \right\}. \tag{5.19}$$

The surfaces $D_2^a$ and $D_2^{a^2}$ are mapped to one another upon conjugation by $D_2^b$, therefore

$$\left. Q_2^{[a]} \right|_{\text{Neu}(\mathbb{Z}_2),\omega} = \left\{ (D_2^A, \mathcal{E}_1^A) \right\}, \tag{5.20}$$

where $D_2^A$ is an indecomposable defect on $\mathfrak{B}_{\text{Neu}(\mathbb{Z}_2),\omega}$ obtained from $D_2^a \oplus D_2^{a^2}$ on $\mathfrak{B}_{\text{Dir}}$. This non-invertible non-condensation defect has fusion rules

$$D_2^A \otimes D_2^A = D_2^{\text{id}} \oplus \frac{D_2^{\text{id}}}{D_1^{\text{id}} \oplus D_1^{\widehat{b}}}. \tag{5.21}$$

Similarly, $D_2^{ab}$ and $D_2^{a^2 b}$ are mapped to one another upon conjugation by $D_2^b$ therefore

$$\left. Q_2^{[b]} \right|_{\text{Neu}(\mathbb{Z}_2),\omega} = \left\{ \mathcal{E}_1^b, (D_2^A, \mathcal{E}_1^{Ab}) \right\}, \tag{5.22}$$

where $\mathcal{E}_1^{Ab}$ is an indecomposable twisted sector line on $\mathfrak{B}_{\text{Neu}(\mathbb{Z}_2),\omega}$ obtained from $\mathcal{E}_1^{ab} \oplus \mathcal{E}_1^{a^2 b}$ on $\mathfrak{B}_{\text{Dir}}$. The line $\mathcal{E}_1^b$ has F-symbols given by $\omega \in H^3(\mathbb{Z}_2, U(1))$. In summary, this

boundary carries a symmetry corresponding to the 2-representations of the 2-group generated by a (non-condensation) non-invertible 0-form symmetry generator $D_2^A$ and a $\mathbb{Z}_2$ 1-form generator $D_1^{\widehat{b}}$. This $\mathfrak{B}_{\text{Neu}(\mathbb{Z}_2),\omega}$ boundary condition is depicted as

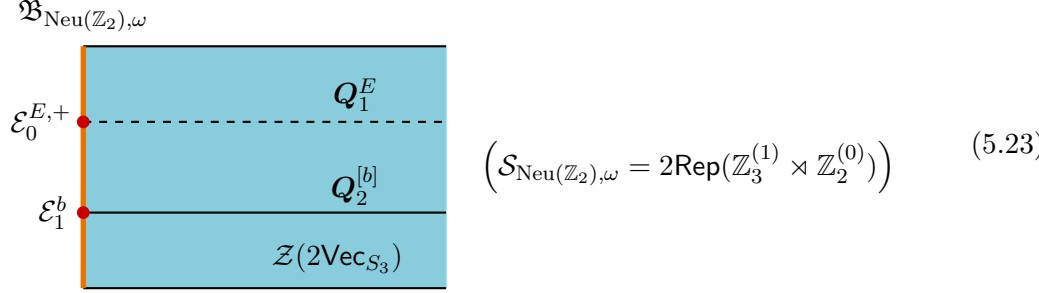

$$\left(\mathcal{S}_{\text{Neu}(\mathbb{Z}_2),\omega} = 2\text{Rep}(\mathbb{Z}_3^{(1)} \rtimes \mathbb{Z}_2^{(0)})\right) \qquad (5.23)$$

- **Neumann $S_3$ Boundary Conditions.** This boundary condition is obtained by gauging the full $S_3$ symmetry on $\mathfrak{B}_{\text{Dir}}$. The SymTFT lines $\boldsymbol{Q}_1^R$ end on twisted sector local operators

$$\boldsymbol{Q}_1^R\bigg|_{\text{Neu}(S_3),\omega} = (D_1^R, \mathcal{E}_0^R)\,, \qquad (5.24)$$

where the $D_1^R$ generate a $\text{Rep}(S_3)$ 1-form symmetry. The SymTFT surfaces end on single untwisted sector lines

$$\boldsymbol{Q}_2^{[g]}\bigg|_{\text{Neu}(S_3),\omega} = \mathcal{E}_1^{[g]}\,. \qquad (5.25)$$

The line $\mathcal{E}_1^{[g]}$ has a braiding phase of $\chi_R([g])$ with $D_1^R$. The lines $\mathcal{E}_1^{[g]}$ have F-symbols that are determined by $\omega \in H^3(S_3, U(1))$ and are precisely those of the 3d $S_3$ Dijkgraaf-Witten theory with topological action $\omega$ [80]. All other symmetry defects on this boundary can be obtained as condensation defects constructed from the $\text{Rep}(S_3)$ lines. The boundary condition can be summarized as (only the bulk defects with untwisted ends are shown)

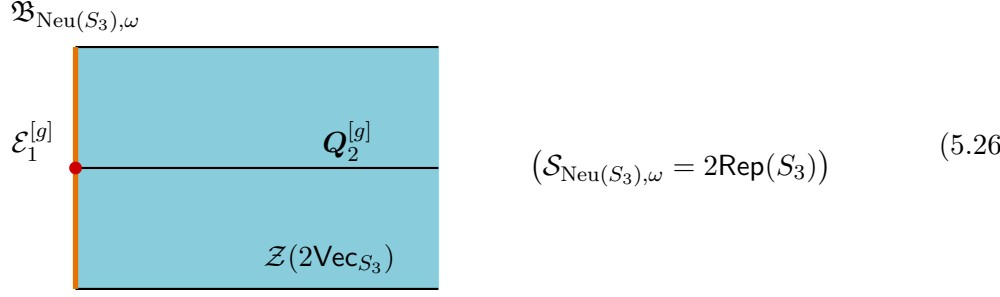

$$\left(\mathcal{S}_{\text{Neu}(S_3),\omega} = 2\text{Rep}(S_3)\right) \qquad (5.26)$$

| $H$ | $N$ | $H/N$ | $H^3(H,U(1))$ | $H^{\text{diag}}$ | $\boldsymbol{Q}_p$ that can end on $\mathcal{I}_{(H,N,\omega)}$ |
|---|---|---|---|---|---|
| $S_3$ | $1$ | $S_3$ | $\mathbb{Z}_6$ | $\langle(g,g)\rangle$ | $\begin{cases}\boldsymbol{Q}_2^{[\text{id}]} \\ \boldsymbol{Q}_1^1\end{cases}$ |
| $\mathbb{Z}_3$ | $1$ | $\mathbb{Z}_3$ | $\mathbb{Z}_3$ | $\langle(a,m),\ (a^2,m^2)\rangle$ | $\begin{cases}\boldsymbol{Q}_2^{[\text{id}]} \\ \boldsymbol{Q}_1^1 \oplus \boldsymbol{Q}_1^P\end{cases}$ |
| $S_3$ | $\mathbb{Z}_3$ | $\mathbb{Z}_2$ | $\mathbb{Z}_6$ | $\langle(a^i,1),\ (a^ib,m)\rangle$ | $\begin{cases}\boldsymbol{Q}_2^{[\text{id}]} \oplus \boldsymbol{Q}_2^{[a]} \\ \boldsymbol{Q}_1^1 \oplus \boldsymbol{Q}_1^{[a],1}\end{cases}$ |
| $\mathbb{Z}_2$ | $1$ | $\mathbb{Z}_2$ | $\mathbb{Z}_2$ | $\langle(b,m)\rangle$ | $\begin{cases}\boldsymbol{Q}_2^{[\text{id}]} \\ \boldsymbol{Q}_1^1 \oplus \boldsymbol{Q}_1^E\end{cases}$ |
| $1$ | $1$ | $1$ | $1$ | $\langle(1,1)\rangle$ | $\begin{cases}\boldsymbol{Q}_2^{[\text{id}]}, \\ \boldsymbol{Q}_1^1 \oplus \boldsymbol{Q}_1^P \oplus 2\,\boldsymbol{Q}_1^E\end{cases}$ |
| $\mathbb{Z}_3$ | $\mathbb{Z}_3$ | $1$ | $\mathbb{Z}_3$ | $\langle(a^i,1)\rangle$ | $\begin{cases}\boldsymbol{Q}_2^{[\text{id}]} \oplus \boldsymbol{Q}_2^{[a]} \\ \boldsymbol{Q}_1^1 \oplus \boldsymbol{Q}_1^P \oplus 2\,\boldsymbol{Q}_1^{[a],1}\end{cases}$ |
| $\mathbb{Z}_2$ | $\mathbb{Z}_2$ | $1$ | $\mathbb{Z}_2$ | $\langle(b,1)\rangle$ | $\begin{cases}\boldsymbol{Q}_2^{[\text{id}]} \oplus \boldsymbol{Q}_2^{[b]} \\ \boldsymbol{Q}_1^1 \oplus \boldsymbol{Q}_1^E \oplus \boldsymbol{Q}_1^{[b],+}\end{cases}$ |
| $S_3$ | $S_3$ | $1$ | $\mathbb{Z}_6$ | $\langle(g,1)\rangle$ | $\begin{cases}\boldsymbol{Q}_2^{[\text{id}]} \oplus \boldsymbol{Q}_2^{[a]} \oplus \boldsymbol{Q}_2^{[b]} \\ \boldsymbol{Q}_1^1 \oplus \boldsymbol{Q}_1^{[a],1} \oplus \boldsymbol{Q}_1^{[b],+}\end{cases}$ |

Table 6: Data for the interfaces of $\mathcal{Z}(S_3)$ (for surfaces). The reduced TO is $\mathcal{Z}(2\mathsf{Vec}_{H/N}^{\pi})$, although $\pi \in H^4(H/N, U(1))$ is trivial for all the present cases. The four cases with $H = N$ are gapped boundary conditions of the SymTFT and were discussed in [2].

## 5.3   Minimal Interfaces

A condensable algebra $(H, N, \pi, \omega)$ defines a topological interface between $\mathcal{Z}(2\mathsf{Vec}_G)$ and $\mathcal{Z}(2\mathsf{Vec}_{H/N}^\pi)$. In the present example there will be no topological action in the reduced TO since the $H^4$ cohomology class is trivial for all choices of $H$ and $N$, therefore $\pi = 0$ in the following. Consequently, the minimal interfaces in the $S_3$ SymTFT can be labeled by the datum $(H, N, \omega)$ where $H$ is a subgroup of $G$, $N$ is a normal subgroup of $H$ and $\omega \in H^3(H, U(1))$. We will denote this interface as

$$\mathcal{I}_{(H,N,\omega)} \,. \tag{5.27}$$

The different kinds of interfaces from $\mathcal{Z}(2\mathsf{Vec}_{S_3})$ to $\mathcal{Z}(2\mathsf{Vec}_{H/N})$ are summarized in Table 6. There are two kinds of non-trivial reduced TOs obtainable from DW$(S_3)$. These are DW$(\mathbb{Z}_2)$ and DW$(\mathbb{Z}_3)$. We now consider them in turn.

There are two classes of minimal interfaces between DW$(S_3)$ and DW$(\mathbb{Z}_2)$ and on to DW$(\mathbb{Z}_3)$, which we now discuss:

### 5.3.1   $Q_2^{[a]}$ Condensed Interface

We first consider an interface to DW$(\mathbb{Z}_2)$ for which the only non-trivial defect that can end is $\boldsymbol{Q}_2^{[a]}$. In terms of the aforementioned data, this corresponds to the choice $H = S_3$ and $N = \mathbb{Z}_3$. For this choice the projection map $\kappa : \mathsf{Rep}(S_3) \to \mathsf{Rep}(S_3)$ is just the identity, with a trivial kernel. Therefore, we recover that none of the SymTFT lines can end on this interface. The $\boldsymbol{Q}_2^{[a]}$ surface can however end on this interface as $[a] \in H$. The reduced topological order is

$$\mathcal{Z}(2\mathsf{Vec}_{S_3/\mathbb{Z}_3}) \equiv \mathcal{Z}(2\mathsf{Vec}_{\mathbb{Z}_2}) \,. \tag{5.28}$$

We will denote the non-trivial surface and line defects in the reduced TO as $\boldsymbol{Q}_2^m$ and $\boldsymbol{Q}_1^e$ respectively. Since $\boldsymbol{Q}_1^E$ braids non-trivially with $\boldsymbol{Q}_2^{[a]}$, it is confined on the interface. In contrast, $\boldsymbol{Q}_1^P$ braids trivially with $\boldsymbol{Q}_2^{[a]}$ and therefore can pass through the defect. It becomes the line $\boldsymbol{Q}_1^e$ of the reduced topological order. Finally, $\boldsymbol{Q}_2^{[b]}$ becomes $\boldsymbol{Q}_2^m$ in the reduced TO. In

summary, the map from defects on $\mathcal{Z}(2\mathsf{Vec}_{\mathbb{Z}_2})$ to $\mathcal{Z}(2\mathsf{Vec}_{S_3})$ is

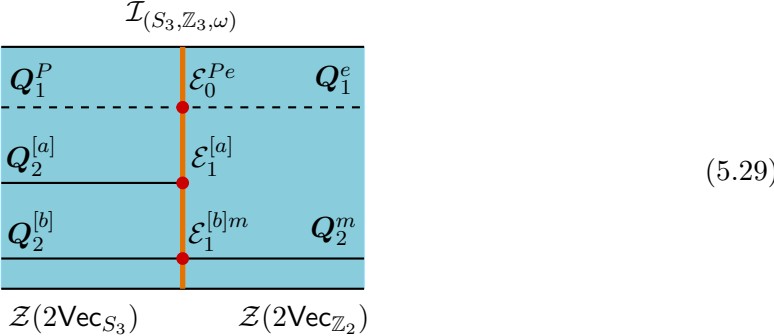

$$(5.29)$$

There is a further choice of discrete torsion valued in $(p,t) \in \mathbb{Z}_3 \times \mathbb{Z}_2 = H^3(S_3, U(1))$ when defining this interface. We denote the end of the $\boldsymbol{Q}_2^{[a]}$ surface on $\mathcal{E}_1^{[a]}$ and the line where the surfaces $\boldsymbol{Q}_2^{[b]}$ and $\boldsymbol{Q}_2^m$ meet on the interface as $\mathcal{E}_1^{[b]m}$. The choice of discrete $(p,t)$ translates to F-symbols for the lines $\mathcal{E}_1^{[a]}$ and $\mathcal{E}_1^{m[b]}$. More specifically, these are precisely the F-symbols for the flux lines in the $(p,t)$ twisted $S_3$ Dijkgraaf-Witten theory [80,81].

### 5.3.2 $Q_1^E$ Condensed Interface

Consider the interface where $\boldsymbol{Q}_1^E$ can end, which also is an interface to DW($\mathbb{Z}_2$). This corresponds to the choice $H = \mathbb{Z}_2$ and $N = 1$. Therefore, we consider the projection

$$\kappa : \mathsf{Rep}(S_3) \longrightarrow \mathsf{Rep}(\mathbb{Z}_2), \qquad \kappa : (\boldsymbol{Q}_1^P, \boldsymbol{Q}_1^E) \longmapsto (\boldsymbol{Q}_1^e, \boldsymbol{Q}_1^{\mathrm{id}} \oplus \boldsymbol{Q}_1^e). \qquad (5.30)$$

This implies that $\boldsymbol{Q}_1^E$ has a single end on this interface. Meanwhile $\boldsymbol{Q}_2^{[a]}$ gets confined as it braids non-trivially with $\boldsymbol{Q}_1^E$. The surface $\boldsymbol{Q}_2^{[b]}$ becomes $\boldsymbol{Q}_2^m$ in the reduced TO. In summary, the map from defects on $\mathcal{Z}(2\mathsf{Vec}_{\mathbb{Z}_2})$ to $\mathcal{Z}(2\mathsf{Vec}_{S_3})$ is

$$\mathcal{I}_{(\mathbb{Z}_2,1,\omega)}$$

| $\boldsymbol{Q}_1^P, \boldsymbol{Q}_1^E$ | $\mathcal{E}_0^{Pe/Ee}$ $\boldsymbol{Q}_1^e$ |
| $\boldsymbol{Q}_1^E$ | $\mathcal{E}_0^E$ |
| $\boldsymbol{Q}_2^{[b]}$ | $\mathcal{E}_1^{[b]m}$ $\boldsymbol{Q}_2^m$ |

$$\mathcal{Z}(2\mathsf{Vec}_{S_3}) \qquad \mathcal{Z}(2\mathsf{Vec}_{\mathbb{Z}_2}) \qquad\qquad (5.31)$$

The choice of discrete torsion $\omega$ determines the F-symbols of the lines $\mathcal{E}_1^{[b]m}$.

### 5.3.3   $Q_1^P$ Condensed Interface

There is a single class of minimal interfaces between DW($S_3$) and DW($\mathbb{Z}_3$). This interface corresponds to the choice $H = \mathbb{Z}_3$ and $N = 1$. Therefore, we consider the projection

$$\kappa : \mathsf{Rep}(S_3) \longrightarrow \mathsf{Rep}(\mathbb{Z}_3), \qquad \kappa : (D_1^P, D_1^E) \longmapsto (D_1^{\mathrm{id}}, D_1^\omega \oplus D_1^\omega). \tag{5.32}$$

This implies that $D_1^P$ can end on this interface. Furthermore, since $N = 1$, none of the bulk surfaces in $\mathcal{Z}(2\mathsf{Vec}_{S_3})$ can end on the interface. The reduced TO is $\mathcal{Z}(2\mathsf{Vec}_{\mathbb{Z}_3})$. Let us denote the topological lines and surfaces in $\mathcal{Z}(2\mathsf{Vec}_{\mathbb{Z}_3})$ as $\boldsymbol{Q}_1^{e^p}$ and $\boldsymbol{Q}_2^{m^q}$ respectively where $p, q = 0, 1, 2$. Since $\boldsymbol{Q}_1^P$ is condensed on the interface and $\boldsymbol{Q}_1^P \otimes \boldsymbol{Q}_1^E = \boldsymbol{Q}_1^E$, there is an extra local operator on $\boldsymbol{Q}_1^E$. Therefore this line splits as it passes through the interface. In the reduced TO $\boldsymbol{Q}_1^E$ becomes $\boldsymbol{Q}_1^e \oplus Q_1^{e^2}$. $\boldsymbol{Q}_2^{[a]}$ also passes through the interface as it braids trivially with $\boldsymbol{Q}_1^P$. Moreover, it also splits as it can absorb $\boldsymbol{Q}_1^P$. As a consequence in the reduced TO, this surface becomes $\boldsymbol{Q}_2^m \oplus \boldsymbol{Q}_2^{m^2}$. The surface, $\boldsymbol{Q}_2^{[b]}$ is confined on the interface as it braids non-trivially with $\boldsymbol{Q}_1^P$.

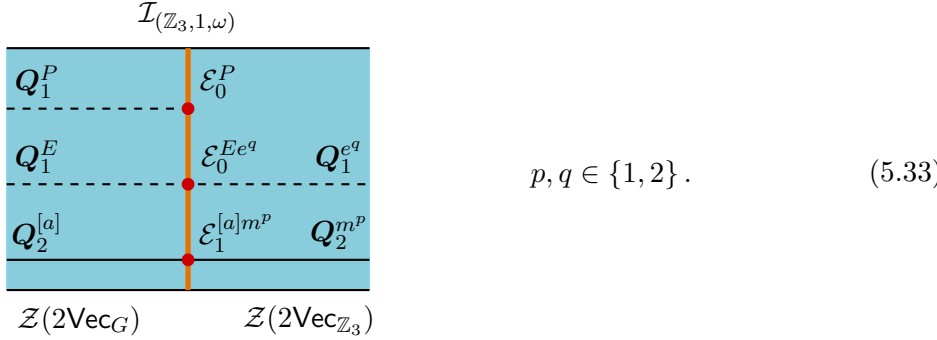

$$p, q \in \{1, 2\}. \tag{5.33}$$

Finally there is a choice of discrete torsion $p \in H^3(\mathbb{Z}_3, U(1))$. The choice of $p$ translates to the associators of two lines $\mathcal{E}_1^{[a]m}$ and $\mathcal{E}_1^{[a]m^2}$.

### 5.3.4   Hasse Diagram

Starting from a certain phase $\mathcal{P}$ defined via a set of condensed charges $\mathcal{Q}_\mathcal{P}$, a further deconfined charge, i.e., an element of $\mathcal{Q}_\mathcal{D}$ can be condensed without un-condensing any of the condensed charges. Such an additional condensation produces a less gapless phase $\mathcal{P}'$. Specifically we mean that the number of dynamical pr deconfined generalized charges in $\mathcal{P}'$ is fewer than in $\mathcal{P}$. A gapped phase is one where there are no deconfined charges. Therefore one obtains a partial ordering on phases, given by the inclusion of algebras. The complete set of condensable algebras along with the partial ordering can be conveniently organized into a Hasse diagram. For $\mathcal{Z}(2\mathsf{Vec}_{S_3})$, the Hasse diagram has the form in figure 6.

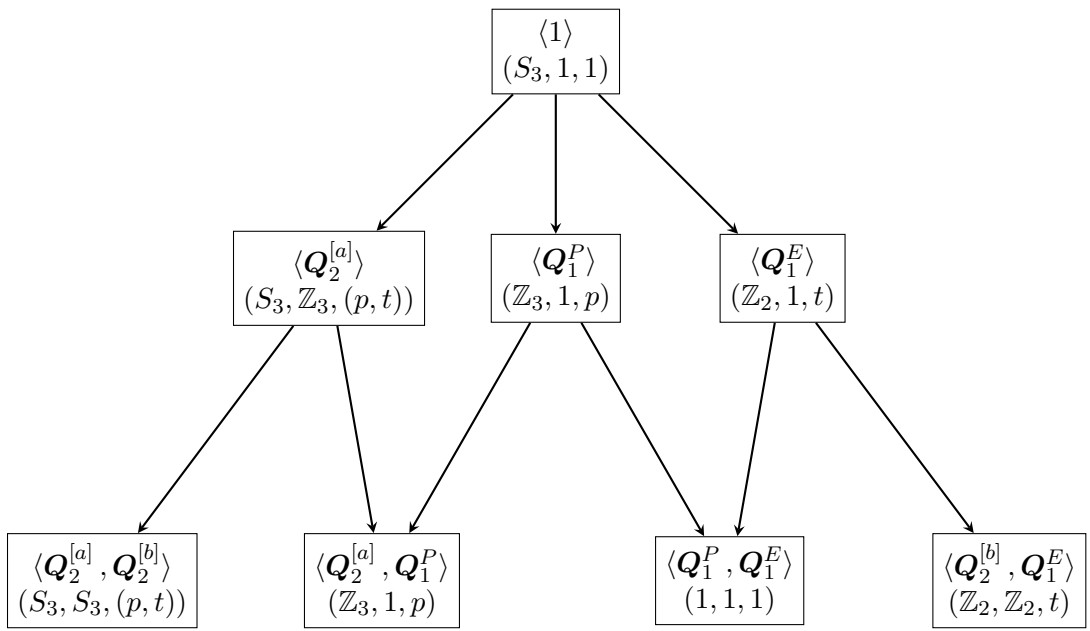

Figure 6: The set of condensable 2-algebras in $\mathcal{Z}(2\mathsf{Vec}_{S_3})$ has a partial ordering given by inclusion of algebras. These algebras form a Hasse diagram depicted above. Each box contains a condensable algebra, for which we provide generating objects and the data $(H, N, \omega)$.

## 5.4 Non-Minimal Interfaces

Non-minimal interfaces have the property that they contain line operators that are not obtainable via a boundary projection of a bulk SymTFT line. Equivalently, there are boundary lines for which there is no topological local operator that connects them to a bulk line. Non-minimal interfaces in $\mathcal{Z}(2\mathsf{Vec}_{S_3})$ are described by the datum $(H, N, \mathcal{A})$ where $\mathcal{A}$ is an $H$-graded fusion category. Note that minimal interfaces are recovered by setting $\mathcal{A} = \mathsf{Vec}_H^\omega$. Specifically, an interface labeled as $\mathcal{I}_{(H,N,\mathcal{A})}$ has the property that the category of lines on it forms $\mathcal{Z}(\mathcal{A})$. There is a $\mathsf{Rep}(H)$ worth of lines on this interface given by the boundary projection of lines in the SymTFT $\mathcal{Z}(2\mathsf{Vec}_{S_3})$ via the functor

$$\kappa : \mathsf{Rep}(G) \longrightarrow \mathsf{Rep}(H) \,. \tag{5.34}$$

These lines can be used to grade $\mathcal{Z}(\mathcal{A})$ as

$$\mathcal{Z}(\mathcal{A}) = \bigoplus_{[h]} \widetilde{\mathcal{M}}^{[h]} \,, \tag{5.35}$$

where the sum is over conjugacy classes in $H$. Specifically the grading is given by the braiding of a line in $\mathsf{Rep}(H)$ with the lines in $\mathcal{Z}(\mathcal{A})$. Any line in $\widetilde{\mathcal{M}}^{[h]}$ has a braiding of $\chi_R([h])$ with a line $D_1^R$ for $R \in \mathsf{Rep}(H)$. Clearly, $\mathsf{Rep}(H)$ is the Müger center of the grade $\widetilde{\mathcal{M}}^{[\mathrm{id}]}$. Only the

lines in the trivial grade $\widetilde{\mathcal{M}}^{[\mathrm{id}]}$ are unattached to a bulk surface. The lines in all other grades are attached to bulk surfaces. For $h \in \ker(p) \cong N$, the lines $\widetilde{\mathcal{M}}^{[n]}$ arise at the end of the $\boldsymbol{Q}_2^{[n]}$ surfaces in $\mathcal{Z}(2\mathsf{Vec}_G)$. For $h \in \mathrm{im}(p) \cong H/N$, the lines $\widetilde{\mathcal{M}}^{[h]}$ are attached to the $\boldsymbol{Q}_2^{[h]}$ surfaces in $\mathcal{Z}(2\mathsf{Vec}_G)$ and $\boldsymbol{Q}_2^{[p(h)]}$ surfaces in $\mathcal{Z}(2\mathsf{Vec}_{H/N})$.

**Non-Minimal Interfaces from Non-Minimal Gauging.** Non-minimal interfaces between $\mathcal{Z}(2\mathsf{Vec}_{S_3})$ and $\mathcal{Z}(2\mathsf{Vec}_{\mathbb{Z}_n})$ for $n = 2, 3$ can be obtained via a modification of the "folding and gauging approach. Wherein, we stack the Dirichlet boundary condition of the folded theory $\overline{\mathrm{DW}(S_3)} \boxtimes \mathrm{DW}(\mathbb{Z}_n)$ with an $H$ symmetric topological order $\mathfrak{T}_H$ to obtain

$$\mathfrak{B}_{\mathrm{Dir}}^{\mathfrak{T}_H} = \mathfrak{B}_{\mathrm{Dir}} \boxtimes \mathfrak{T}_H. \tag{5.36}$$

The underlying category of lines of $\mathfrak{T}_H$ is described by an MTC $\mathcal{M}$ and $H$ action is described via a $H$-crossed braided extension of $\mathcal{M}$, denoted as $\mathcal{M}_H^\times$. In terms of the above defined $H$-graded fusion category $\mathcal{A}$

$$\mathcal{M} = \mathcal{Z}(\mathcal{A}^{\mathrm{id}}), \tag{5.37}$$

where $\mathcal{A}^{\mathrm{id}}$ is the identity grade. $\mathcal{A}$ is the category of lines on the Dirichlet boundary of $\mathfrak{T}_H$, where the $H$-grading is given by the lines on the ends of $H$-condensation surfaces on the boundary. Then we gauge $H^{\mathrm{diag}}$ [22] defined via $\mathcal{A}$ (see section 3) and unfold to obtain a non-minimal interface $\mathcal{I}_{(H,N,\mathcal{A})}$

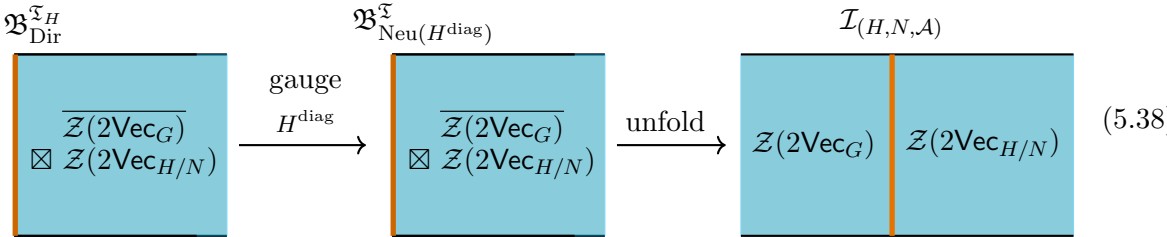

$$\tag{5.38}$$

### 5.4.1 Non-Minimal Interfaces to $\mathrm{DW}(\mathbb{Z}_2)$

There are two classes on non-minimal interfaces to the reduced TO $\mathrm{DW}(\mathbb{Z}_2)$, which are the non-minimal extensions of the interfaces with $\boldsymbol{Q}_2^{[a]}$ and $\boldsymbol{Q}_1^E$ condensed respectively. Let us describe these in turn

**Non-minimal $\boldsymbol{Q}_2^{[a]}$ Condensed Interface.** Recall that this interface corresponds to $H = S_3$ and $N = 1$. Correspondingly, we stack an $S_3$-symmetric 3d TFT $\mathcal{M} = \mathcal{Z}(\mathcal{A}^{\mathrm{id}})$ on $\mathfrak{B}_{\mathrm{Dir}}$

---

[22]We are abusing notation here as by $H^{\mathrm{diag}}$, we really mean a diagonal between $H^{\mathrm{diag}}$ and the $H$ symmetry action on $\mathfrak{T}_H$.

and subsequently gauge the diagonal $S_3$. Let us denote the $S_3$ crossed braided extension of $\mathcal{M}$ as

$$\mathcal{M}_{S_3}^{\times} = \bigoplus_{g \in S_3} \mathcal{M}^g \,, \tag{5.39}$$

where the grade $\mathcal{M}^g$ contains the topological line defects at the end of the condensation surface that implements the $g \in S_3$ symmetry in $\mathfrak{T}$. After gauging $S_3$, this grade combines with other grades in the $[g]$ conjugacy class. Moreover, these become untwisted topological line operators that are charged under the dual non-invertible $\mathsf{Rep}(S_3)$ 1-form symmetry. Specifically, after gauging $S_3$, one obtains

$$\widetilde{\mathcal{M}} = \widetilde{\mathcal{M}}^{[\mathrm{id}]} \oplus \widetilde{\mathcal{M}}^{[a]} \oplus \widetilde{\mathcal{M}}^{[b]} \,, \tag{5.40}$$

where

$$\widetilde{\mathcal{M}}^{[\mathrm{id}]} = \frac{\mathcal{M}^{\mathrm{id}}}{S_3} \,, \quad \widetilde{\mathcal{M}}^{[a]} = \frac{\mathcal{M}^a \oplus \mathcal{M}^{a^2}}{S_3} \,, \quad \widetilde{\mathcal{M}}^{[b]} = \frac{\mathcal{M}^b \oplus \mathcal{M}^{ab} \oplus \mathcal{M}^{a^2 b}}{S_3} \,. \tag{5.41}$$

Let us describe the physical meaning of $\widetilde{\mathcal{M}}^{[g]}$ in terms of defects on the interface, SymTFT and reduced TO. On the interface, $\mathcal{I}_{(S_3, \mathbb{Z}_3, \mathcal{A})}^{\mathfrak{T}}$, $\widetilde{\mathcal{M}}^{[\mathrm{id}]}$ forms a braided fusion category of lines that are unattached to any bulk surfaces either from the SymTFT or the reduced TO. The Müger center of $\widetilde{\mathcal{M}}^{[\mathrm{id}]}$ is

$$\mathcal{Z}_2(\widetilde{\mathcal{M}}^{[\mathrm{id}]}) = \mathsf{Rep}(S_3) \,. \tag{5.42}$$

Specifically, the lines that generate $\mathcal{Z}_2(\widetilde{\mathcal{M}}^{[\mathrm{id}]})$ are obtainable via the projection of $\boldsymbol{Q}_1^P$ and $\boldsymbol{Q}_1^E$ onto the interface. Next, the lines in $\widetilde{\mathcal{M}}^{[a]}$ are attached to the $\boldsymbol{Q}_2^{[a]}$ surface in DW$(S_3)$. This is the non-minimal generalization of $\mathcal{E}_1^{[a]}$. Finally, the lines in $\widetilde{\mathcal{M}}^{[b]}$ are attached to the $\boldsymbol{Q}_2^{[b]}$ surface in DW$(S_3)$ and $\boldsymbol{Q}_2^m$ surface in DW$(\mathbb{Z}_2)$. This is the non-minimal generalization of $\mathcal{E}_1^{[b]m}$. This interface can be depicted as

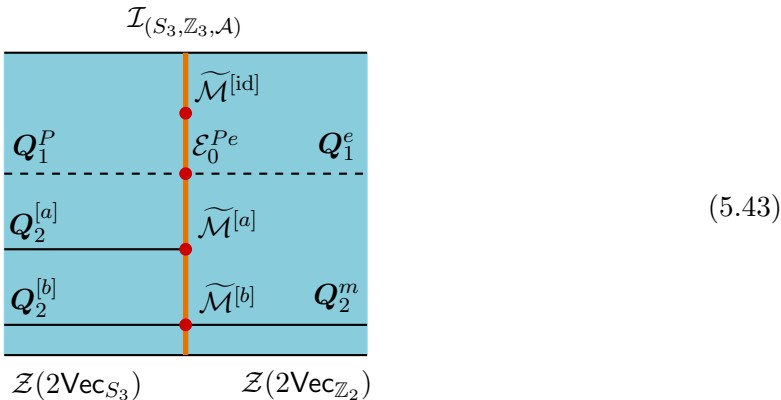

$$\tag{5.43}$$

The associators and other remaining data of these lines are internal to $\widetilde{\mathcal{M}} = \mathcal{Z}(\mathcal{A})$.

**Non-minimal $Q_1^E$ Condensed Interface.**   Following the same procedure, we now obtain minimal interface, on which the category of lines forms the MTC

$$\mathcal{Z}(\mathcal{A}) \,, \tag{5.44}$$

where $\mathcal{A}$ is a $\mathbb{Z}_2$ graded fusion category. In this case, $\mathcal{Z}(\mathcal{A})$ has an order 2 Bosonic line $D_1^P$ obtained by the projection of $\boldsymbol{Q}_1^P$. This line provides a grading on $\mathcal{Z}(\mathcal{A})$ as

$$\mathcal{Z}(\mathcal{A}) = \widetilde{\mathcal{M}}^{\mathrm{id}} \oplus \widetilde{\mathcal{M}}^p \,, \tag{5.45}$$

such that the lines in $\widetilde{\mathcal{M}}^{\mathrm{id}}$ braid trivially with $D_1^P$ while the lines in $\widetilde{\mathcal{M}}^p$ have a -1 braiding with $D_1^P$. The properties of this interface are depicted as follows

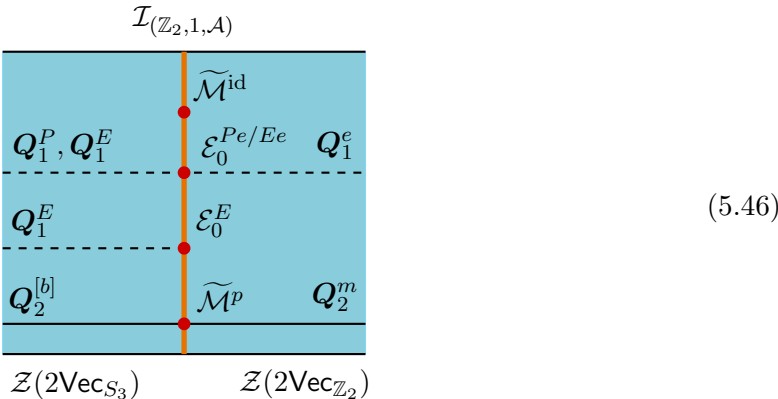

$$\tag{5.46}$$

### 5.4.2   Non-Minimal Interfaces to $\mathrm{DW}(\mathbb{Z}_3)$

**Non-minimal $Q_1^P$ Condensed Interface.**   We now describe the non-minimal interface, on which the $\boldsymbol{Q}_1^P$ line can end. This is described by $H = \mathbb{Z}_3$, $N = 1$ and $\mathcal{A}$ a $\mathbb{Z}_3$ graded fusion category. The category of lines on this interface forms the MTC

$$\mathcal{Z}(\mathcal{A}) \,. \tag{5.47}$$

In this case, $\mathcal{Z}(\mathcal{A})$ has a $\mathsf{Rep}(\mathbb{Z}_3)$ category of lines obtained by projecting the $\mathrm{DW}(S_3)$ lines via the projection functor

$$\kappa : \mathsf{Rep}(S_3) \longrightarrow \mathsf{Rep}(\mathbb{Z}_3) \,. \tag{5.48}$$

In particular, the lines $D_1^\omega$ and $D_1^{\omega^2}$ are obtained by the projection of $\boldsymbol{Q}_1^E$. The $\mathsf{Rep}(\mathbb{Z}_3)$ lines provide a grading on $\mathcal{Z}(\mathcal{A})$ as

$$\mathcal{Z}(\mathcal{A}) = \bigoplus_{h \in \mathbb{Z}_3} \widetilde{\mathcal{M}}^h \,, \tag{5.49}$$

such that the braiding of the lines in $\widetilde{\mathcal{M}}^h$ with $D_1^\omega$ is $\exp\{2\pi i h/3\}$. The properties of this interface are depicted as follows

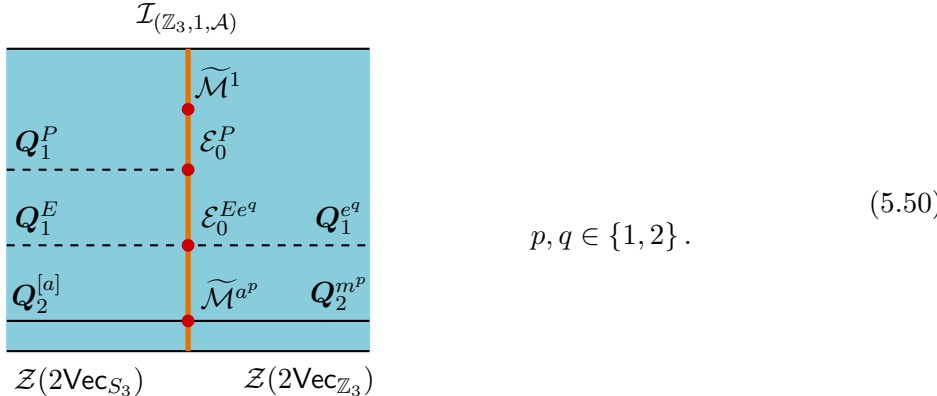

$$p, q \in \{1, 2\}\,. \tag{5.50}$$

## 5.5 Club Quiches

We now pick the symmetry boundary $\mathfrak{B}_{\mathrm{sym}}$ to be one of the minimal gapped boundaries of $\mathrm{DW}(S_3)$ such that the 2-category of defects on it determine the symmetry $\mathcal{S}$. Then, upon inserting a topological interface (minimal or non-minimal) to a reduced TO and compactifying the interval occupied by the SymTFT $\mathrm{DW}(S_3)$, one obtains a (possibly decomposable) boundary condition $\mathfrak{B}'$ of the reduced TO. Let the symmetry carried by $\mathfrak{B}'$ be denoted as $\mathcal{S}'$. We determine the 2-functor

$$\mathcal{F} : \mathcal{S} \longrightarrow \mathcal{S}'\,, \tag{5.51}$$

for different choices of symmetry boundary and interface.

### 5.5.1 2Vec($S_3$) Symmetric Club Quiches

Consider $\mathfrak{B}_{\mathrm{sym}} = \mathfrak{B}_{\mathrm{Dir}}$, on which all the SymTFT lines can end. This boundary carries an $S_3$ 0-form symmetry on it. Now let us in turn choose various possible minimal interfaces

- $\boldsymbol{Q}_2^{[a]}$ **Condensed Interface:** As described previously, this is an interface to the $\mathbb{Z}_2$ reduced TO. Since $\boldsymbol{Q}_1^P$ can end on $\mathfrak{B}_{\mathrm{sym}}$ and $\boldsymbol{Q}_1^P$ is connected to $\boldsymbol{Q}_1^e$ across the interface, after compactifying the interval occupied by $\mathcal{Z}(2\mathsf{Vec}_{S_3})$, we obtain a boundary condition $\mathfrak{B}'$ for the $\mathbb{Z}_2$ reduced TO on which $\boldsymbol{Q}_1^e$ can end. This is therefore the Dirichlet boundary condition.

  The $\boldsymbol{Q}_1^E$ line in the $S_3$ SymTFT gets confined on this interface and therefore becomes a line isomorphic to the identity line on compactifying the $S_3$ SymTFT. The $D_2^a$ symmetry

generating the $\mathbb{Z}_3$ subgroup of $S_3$ has a twisted sector operator obtained from compactifying the interval occupied by $\mathcal{Z}(2\mathsf{Vec}_{S_3})$ with the $\boldsymbol{Q}_2^{[a]}$ surface stretched between the interface and symmetry boundary. Specifically, after taking the club quiche one obtains two twisted sector lines

$$(D_2^a, \mathcal{L}^a) \quad \text{and} \quad (D_2^{a^2}, \mathcal{L}^{a^2}). \tag{5.52}$$

These satisfy $\mathbb{Z}_3$ fusion rules and their associator depends on $p$ where $\omega = (p, t) \in \mathbb{Z}_3 \times \mathbb{Z}_3 \cong \in H^3(S_3, U(1))$ These are the properties of a $\mathbb{Z}_3$ SPT labeled by $p$. The underlying 3d TFT corresponding to a $\mathbb{Z}_3$ SPT is simply the Trivial theory.

Let us finally describe the $\mathbb{Z}_2$ symmetry on $\mathfrak{B}'$. The $\boldsymbol{Q}_2^m$ surface becomes the $\boldsymbol{Q}_2^{[b]}$ surface which ends on $\mathfrak{B}_{\mathrm{sym}}$ on a twisted sector line, attached to a $\mathbb{Z}_2$ symmetry defect. This $\mathbb{Z}_2$ defect has two relevant properties — (i) the end of $\boldsymbol{Q}_1^P$ is charged under it and (ii) it swaps the two ends of the $\boldsymbol{Q}_2^{[a]}$ surface which form a generalized 1-charge of $\mathbb{Z}_2$. Consequently after taking the club quiche, i.e., compactifying the interval occupied by $\mathcal{Z}(2\mathsf{Vec}_{S_3})$, we find that the end of $\boldsymbol{Q}_2^m$ on $\mathfrak{B}'$ is attached to a $\mathbb{Z}_2$ defect that satisfies the following two properties. (i) the end of $\boldsymbol{Q}_1^e$ is charged under it and (ii) it swaps the twisted sector lines in (5.52). The boundary condition is simply

$$\mathfrak{B}' = \mathfrak{B}_e. \tag{5.53}$$

Since the 3d TFT underlying an SPT is trivial, i.e., it does not have any non-identity defects, the symmetry category on $\mathfrak{B}'$ is simply

$$\mathcal{S}' = 2\mathsf{Vec}_{\mathbb{Z}_2}. \tag{5.54}$$

The club quiche furnishes a 2-functor mapping between the symmetry categories

$$\mathcal{F} : 2\mathsf{Vec}_{S_3} \longrightarrow 2\mathsf{Vec}_{\mathbb{Z}_2}, \tag{5.55}$$

under which $\mathcal{F}(D_2^b) = D_2^m$ and $\mathcal{F}(D_2^a) = D_2^{\mathrm{id}}$. The associators of $D_2^m$ are modified by $t \in H^3(\mathbb{Z}_2, U(1))$.

**Non-minimal generalization.** Consider the non-minimal generalization of the interface $\mathcal{I}_{(S_3, \mathbb{Z}_3, \mathcal{A})}$, where $\mathcal{A}$ is an $S_3$ graded fusion category. By picking $\mathcal{A} = \mathsf{Vec}_{S_3}^\omega$ and viewing it as an $S_3$ graded fusion category, we recover the minimal case. Now let us consider a more general $S_3$ graded algebra. The corresponding interface was described

previously. The new non-minimal aspect of this interface is that it hosts an MTC of line defects which is

$$\mathcal{Z}(\mathcal{A}) = \bigoplus_{[h]} \widetilde{\mathcal{M}}^{[h]} \, . \tag{5.56}$$

Among these, the lines in the grade $\widetilde{\mathcal{M}}^{[\mathrm{id}]}$ are unattached to any SymTFT surface while lines in the remaining grades are attached to surfaces in the SymTFT or reduced TO.

Upon compactifying the interval occupied by $\mathcal{Z}(2\mathsf{Vec}_{S_3})$, we obtain a non-minimal boundary condition $\mathfrak{B}'$ of $\mathcal{Z}(2\mathsf{Vec}_{\mathbb{Z}_2})$, which hosts the genuine lines $\widetilde{\mathcal{M}}^{[\mathrm{id}]} = \mathcal{Z}(\mathcal{A}^{\mathrm{id}})$. Furthermore, compactifying the SymTFT interval occupied by $\mathcal{Z}(2\mathsf{Vec}_{S_3})$ provides $\mathbb{Z}_3$ twist defect order parameters. $\mathfrak{B}'$ has a $\mathbb{Z}_2^{(0)}$ symmetry implemented by the parallel projection of $\boldsymbol{Q}_2^m$. For reasons similar to the minimal case described above, the $\mathbb{Z}_2^{(0)}$ defect acts on the local operator at the end of the $\boldsymbol{Q}_1^e$ line and also implements a $\mathbb{Z}_2$ action on the topological order. The boundary condition is identified as

$$\mathfrak{B}' = \frac{\left( \dfrac{\mathfrak{B}_e}{\mathbb{Z}_2^{(0)}} \boxtimes \dfrac{\mathcal{Z}(\mathcal{A}^{\mathrm{id}})}{\mathbb{Z}_2^{(0)}} \right)}{\mathbb{Z}_2^{(1),\mathrm{diag}}} \, . \tag{5.57}$$

The category of defects on this boundary is

$$\mathcal{S}' = 2\mathsf{Vec}_{\mathbb{Z}_2} \boxtimes \Sigma(\mathcal{Z}(\mathcal{A}^{\mathrm{id}})) \tag{5.58}$$

The $\mathbb{Z}_3$ 0-form symmetry is realized as a condensation defect that implements the $\mathbb{Z}_3 \subset S_3$ symmetry of the MTC. The $\mathbb{Z}_2^{(0)}$ symmetry is realized as the diagonal $D_2^m \otimes D_2^\phi$, where $D_2^m$ is the non-trivial object in $2\mathsf{Vec}_{\mathbb{Z}_2}$, while $D_2^\phi$ is the condensation defect implementing $\mathbb{Z}_2 \subset S_3$ symmetry.

- $\boldsymbol{Q}_1^E$ **Condensed interface:** This corresponds to an interface to the $\mathbb{Z}_2$ reduced TO. The SymTFT line $\boldsymbol{Q}_1^E$ has two ends on $\mathfrak{B}_{\mathrm{sym}}$ and a single end on the interface $\mathcal{I}_{(H,N,\omega)}$. After compactifying the interval occupied by the $S_3$ SymTFT, we obtain a topological boundary condition $\mathfrak{B}'$ with three topological local operators

$$1, \quad \mathcal{O}^1, \quad \mathcal{O}^2, \tag{5.59}$$

and consequently the 2-category of defects on $\mathfrak{B}'$ is a multifusion category that decomposes into a direct sum of 3 indecomposable fusion 2-categories. The operators $\mathcal{O}^j$ with $j = 1, 2$ come from compactifying the $\boldsymbol{Q}_1^E$ line extended between $\mathfrak{B}_{\mathrm{sym}}$ and the interface with the two possible choices of ends on $\mathfrak{B}_{\mathrm{sym}}$. Specifically, the local operator $\mathcal{O}^j$ comes

from fixing the end of $\boldsymbol{Q}_1^E$ line to be $\mathcal{E}_0^{E,j}$ and $\mathcal{E}_0^E$ on the symmetry boundary and the interface respectively:

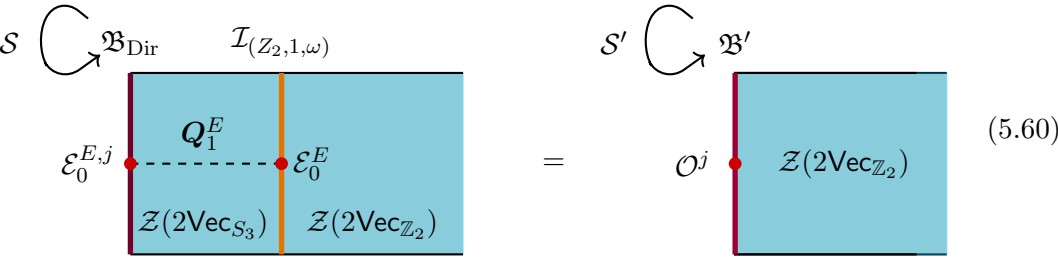

$$(5.60)$$

The ends $\mathcal{E}_0^{E,j}$ transform as a doublet in the $E$ representation of the $S_3$ symmetry on $\mathfrak{B}_{\text{sym}}$

$$D_2^a : \mathcal{E}_0^{E,j} \longmapsto \omega^j \mathcal{E}_0^{E,j}\,, \quad D_2^b : \mathcal{E}_0^{E,1} \longleftrightarrow \mathcal{E}_0^{E,2}\,. \tag{5.61}$$

Requiring consistency between the fusion of $\boldsymbol{Q}_1^E$ and its ends on $\mathfrak{B}_{\text{sym}}$ and the interface imposes the operators to satisfy the following $\mathbb{Z}_3$ fusion rules

$$\mathcal{O}^1 \times \mathcal{O}^2 = \mathcal{O}^2 \times \mathcal{O}^1 = 1\,, \quad \left(\mathcal{O}^1\right)^2 = \mathcal{O}^2\,, \quad \left(\mathcal{O}^2\right)^2 = \mathcal{O}^1\,. \tag{5.62}$$

Therefore the components of multifusion category $\mathcal{S}'$ are labeled by idempotents or projectors onto these components, which are obtained via a $\mathbb{Z}_3$ Fourier transform to be

$$\Pi_j = \frac{1 + \omega^j \mathcal{O}^1 + \omega^{2j} \mathcal{O}^2}{3}\,. \tag{5.63}$$

After performing the club quiche, i.e, compactifying the interval occupied by the $S_3$ SymTFT, one obtains the boundary condition

$$\mathfrak{B}' = (\mathfrak{B}_e)_0 \oplus (\mathfrak{B}_e)_1 \oplus (\mathfrak{B}_e)_2\,, \tag{5.64}$$

where $\mathfrak{B}_e$ is the $e$-condensed or Dirichlet boundary condition of the $\mathbb{Z}_2$ SymTFT. The category of defects is multifusion category, that may be viewed as a 3-category whose objects are connected components. The 2-category of endomorphisms of the $i^{\text{th}}$ component form the symmetry on $(\mathfrak{B}_e)_i$ which is $2\mathsf{Vec}_{\mathbb{Z}_2}$. The 2-category of morphisms between the $i^{\text{th}}$ and $j^{\text{th}}$ components is the regular $2\mathsf{Vec}_{\mathbb{Z}_2}$ module that is $2\mathsf{Vec}_{\mathbb{Z}_2}$. Therefore we obtain

$$\mathcal{S}' = \text{Mat}_3(2\mathsf{Vec}_{\mathbb{Z}_2})\,. \tag{5.65}$$

Let us now describe the functor

$$\mathcal{F} : 2\mathsf{Vec}(S_3) \longrightarrow \text{Mat}_3(2\mathsf{Vec}_{\mathbb{Z}_2})\,. \tag{5.66}$$

The $S_3$ action on the components of $\mathfrak{B}'$ is straightforwardly deduced from the linking action

$$\mathcal{F}(D_2^a) = 1_{01} \oplus 1_{12} \oplus 1_{20} \,,$$
$$\mathcal{F}(D_2^b) = (D_2^m)_{00} \oplus (D_2^m)_{12} \oplus (D_2^m)_{21} \,, \tag{5.67}$$

where $1_{ij}$ is the invertible surface that has the the following linking action on the idempotents

$$1_{ij}\Pi_k = \delta_{jk}\Pi_i \,. \tag{5.68}$$

The fusion of surfaces satisfies

$$1_{ij}1_{kl} = \delta_{jk}1_{il} \,, \qquad (D_2^m)_{ij} = 1_{ij} \otimes (D_2^m)_{jj} = (D_2^m)_{ii} \otimes 1_{ij} \,. \tag{5.69}$$

From these it can be verified that $\mathcal{F}(D_2^a)$ and $\mathcal{F}(D_2^b)$ provide an $S_3$ representation on the surfaces in $\mathcal{S}'$. This choice of interface will correspond to gapless SSB phases upon completing the club quiche to a club sandwich.

**Non-minimal generalization.** Now we consider the non-minimal generalization of the $\boldsymbol{Q}_1^E$ condensed interface. This interface can be defined via a $\mathbb{Z}_2$ graded algebra $\mathcal{A} = \mathcal{A}^{\mathrm{id}} \oplus \mathcal{A}^p$. The category of lines on the interface is

$$\mathcal{Z}(\mathcal{A}) = \widetilde{\mathcal{M}}^{\mathrm{id}} \oplus \widetilde{\mathcal{M}}^p \,, \tag{5.70}$$

where $\widetilde{\mathcal{M}}^{\mathrm{id}}$ is a braided fusion category with center $\mathsf{Rep}(Z_2)$ generated by the interface projection of the line $\boldsymbol{Q}_1^P$. While $\widetilde{\mathcal{M}}^p$ are the lines attached to $\boldsymbol{Q}_2^{[b]}$ and $\boldsymbol{Q}_2^m$ in $\mathcal{Z}(2\mathsf{Vec}_{S_3})$ and $\mathcal{Z}(2\mathsf{Vec}_{\mathbb{Z}_2})$ respectively. Upon compactifying the interval occupied by $\mathcal{Z}(2\mathsf{Vec}_{S_3})$, we again get a three component multifusion category. The symmetry category on this boundary is

$$\mathcal{S}' = \mathrm{Mat}_3\left(2\mathsf{Vec}_{\mathbb{Z}_2} \boxtimes \Sigma[\mathcal{Z}(\mathcal{A}^{\mathrm{id}})]\right) \,. \tag{5.71}$$

The functor is very similar to the one described in (5.67) except with the replacement

$$(D_2^m)_{ij} \longrightarrow \left(D_2^{\phi m}\right)_{ij} \equiv \left(D_2^m \otimes D_2^\phi\right)_{ii} \otimes 1_{ij} \equiv 1_{ij} \otimes \left(D_2^m \otimes D_2^\phi\right)_{jj} \,, \tag{5.72}$$

where $\left(D_2^\phi\right)_{ii}$ is a condensation defect that implements the $\mathbb{Z}_2$ symmetry on $\mathcal{Z}(A^{\mathrm{id}})$ in the $i^{\mathrm{th}}$ component.

- $\boldsymbol{Q}_1^P$ **Condensed interface:** This is an interface to the $\mathbb{Z}_3$ reduced TO. Since $\boldsymbol{Q}_1^P$ can end on both the interface and the symmetry boundary, one obtains an addition local

operator (denoted $\mathcal{O}$) after compactifying the interval occupied by the $S_3$ SymTFT.

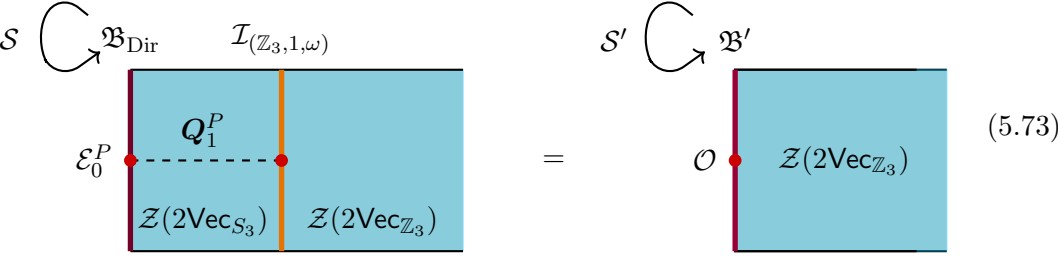

$$(5.73)$$

The topological local operator has the OPE $\mathcal{O} \times \mathcal{O} = 1$. Therefore $\mathcal{S}'$ is a multifusion 2-category with two components, such that the projectors onto these components are

$$\Pi_0 = \frac{1 + \mathcal{O}}{2}, \qquad \Pi_1 = \frac{1 - \mathcal{O}}{2}. \tag{5.74}$$

Next, $\boldsymbol{Q}_1^E$ has two local (untwisted) ends on symmetry boundary $\mathfrak{B}_{\text{sym}}$. These become the ends for $\boldsymbol{Q}_1^e$ and $\boldsymbol{Q}_1^{e^2}$ after we compactify the interval occupied by the $S_3$ SymTFT. Therefore the boundary condition obtained after compactifying the SymTFT is

$$\mathfrak{B}' = (\mathfrak{B}_e)_0 \oplus (\mathfrak{B}_e)_1. \tag{5.75}$$

The 2-category of topological defects on this boundary is

$$\mathcal{S}' = \text{Mat}_2(2\text{Vec}(\mathbb{Z}_3)). \tag{5.76}$$

The club quiche defines a 2-functor that prescribes the $\mathcal{S}$ symmetry realization on $\mathfrak{B}'$

$$\mathcal{F} : 2\text{Vec}(S_3) \longrightarrow \text{Mat}_2(2\text{Vec}(\mathbb{Z}_3)), \tag{5.77}$$

where

$$\begin{aligned}
\mathcal{F}(D_2^a) &= (D_2^m)_{00} \oplus \left(D_2^{m^2}\right)_{11}, \\
\mathcal{F}(D_2^b) &= 1_{01} \oplus 1_{10}.
\end{aligned} \tag{5.78}$$

Upon fixing a gapless physical boundary, this choice of interface would correspond to a gapless SSB phase with two dynamically disconnected sectors/universes.

**Non-minimal generalization.** Now we consider the non-minimal generalization of the $\boldsymbol{Q}_1^P$ condensed interface. This interface can be defined via a $\mathbb{Z}_3$ graded algebra $\mathcal{A}$. The category of lines on the interface is

$$\mathcal{Z}(\mathcal{A}) = \widetilde{\mathcal{M}}^{\text{id}} \oplus \widetilde{\mathcal{M}}^a \oplus \widetilde{\mathcal{M}}^{a^2}, \tag{5.79}$$

where $\widetilde{\mathcal{M}}^{\mathrm{id}}$ is a braided fusion category with center $\mathsf{Rep}(Z_3)$ generated by the interface projection of the line $\boldsymbol{Q}_1^E$. While $\widetilde{\mathcal{M}}^a$ and $\widetilde{\mathcal{M}}^{a^2}$ are the lines attached to $\boldsymbol{Q}_2^{[a]}$ and $\boldsymbol{Q}_2^m$ and $\boldsymbol{Q}_2^{m^2}$ in $\mathcal{Z}(2\mathsf{Vec}_{S_3})$ and $\mathcal{Z}(2\mathsf{Vec}_{\mathbb{Z}_2})$ respectively. Upon compactifying the interval occupied by $\mathcal{Z}(2\mathsf{Vec}_{S_3})$, we again get a two component multifusion category. The symmetry category on this boundary is

$$\mathcal{S}' = \mathrm{Mat}_2\left(2\mathsf{Vec}_{\mathbb{Z}_3} \boxtimes \Sigma[\mathcal{Z}(\mathcal{A}^{\mathrm{id}})]\right). \tag{5.80}$$

The functor is very similar to the one described in (5.78) except with the replacement

$$\left(D_2^{m^j}\right)_{ii} \longrightarrow \left(D_2^{\phi^j m^j}\right)_{ii}, \tag{5.81}$$

where $\left(D_2^\phi\right)_{ii}$ is a condensation defect that implements the $\mathbb{Z}_3$ symmetry on $\mathcal{Z}(A^{\mathrm{id}})$ in the $i^{\mathrm{th}}$ component.

## 5.5.2 $2\mathsf{Vec}(\mathbb{Z}_3^{(1)} \rtimes \mathbb{Z}_2^{(0)})$ Symmetric Club Quiches

Now we describe the club quiches with the symmetry boundary chosen as

$$\mathfrak{B}_{\mathrm{sym}} = \mathfrak{B}_{\mathrm{Neu}(\mathbb{Z}_3),p}. \tag{5.82}$$

These can be constructed by starting with the club quiches obtained with $2\mathsf{Vec}(S_3)$ symmetry and gauging the $\mathbb{Z}_3$ sub-symmetry on the symmetry boundary.

- $\boldsymbol{Q}_2^{[a]}$ **Condensed Interface:** Let us consider the interface labeled by

$$H = S_3, \quad N = \mathbb{Z}_3, \quad \omega = (p', t') \in \mathbb{Z}_3 \times \mathbb{Z}_2 = H^3(S_3, U(1)). \tag{5.83}$$

The $\boldsymbol{Q}_2^{[a]}$ surface can end on both the symmetry boundary and the interface. This surface has two untwisted ends on the symmetry boundary and one on the interface therefore after compactifying the interval occupied by the $S_3$ SymTFT with the $\boldsymbol{Q}_2^{[a]}$ stretched between the interface and $\mathfrak{B}_{\mathrm{sym}}$, we obtain condensed lines $\mathcal{L}^a$ and $\mathcal{L}^{a^2}$

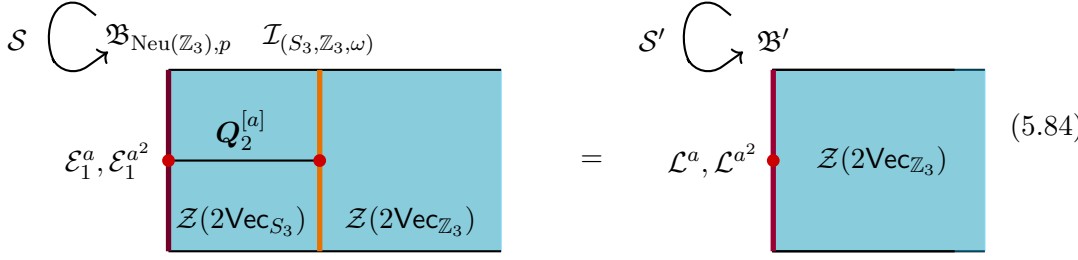

$$\tag{5.84}$$

These lines are charged under the $\mathbb{Z}_3^{(1)}$ symmetry on $\mathfrak{B}_{\text{sym}}$ generated by $D_1^\omega$ and $D_1^{\omega^2}$, which are obtained from the boundary projections of $\boldsymbol{Q}_1^E$. Furthermore, the $F$-symbols of these lines is given by

$$p - p' \in H^3(\mathbb{Z}_3, U(1)) \,. \tag{5.85}$$

The $\boldsymbol{Q}_1^P$ line can end on $\mathfrak{B}_{\text{sym}}$. Since $\boldsymbol{Q}_1^P$ is attached to the $\boldsymbol{Q}_1^e$ line across the interface, this boundary condition becomes a non-minimal Dirichlet boundary condition $\mathfrak{B}_e$ of $\mathcal{Z}(2\text{Vec}_{\mathbb{Z}_2})$ after compactifying the club quiche. $\boldsymbol{Q}_2^m \in \mathcal{Z}(2\text{Vec}_{\mathbb{Z}_2})$ becomes the $\boldsymbol{Q}_2^{[b]} \in \mathcal{Z}(2\text{Vec}_{S_3})$ across the interface which further becomes a line $\mathcal{E}_1^{[b]}$ in the twisted sector of the $D_2^b$ (upto condensation). This $D_2^b$ symmetry defect acts as the $\mathbb{Z}_2$ symmetry on the $\mathfrak{B}_e$ with the additional property that it also implements a charge conjugation $\mathbb{Z}_2$ symmetry.

In summary, the category of defects on $\mathfrak{B}'$ is

$$\mathfrak{B}' = \frac{\mathfrak{B}_e/\mathbb{Z}_2^{(0)} \boxtimes \mathfrak{T}_{\text{DW}(\mathbb{Z}_3),\text{p}-\text{p}'}/\mathbb{Z}_2^{(0)}}{\mathbb{Z}_2^{(1),\text{diag}}} \tag{5.86}$$

where $\mathfrak{T}_{\text{DW}(\mathbb{Z}_3),\text{p}-\text{p}'}$ is the $\mathbb{Z}_3$ Dijkgraaf-Witten theory whose topological action is given by $[p - p'] \in H^3(\mathbb{Z}_3, U(1))$. The multifusion 2-category of defects on this boundary has the following structure. There is a single topological local operator, i.e., it is fusion. Upto condensations, there is a single non-trivial surface, which we denote as $D_2^m$. The 1-endomorphisms of the identity object form a modular tensor category describing the lines of the $\mathbb{Z}_3$ Dijkgraaf-Witten theory. We denote this MTC as $\text{DW}(\mathbb{Z}_3)_{p-p'}$ and its Karoubi completion [76, 82] into a 2-category as

$$\Sigma\left(\text{DW}(\mathbb{Z}_3)_{p-p'}\right) \,. \tag{5.87}$$

More precisely, the category of defects on $\mathfrak{B}'$ form the 2-category

$$\mathcal{S}' = 2\text{Vec}_{\mathbb{Z}_2} \boxtimes \Sigma\left(\text{DW}(\mathbb{Z}_3)_{p-p'}\right) \,. \tag{5.88}$$

The club quiche furnishes the functor

$$\mathcal{F} : 2\text{Vec}(\mathbb{Z}_3^{(1)} \rtimes \mathbb{Z}_2^{(0)}) \longrightarrow 2\text{Vec}_{\mathbb{Z}_2} \boxtimes \Sigma\left(\text{DW}(\mathbb{Z}_3)_{p-p'}\right) \,. \tag{5.89}$$

Under this functor, $D_2^b$ maps to the $\mathbb{Z}_2$ generator in $2\text{Vec}_{\mathbb{Z}_2}$ stacked with the charge conjugation condensation defect [26] in the Dijkgraaf-Witten theory. While the $\mathbb{Z}_3^{(1)}$ generators are mapped to the to the topological Wilson lines in the Dijkgraaf-Witten theory. Finally we note that the lines in the twisted sector of the 0-form $\mathbb{Z}_2$ symmetry

generators on this boundary have associators that are determined by the datum $t \in H^3(\mathbb{Z}_2, U(1))$ on the interface. Upon choosing a gapless physical boundary, this choice of interface will correspond to a $\mathbb{Z}_3^{(1)} \rtimes \mathbb{Z}_2^{(0)}$ symmetric gapless phase where the $\mathbb{Z}_3^{(1)}$ symmetry is spontaneously broken, i.e., a gapless 1-form SSB.

**Non-minimal generalization.** The non-minimal generalization of this gapless 1-form SSB phase can be described in terms of an $S_3$ symmetric MTC $\mathcal{Z}(\mathcal{A}^{\mathrm{id}})$ that was obtained from the non-minimal $\boldsymbol{Q}_2^{[a]}$ condensed interface and $\mathfrak{B}_{\mathrm{sym}} = \mathfrak{B}_{\mathrm{Dir}}$. Upon gauging the $\mathbb{Z}_3$ 0-form subsymmetry on $\mathfrak{B}_{\mathrm{Dir}}$ one obtains a non-minimal $\mathbb{Z}_3^{(1)}$ SSB phase, denoted as $\mathcal{Z}(\mathcal{A}^{\mathrm{id}} \oplus \mathcal{A}^a \oplus \mathcal{A}^{a^2})$. This 3d TFT theory has a $\mathbb{Z}_2^{(0)}$ left behind after the $\mathbb{Z}_3$ gauging. We denote the condensation defect implementing this $\mathbb{Z}_2^{(0)}$ symmetry as $D_2^\phi$. Then the non-minimal symmetry category is

$$\mathcal{S}' = 2\mathsf{Vec}_{\mathbb{Z}_2} \boxtimes \Sigma \left[ \mathcal{Z}(\mathcal{A}^{\mathrm{id}} \oplus \mathcal{A}^a \oplus \mathcal{A}^{a^2}) \right] . \tag{5.90}$$

There is a 2-functor $\mathcal{F} : \mathcal{S} \to \mathcal{S}'$ which maps the $\mathbb{Z}_3^{(1)}$ generators to the (broken) $\mathbb{Z}_3^{(1)}$ generators in $\mathcal{Z}(\mathcal{A}^{\mathrm{id}} \oplus \mathcal{A}^a \oplus \mathcal{A}^{a^2})$ and the $\mathbb{Z}_2^{(0)}$ generator to $D_2^m \otimes D_2^\phi$.

- $\boldsymbol{Q}_1^E$ **interface:** With this choice of interface, the $\boldsymbol{Q}_1^E$ line can end on both the interface and the symmetry boundary. This line has two ends on the symmetry boundary. After compactifying the interval occupied by the $S_3$ SymTFT, one obtains local operators in the $\mathbb{Z}_3^{(1)}$ twisted sectors.

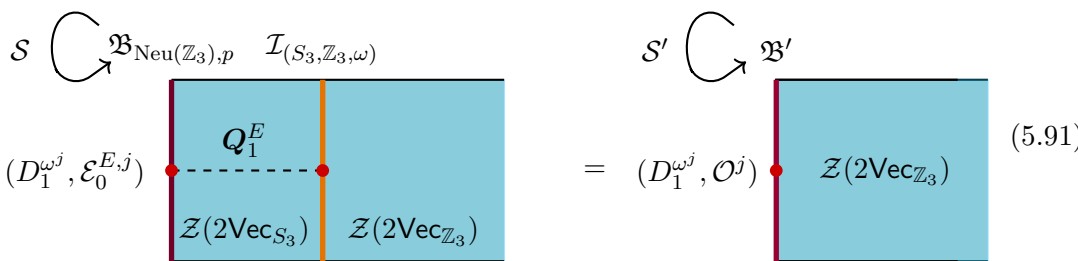

$$\tag{5.91}$$

where by $(D_1^{\omega^j}, \mathcal{E}_0^{E,j})$, we denote a topological local operator $\mathcal{E}_0^{E,j}$ attached to a topological line $D_1^{\omega^j}$. Therefore the $\mathbb{Z}_3^{(1)}$ is represented trivially after the club quiche. The $\boldsymbol{Q}_1^P$ line can end on the symmetry boundary. This line continues across the interface as the $\boldsymbol{Q}_1^e$ line in $\mathcal{Z}(2\mathsf{Vec}_{\mathbb{Z}_2})$. Therefore upon taking the club quiche, one obtains the boundary condition

$$\mathfrak{B}' = \mathfrak{B}_e , \tag{5.92}$$

for $\mathcal{Z}(2\mathsf{Vec}_{\mathbb{Z}_2})$. The category of defects on this boundary simply realize a $\mathbb{Z}_2$ 0-form symmetry generated by $D_2^m$. This club quiche furnishes a functor

$$\mathcal{F} : 2\mathsf{Vec}(\mathbb{Z}_3^{(1)} \rtimes \mathbb{Z}_2^{(0)}) \longrightarrow 2\mathsf{Vec}(\mathbb{Z}_2) \,, \tag{5.93}$$

under which

$$\mathcal{F}(D_2^b) = D_2^m \,, \quad \mathcal{F}(D_1^{\omega^j}) \cong D_1^{\mathrm{id}} \,. \tag{5.94}$$

**Non-minimal generalization.** Since the $\mathbb{Z}_3^{(1)}$ is trivial in this gapless phase, the minimal extension is rather straightforward. It simply involves stacking with an $\mathbb{Z}_2$ symmetric MTC and implementing the $\mathbb{Z}_2$ symmetry via a digonal defect $D_2^m \otimes D_2^\phi$ where $D_2^\phi$ implements the $\mathbb{Z}_2^{(0)}$ symmetry in the MTC.

- $\boldsymbol{Q}_1^P$ **Condensed Interface:** For this choice, $\boldsymbol{Q}_1^P$ can end on both the symmetry boundary and the interface. After compactifying the interval occupied by $\boldsymbol{Q}_1^P$, one obtains a local operator $\mathcal{O}$ that is charged under the $\mathbb{Z}_2$ 0-form subsymmetry on $\mathfrak{B}_{\mathrm{sym}}$ as in (5.73). Furthermore, $\boldsymbol{Q}_2^{[a]}$ has two untwisted ends $\mathcal{E}_1^a$ and $\mathcal{E}_1^{a^2}$ on the symmetry boundary. The $\boldsymbol{Q}_2^{[a]}$ surface is connected to $D_2^m \oplus D_2^{m^2}$ across the interface. Therefore after compactifying the interval region occupied by $\mathcal{Z}(2\mathsf{Vec}(S_3))$, we obtain the boundary condition

$$\mathfrak{B}' = (\mathfrak{B}_m)_0 \ \oplus \ (\mathfrak{B}_{m^2})_1 \,. \tag{5.95}$$

The 2-category of topological defects on this boundary form

$$\mathcal{S}' = \mathrm{Mat}_2 \left( 2\mathsf{Rep}(\mathbb{Z}_3) \right) \,. \tag{5.96}$$

The two components in this multifusion category correspond to the projectors

$$\Pi_0 = \frac{1 + \mathcal{O}}{2} \,, \quad \Pi_1 = \frac{1 - \mathcal{O}}{2} \,. \tag{5.97}$$

The 2-group symmetry is represented on this boundary via the functor

$$\mathcal{F} : 2\mathsf{Vec}(\mathbb{Z}_3^{(1)} \rtimes \mathbb{Z}_2^{(0)}) \longrightarrow \mathrm{Mat}_2 \left( 2\mathsf{Rep}(\mathbb{Z}_3) \right) \,. \tag{5.98}$$

such that

$$\mathcal{F}(D_2^b) = 1_{01} \oplus 1_{10} \,, \quad \mathcal{F}(D_1^\omega) = (D_1^e)_{00} \oplus (D_1^{e^2})_{11} \,. \tag{5.99}$$

Upon choosing a gapless physical boundary, this will correspond to a $\mathbb{Z}_3^{(1)} \rtimes \mathbb{Z}_2^{(0)}$ gapless SSB phase where $\mathbb{Z}_2^{(0)}$ is spontaneously broken and therefore there are two gapless vacua related by $\mathbb{Z}_2^{(0)}$.

**Non-minimal generalization.** In this case, one obtains a multifusion category, such that category of defects within each component is given by the Karoubi completion of a braided fusion category $\widetilde{\mathcal{M}}^{\text{id}}$ descibed in (5.79) into a 2-category

$$\Sigma_2(\widetilde{\mathcal{M}}^{\text{id}})\,. \tag{5.100}$$

The full symmetry on the non-minimal boundary obtained after compactifying the interval occupied by $\mathcal{Z}(2\text{Vec}_{S_3})$ is

$$\mathcal{S}' = \text{Mat}_2\left(\Sigma_2(\widetilde{\mathcal{M}}^{\text{id}})\right)\,. \tag{5.101}$$

Then the functor $\mathcal{F}: \mathcal{S} \to \mathcal{S}'$ is again given by (5.99) where $(D_1^e)_{ii}$ denotes the generator of the Müger center of $\widetilde{\mathcal{M}}^{\text{id}})$ in the $i^{\text{th}}$ component.

### 5.5.3 $2\text{Rep}(\mathbb{Z}_3^{(1)} \rtimes \mathbb{Z}_2^{(0)})$ Symmetric Club Quiches

We now move on to the club quiches obtained by considering the symmetry boundary to be

$$\mathfrak{B}_{\text{sym}} = \mathfrak{B}_{\text{Neu}(\mathbb{Z}_2),t}\,. \tag{5.102}$$

- $Q_2^{[a]}$ **Condensed Interface:** $D_2^A$ is trivialized as we get a condensed twisted sector order parameter after compactifying the interval occupied by $\mathcal{Z}(2\text{Vec}_{S_3})$. The 2-category of defects on $\mathfrak{B}'$ has a single component as none of SymTFT lines can end on both the interface and the symmetry boundary. The surface $Q_2^m$ in the reduced TO $\mathcal{Z}(2\text{Vec}_{\mathbb{Z}_2})$ continues across the interface as $Q_2^{[b]}$ which has a single untwisted end on $\mathfrak{B}_{\text{sym}}$. Therefore the boundary $\mathfrak{B}'$ is the $Q_2^m$ condensed (or $\text{Neu}(\mathbb{Z}_2)$) boundary of the reduced TO with discrete torsion $t \in H^3(\mathbb{Z}_2, U(1))$

$$\mathfrak{B}' = \mathfrak{B}_{m,t}\,. \tag{5.103}$$

The symmetry $\mathcal{S}'$ on this boundary is a $\mathbb{Z}_2$ 1-form symmetry which is corresponds to

$$\mathcal{S}' = 2\text{Rep}(\mathbb{Z}_2^{(0)})\,. \tag{5.104}$$

The club quiche furnishes a monoidal 2-functor

$$\mathcal{F}: 2\text{Rep}(\mathbb{Z}_3^{(1)} \rtimes \mathbb{Z}_2^{(0)}) \longrightarrow 2\text{Rep}(\mathbb{Z}_2^{(0)})\,. \tag{5.105}$$

Such that

$$\mathcal{F}(D_2^A) \cong 2D_2^{\text{id}}\,, \quad \mathcal{F}(D_1^{\widehat{b}}) = D_1^e\,. \tag{5.106}$$

**Non-minimal generalization.** To obtain the non-minimal generalization, we start with the $S_3$ symmetric TO $\mathcal{Z}(\mathcal{A}^{\text{id}})$ obtained from taking the club quiche with $\mathfrak{B}_{\text{sym}} = \mathfrak{B}_{\text{Dir}}$. Then we gauge the $\mathbb{Z}_2^{(0)}$ symmetry on the symmetry boundary with the choice of discrete torsion $t \in H^3(\mathbb{Z}_2, U(1))$. This produces a non-minimal boundary condition of $\mathcal{Z}(2\mathsf{Vec}_{\mathbb{Z}_2})$

$$\mathfrak{B}' = \frac{\mathfrak{B}_e \boxtimes \mathcal{Z}(\mathcal{A}^{\text{id}})}{(\mathbb{Z}_2^{(0)}, t)} \, . \tag{5.107}$$

The category of defects on this boundary is

$$\mathcal{S}' = \Sigma \left( \mathcal{Z}(\mathcal{A}^{\text{id}})/\mathbb{Z}_2^{(0)} \right) = \Sigma \left( \widetilde{\mathcal{M}}^0 \right) \, . \tag{5.108}$$

The Müger center of the braided fusion category $\widetilde{\mathcal{M}}^0$ is $\mathsf{Rep}(\mathbb{Z}_2)$ generated by the boundary projection of $\boldsymbol{Q}_1^e$. We denote the corresponding defect in $\Sigma(\widetilde{\mathcal{M}}^0)$ as $D_1^e$. Also recall that there was a $\mathbb{Z}_3$ symmetry implemented on $\mathcal{Z}(\mathcal{A}^{\text{id}}$[23]. Let us denote the corresponding $\mathbb{Z}_3$ condensation defects as $D_2^{\phi_a}$ and $D_2^{\phi_{a^2}}$. These are exchanged by the $\mathbb{Z}_2$ action and therefore combine into a single non-invertible symmetry defect

$$D_2^{\phi_A} = \left( D_2^{\phi_a} \oplus D_2^{\phi_{a^2}} \right)/\mathbb{Z}_2 \, . \tag{5.109}$$

Now we are ready to describe the functor

$$\mathcal{F} : 2\mathsf{Rep}(\mathbb{Z}_3^{(1)} \rtimes \mathbb{Z}_2^{(0)}) \longrightarrow \Sigma(\widetilde{\mathcal{M}}^0) \, . \tag{5.110}$$

Under this functor

$$\mathcal{F}(D_2^A) = D_2^{\phi_A} \, , \qquad \mathcal{F}(D_1^{\widehat{b}}) = D_1^e \, . \tag{5.111}$$

Finally, note that the bulk surface $\boldsymbol{Q}_2^m \in \mathcal{Z}(2\mathsf{Vec}_{\mathbb{Z}_2})$ ends on $\mathfrak{B}'$ on the category of lines $\widetilde{\mathcal{M}}^1$ whose associators have been modified by $t \in H^3(\mathbb{Z}_2, U(1))$.

- $\boldsymbol{Q}_1^E$ **Condensed Interface:** The $\boldsymbol{Q}_1^E$ line can end (and has a single end) on both the interface and the symmetry boundary. Therefore are compactifying the interval occupied by $\mathrm{DW}(S_3)$, we obtain an additional local operator $\mathcal{O}$ on $\mathfrak{B}'$. The fusion rules of $\mathcal{O}$ are (see Sec 3.5.3 in [2])

$$\mathcal{O} \times \mathcal{O} = 2 + \mathcal{O} \, . \tag{5.112}$$

Since $\mathfrak{B}'$ has two local operators, the category of defects on it forms a multifusion 2-category with two components. The projectors onto these components are the projectors

$$\Pi_0 = \frac{2 - \mathcal{O}}{3} \, , \qquad \Pi_1 = \frac{1 + \mathcal{O}}{3} \, . \tag{5.113}$$

---

[23] Defined either via a choice of $\mathbb{Z}_3$ restriction of $\mathcal{A}$ or the $\mathbb{Z}_3$ restriction of $\widetilde{\mathcal{M}}_{S_3}^\times$.

We can describe the boundary condition $\mathfrak{B}'$ by directly gauging the $\mathbb{Z}_2^{(0)}$ symmetry on (5.64). Since $\mathbb{Z}_2^{(0)}$ acts within $(\mathfrak{B}_e)_0$ while it exchanges $(\mathfrak{B}_e)_1$ and $(\mathfrak{B}_e)_2$, we obtain

$$\mathfrak{B}' = \frac{(\mathfrak{B}_e)_0}{\mathbb{Z}_2^{(0)}} \; \boxplus \; \left( \frac{(\mathfrak{B}_e)_1 \; \boxplus \; (\mathfrak{B}_e)_2}{\mathbb{Z}_2^{(0)}} \right) , \tag{5.114}$$

The symmetry category on this boundary is a multifusion 2-category with two components. The component labeled 0 (descending from $(\mathfrak{B}_e)_0$ upon gauging) carries the $2\mathsf{Rep}(\mathbb{Z}_2)$, while the component labeled 1 (descending from $(\mathfrak{B}_e)_1 \boxplus (\mathfrak{B}_e)_2$ upon gauging) carries $2\mathsf{Vec}_{\mathbb{Z}_2}$. Additionally there are morphisms between these two components given by $2\mathsf{Vec}$. The interface defects between distinct components $i$ and $j$ are denoted as $B_{ij}$. In summary, the symmetry category can be described by the matrix of 2-categories

$$\mathcal{S}' = \begin{pmatrix} 2\mathsf{Rep}(\mathbb{Z}_2) & 2\mathsf{Vec} \\ 2\mathsf{Vec} & 2\mathsf{Vec}_{\mathbb{Z}_2} \end{pmatrix} . \tag{5.115}$$

The club quiche determines a functor $\mathcal{F} : 2\mathsf{Rep}(\mathbb{Z}_3^{(1)} \rtimes \mathbb{Z}_2^{(0)}) \to \mathcal{S}'$, such that

$$\mathcal{F}(D_2^A) = B_{01} \; \oplus \; B_{10} \; \oplus \; 1_{11} , \qquad \mathcal{F}(D_1^{\widehat{b}}) = (D_1^e)_{00} \; \oplus \; D_1^{\mathrm{id}} . \tag{5.116}$$

- $\boldsymbol{Q}_1^P$ **Condensed Interface:** After compactifying the interval occupied by $\mathcal{Z}(2\mathsf{Vec}_{S_3})$, one obtains a $D_1^{\widehat{b}}$ twisted order parameter, therefore the $\mathbb{Z}_2^{(1)}$ sub-symmetry is trivialized in this gapless phase. There are no topological local operators on $\mathfrak{B}'$ other than the identity. The surface $\boldsymbol{Q}_2^{[a]}$ that goes to the $D_2^A$ twisted sector on $\mathfrak{B}_{\mathrm{sym}}$ becomes $\boldsymbol{Q}_2^m \oplus \boldsymbol{Q}_2^{m^2}$ across the interface. Therefore after collapsing the interface on $\mathfrak{B}_{\mathrm{sym}}$, one obtains the Dirichlet boundary condition for $\mathcal{Z}(2\mathsf{Vec}_{\mathbb{Z}_3})$, i.e., $\mathfrak{B}' = \mathfrak{B}_m$. This boundary carries a $\mathbb{Z}_3$ 0-form symmetry.

$$\mathcal{F} : 2\mathsf{Rep}(\mathbb{Z}_3^{(1)} \rtimes \mathbb{Z}_2^{(0)}) \longrightarrow 2\mathsf{Vec}_{\mathbb{Z}_3} , \tag{5.117}$$

such that

$$\mathcal{F}(D_2^A) = D_2^m \oplus D_2^{m^2} , \qquad \mathcal{F}(D_1^{\widehat{b}}) \cong D_1^{\mathrm{id}} . \tag{5.118}$$

**Non-minimal generalization.** The category of lines on the non-minimal generalization of the $\boldsymbol{Q}_1^P$ condensed interface forms the modular tensor category

$$\mathcal{Z}(\mathcal{A}) = \widetilde{\mathcal{M}} = \widetilde{\mathcal{M}}^1 \; \oplus \; \widetilde{\mathcal{M}}^a \; \oplus \; \widetilde{\mathcal{M}}^{a^2} , \tag{5.119}$$

where $\mathcal{A}$ is a $\mathbb{Z}_3$ graded fusion category. The boundary condition obtained after compactifying the interval occupied by $\mathcal{Z}(2\mathsf{Vec}_{S_3})$ is

$$\mathfrak{B}' = \frac{\mathfrak{B}_e/\mathbb{Z}_3^{(0)} \; \boxtimes \; \mathfrak{T}_{\mathbb{Z}_3}/\mathbb{Z}_3^{(0)}}{\mathbb{Z}_3^{(1)}} , \tag{5.120}$$

where $\mathfrak{T}_{\mathbb{Z}_3}$ is the 3d TFT whose underlying MTC is $\mathcal{Z}(\mathcal{A}^{\mathrm{id}})$ and $\mathbb{Z}_3$ symmetry action is defined via $\mathcal{A}$. The 2-category of defects on $\mathfrak{B}'$ is given by the Karoubi completion of $\widetilde{\mathcal{M}}^{[\mathrm{id}]}$ into a 2-category.

$$\mathcal{S}' = \Sigma(\widetilde{\mathcal{M}}^{\mathrm{id}}). \tag{5.121}$$

Let us denote the condensation defects implementing the $\mathbb{Z}_3$ 0-form on $\mathfrak{T}_{\mathbb{Z}_3}$ as $D_2^{\phi_a}$ and $D_2^{\phi_{a^2}}$. Then the club quiche furnishes a functor

$$\mathcal{F} : 2\mathsf{Rep}(\mathbb{Z}_3^{(1)} \rtimes \mathbb{Z}_2^{(0)}) \longrightarrow \Sigma(\widetilde{\mathcal{M}}^{\mathrm{id}}), \tag{5.122}$$

under which

$$\mathcal{F}(D_1^{\widehat{b}}) \cong D_1^{\mathrm{id}}, \qquad \mathcal{F}(D_2^A) \cong D_2^{\phi_a} \oplus D_2^{\phi_{a^2}}. \tag{5.123}$$

### 5.5.4 $2\mathsf{Rep}(S_3)$ Symmetry

We now consider the symmetry boundary to be

$$\mathfrak{B}_{\mathrm{sym}} = \mathfrak{B}_{\mathrm{Neu}(S_3),\omega}, \tag{5.124}$$

which carries a $2\mathsf{Rep}(S_3)$ symmetry on it. We will denote $\omega = (p, t) \in \mathbb{Z}_3 \times \mathbb{Z}_2 \cong H^3(S_3, U(1))$.

- $\boldsymbol{Q}_2^{[a]}$ **Condensed Interface:** Consider the $\boldsymbol{Q}_2^{[a]}$ Condensed Interface labeled by $H = S_3$, $N = \mathbb{Z}_3$ and $(p', t') \in H^3(S_3, U(1))$. The $\boldsymbol{Q}_2^{[a]}$ surface ends on $\mathfrak{B}_{\mathrm{sym}}$ on a untwisted line $\mathcal{E}_1^{[a]}$ which is charged under (braids non-trivially with) $D_1^E$ that generates the non-invertible $\mathsf{Rep}(S_3)$ 1-form symmetry. Therefore, this line is added to the symmetry category $\mathcal{S}'$ on $\mathfrak{B}'$. The reduced TO after taking the club quiche is $\mathcal{Z}(2\mathsf{Vec}_{\mathbb{Z}_2})$. The line $\boldsymbol{Q}_1^e$ in the reduced TO connects to $\boldsymbol{Q}_1^P$ in $\mathcal{Z}(2\mathsf{Vec}_{S_3})$ that ends on a local operator attached to $D_1^P$ on $\mathfrak{B}_{\mathrm{sym}}$. Therefore upon compactifying the interval occupied by $\mathcal{Z}(2\mathsf{Vec}_{S_3})$, $\boldsymbol{Q}_1^e$ has a twisted sector end attached to $D_1^e$ which generates a $\mathbb{Z}_2$ 1-form symmetry. The surface $\boldsymbol{Q}_2^m$ in the reduced TO becomes the $\boldsymbol{Q}_2^{[b]}$ surface in $\mathcal{Z}(2\mathsf{Vec}_{S_3})$ that can end on a single untwisted line $\mathcal{E}_1^{[b]}$ on $\mathfrak{B}_{\mathrm{sym}}$ whose $F$ symbols are determined by $t - t' \in H^3(\mathbb{Z}_2, U(1))$. We identify the boundary condition after taking the club sandwich as

$$\mathfrak{B}' = \frac{\mathrm{DW}(\mathbb{Z}_3)_{p-p'} \boxtimes \mathfrak{B}_e}{(\mathbb{Z}_2^{(0),\mathrm{diag}}, t - t')}. \tag{5.125}$$

Here we have stacked $\mathrm{DW}(\mathbb{Z}_3)_{p-p'}$, the 3d $\mathbb{Z}_3$ Dijkgraaf-Witten theory with topological action $p \in H^3(\mathbb{Z}_3, U(1))$ on the Dirichlet boundary condition and gauged the diagonal $\mathbb{Z}_2^{(0)}$ that acts as charge conjugation on $\mathrm{DW}(\mathbb{Z}_3)_{p-p'}$ and via $D_2^m$ on $\mathfrak{B}_{\mathrm{Dir}}$. The category of all line defects on $\mathfrak{B}'$ form

$$\frac{\mathrm{DW}(\mathbb{Z}_3)_{p-p'}}{(\mathbb{Z}_2^{(0)}, t - t')} \cong \mathrm{DW}(S_3)_{(p-p',t-t')}. \tag{5.126}$$

However some of these are attached to bulk surfaces in $\mathcal{Z}(2\mathsf{Vec}_{\mathbb{Z}_2})$. Specifically the lines that are charged under $\mathsf{D}_1^e$ are attached to bulk surface $\boldsymbol{Q}_2^m$. Therefore the symmetry category on $\mathfrak{B}'$ is obtained by removing the lines that braid non-trivially with $D_1^e$ from $\mathrm{DW}(S_3)_{(p-p',t-t')}$. We can give $\mathrm{DW}(S_3)_{(p-p',t-t')}$ a $\mathbb{Z}_2$ grading

$$\mathrm{DW}(S_3)_{(p-p',t-t')} = \widetilde{\mathcal{N}}^1 \oplus \widetilde{\mathcal{N}}^{\widehat{e}}, \tag{5.127}$$

according to the braiding with the $D_1^e$ line. Then the symmetry category on $\mathfrak{B}'$ is

$$\mathcal{S}' = \Sigma(\widetilde{\mathcal{N}}^1). \tag{5.128}$$

The club quiche furnishes a natural 2-functor

$$\mathcal{F} : 2\mathsf{Rep}(S_3) \longrightarrow \Sigma(\widetilde{\mathcal{N}}^1), \tag{5.129}$$

whose image are the Wilson lines in $\mathrm{DW}(S_3)_{(p-p',t-t')}$ (all of which are in $\widetilde{\mathcal{N}}^1$) and their condensation defects.

- $\boldsymbol{Q}_1^E$ **Condensed Interface:** Consider the $\boldsymbol{Q}_1^E$ Condensed interface labeled by $H = Z_2$, $N = 1$ and $t' \in H^3(\mathbb{Z}_2, U(1))$. Upon compactifying the interval occupied by $\mathcal{Z}(2\mathsf{Vec}_{S_3})$, one obtains an order parameter in the twisted sector of the $D_1^E$ symmetry line. Therefore in this gapless phase $D_1^E \cong D_1^{\mathrm{id}} \oplus D_1^P$. The surface $\boldsymbol{Q}_2^m$ in the reduced TO is attached to the $\boldsymbol{Q}_2^{[b]}$ across the interface that can end on the symmetry boundary. Therefore upon taking the club quiche, one obtains a boundary of the reduced TO $\mathcal{Z}(2\mathsf{Vec}_{\mathbb{Z}_2})$ on which $\boldsymbol{Q}_2^m$ can end on an untwisted line $\mathcal{E}_1^m$. This is the $\boldsymbol{Q}_2^m$ condensed (Neumann $\mathbb{Z}_2$) boundary condition

$$\mathfrak{B}' = \mathfrak{B}_{m,t-t'} \tag{5.130}$$

which carries a symmetry category

$$\mathcal{S}' = 2\mathsf{Rep}(\mathbb{Z}_2). \tag{5.131}$$

The F-symbols of $\mathcal{E}_1^m$ are determined by the $\mathbb{Z}_2$ restriction of $t - t' \in H^3(\mathbb{Z}_2, U(1))$. The club quiche furnishes a functor

$$\mathcal{F} : 2\mathsf{Rep}(S_3) \longrightarrow 2\mathsf{Rep}(\mathbb{Z}_2), \quad \mathcal{F}(D_1^E) \cong D_1^{\mathrm{id}} \oplus D_1^e, \quad \mathcal{F}(D_1^P) \cong D_1^e. \tag{5.132}$$

- $\boldsymbol{Q}_1^P$ **Condensed Interface:** Consider the $\boldsymbol{Q}_1^P$ Condensed interface labeled by $H = Z_3$, $N = 1$ and $p' \in H^3(\mathbb{Z}_2, U(1))$. Upon compactifying the interval occupied by $\mathcal{Z}(2\mathsf{Vec}_{S_3})$, one obtains an order parameter in the twisted sector of the $D_1^P$ symmetry line. Therefore in this gapless phase $D_1^P \cong D_1^{\mathrm{id}}$. The $\boldsymbol{Q}_2^m$ and $\boldsymbol{Q}_2^{m^2}$ surface in the reduced TO

$(\mathcal{Z}(2\mathsf{Vec}_{\mathbb{Z}_3}))$ are attached to the $\boldsymbol{Q}_2^{[a]}$ across the interface that can end on the symmetry boundary. Therefore upon taking the club quiche, one obtains the Neumann boundary of the reduced TO $\mathcal{Z}(2\mathsf{Vec}_{\mathbb{Z}_3})$

$$\mathfrak{B}' = \mathfrak{B}_{m,p-p'} \tag{5.133}$$

which carries a symmetry category

$$\mathcal{S}' = 2\mathsf{Rep}(\mathbb{Z}_3) \,. \tag{5.134}$$

The F-symbols of the ends $\mathcal{E}_1^{m^j}$ of $\boldsymbol{Q}_2^{m^j}$ (for $j = 1, 2$) are given by $p - p' \in H^3(\mathbb{Z}_3, U(1))$. The club quiche furnishes a functor

$$\mathcal{F} : 2\mathsf{Rep}(S_3) \longrightarrow 2\mathsf{Rep}(\mathbb{Z}_3), \quad \mathcal{F}(D_1^E) \cong D_1^\omega \oplus D_1^{\omega^2}, \quad \mathcal{F}(D_1^P) \cong D_1^{\mathrm{id}}\,. \tag{5.135}$$

## 5.6  Gapless Phases

Now we describe the theory obtained upon closing the club quiche by slotting in a physical boundary. This will be referred to as a club sandwich. This construction takes in an $\mathcal{S}'$ symmetric quantum field theory and outputs an $\mathcal{S}$ symmetric quantum field theory. This construction has the merit as it can be used to realize transitions in systems with more complicated symmetry structures, using simpler ones. This is a generalization of the Kennedy-Tasaki transformation that was originally formulated for $\mathbb{Z}_2 \times \mathbb{Z}_2$ symmetric quantum spin chains [83]. It has subsequently been generalized to various contexts including for the study of transitions and gapless phases with fusion categorical symmetries [17, 24, 38, 84, 85]. Here we use it for the study of transitions and gapless phases in theories with fusion 2-categorical symmetries.

### 5.6.1  KTs from $\mathbb{Z}_2^{(0)}$ to $S_3$

We now describe the KT transformations that takes a theory with $\mathbb{Z}_2$ 0-form symmetry and outputs a theory with $S_3$ 0-form. In the SymTFT picture we have

$$\mathfrak{B}_{\mathrm{sym}} = \mathfrak{B}_{\mathrm{Dir}}\,, \tag{5.136}$$

and we consider the quiches of section 5.5.1. There are two classes of interfaces between $\mathcal{Z}(2\mathsf{Vec}_{S_3})$ and $\mathcal{Z}(2\mathsf{Vec}_{\mathbb{Z}_2})$ and correspondingly two classes of KT transformations.

**KT from the $\boldsymbol{Q}_2^{[a]}$ Condensed Interface.**  Let us consider the club quiche corresponding to an $\boldsymbol{Q}_2^{[a]}$ condensed interface, section 5.3.1. This produced a boundary

$$\mathfrak{B}' = \mathfrak{B}_e\,, \tag{5.137}$$

with symmetry $\mathcal{S}' = 2\mathsf{Vec}_{\mathbb{Z}_2}$. We may input a $\mathbb{Z}_2$ 0-form symmetric theory on the physical boundary. The $S_3$ symmetry is realized on this theory via the projection

$$2\mathsf{Vec}_{S_3} \longrightarrow 2\mathsf{Vec}_{\mathbb{Z}_2} . \tag{5.138}$$

In particular, the kernel of this projection is $2\mathsf{Vec}_{\mathbb{Z}_3}$ for which, this theory realizes a $\mathbb{Z}_3$ SPT labeled by $p \in H^3(\mathbb{Z}_3, U(1))$. Recall $p$ was the $\mathbb{Z}_3$ projection of $\omega = (p, t) \in H^3(S_3, U(1))$. The data of $t \in H^3(\mathbb{Z}_2, U(1))$ would modify the F-symbols of the ends of the $\mathbf{Q}_2^m$ surface on $\mathfrak{B}_{\text{phys}}$. To summarize the $S_3$ symmetry is realized as

$$\mathfrak{T}_{\mathbb{Z}_2} \overset{\curvearrowleft}{\frown} \quad 2\mathsf{Vec}_{\mathbb{Z}_2} \longleftarrow 2\mathsf{Vec}_{S_3} . \tag{5.139}$$

Now we choose $\mathfrak{T}_{\mathbb{Z}_2}$ as the Ising CFT that realizes a second-order phase transition between the $\mathbb{Z}_2^{(0)}$ Trivial and the $\mathbb{Z}_2^{(0)}$ SSB Phase. The KT transformation converts this $\mathbb{Z}_2$-symmetric transition into an $S_3$-symmetric transition. The $\mathbb{Z}_2$-symmetric gapped phases on the two sides of the transition correspond the $\mathfrak{B}_{\text{Dir}}$ and $\mathfrak{B}_{\text{Neu}}$ gapped physical boundaries of the reduced $\mathbb{Z}_2$ TO. The KT transformation converts these to the $\mathfrak{B}_{\text{Neu}(\mathbb{Z}_3),p}$ and $\mathfrak{B}_{\text{Neu}(S_3),(p,t)}$ respectively.

$$\boxed{\mathbb{Z}_2 \text{ SSB} \boxtimes \text{SPT}(\mathbb{Z}_3)_p} \quad \longleftarrow \quad \boxed{\text{3d Ising}} \quad \longrightarrow \quad \boxed{\text{SPT}(S_3)_{(p,t)}} \tag{5.140}$$

**KT from the $\mathbf{Q}_1^E$ Condensed Interface.** We now consider the club quiche corresponding to the $\mathbf{Q}_1^E$ condensed interface, section 5.3.2. This produced a boundary

$$\mathfrak{B}' = (\mathfrak{B}_e)_0 \oplus (\mathfrak{B}_e)_1 \oplus (\mathfrak{B}_e)_2 , \tag{5.141}$$

with symmetry $\mathcal{S}' = \mathrm{Mat}_3(2\mathsf{Vec}_{\mathbb{Z}_2})$. We may input a $\mathbb{Z}_2$ 0-form symmetric theory on the physical boundary. The $S_3$ symmetry is realized on this theory via the projection

$$2\mathsf{Vec}_{S_3} \longrightarrow \mathrm{Mat}_3(2\mathsf{Vec}_{\mathbb{Z}_2}) . \tag{5.142}$$

In particular, the Hilbert space of the theory obtained after compactifying the SymTFT decomposes into a direct sum of three blocks or *universes* corresponding to the three components of $\mathfrak{B}'$. The $\mathbb{Z}_3$ 0-form symmetry permutes these components while the $\mathbb{Z}_2^{(0)}$ symmetry acts as (5.67) To summarize the $S_3$ symmetry is realized as

$$\tag{5.143}$$

Finally, the data of $t \in H^3(\mathbb{Z}_2, U(1))$ labeling the interface would modify the F-symbols of $\mathbb{Z}_2^{a^j b}$ twisted sector lines if these lines exist within the $j^{\text{th}}$ component in the low energy theory.

Now we choose $\mathfrak{T}_{\mathbb{Z}_2}$ as the Ising CFT. Again, the KT transformation converts this $\mathbb{Z}_2$-symmetric transition into an $S_3$-symmetric transition. The $\mathbb{Z}_2$-symmetric gapped phases on the two sides of the transition correspond the $\mathfrak{B}_{\text{Dir}}$ and $\mathfrak{B}_{\text{Neu}}$ gapped physical boundaries of the reduced $\mathbb{Z}_2$ TO. The KT transformation converts these to the $\mathfrak{B}_{\text{Dir}}$ and $\mathfrak{B}_{\text{Neu}(\mathbb{Z}_2),t}$ respectively. The $\mathfrak{B}_{\text{Dir}}$ and $\mathfrak{B}_{\text{Neu}(\mathbb{Z}_2),t}$ as physical boundaries correspond to $S_3$ SSB and $\mathbb{Z}_3$ SSB $\times$ $\mathbb{Z}_2$ SPT phase [2]. Therefore

$$\boxed{S_3 \text{ SSB}} \quad \longleftarrow \quad \boxed{\text{Ising} \boxplus \text{Ising} \boxplus \text{Ising}} \quad \longrightarrow \quad \boxed{\mathbb{Z}_3 \text{ SSB} \boxtimes \text{SPT}(\mathbb{Z}_2)_t} \qquad (5.144)$$

## 5.6.2 KT from $\mathbb{Z}_3^{(0)}$ to $S_3$

In the SymTFT picture we have

$$\mathfrak{B}_{\text{sym}} = \mathfrak{B}_{\text{Dir}}, \qquad (5.145)$$

and we consider the quiches of section 5.5.1 that interface to DW($\mathbb{Z}_3$).

**KT from the $Q_1^P$ Condensed Interface.** Similarly, an $S_3$ symmetric theory can be obtained from a $\mathbb{Z}_3$ symmetric theory using a generalized KT transformation. For this we consider the club quiche with $\mathfrak{B}_{\text{sym}} = \mathfrak{B}_{\text{Dir}}$ (carrying an $S_3$ 0-form symmetry) and the $Q_1^P$ condensed interface such that the reduced TO is $\mathcal{Z}(2\mathsf{Vec}\mathbb{Z}_3)$. This club quiche produces a boundary condition for $\mathcal{Z}(2\mathsf{Vec}\mathbb{Z}_3)$

$$\mathfrak{B}' = (\mathfrak{B}_e)_0 \ \oplus \ (\mathfrak{B}_e)_1, \qquad (5.146)$$

where $\mathfrak{B}_e$ is the Dirichlet boundary condition that carries a $\mathbb{Z}_3$ 0-form symmetry. The symmetry on this boundary is $\mathcal{S}' = \text{Mat}_2(2\mathsf{Vec}_{\mathbb{Z}_3})$. We now input a $\mathbb{Z}_3$ 0-form symmetric theory on the physical boundary. The $S_3$ symmetry is realized on this theory via the projection

$$2\mathsf{Vec}_{S_3} \longrightarrow \text{Mat}_2(2\mathsf{Vec}_{\mathbb{Z}_3}). \qquad (5.147)$$

In particular, the Hilbert space of the theory obtained after compactifying the SymTFT decomposes into a direct sum of two *universes* corresponding to the two components of $\mathfrak{B}'$. The $\mathbb{Z}_2$ 0-form symmetry exchanges these universes/components while the $\mathbb{Z}_3^{(0)}$ symmetry acts within the components as (5.78) To summarize the $S_3$ symmetry is realized as

$$\mathbb{Z}_3^{a^2} \ \curvearrowright \mathfrak{T}_{\mathbb{Z}_3} \ \boxplus \ \mathfrak{T}_{\mathbb{Z}_3} \curvearrowleft \ \mathbb{Z}_3^a$$
$$b$$

$$(5.148)$$

Lastly, the data $p \in H^3(\mathbb{Z}_3, U(1))$ on the interface, modifies the associator of the $\mathbb{Z}_3^a$ twisted sector lines in the two vacua/components after compactifying the SymTFT.

Now we choose $\mathfrak{T}_{\mathbb{Z}_3}$ as the 3d 3-State Potts model that realizes a first order transition between the $\mathbb{Z}_3$ broken and the $\mathbb{Z}_3$ trivial phase. The KT transformation converts this $\mathbb{Z}_3$-symmetric transition into an $S_3$-symmetric transition. The $\mathbb{Z}_3$ trivial phase is mapped via the KT transformation to the 2 vacua $\mathbb{Z}_2$ SSB phase, while the $\mathbb{Z}_3$ SSB phase is mapped to the 6 vacua $S_3$ SSB phase.

$$\boxed{S_3 \text{ SSB}} \quad \longleftarrow \quad \boxed{\text{3-Potts} \boxplus \text{3-Potts}} \quad \longrightarrow \quad \boxed{\mathbb{Z}_2 \text{ SSB}} \tag{5.149}$$

### 5.6.3   KTs from $\mathbb{Z}_2^{(0)}$ to $\mathbb{Z}_3^{(1)} \rtimes \mathbb{Z}_2^{(0)}$

There are two classes of generalized KT transformations that take a $\mathbb{Z}_2^{(0)}$ symmetric theory $\mathfrak{T}_{\mathbb{Z}_2}$ and outputs a $\mathbb{Z}_3^{(1)} \rtimes \mathbb{Z}_2^{(0)}$ symmetric theory. These correspond to the two classes of interfaces from $\mathcal{Z}(2\mathsf{Vec}_{S_3})$ to $\mathcal{Z}(2\mathsf{Vec}_{\mathbb{Z}_2})$. The symmetry boundary is chosen as

$$\mathfrak{B}_{\text{sym}} = \mathfrak{B}_{\text{Neu}(\mathbb{Z}_3), p}, \tag{5.150}$$

where $p \in H^3(\mathbb{Z}_3, U(1))$ which carries the 2-group symmetry on it, and we consider the club quiches from section 5.5.2.

**KT from the $Q_2^{[a]}$ Condensed Interface.**   This club quiche produces a boundary condition (5.86) for $\mathcal{Z}(2\mathsf{Vec}_{\mathbb{Z}_2})$ which carries the symmetry (5.88). The $\mathbb{Z}_3^{(1)}$ symmetry is spontaneously broken in this gapless phase. The 2-group symmetric theory thus obtained is

$$\mathfrak{T}_{\mathbb{Z}_2} \; \boxtimes \; \text{DW}(\mathbb{Z}_3)_{p-p'} \curvearrowleft \; 2\mathsf{Vec}_{\mathbb{Z}_2}. \tag{5.151}$$

where $\omega = (p', t') \in \mathbb{Z}_3 \times \mathbb{Z}_2 \in H^3(S_3, U(1))$ and $\omega$ labels the discrete torsion on the interfaces. The $\mathbb{Z}_2^{(0)}$ symmetry acts diagonally in this theory, i.e., it is generated by a product of the $\mathbb{Z}_2$ symmetry defect in $\mathfrak{T}_{\mathbb{Z}_2}$ and the charge conjugation condensation defect $\text{DW}(\mathbb{Z}_3)_{p-p'}$. If the $\mathbb{Z}_2^{(0)}$ symmetry is preserved, i.e. there are $\mathbb{Z}_2$ twisted sector lines in the IR, then the associators of those lines are modified by the data $t' \in H^3(\mathbb{Z}_2, U(1))$ coming from the interface.

Choosing $\mathfrak{T}_{\mathbb{Z}_2}$ to be the Ising CFT, we realize a transition between two phases, one in which the full 2-group symmetry is spontaneously broken and another in which only the $\mathbb{Z}_3^{(1)}$ is broken.

$$\boxed{\text{DW}(\mathbb{Z}_3)_{p-p'} \boxplus \text{DW}(\mathbb{Z}_3)_{p-p'}} \quad \longleftarrow \quad \boxed{\text{Ising} \boxtimes \text{DW}(\mathbb{Z}_3)_{p-p'}} \quad \longrightarrow \boxed{\text{DW}(\mathbb{Z}_3)_{p-p'}}$$

$$\tag{5.152}$$

**KT from the $Q_1^E$ Condensed Interface.** The club quiche with $\mathfrak{B}_{\text{sym}} = \mathfrak{B}_{\text{Neu}(\mathbb{Z}_3),p}$ and the minimal $Q_1^E$ condensed interface (labeled by $H = \mathbb{Z}_2, N = 1, t \in H^3(\mathbb{Z}_2, U(1))$) produces a Dirichlet boundary condition for the reduced TO $\mathcal{Z}(2\text{Vec}_{\mathbb{Z}_2})$. The $\mathbb{Z}_3^{(1)}$ symmetry is trivial, i.e., the lines that generate this symmetry can end on topological local operators. We pick a $\mathbb{Z}_2^{(0)}$ symmetric theory $\mathfrak{T}_{\mathbb{Z}_2}$ as the physical boundary. This straightforwardly produces a $\mathbb{Z}_3^{(1)} \rtimes \mathbb{Z}_2^{(0)}$ theory on which only the $\mathbb{Z}_2^{(0)}$ sub-symmetry acts

$$\mathfrak{T}_{\mathbb{Z}_2} \; \circlearrowleft \quad 2\text{Vec}_{\mathbb{Z}_2} \longleftarrow 2\text{Vec}_{\mathbb{Z}_3^{(1)} \rtimes \mathbb{Z}_2^{(0)}} \; . \tag{5.153}$$

The data $t \in H^3(\mathbb{Z}_2, U(1))$ from the interface modifies the associators of the $\mathbb{Z}_2^{(0)}$ twisted sector lines in the IR phase.

Choosing $\mathfrak{T}_{\mathbb{Z}_2}$ to be the Ising CFT, we realize a transition between a 2-group SPT and a $\mathbb{Z}_2^{(0)}$ spontaneously broken phase.

$$\boxed{\mathbb{Z}_3^{(1)} \rtimes \mathbb{Z}_2^{(0)} \text{ SPT}_t} \qquad \longleftarrow \qquad \boxed{\text{Ising}} \qquad \longrightarrow \qquad \boxed{\mathbb{Z}_2^{(0)} \text{ SSB}} \tag{5.154}$$

### 5.6.4   KTs from $\mathbb{Z}_3^{(0)}$ to $\mathbb{Z}_3^{(1)} \rtimes \mathbb{Z}_2^{(0)}$

**KT from the $Q_1^P$ Condensed Interface.** Similarly, an $\mathbb{Z}_3^{(1)} \rtimes \mathbb{Z}_2^{(0)}$ symmetric theory can be obtained from a $\mathbb{Z}_3$ symmetric theory using a generalized KT transformation. For this we consider the club quiche with $\mathfrak{B}_{\text{sym}} = \mathfrak{B}_{\text{Neu}(\mathbb{Z}_3),p}$ and the $Q_1^P$ condensed interface such that the reduced TO is $\mathcal{Z}(2\text{Vec}\mathbb{Z}_3)$. This club quiche produces a boundary condition for $\mathcal{Z}(2\text{Vec}_{\mathbb{Z}_3})$

$$\mathfrak{B}' = (\mathfrak{B}_m)_0 \; \oplus \; (\mathfrak{B}_{m^2})_1 \; , \tag{5.155}$$

where $\mathfrak{B}_m$ is the Neumann boundary condition that carries a $\mathbb{Z}_3$ 1-form symmetry. The symmetry on this boundary is $\mathcal{S}' = \text{Mat}_2(2\text{Rep}(\mathbb{Z}_3))$. We now input a $\mathbb{Z}_3$ 0-form symmetric theory $\mathfrak{T}_{\mathbb{Z}_3}$ on the physical boundary. The 2-group symmetry is realized on this theory via the 2-functor (see (5.98))

$$2\text{Vec}(\mathbb{Z}_3^{(1)} \rtimes \mathbb{Z}_2^{(0)}) \longrightarrow \text{Mat}_2(2\text{Vec}_{\mathbb{Z}_3}) \; . \tag{5.156}$$

In particular, the Hilbert space of the theory obtained after compactifying the SymTFT decomposes into a direct sum of two *universes* corresponding to the two components of $\mathfrak{B}'$. The $\mathbb{Z}_2$ 0-form symmetry exchanges these universes/components while the $\mathbb{Z}_3^{(1)}$ symmetry acts

within the components as (5.78) To summarize the 2-group symmetry is realized as

$$\mathbb{Z}_3^{(1)} = \langle D_1^{\omega^2} \rangle \,\,\, \overset{\curvearrowright}{\phantom{.}} \,\,\, \mathfrak{T}_{\mathbb{Z}_3}/\mathbb{Z}_3^{(0)} \quad \boxplus \quad \mathfrak{T}_{\mathbb{Z}_3}/\mathbb{Z}_3^{(0)} \,\,\, \overset{\curvearrowleft}{\phantom{.}} \,\,\, \mathbb{Z}_3^{(1)} = \langle D_1^{\omega} \rangle \tag{5.157}$$

$$b$$

Lastly, the data $p \in H^3(\mathbb{Z}_3, U(1))$ on the interface, modifies the associator of the $\mathbb{Z}_3^{(1)}$ charged lines in the two vacua/components after compactifying the SymTFT.

Now choosing $\mathfrak{T}_{\mathbb{Z}_3}$ to be the 3-Potts models, we realize a 1st order transition between a a phase where the full 2-group symmetry is spontaneously broken and a $\mathbb{Z}_2^{(0)}$ spontaneously broken phase. The underlying TFTs for these two gapped phases is $\mathrm{DW}(\mathbb{Z}_3)_p \boxplus \mathrm{DW}(\mathbb{Z}_3)_p$ and $\mathrm{Triv} \boxplus \mathrm{Triv}$ respectively, with the $\mathbb{Z}_2^{(0)}$ symmetry exchanging the two components/universes

$$\boxed{\mathbb{Z}_3^{(1)} \rtimes \mathbb{Z}_2^{(0)} \text{ SSB}} \quad \longleftarrow \quad \boxed{\tfrac{\text{3-Potts}}{\mathbb{Z}_3} \boxplus \tfrac{\text{3-Potts}}{\mathbb{Z}_3}} \quad \longrightarrow \quad \boxed{\mathbb{Z}_2^{(0)} \text{ SSB}} \tag{5.158}$$

### 5.6.5   KTs from $\mathbb{Z}_2^{(0)}$ to $2\mathsf{Rep}(\mathbb{Z}_3^{(1)} \rtimes \mathbb{Z}_2^{(0)})$

We consider now the symmetry boundary to be

$$\mathfrak{B}_{\mathrm{sym}} = \mathfrak{B}_{\mathrm{Neu}(\mathbb{Z}_2),t} \tag{5.159}$$

for the symmetry $2\mathsf{Rep}(\mathbb{Z}_3^{(1)} \rtimes \mathbb{Z}_2^{(0)})$, we apply the club quiches of section 5.5.3 to $\mathbb{Z}_2$ DW.

**KT from the $Q_2^{[a]}$ Condensed Interface.**   For this we consider the club quiche with $\mathfrak{B}_{\mathrm{sym}} = \mathfrak{B}_{\mathrm{Neu}(\mathbb{Z}_2),t}$ and the $Q_2^{[a]}$ condensed interface. The club quiche produces a boundary condition for $\mathcal{Z}(2\mathsf{Vec}_{\mathbb{Z}_2})$

$$\mathfrak{B}' = \mathfrak{B}_{m,t} \tag{5.160}$$

which carries a $2\mathsf{Rep}(\mathbb{Z}_2)$ symmetry on it. Then inserting a $\mathbb{Z}_2^{(0)}$ symmetric theory $\mathfrak{T}_{\mathbb{Z}_2}$ on the physical boundary and compactifying the SymTFT produces the 3d theory

$$(\mathfrak{T}_{\mathbb{Z}_2})_0/\mathbb{Z}_2^{(0)}\,, \tag{5.161}$$

on which the dual $\mathbb{Z}_2^{(1)}$ symmetry acts. The $2\mathsf{Rep}(\mathbb{Z}_3^{(1)} \rtimes \mathbb{Z}_2^{(0)})$ is given by the projection functor (5.106) and is depicted as

$$\mathfrak{T}_{\mathbb{Z}_2}/\mathbb{Z}_2^{(0)} \,\,\, \overset{\curvearrowleft}{\phantom{.}} \,\,\, 2\mathsf{Rep}\mathbb{Z}_2 \longleftarrow 2\mathsf{Rep}(\mathbb{Z}_3^{(1)} \rtimes \mathbb{Z}_2^{(0)})\,. \tag{5.162}$$

Now choosing $\mathfrak{T}_{\mathbb{Z}_2}$ to be the Ising CFT, we realize a transition between a $2\mathsf{Rep}(\mathbb{Z}_3^{(1)} \rtimes \mathbb{Z}_2^{(0)})$ trivial phase a $\mathbb{Z}_2^{(1)}$ spontaneously broken phase.

$$\boxed{2\mathsf{Rep}(\mathbb{Z}_3^{(1)} \rtimes \mathbb{Z}_2^{(0)})\ \text{Triv}} \qquad \longleftarrow \qquad \boxed{\frac{\text{Ising}}{\mathbb{Z}_2^{(0)}}} \qquad \longrightarrow \qquad \boxed{\mathbb{Z}_2^{(1)}\ \text{SSB}} \qquad (5.163)$$

**KT from the $\boldsymbol{Q}_1^E$ Condensed Interface.** A $2\mathsf{Rep}(\mathbb{Z}_3^{(1)} \rtimes \mathbb{Z}_2^{(0)})$ symmetric theory can be obtained from a $\mathbb{Z}_2$ symmetric theory using a generalized KT transformation. For this we consider the club quiche with $\mathfrak{B}_{\mathrm{sym}} = \mathfrak{B}_{\mathrm{Neu}(\mathbb{Z}_2),t}$ and the $\boldsymbol{Q}_1^E$ condensed interface such that the reduced TO is $\mathcal{Z}(2\mathsf{Vec}_{\mathbb{Z}_2})$. This club quiche produces a boundary condition for $\mathcal{Z}(2\mathsf{Vec}_{\mathbb{Z}_2})$

$$\mathfrak{B}' = \frac{(\mathfrak{B}_e)_0}{\mathbb{Z}_2^{(0)}}\ \boxplus\ \left( \frac{(\mathfrak{B}_e)_1\ \boxplus\ (\mathfrak{B}_e)_2}{\mathbb{Z}_2^{(0)}} \right), \qquad (5.164)$$

Then inserting a $\mathbb{Z}_2^{(0)}$ symmetric theory $\mathfrak{T}_{\mathbb{Z}_2}$ on the physical boundary and compactifying the SymTFT produces the 3d theory

$$\frac{(\mathfrak{T}_{\mathbb{Z}_2})_0}{\mathbb{Z}_2^{(0)}}\ \boxplus\ \left( \frac{(\mathfrak{T}_{\mathbb{Z}_2})_1\ \boxplus\ (\mathfrak{T}_{\mathbb{Z}_2})_2}{\mathbb{Z}_2^{(0)}} \right), \qquad (5.165)$$

where the $\mathbb{Z}_2^{(0)}$ acts within $(\mathfrak{T}_{\mathbb{Z}_2})_0$ and exchanges $(\mathfrak{T}_{\mathbb{Z}_2})_1$ and $(\mathfrak{T}_{\mathbb{Z}_2})_2$. The $2\mathsf{Rep}(\mathbb{Z}_3^{(1)} \rtimes \mathbb{Z}_2^{(0)})$ is given by (5.116) and is depicted as

$$\mathbb{Z}_2^{(1)}\ \circlearrowright\ \frac{(\mathfrak{T}_{\mathbb{Z}_2})_0}{\mathbb{Z}_2^{(0)}}\ \boxplus\ \left( \frac{(\mathfrak{T}_{\mathbb{Z}_2})_1\ \boxplus\ (\mathfrak{T}_{\mathbb{Z}_2})_2}{\mathbb{Z}_2^{(0)}} \right)\ \circlearrowleft\ \mathbb{Z}_2^{(1)}\ \text{Triv}\ (\mathrm{D}_1^{\widehat{b}} \cong \mathrm{D}_1^{\mathrm{id}})$$

$$D_2^A$$

$$(5.166)$$

Now choosing $\mathfrak{T}_{\mathbb{Z}_2}$ to be the Ising CFT, we realize a transition between a $2\mathsf{Rep}(\mathbb{Z}_3^{(1)} \rtimes \mathbb{Z}_2^{(0)})$ symmetric phases. The Ising CFT has deformations to the $\mathbb{Z}_2$ broken and $\mathbb{Z}_2$ preserving phases. The $\mathbb{Z}_2$ preserving phase (corresponding to the $\mathfrak{B}_m$ boundary condition of the reduced TO), produces a gapped phase

$$\frac{\text{Triv}}{\mathbb{Z}_2^{(0)}}\ \boxplus\ \text{Triv} = \mathcal{Z}(\mathsf{Vec}_{\mathbb{Z}_2})\ \boxplus\ \text{Triv}, \qquad (5.167)$$

where we have used (Triv $\boxplus$ Triv)/$\mathbb{Z}_2 \cong$ Triv. This is the $2\mathsf{Rep}(\mathbb{Z}_3^{(1)} \rtimes \mathbb{Z}_2^{(0)})$ SSB phase (dubbed "superstar phase" in [2]) and has the interesting feature that one of its vacua is trivial ($\mathbb{Z}_2^{(1)}$ preserving), while the other is topologically ordered ($\mathbb{Z}_2^{(1)}$ breaking). Conversely the deformation with the opposite flows to the symmetry broken phase, for which the underlying

3d TFT is $\mathfrak{T}_{\mathbb{Z}_2} = \text{Triv} \boxplus \text{Triv}$, with the $\mathbb{Z}_2$ symmetry exchanging the two components. The $2\text{Rep}(\mathbb{Z}_3^{(1)} \rtimes \mathbb{Z}_2^{(0)})$ symmetric theory obtained by KT transforming this theory is

$$\left(\frac{\text{Triv} \boxplus \text{Triv}}{\mathbb{Z}_2}\right) \boxplus \left(\frac{\text{Triv} \boxplus \text{Triv} \boxplus \text{Triv} \boxplus \text{Triv}}{\mathbb{Z}_2}\right) = \text{Triv} \boxplus \text{Triv} \boxplus \text{Triv}. \qquad (5.168)$$

This gapped phase is very similar to a $\mathbb{Z}_3$ SSB phase, where the three vacua are cyclically permuted by the symmetry action, except that the non-invertible symmetry $D_2^A$ acts as the direct sum of $\mathbb{Z}_3^a$ generators $D_2^a \oplus D_2^{a^2}$. In this phase the $\mathbb{Z}_2^{(1)}$ form symmetry is trivial while the $D_2^A$ non-invertible symmetry is spontaneously broken, therefore we refer to it as $2\text{Rep}(\mathbb{Z}_3^{(1)} \rtimes \mathbb{Z}_2^{(0)})/\mathbb{Z}_2^{(1)}$ SSB Phase.

$$\boxed{2\text{Rep}(\mathbb{G}) \text{ SSB}} \longleftarrow \boxed{\left(\frac{\text{Ising}}{\mathbb{Z}_2^{(0)}}\right) \boxplus \text{Ising}} \longrightarrow \boxed{2\text{Rep}(\mathbb{G})/\mathbb{Z}_2^{(1)} \text{ SSB}} \qquad (5.169)$$

where $\mathbb{G} = \mathbb{Z}_3^{(1)} \rtimes \mathbb{Z}_2^{(0)}$.

### 5.6.6 KT from $\mathbb{Z}_3^{(0)}$ to $2\text{Rep}(\mathbb{Z}_3^{(1)} \rtimes \mathbb{Z}_2^{(0)})$

We consider now the symmetry boundary to be

$$\mathfrak{B}_{\text{sym}} = \mathfrak{B}_{\text{Neu}(\mathbb{Z}_2),t} \qquad (5.170)$$

for the symmetry $2\text{Rep}(\mathbb{Z}_3^{(1)} \rtimes \mathbb{Z}_2^{(0)})$, we apply the club quiches of section 5.5.3 to $\mathbb{Z}_3$ DW.

**KT from the $Q_1^P$ Condensed Interface.** Consider the club quiche with $\mathfrak{B}_{\text{sym}} = \mathfrak{B}_{\text{Neu}(\mathbb{Z}_2),t}$ and the $Q_1^P$ condensed interface such that the reduced TO is $\mathcal{Z}(2\text{Vec}\mathbb{Z}_3)$. This club quiche produces a boundary condition $\mathfrak{B}' = \mathfrak{B}_e$ for $\mathcal{Z}(2\text{Vec}_{\mathbb{Z}_3})$ which carries a $2\text{Vec}_{\mathbb{Z}_3}$ symmetry on it

$$\mathfrak{B}' = \mathfrak{B}_m, \qquad (5.171)$$

Then inserting a $\mathbb{Z}_3^{(0)}$ symmetric theory $\mathfrak{T}_{\mathbb{Z}_2}$ on the physical boundary and compactifying the SymTFT produces the 3d theory

$$\mathfrak{T}_{\mathbb{Z}_3} \quad \curvearrowleft \quad 2\text{Vec}\mathbb{Z}_3 \longleftarrow 2\text{Rep}(\mathbb{Z}_3^{(1)} \rtimes \mathbb{Z}_2^{(0)}). \qquad (5.172)$$

The $\mathbb{Z}_2^{(1)}$ is trivialized in this gapless phase, while the non-invertible symmetry is realized as $D_2^m \oplus D_2^{m^2}$.

Now picking the 3d theory $\mathfrak{T}_{\mathbb{Z}_3}$ to be the 3-Potts model, we realize a 1st order transition between a 3 vacuum phase which is the $2\mathsf{Rep}(\mathbb{G})/\mathbb{Z}_2^{(1)}$ SSB described above and a single vacuum $2\mathsf{Rep}(\mathbb{G}^{(2)})$ SSB phase.

$$\boxed{2\mathsf{Rep}(\mathbb{G})\ \mathrm{Triv}} \longleftarrow \boxed{\text{3-Potts}} \longrightarrow \boxed{2\mathsf{Rep}(\mathbb{G})/\mathbb{Z}_2^{(1)}\ \mathrm{SSB}} \quad (5.173)$$

### 5.6.7   KTs from $\mathbb{Z}_2^{(1)}$ to $2\mathsf{Rep}(S_3)$

Finally, consider

$$\mathfrak{B}_{\mathrm{sym}} = \mathfrak{B}_{\mathrm{Neu}(S_3),(p-p',t-t')} \tag{5.174}$$

for the $2\mathsf{Rep}(S_3)$ symmetry, and the club quiches from section 5.5.4.

**KT from the $Q_2^{[a]}$ Condensed Interface.**   A $2\mathsf{Rep}(S_3)$ symmetric theory can be obtained from a $\mathbb{Z}_2$ symmetric theory using a generalized KT transformation. and the $Q_2^{[a]}$ condensed interface of section 5.3.1 such that the reduced TO is $\mathcal{Z}(2\mathsf{Vec}_{\mathbb{Z}_2})$. This club quiche produces a boundary condition for $\mathcal{Z}(2\mathsf{Vec}_{\mathbb{Z}_2})$

$$\mathfrak{B}' = \frac{\mathrm{DW}(\mathbb{Z}_3)_{p-p'} \boxtimes \mathfrak{B}_e}{(\mathbb{Z}_2^{(0),\mathrm{diag}}, t-t')} . \tag{5.175}$$

Then inserting a $\mathbb{Z}_2^{(0)}$ symmetric theory $\mathfrak{T}_{\mathbb{Z}_2}$ on the physical boundary and compactifying the SymTFT produces the 3d theory

$$\frac{\mathrm{DW}(\mathbb{Z}_3)_{p-p'} \boxtimes \mathfrak{T}_{\mathbb{Z}_2}}{(\mathbb{Z}_2^{(0),\mathrm{diag}}, t-t')} . \tag{5.176}$$

The $D_1^E$ symmetry is realized as the non-invertible line in $\mathrm{DW}(\mathbb{Z}_3)_p/\mathbb{Z}_2^{(0)}$. $D_1^P$ is the dual $\mathsf{Rep}(Z_2)$ symmetry generator obtained upon gauging $\mathbb{Z}_2^{(0),\mathrm{diag}}$.

Picking $\mathfrak{T}_{\mathbb{Z}_2}$ to be Ising, produces a second-order transition between two $2\mathsf{Rep}(S_3)$ symmetric phases. The Ising transition has deformations to the $\mathbb{Z}_2$ trivial and the $\mathbb{Z}_2$ broken phase. Performing the KT transformation on the $\mathbb{Z}_2$ trivial phase produces

$$\frac{\mathrm{DW}(\mathbb{Z}_3)_{p-p'} \boxtimes \mathrm{Triv}}{(\mathbb{Z}_2^{(0),\mathrm{diag}}, t-t')} \cong \mathrm{DW}(S_3)_{(p-p',t-t')} . \tag{5.177}$$

This is the gapped phase in which the full $\mathsf{Rep}(S_3)$ 1-form symmetry is spontaneously broken. Instead, performing the KT transformation on the $\mathbb{Z}_2$ broken phase produces

$$\frac{\mathrm{DW}(\mathbb{Z}_3)_{p-p'} \boxtimes \mathrm{DW}(\mathbb{Z}_3)_{p-p'}}{(\mathbb{Z}_2^{(0),\mathrm{diag}}, t-t')} \cong \mathrm{DW}(Z_3)_{p-p'} . \tag{5.178}$$

This is the gapped phase in which $D_1^P$ remains unbroken, i.e., it can end while $D_1^E$ is realized as $D_1^\omega \oplus D_1^{\omega^2}$ which braids with the other flux lines. Therefore, the $\mathsf{Rep}(S_3)/\mathbb{Z}_2$ 1-form symmetry is spontaneously broken. Hence we find the following $2\mathsf{Rep}(S_3)$-symmetric second order transition:

$$\boxed{\mathsf{Rep}(S_3)/\mathbb{Z}_2 \text{ 1-form SSB}} \longleftarrow \boxed{\frac{\mathrm{DW}(\mathbb{Z}_3)_{p-p'} \boxtimes \text{ Ising}}{(\mathbb{Z}_2^{(0),\mathrm{diag}}, t-t')}} \longrightarrow \boxed{\mathsf{Rep}(S_3) \text{ 1-form SSB}}$$

(5.179)

This is an **igSSB** for the 1-form symmetry, as it cannot be gapped without increasing the number of charged lines. One way to see this is that for this gapless phase we have the lines $\boldsymbol{Q}_1^1 \oplus \boldsymbol{Q}_1^{[a]}$ (see table 6, with $H = S_3$ and $N = \mathbb{Z}_3$). For gapped phases, these get enhanced to either $\boldsymbol{Q}_1^1 \oplus 2\boldsymbol{Q}_1^{[a]} \oplus \boldsymbol{Q}_1^P$ or $\boldsymbol{Q}_1^1 \oplus \boldsymbol{Q}_1^{[a]} \oplus \boldsymbol{Q}_1^{[b]}$ (see table 4 of [2] for the lines in the minimal gapped BC of $\mathrm{DW}(S_3)$). So the number of $2\mathsf{Rep}(S_3)$ charged lines i.e. those arising from bulk surfaces ending on both $\mathfrak{B}_{\mathrm{sym}}$ and $\mathfrak{B}_{\mathrm{phys}}$ increase in the gapped phase. Thus we have an igSSB for the 1-form symmetry.

**KT from the $\boldsymbol{Q}_1^E$ Condensed Interface.** Consider the club quiche with $\mathfrak{B}_{\mathrm{sym}} = \mathfrak{B}_{\mathrm{Neu}(S_3),(p,t)}$ and the $\boldsymbol{Q}_1^E$ condensed interface of section 5.3.2 such that the reduced TO is $\mathcal{Z}(2\mathsf{Vec}_{\mathbb{Z}_2})$. This club quiche produces a boundary condition $\mathfrak{B}' = \mathfrak{B}_{m,t'}$ for $\mathcal{Z}(2\mathsf{Vec}_{\mathbb{Z}_2})$. The symmetry category on this boundary is $2\mathsf{Rep}(\mathbb{Z}_2)$ and the $2\mathsf{Rep}(S_3)$ symmetry is implemented via a projection from $2\mathsf{Rep}(S_3)$ to $2\mathsf{Rep}(\mathbb{Z}_2)$. Then inserting a $\mathbb{Z}_2^{(0)}$ symmetric theory $\mathfrak{T}_{\mathbb{Z}_2}$ on the physical boundary and compactifying the SymTFT produces the 3d theory

$$\mathfrak{T}_{\mathbb{Z}_2}/(\mathbb{Z}_2^{(0)}, t-t').$$

(5.180)

The $2\mathsf{Rep}(S_3)$ acts as

$$\frac{\mathfrak{T}_{\mathbb{Z}_2}}{(\mathbb{Z}_2^{(0)}, t-t')} \,\circlearrowleft \qquad 2\mathsf{Rep}\mathbb{Z}_2 \longleftarrow 2\mathsf{Rep}(S_3).$$

(5.181)

By picking $\mathfrak{T}_{\mathbb{Z}_2}$ to be Ising, we immediately find a second-order phase transition

$$\boxed{\mathsf{Rep}(S_3) \text{ 1-form Triv}} \longleftarrow \boxed{\frac{\text{Ising}}{(\mathbb{Z}_2^{(0)}, t-t')}} \longrightarrow \boxed{\mathsf{Rep}(Z_2) \text{ 1-form SSB}}$$

(5.182)

### 5.6.8 KTs from $\mathbb{Z}_3^{(1)}$ to $2\mathsf{Rep}(S_3)$

**KT from the $\boldsymbol{Q}_1^P$ Condensed Interface.** Consider the club quiche with the $\boldsymbol{Q}_1^P$ condensed interface (section 5.3.3) such that the reduced TO is $\mathcal{Z}(2\mathsf{Vec}\mathbb{Z}_3)$. This club quiche produces

a boundary condition $\mathfrak{B}' = \mathfrak{B}_{m,p'}$ for $\mathcal{Z}(2\mathsf{Vec}_{\mathbb{Z}_3})$. The symmetry category on this boundary is $2\mathsf{Rep}(\mathbb{Z}_3)$ and the $2\mathsf{Rep}(S_3)$ symmetry is implemented via a projection from $2\mathsf{Rep}(S_3)$ to $2\mathsf{Rep}(\mathbb{Z}_3)$. Specifically, $D_1^P \cong D_1^{\mathrm{id}}$ and $D_1^E \cong D_1^\omega \oplus D_1^{\omega^2}$. Inserting a $\mathbb{Z}_3^{(0)}$ symmetric theory $\mathfrak{T}_{\mathbb{Z}_3}$ on the physical boundary and compactifying the SymTFT produces the 3d theory

$$\mathfrak{T}_{\mathbb{Z}_3}/(\mathbb{Z}_3^{(0)}, p - p') \, . \tag{5.183}$$

The $2\mathsf{Rep}(S_3)$ acts as

$$\frac{\mathfrak{T}_{\mathbb{Z}_3}}{(\mathbb{Z}_3^{(0)}, p-p')} \;\;\circlearrowleft\;\; \quad 2\mathsf{Rep}\mathbb{Z}_3 \longleftarrow 2\mathsf{Rep}(S_3) \, . \tag{5.184}$$

By picking $\mathfrak{T}_{\mathbb{Z}_3}$ to be 3-Potts, we find a first-order phase transition

$$
\boxed{\mathsf{Rep}(S_3) \text{ 1-form Triv}} \longleftarrow \boxed{\frac{\text{3-Potts}}{(\mathbb{Z}_3^{(0)}, p-p')}} \longrightarrow \boxed{\mathsf{Rep}(S_3)/\mathbb{Z}_2 \text{ 1-form SSB}}
\tag{5.185}
$$

# 6   Gapless Phases from $D_8$ DW Theory

In this section we turn to symmetries that are realized as the fusion 2-categories of topological defects on gapped boundary conditions of the (3+1)d $D_8$ Dijkgraaf-Witten theory. Such gapped boundary conditions and gapped phases were discussed in [2]. After a brief review of the $D_8$ (3+1)d gauge theory, we discuss gapless (2+1)d phases with symmetries obtainable from $D_8$ 0-form symmetry via generalized gaugings.

## 6.1   The SymTFT

We present the group $D_8$ (of symmetries of a square, sometimes referred to as $D_4$) as

$$D_8 = \mathbb{Z}_4 \rtimes \mathbb{Z}_2 = \left\langle r\, , \; x \mid r^4 = x^2 = \mathrm{id}\, , \; xrx = r^3 \right\rangle . \tag{6.1}$$

The $D_8$ conjugacy classes are

$$[\mathrm{id}]\, , \quad [r^2]\, , \quad [r] = \{r,\, r^3\}\, , \quad [x] = \{x,\, xr^2\}\, , \quad [xr] = \{xr,\, xr^3\}\, , \tag{6.2}$$

each of which labels a topological non-condensation surface $\boldsymbol{Q}_2^{[g]}$:

$$\boldsymbol{Q}_2^{[\mathrm{id}]}\, , \quad \boldsymbol{Q}_2^{[r^2]}\, , \quad \boldsymbol{Q}_2^{[r]}\, , \quad \boldsymbol{Q}_2^{[x]}\, , \quad \boldsymbol{Q}_2^{[xr]}\, . \tag{6.3}$$

Topological lines on each SymTFT surface are labeled by irreducible representations of the centralizer $H_g = \{h \in D_8 \mid hg = gh\}$ for $g$ a representative of $[g]$, and are [2]:

$$
\begin{aligned}
\boldsymbol{Q}_2^{[\mathrm{id}]} : \quad & 1\mathrm{End}(\boldsymbol{Q}_2^{[\mathrm{id}]}) = \left\{\boldsymbol{Q}_1^R;\ R = 1, 1_r, 1_x, 1_{xr}, E\right\} \cong \mathsf{Rep}(D_8) \\
\boldsymbol{Q}_2^{[r^2]} : \quad & 1\mathrm{End}(\boldsymbol{Q}_2^{[r^2]}) = \left\{\boldsymbol{Q}_1^{[r^2],R};\ R = 1, 1_r, 1_x, 1_{xr}, E\right\} \cong \mathsf{Rep}(D_8) \\
\boldsymbol{Q}_2^{[xr]} : \quad & 1\mathrm{End}(\boldsymbol{Q}_2^{[xr]}) = \left\{\boldsymbol{Q}_1^{[xr],R};\ R = (\epsilon_1, \epsilon_2),\ \epsilon_i = \pm\right\} \cong \mathsf{Rep}(\mathbb{Z}_2 \times \mathbb{Z}_2) \qquad (6.4) \\
\boldsymbol{Q}_2^{[x]} : \quad & 1\mathrm{End}(\boldsymbol{Q}_2^{[x]}) = \left\{\boldsymbol{Q}_1^{[x],R};\ R = (\epsilon_1, \epsilon_2),\ \epsilon_i = \pm\right\} \cong \mathsf{Rep}(\mathbb{Z}_2 \times \mathbb{Z}_2) \\
\boldsymbol{Q}_2^{[r]} : \quad & 1\mathrm{End}(\boldsymbol{Q}_2^{[r]}) = \left\{\boldsymbol{Q}_1^{[r],R};\ R = 1, i, -1, -i\right\} \cong \mathsf{Rep}(\mathbb{Z}_4)\,.
\end{aligned}
$$

In particular, the topological lines on the identity surface (and thus genuine bulk topological lines), $\boldsymbol{Q}_1^{[\mathrm{id}],R} \equiv \boldsymbol{Q}_1^R$, are labeled by $R \in \mathsf{Rep}(D_8)$: we denote by $1_k$ for $k \in \{r, x, xr\}$ a 1-dimensional irrep that is 1 on $[1], [r^2], [k]$ and $-1$ otherwise, whereas for the 2-dimensional $E$ we chose the following matrix representation:

$$
\mathcal{D}_E(r) = \begin{pmatrix} i & 0 \\ 0 & -i \end{pmatrix}, \qquad \mathcal{D}_E(x) = \begin{pmatrix} 0 & 1 \\ 1 & 0 \end{pmatrix}. \qquad (6.5)
$$

## 6.2 Symmetries and Gapped Boundary Conditions

The topological boundary conditions of the $(3+1)$d $D_8$ SymTFT were studied in detail in [2] and are summarized in table 7. A convenient way to obtain any minimal boundary condition is to start with the Dirichlet boundary condition $\mathfrak{B}_{\mathrm{Dir}}$, which gives rise to $D_8$ 0-form symmetry $2\mathsf{Vec}_{D_8}$, and gauge a sub-group $H$ of $D_8$. There is an additional choice of discrete torsion $\omega \in H^3(H, U(1))$ in gauging $H$, which can be understood as stacking an $H$-SPT before gauging. The boundary condition obtained by gauging $H$ with discrete torsion $\omega$ is the $H$-Neumann boundary condition denoted as

$$
\mathfrak{B}_{\mathrm{Neu}(H),\omega} = \frac{\mathfrak{B}_{\mathrm{Dir}}}{(H, \omega)}\,. \qquad (6.6)
$$

The non-invertible symmetry web for $D_8$, which includes (anomalous) 2-groups and 2-representation categories of 2-groups, was described in [45] and the SymTFT derivation provided in [2]. Here we will briefly summarize some specific cases.

### 6.2.1 Dirichlet Boundary Condition

Each SymTFT line $\boldsymbol{Q}_1^R$ for $R \in \mathsf{Rep}(D_8)$ has $\dim(R)$ ends on $\mathfrak{B}_{\mathrm{Dir}}$ which we denote as $\mathcal{E}_0^{R,i}$ with $i = 1, \ldots, \dim(R)$. We denote this as

$$
\boldsymbol{Q}_1^R\Big|_{\mathrm{Dir}} = \left\{\mathcal{E}_0^{R,i} \,\Big|\, i = 1, \ldots, \dim(R)\right\}. \qquad (6.7)
$$

| $H = N$ | $H^3(H, U(1))$ | $H^{\text{diag}}$ | $Q_p$ that can end on $\mathfrak{B}_{\text{Neu}(H),\omega}$ |
|---|---|---|---|
| $1$ | $1$ | $\langle(1,1)\rangle$ | $\begin{cases} Q_2^{[\text{id}]} \\ Q_1^1 \oplus Q_1^{1r} \oplus Q_1^{1x} \oplus Q_1^{1xr} \oplus 2Q_1^E \end{cases}$ |
| $\mathbb{Z}_2^{r^2}$ | $\mathbb{Z}_2$ | $\langle(r^2,1)\rangle$ | $\begin{cases} Q_2^{[\text{id}]} \oplus Q_2^{[r^2]} \\ Q_1^1 \oplus Q_1^{1r} \oplus Q_1^{1x} \oplus Q_1^{1xr} \oplus \\ Q_1^{r^2,1} \oplus Q_1^{r^2,1_r} \oplus Q_1^{r^2,1_x} \oplus Q_1^{r^2,1_{xr}} \end{cases}$ |
| $\mathbb{Z}_4^r$ | $\mathbb{Z}_4$ | $\langle(r,1)\rangle$ | $\begin{cases} Q_2^{[\text{id}]} \oplus Q_2^{[r^2]} \oplus Q_2^{[r]} \\ Q_1^1 \oplus Q_1^{1r} \oplus Q_1^{[r^2],1} \oplus Q_1^{[r^2],1_r} \oplus 2Q_1^{[r],1} \end{cases}$ |
| $\mathbb{Z}_2^{r^2} \times \mathbb{Z}_2^{xr}$ | $\mathbb{Z}_2 \times \mathbb{Z}_2 \times \mathbb{Z}_2$ | $\langle(r^2,1),\ (xr,1)\rangle$ | $\begin{cases} Q_2^{[\text{id}]} \oplus Q_2^{[r^2]} \oplus Q_2^{[xr]} \\ Q_1^1 \oplus Q_1^{1xr} \oplus Q_1^{r^2,1} \oplus Q_1^{r^2,1_{xr}} \oplus 2Q_1^{xr,1_{++}} \end{cases}$ |
| $\mathbb{Z}_2^{r^2} \times \mathbb{Z}_2^x$ | $\mathbb{Z}_2 \times \mathbb{Z}_2 \times \mathbb{Z}_2$ | $\langle(r^2,1),\ (x,1)\rangle$ | $\begin{cases} Q_2^{[\text{id}]} \oplus Q_2^{[r^2]} \oplus Q_2^{[x]} \\ Q_1^1 \oplus Q_1^{1x} \oplus Q_1^{r^2,1} \oplus Q_1^{r^2,1_x} \oplus 2Q_1^{x,1_{++}} \end{cases}$ |
| $\mathbb{Z}_2^x$ | $\mathbb{Z}_2$ | $\langle(x,1)\rangle$ | $\begin{cases} Q_2^{[\text{id}]} \oplus Q_2^{[x]} \\ Q_1^1 \oplus Q_1^{1x} \oplus Q_1^E \oplus Q_1^{x,1_{++}} \oplus 2Q_1^{x,1_{+-}} \end{cases}$ |
| $\mathbb{Z}_2^{xr}$ | $\mathbb{Z}_2$ | $\langle(xr,1)\rangle$ | $\begin{cases} Q_2^{[\text{id}]} \oplus Q_2^{[xr]} \\ Q_1^1 \oplus Q_1^{1xr} \oplus Q_1^E \oplus Q_1^{xr,1_{++}} \oplus 2Q_1^{xr,1_{+-}} \end{cases}$ |
| $D_8$ | $\mathbb{Z}_2 \times \mathbb{Z}_2 \times \mathbb{Z}_4$ | $\langle(r,1),\ (x,1)\rangle$ | $\begin{cases} Q_2^{[\text{id}]} \oplus Q_2^{[r^2]} \oplus Q_2^{[r]} \oplus Q_2^{[x]} \oplus Q_2^{[xr]} \\ Q_1^1 \oplus Q_1^{[r^2],1} \oplus Q_1^{[r],1} \oplus Q_1^{[x],1_{++}} \oplus Q_1^{[xr],1_{++}} \end{cases}$ |

Table 7: Data for (2+1)d minimal gapped boundaries of $\mathcal{Z}(2\mathsf{Vec}_{D_8})$, specified by a subgroup $H = N$ (up to conjugation) and an $H$-SPT $\omega \in H^3(H, U(1))$ [2].

Each bulk surface $\boldsymbol{Q}_2^{[g]}$ splits into a direct sum of surfaces $D_2^g$ for $g \in [g]$ on $\mathfrak{B}_{\text{Dir}}$, which fuse according to the group multiplication in $D_8$. These are the generators of the $D_8$ 0-form symmetry on this boundary. More specifically, the bulk surface $\boldsymbol{Q}_2^{[g]}$ has twisted sector ends $\mathcal{E}_1^g$ for all $g \in [g]$, where the line $\mathcal{E}_1^g$ is in the twisted sector of the $D_2^g$ defect on $\mathfrak{B}_{\text{Dir}}$. We denote this as

$$\boldsymbol{Q}_2^{[g]}\bigg|_{\text{Dir}} = \left\{ (D_2^g , \mathcal{E}_1^g) \;\; \bigg| \;\; g \in [g] \right\} . \tag{6.8}$$

The topological operators $\{\mathcal{E}_0^{R,i}\}$ transform as an $R$-multiplet under the $D_8$ symmetry. We summarize the $\mathfrak{B}_{\text{Dir}}$ boundary condition as (only the bulk defects with untwisted ends are shown) the quiche:

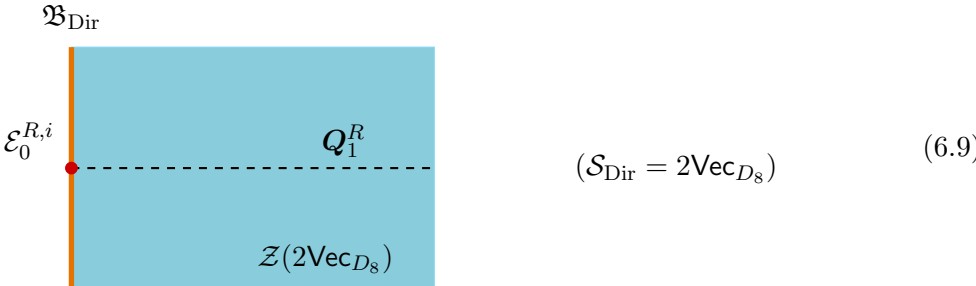

$$(\mathcal{S}_{\text{Dir}} = 2\text{Vec}_{D_8}) \tag{6.9}$$

## 6.2.2 Neumann $\mathbb{Z}_2^x$ Boundary Condition

This boundary conditions is obtained by gauging the non-normal $\mathbb{Z}_2^x$ on $\mathfrak{B}_{\text{Dir}}$ with discrete torsion $\omega \in H^3(\mathbb{Z}_2, U(1)) = \mathbb{Z}_2$. The resulting boundary $\mathfrak{B}_{\text{Neu}(\mathbb{Z}_2^x),\omega}$ has a 1-form symmetry whose generator we denote by $D_1^{1r}$, an invertible 0-form symmetry generated by $D_2^{r^2}$ and a non-invertible 0-form symmetry generated by a surface we will denote by $D_2^{[r]}$, which on $\mathfrak{B}_{\text{Dir}}$ was the composite object $D_2^r \oplus D_2^{r^3}$, constituting an orbit under $\mathbb{Z}_2^x$. It obeys the fusion [45]:

$$D_2^{[r]} \otimes D_2^{[r]} = \frac{D_2^{\text{id}}}{D_1^1 \oplus D_1^{1r}} \oplus \frac{D_2^{r^2}}{D_1^1 \oplus D_1^{1r}} . \tag{6.10}$$

Let us summarize the topological defects on this boundary, following [2]. Since the genuine local operator $\mathcal{E}_0^{1x}$ at the end of $\boldsymbol{Q}_1^{1x}$ was untwisted and uncharged under the $\mathbb{Z}_2^x$ symmetry, it remains unaltered after gauging and ends on this boundary.

The ends $\mathcal{E}_0^{E,1}$ and $\mathcal{E}_0^{E,2}$ of $\boldsymbol{Q}_1^E$ on $\mathfrak{B}_{\text{Dir}}$, can be decomposed into linear combinations $\mathcal{E}_0^{E,\pm} = \mathcal{E}_0^{E,1} \pm \mathcal{E}_0^{E,2}$ with $\mathcal{E}_0^{E,+}$ and $\mathcal{E}_0^{E,-}$ respectively uncharged and charged under $D_2^x$. Therefore, upon gauging $\mathbb{Z}_2^x$, one obtains

$$\boldsymbol{Q}_1^E\bigg|_{\text{Neu}(\mathbb{Z}_2^x),\omega} = (\mathcal{E}_0^{E,+} , (D_1^{1r} , \mathcal{E}_0^{E,-})) . \tag{6.11}$$

The surfaces $D_2^r$ and $D_2^{r^3}$ are mapped to one another upon conjugation by $D_2^x$, therefore

$$\left. \boldsymbol{Q}_2^{[r]} \right|_{\mathrm{Neu}(\mathbb{Z}_2),\omega} = \left\{ (D_2^{[r]}, \mathcal{E}_1^{[r]}) \right\}, \tag{6.12}$$

where $D_2^{[r]}$ is an indecomposable defect on $\mathfrak{B}_{\mathrm{Neu}(\mathbb{Z}_2),\omega}$ obeying the fusion (6.10).

The bulk SymTFT surface $\boldsymbol{Q}_2^{[x]}$ can now end on this boundary, and gives rise to a line line $\mathcal{E}_1^x$ charged under $D_1^{1r}$ and with F-symbols given by $\omega \in H^3(\mathbb{Z}_2, U(1))$. In summary, this boundary carries a symmetry corresponding to the 2-representations of the 2-group generated by a (non-condensation) non-invertible 0-form symmetry generator $D_2^{[r]}$, an invertible $\mathbb{Z}_2$ 0-form symmetry generator $D_2^{r^2}$ and a $\mathbb{Z}_2$ 1-form generator $D_1^{1r}$. This $\mathfrak{B}_{\mathrm{Neu}(\mathbb{Z}_2),\omega}$ boundary condition is depicted as the quiche[24]

$$\left( \mathcal{S}_{\mathrm{Neu}(\mathbb{Z}_2),\omega} = 2\mathsf{Rep}(\mathbb{Z}_4^{(1)} \rtimes \mathbb{Z}_2^{(0)}) \right) \tag{6.13}$$

### 6.2.3 Neumann $D_8$ Boundary Conditions

This boundary condition is obtained by gauging the full $D_8$ symmetry on $\mathfrak{B}_{\mathrm{Dir}}$. The SymTFT lines $\boldsymbol{Q}_1^R$ end on twisted sector local operators

$$\left. \boldsymbol{Q}_1^R \right|_{\mathrm{Neu}(D_8),\omega} = (D_1^R, \mathcal{E}_0^R), \tag{6.14}$$

where the $D_1^R$ generate a $\mathsf{Rep}(D_8)$ 1-form symmetry and are:

$$D_1^1, \quad D_1^{1x}, \quad D_1^{1r}, \quad D_1^{1xr}, \quad D_1^E, \tag{6.15}$$

whose fusions follow from the tensor products of irreps (for $1_k = 1_x, 1_r, 1_{xr}$):

$$D_1^{1x} \otimes D_1^{1r} = D_1^{1xr}, \quad D_1^{1k} \otimes D_1^{1k} = D_1^1,$$
$$D_1^E \otimes D_1^E = D_1^1 \oplus D_1^{1x} \oplus D_1^{1r} \oplus D_1^{1xr}. \tag{6.16}$$

The 1-dimensional irreps thus generate an invertible $\mathbb{Z}_2 \times \mathbb{Z}_2$ 1-form symmetry, whereas $D_1^E$ is non-invertible. The SymTFT surfaces end on single untwisted sector lines

$$\left. \boldsymbol{Q}_2^{[g]} \right|_{\mathrm{Neu}(D_8),\omega} = \mathcal{E}_1^{[g]}. \tag{6.17}$$

---

[24] Again, in the 2d project, lines are dashed, surfaces solid lines.

The line $\mathcal{E}_1^{[g]}$ has a braiding phase of $\chi_R([g])$ with $D_1^R$. The lines $\mathcal{E}_1^{[g]}$ have F-symbols that are determined by $\omega \in H^3(D_8, U(1))$ and are those of the 3d $D_8$ Dijkgraaf-Witten theory with topological action $\omega$ [80]. All other symmetry defects on this boundary can be obtained as condensation defects constructed from the $\mathsf{Rep}(D_8)$ lines. The boundary condition can be summarized as (only the bulk defects with untwisted ends are shown)

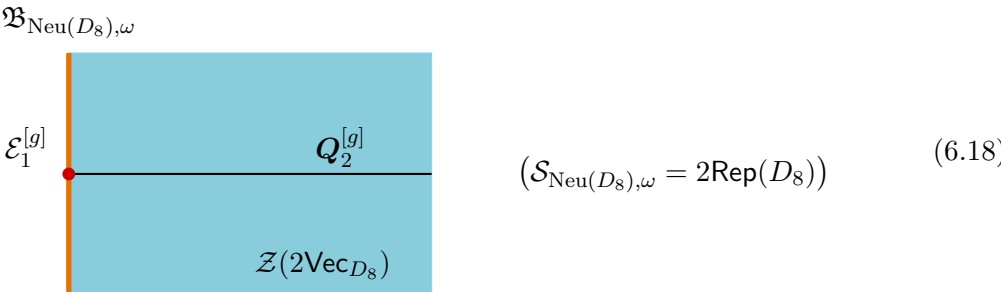

$$\left( \mathcal{S}_{\mathrm{Neu}(D_8),\omega} = 2\mathsf{Rep}(D_8) \right) \tag{6.18}$$

All minimal (2+1)d gapped phases with $2\mathsf{Vec}_{D_8}$, $2\mathsf{Rep}(\mathbb{Z}_4^{(1)} \rtimes \mathbb{Z}_2^{(0)})$ and $2\mathsf{Rep}(D_8)$ symmetry were described in [2].

## 6.3    Minimal Interfaces

We now turn to the minimal interfaces from the $D_8$ TO to a reduced TO. These are given in terms of the usual data $(H, N, \pi, \omega)$, specifying an interface between $\mathcal{Z}(2\mathsf{Vec}_{D_8})$ and $\mathcal{Z}(2\mathsf{Vec}_{H/N}^\pi)$. We have listed these in table 8. We note that if $N = 1$, then $H = H/N$ so one must choose trivial twist $\pi \in H^4(H/N, U(1))$ for it to be trivial when pulled-back to $H$. Instead, for $H = D_8$ and $N = \mathbb{Z}_2^{r^2}$, we can pick a non-trivial $\pi \in H^4(\mathbb{Z}_2 \times \mathbb{Z}_2, U(1)) = \mathbb{Z}_2 \times \mathbb{Z}_2$, as we will discuss below.

$H = D_8$ and $N = 1$, describes the transparent interface within $\mathcal{Z}(2\mathsf{Vec}_{D_8})$. Let us therefore analyze the other possibilities, corresponding to interfaces between $\mathcal{Z}(2\mathsf{Vec}_{D_8})$ and $\mathcal{Z}(2\mathsf{Vec}_{H/N}^\pi)$, which will then give rise to gapless phases upon fixing $\mathfrak{B}_{\mathrm{sym}}$ and $\mathfrak{B}_{\mathrm{phys}}$ boundary conditions.

### 6.3.1    Interface to $\mathrm{DW}(\mathbb{Z}_4)$

There is a single class of minimal interface between $\mathrm{DW}(D_8)$ and $\mathrm{DW}(\mathbb{Z}_4)$. We denote the topological defects of $\mathrm{DW}(\mathbb{Z}_4)$ as:

$$\boldsymbol{Q}_1^{e^p}, \qquad \boldsymbol{Q}_2^{m^p}, \qquad p \in \{0, 1, 2, 3\}. \tag{6.19}$$

$\boldsymbol{Q}_1^{1^r}$ **Condensed Interface.**    This class of interfaces is characterized by $H = \mathbb{Z}_4^r$, $N = 1$. Since $N$ is trivial, no (non-identity) surfaces can end on this interface. For the topological

| $H$ | $N$ | $H^3(H,U(1))$ | $H/N$ | $H^4(H/N,U(1))$ | $H^{\text{diag}}$ | $Q_p$ that can end on $\mathcal{I}_{(H,N,\pi,\omega)}$ |
|---|---|---|---|---|---|---|
| $D_8$ | $1$ | $\mathbb{Z}_2 \times \mathbb{Z}_2 \times \mathbb{Z}_4$ | $D_8$ | $(\mathbb{Z}_2 \times \mathbb{Z}_2)$ | $\langle(r,r),\ (x,x)\rangle$ | $Q_2^{[\text{id}]}$ <br> $Q_1^1$ |
| $\mathbb{Z}_4^r$ | $1$ | $\mathbb{Z}_4$ | $\mathbb{Z}_4$ | $1$ | $\langle(r,m)\rangle$ | $Q_2^{[\text{id}]}$ <br> $Q_1^1 \oplus Q_1^{1r}$ |
| $\mathbb{Z}_2^{r^2} \times \mathbb{Z}_2^{xr}$ | $1$ | $\mathbb{Z}_2 \times \mathbb{Z}_2 \times \mathbb{Z}_2$ | $\mathbb{Z}_2 \times \mathbb{Z}_2$ | $(\mathbb{Z}_2 \times \mathbb{Z}_2)$ | $\langle(r^2,m_1),\ (xr,m_2)\rangle$ | $Q_2^{[\text{id}]}$ <br> $Q_1^1 \oplus Q_1^{1xr}$ |
| $\mathbb{Z}_2^{r^2} \times \mathbb{Z}_2^{x}$ | $1$ | $\mathbb{Z}_2 \times \mathbb{Z}_2 \times \mathbb{Z}_2$ | $\mathbb{Z}_2 \times \mathbb{Z}_2$ | $(\mathbb{Z}_2 \times \mathbb{Z}_2)$ | $\langle(r^2,m_1),\ (x,m_2)\rangle$ | $Q_2^{[\text{id}]}$ <br> $Q_1^1 \oplus Q_1^{1x}$ |
| $D_8$ | $\mathbb{Z}_2^{r^2}$ | $\mathbb{Z}_2 \times \mathbb{Z}_2 \times \mathbb{Z}_4$ | $\mathbb{Z}_2 \times \mathbb{Z}_2$ | $\mathbb{Z}_2 \times \mathbb{Z}_2$ | $\langle(r,m_1),\ (x,m_2)\rangle$ | $Q_2^{[\text{id}]} \oplus Q_2^{[r^2]}$ <br> $Q_1^1 \oplus Q_1^{[r^2],1}$ |
| $\mathbb{Z}_2^{r^2}$ | $1$ | $\mathbb{Z}_2$ | $\mathbb{Z}_2$ | $1$ | $\langle(r^2,m)\rangle$ | $Q_2^{[\text{id}]}$ <br> $Q_1^1 \oplus Q_1^{1r} \oplus Q_1^{1x} \oplus Q_1^{1xr}$ |
| $\mathbb{Z}_2^{x}$ | $1$ | $\mathbb{Z}_2$ | $\mathbb{Z}_2$ | $1$ | $\langle(x,m)\rangle$ | $Q_2^{[\text{id}]}$ <br> $Q_1^1 \oplus Q_1^{1x} \oplus Q_1^{E}$ |
| $\mathbb{Z}_2^{xr}$ | $1$ | $\mathbb{Z}_2$ | $\mathbb{Z}_2$ | $1$ | $\langle(xr,m)\rangle$ | $Q_2^{[\text{id}]}$ <br> $Q_1^1 \oplus Q_1^{1xr} \oplus Q_1^{E}$ |
| $\mathbb{Z}_2^{r^2} \times \mathbb{Z}_2^{xr}$ | $\mathbb{Z}_2^{r^2}$ | $\mathbb{Z}_2 \times \mathbb{Z}_2 \times \mathbb{Z}_2$ | $\mathbb{Z}_2$ | $1$ | $\langle(r^2,1),\ (xr,m)\rangle$ | $Q_2^{[\text{id}]} \oplus Q_2^{[r^2]}$ <br> $Q_1^1 \oplus Q_1^{1xr} \oplus Q_1^{[r^2],1} \oplus Q_1^{[r^2],1xr}$ |
| $\mathbb{Z}_2^{r^2} \times \mathbb{Z}_2^{xr}$ | $\mathbb{Z}_2^{xr}$ | $\mathbb{Z}_2 \times \mathbb{Z}_2 \times \mathbb{Z}_2$ | $\mathbb{Z}_2$ | $1$ | $\langle(r^2,m),\ (x,1)\rangle$ | $Q_2^{[\text{id}]} \oplus Q_2^{[xr]}$ <br> $Q_1^1 \oplus Q_1^{1xr} \oplus Q_1^{[xr],1++}$ |
| $\mathbb{Z}_2^{r^2} \times \mathbb{Z}_2^{x}$ | $\mathbb{Z}_2^{r^2}$ | $\mathbb{Z}_2 \times \mathbb{Z}_2 \times \mathbb{Z}_2$ | $\mathbb{Z}_2$ | $1$ | $\langle(r^2,1),\ (x,m)\rangle$ | $Q_2^{[\text{id}]} \oplus Q_2^{[r^2]}$ <br> $Q_1^1 \oplus Q_1^{1x} \oplus Q_1^{[r^2],1} \oplus Q_1^{[r^2],1x}$ |
| $\mathbb{Z}_2^{r^2} \times \mathbb{Z}_2^{x}$ | $\mathbb{Z}_2^{x}$ | $\mathbb{Z}_2 \times \mathbb{Z}_2 \times \mathbb{Z}_2$ | $\mathbb{Z}_2$ | $1$ | $\langle(r^2,m),\ (x,1)\rangle$ | $Q_2^{[\text{id}]} \oplus Q_2^{[x]}$ <br> $Q_1^1 \oplus Q_1^{1x} \oplus Q_1^{[x],1++}$ |
| $\mathbb{Z}_4^r$ | $\mathbb{Z}_2^{r^2}$ | $\mathbb{Z}_4$ | $\mathbb{Z}_2$ | $1$ | $\langle(r,m)\rangle$ | $Q_2^{[\text{id}]} \oplus Q_2^{[r^2]}$ <br> $Q_1^1 \oplus Q_1^{1r} \oplus Q_1^{[r^2],1} \oplus Q_1^{[r^2],1r}$ |
| $D_8$ | $\mathbb{Z}_4^r$ | $\mathbb{Z}_2 \times \mathbb{Z}_2 \times \mathbb{Z}_4$ | $\mathbb{Z}_2$ | $1$ | $\langle(r,1),\ (x,m)\rangle$ | $Q_2^{[\text{id}]} \oplus Q_2^{[r^2]} \oplus Q_2^{[r]}$ <br> $Q_1^1 \oplus Q_1^{[r^2],1} \oplus Q_1^{[r],1}$ |
| $D_8$ | $\mathbb{Z}_2^{r^2} \times \mathbb{Z}_2^{xr}$ | $\mathbb{Z}_2 \times \mathbb{Z}_2 \times \mathbb{Z}_4$ | $\mathbb{Z}_2$ | $1$ | $\langle(r,m),\ (x,1)\rangle$ | $Q_2^{[\text{id}]} \oplus Q_2^{[r^2]} \oplus Q_2^{[xr]}$ <br> $Q_1^1 \oplus Q_1^{[r^2],1} \oplus Q_1^{[xr],1++}$ |
| $D_8$ | $\mathbb{Z}_2^{r^2} \times \mathbb{Z}_2^{x}$ | $\mathbb{Z}_2 \times \mathbb{Z}_2 \times \mathbb{Z}_4$ | $\mathbb{Z}_2$ | $1$ | $\langle(r,m),\ (x,1)\rangle$ | $Q_2^{[\text{id}]} \oplus Q_2^{[r^2]} \oplus Q_2^{[x]}$ <br> $Q_1^1 \oplus Q_1^{[r^2],1} \oplus Q_1^{[x],1++}$ |

Table 8: Data for (2+1)d interfaces $\mathcal{I}_{H,N,\omega,\pi}$ from $\mathcal{Z}(2\mathsf{Vec}_{D_8})$ to the reduced TO $\mathcal{Z}(2\mathsf{Vec}_{H/N}^\pi)$, where $H \subset D_8$ (up to conjugation), $N \lhd H$, $\omega \in H^3(H,U(1))$ and $\pi \in H^4(H/N,U(1))$ such that the pullback of $\pi$ to $D_8$ is trivial: this condition implies that when $N = 1$ we must chose trivial $\pi$, but it can be non-trivial for $H = D_8, N = \mathbb{Z}_2^{r^2}$.

lines, we consider the projection:

$$\kappa: \qquad \mathsf{Rep}(D_8) \qquad \to \qquad \mathsf{Rep}(\mathbb{Z}_4)$$
$$(D_1^{1r}, D_1^{1x}, D_1^{1xr}, D_1^{E}) \mapsto (D_1^{1}, D_1^{e^2}, D_1^{e^2}, D_1^{e} \oplus D_1^{e^3}) \tag{6.20}$$

implying that the line $\boldsymbol{Q}_1^{1r}$ can end on this interface. The deconfined surfaces are labeled by $[g]$ for $g \in \mathbb{Z}_4^r$, in particular, $\boldsymbol{Q}_2^{[r]}$ splits on this interface into $\boldsymbol{Q}_2^{r} \oplus \boldsymbol{Q}_2^{r^3}$. The group $H^{\mathrm{diag}}$ (see table 8), describes the mapping of topological surfaces: $\boldsymbol{Q}_2^{r^p}$ becomes $\boldsymbol{Q}_2^{m^p}$ in DW($\mathbb{Z}_4$).

$$\mathcal{I}_{(\mathbb{Z}_4,1,1,\omega)}$$

| | | |
|---|---|---|
| $\boldsymbol{Q}_1^{1r}$ | $\mathcal{E}_0^{1r}$ | |
| $\boldsymbol{Q}_1^{1k}$ | $\mathcal{E}_0^{1k e^2}$ | $\boldsymbol{Q}_1^{e^2}$ | $k \in \{x, xr\}$ |
| $\boldsymbol{Q}_2^{[r^2]}$ | $\mathcal{E}_1^{[r^2]m^2}$ | $\boldsymbol{Q}_2^{m^2}$ |
| $\boldsymbol{Q}_1^{E}$ | $\mathcal{E}_0^{E e^q}$ | $\boldsymbol{Q}_1^{e^q}$ |
| $\boldsymbol{Q}_2^{[r]}$ | $\mathcal{E}_1^{[r] m^p}$ | $\boldsymbol{Q}_2^{m^p}$ | $p, q \in \{1,3\}$ |

$$\mathcal{Z}(2\mathsf{Vec}_{D_8}) \qquad\qquad \mathcal{Z}(2\mathsf{Vec}_{\mathbb{Z}_4})$$

(6.21)

There is also a choice of $\omega \in H^3(\mathbb{Z}_4, U(1)) = \mathbb{Z}_4$ that translates to the associators of the lines obtained from bulk surfaces intersecting the interface.

### 6.3.2 Interfaces to DW($\mathbb{Z}_2 \times \mathbb{Z}_2$)

Let us now discuss the classes of interfaces to DW($\mathbb{Z}_2 \times \mathbb{Z}_2$), whose topological defects we denote as:

$$\boldsymbol{Q}_1^{e_1^p e_2^q}, \qquad \boldsymbol{Q}_2^{m_1^p m_2^q}, \qquad p, q \in \{0,1\}. \tag{6.22}$$

$\boldsymbol{Q}_1^{1x}$ **Condensed Interface.** This class of interfaces is obtained by setting $H = \mathbb{Z}_2^{r^2} \times \mathbb{Z}_2^{x}$, $N = 1$. Since $N$ is trivial, no (non-identity) surfaces can end on this interface. For the topological lines, we consider the projection:

$$\kappa: \qquad \mathsf{Rep}(D_8) \qquad \to \qquad \mathsf{Rep}(\mathbb{Z}_2 \times \mathbb{Z}_2)$$
$$(D_1^{1r}, D_1^{1x}, D_1^{1xr}, D_1^{E}) \mapsto (D_1^{e_2}, D_1^{1}, D_1^{e_2}, D_1^{e_1} \oplus D_1^{e_1 e_2}) \tag{6.23}$$

implying that the line $\boldsymbol{Q}_1^{1x}$ is condensed on this interface. The deconfined surfaces are labeled by $g \in \mathbb{Z}_2^{r^2} \times \mathbb{Z}_2^{x}$, in particular, $\boldsymbol{Q}_2^{[x]}$ splits on this interface into $\boldsymbol{Q}_2^{x} \oplus \boldsymbol{Q}_2^{xr^2}$. The group $H^{\mathrm{diag}}$

(see table 8), describes the mapping of topological surfaces:

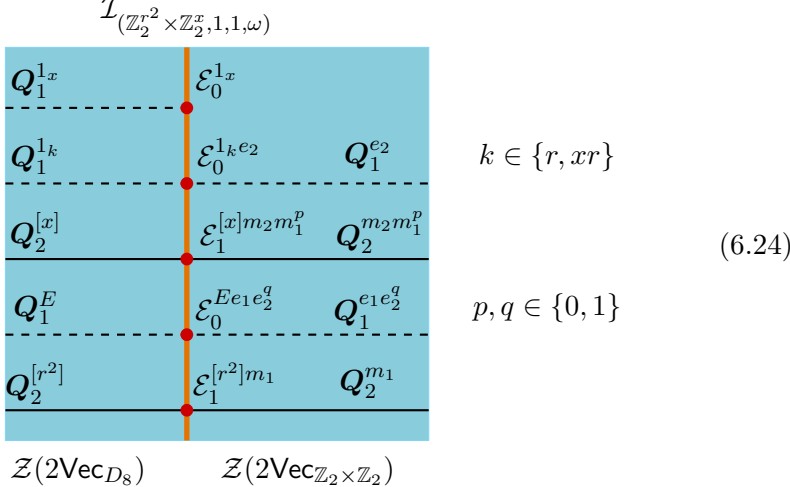

$$\mathcal{I}_{(\mathbb{Z}_2^{r^2} \times \mathbb{Z}_2^x, 1, 1, \omega)}$$

$\mathcal{Z}(2\mathsf{Vec}_{D_8})$    $\mathcal{Z}(2\mathsf{Vec}_{\mathbb{Z}_2 \times \mathbb{Z}_2})$

(6.24)

The choice of $\omega \in H^3(\mathbb{Z}_2 \times \mathbb{Z}_2, U(1)) = \mathbb{Z}_2^3$ translates to the associators of the lines obtained from bulk surfaces intersecting the interface.

$\boldsymbol{Q}_1^{1_{xr}}$ **Condensed Interface.** This is obtained by setting $H = \mathbb{Z}_2^{r^2} \times \mathbb{Z}_2^{xr}$, $N = 1$. Its properties are very similar to the previous case, and can be determined from it by performing the automorphism relating the following equivalent presentations of $D_8$:

$$\begin{aligned}
D_8 = \mathbb{Z}_4^r \rtimes \mathbb{Z}_2^x : & \quad \{\mathrm{id}, r, r^2, r^3, x, xr, xr^2, xr^3\}, \\
D_8 = \mathbb{Z}_4^r \rtimes \mathbb{Z}_2^{xr} : & \quad \{\mathrm{id}, r, r^2, r^3, xr, xr^2, xr^3, x\}.
\end{aligned}$$

(6.25)

$\boldsymbol{Q}_2^{[r^2]}$ **Condensed Interface.** By setting $H = D_8$, $N = \mathbb{Z}_2^{r^2}$ the surface $\boldsymbol{Q}_2^{r^2}$ can end on the interface since $N = \mathbb{Z}_2^{r^2}$, while no (non-identity) topological lines can end since $H = D_8$. $\boldsymbol{Q}_1^E$ is confined since it braids non-trivially with the condensed $\boldsymbol{Q}_2^{r^2}$. The deconfined surfaces are labeled by $g \in D_8/\mathbb{Z}_2^{r^2} \simeq \mathbb{Z}_2^r \times \mathbb{Z}_2^x$ and the group $H^{\mathrm{diag}}$ (see table 8), describes the mapping of topological surfaces. For $\mathrm{DW}(\mathbb{Z}_2 \times \mathbb{Z}_2)$ there is a possible non-trivial 4-cocycle twist $\pi \in H^4(\mathbb{Z}_2 \times \mathbb{Z}_2, U(1)) = \mathbb{Z}_2 \times \mathbb{Z}_2$, which trivializes when pulled-back to $D_8$, to which

we will return in section 6.4. The map of generalized charges is:

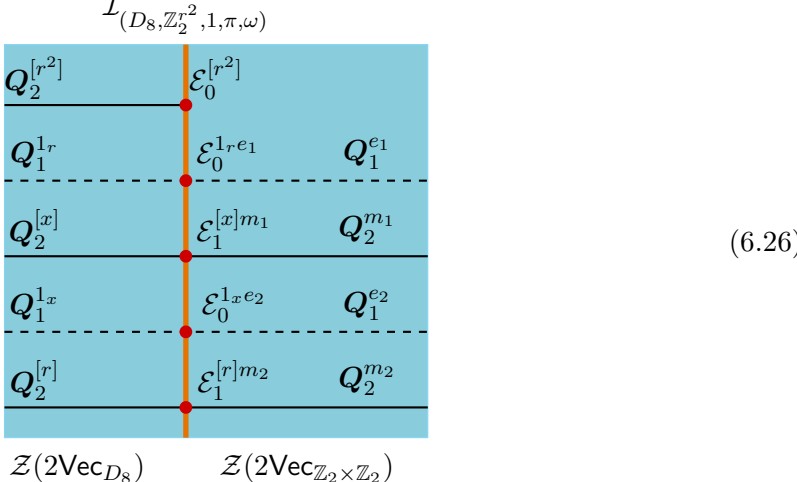

$$\text{(6.26)}$$

The choice of $\omega \in H^3(D_8, U(1)) = \mathbb{Z}_2 \times \mathbb{Z}_2 \times \mathbb{Z}_4$ translates to the associators of the lines obtained from bulk surfaces intersecting the interface.

### 6.3.3 Interfaces to $\mathrm{DW}(\mathbb{Z}_2)$

Let us now turn to the interfaces between $\mathrm{DW}(D_8)$ and $\mathrm{DW}(\mathbb{Z}_2)$, whose topological defects we denote as:

$$\boldsymbol{Q}_1^{e^p}, \qquad \boldsymbol{Q}_2^{m^p}, \qquad p \in \{0, 1\}. \qquad \text{(6.27)}$$

$\boldsymbol{Q}_1^{1r}, \boldsymbol{Q}_1^{1x}, \boldsymbol{Q}_1^{1xr}$ **Condensed Interface.** This is obtained by choosing $H = \mathbb{Z}_2^{r^2}$, $N = 1$. Since $N$ is trivial, no (non-identity) surfaces can end on this interface. For the topological lines, we consider the projection:

$$\begin{aligned}
\kappa: \quad \mathrm{Rep}(D_8) \quad &\to \quad \mathrm{Rep}(\mathbb{Z}_2) \\
(D_1^{1k}, \, D_1^E) &\mapsto (D_1^1, \, 2D_1^e), \qquad k \in \{r, x, xr\},
\end{aligned} \qquad \text{(6.28)}$$

implying that the line $\boldsymbol{Q}_1^{1k}$ for all 1-dim irreps can end on this interface. The surface $\boldsymbol{Q}_2^{[r^2]}$ is deconfined and becomes the magnetic surface $\boldsymbol{Q}_2^m$:

$$\text{(6.29)}$$

There is also a choice of $\omega \in H^3(\mathbb{Z}_2, U(1)) = \mathbb{Z}_2$ that translates to the associators of the line $\mathcal{E}_1^{[r^2]m}$ obtained from the bulk surface $\boldsymbol{Q}_2^{[r^2]}$ intersecting the interface.

$\boldsymbol{Q}_1^{1x}, \boldsymbol{Q}_1^{1E}$ **Condensed Interface.** This is obtained by choosing $H = \mathbb{Z}_2^x$, $N = 1$. Since $N$ is trivial, no (non-identity) surfaces can end on this interface. For the topological lines, we consider the projection:

$$\kappa : \qquad \mathsf{Rep}(D_8) \qquad \to \qquad \mathsf{Rep}(\mathbb{Z}_2)$$
$$(D_1^{1r}, D_1^{1x}, D_1^{1xr}, D_1^E) \mapsto (D_1^e, D_1^1, D_1^e, D_1^1 \oplus D_1^e) \tag{6.30}$$

implying that the lines $\boldsymbol{Q}_1^{1x}$ and $\boldsymbol{Q}_1^E$ can end on this interface. The surface $\boldsymbol{Q}_2^{[x]}$ is deconfined and becomes the magnetic surface $\boldsymbol{Q}_2^m$:

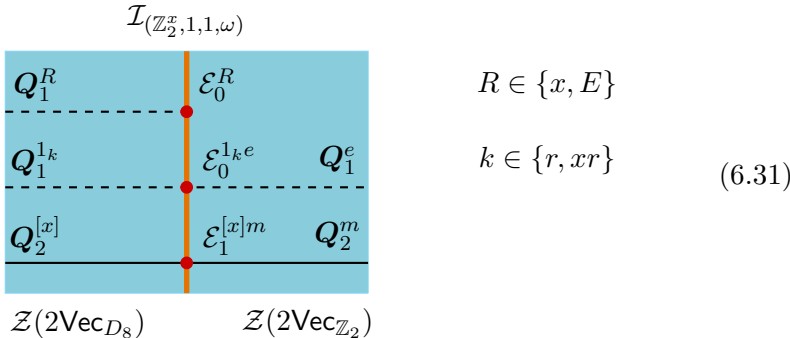

$$
\begin{array}{lll}
\boldsymbol{Q}_1^R & \mathcal{E}_0^R & R \in \{x, E\} \\
\boldsymbol{Q}_1^{1_k} & \mathcal{E}_0^{1_k e} \quad \boldsymbol{Q}_1^e & k \in \{r, xr\} \\
\boldsymbol{Q}_2^{[x]} & \mathcal{E}_1^{[x]m} \quad \boldsymbol{Q}_2^m &
\end{array}
\tag{6.31}
$$

There is also a choice of $\omega \in H^3(\mathbb{Z}_2, U(1)) = \mathbb{Z}_2$ that translates to the associators of the line $\mathcal{E}_1^{[x]m}$ obtained from the bulk surface $\boldsymbol{Q}_2^{[x]}$ intersecting the interface.

$\boldsymbol{Q}_1^{1xr}, \boldsymbol{Q}_1^{1E}$ **Condensed Interface.** This is obtained by setting $H = \mathbb{Z}_2^{xr}$ and $N = 1$. Its properties are very similar to the previous case, and can be determined from it by using the automorphism relating the two presentations of $D_8$ in equation (6.25).

$\boldsymbol{Q}_1^{1x}, \boldsymbol{Q}_2^{[r^2]}$ **Condensed Interface.** Setting $H = \mathbb{Z}_2^{r^2} \times \mathbb{Z}_2^x$, $N = \mathbb{Z}_2^{r^2}$, the topological surface $\boldsymbol{Q}_2^{[r^2]}$ can end on the interface. For the topological lines, we consider the same projection as (6.23) (since it depends only on $H$), implying that the line $\boldsymbol{Q}_1^{1x}$ can end on this interface. The

surface $\boldsymbol{Q}_2^{[x]}$ is deconfined and becomes the magnetic surface $\boldsymbol{Q}_2^m$:

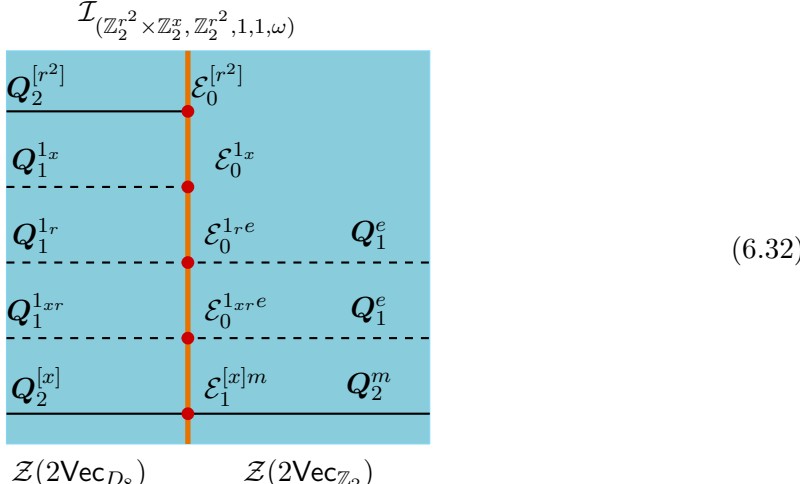

$$\mathcal{I}_{(\mathbb{Z}_2^{r^2} \times \mathbb{Z}_2^x,\, \mathbb{Z}_2^{r^2},\, 1,\, 1,\, \omega)}$$

There is also a choice of $\omega \in H^3(\mathbb{Z}_2 \times \mathbb{Z}_2\, U(1))$ that translates to the associators of the lines obtained from the bulk surfaces intersecting the interface.

$\boldsymbol{Q}_1^{1_{xr}}, \boldsymbol{Q}_2^{[r^2]}$ **Condensed Interface.** Its properties are very similar to the previous case, and can be determined from it by using the automorphism relating the two presentations of $D_8$ in equation (6.25).

$\boldsymbol{Q}_1^{1_x}, \boldsymbol{Q}_2^{[x]}$ **Condensed Interface.** Setting $H = \mathbb{Z}_2^{r^2} \times \mathbb{Z}_2^x$, $N = \mathbb{Z}_2^x$, the topological surface $\boldsymbol{Q}_2^{[x]}$ can end on the interface. For the topological lines, we consider the same projection as (6.23) (since it depends only on $H$), implying that the line $\boldsymbol{Q}_1^{1_x}$ can end on this interface. The surface $\boldsymbol{Q}_2^{[r^2]}$ is deconfined and becomes the magnetic surface $\boldsymbol{Q}_2^m$:

$$\mathcal{I}_{(\mathbb{Z}_2^{r^2} \times \mathbb{Z}_2^x,\, \mathbb{Z}_2^x,\, 1,\, \omega)}$$

| $\boldsymbol{Q}_2^{[x]}$ | $\mathcal{E}_0^{[r^2]}$ | |
| $\boldsymbol{Q}_1^{1_x}$ | $\mathcal{E}_0^{1_x}$ | |
| $\boldsymbol{Q}_1^E$ | $\mathcal{E}_0^{Ee}$ | $2\boldsymbol{Q}_1^e$ |
| $\boldsymbol{Q}_2^{[r^2]}$ | $\mathcal{E}_1^{[r^2]m}$ | $\boldsymbol{Q}_2^m$ |

$$\mathcal{Z}(2\mathsf{Vec}_{D_8}) \qquad \mathcal{Z}(2\mathsf{Vec}_{\mathbb{Z}_2})$$

(6.33)

There is also a choice of $\omega \in H^3(\mathbb{Z}_2 \times \mathbb{Z}_2\, U(1))$ that translates to the associators of the lines obtained from the bulk surfaces intersecting the interface.

$\boldsymbol{Q}_1^{1xr}, \boldsymbol{Q}_2^{[xr]}$ **Condensed Interface.** Its properties are very similar to the previous case, and can be determined from it by using the automorphism relating the two presentations of $D_8$ in equation (6.25).

$\boldsymbol{Q}_1^{1r}, \boldsymbol{Q}_2^{[r^2]}$ **Condensed Interface.** Setting $H = \mathbb{Z}_4^r$, $N = \mathbb{Z}_2^{r^2}$, the topological surface $\boldsymbol{Q}_2^{[r^2]}$ can end on the interface. For the topological lines, we consider the same projection as (6.20) (since it depends only on $H$), implying that the line $\boldsymbol{Q}_1^{1r}$ can end on this interface. The surface $\boldsymbol{Q}_2^{[r]}$ is deconfined and becomes the magnetic surface $\boldsymbol{Q}_2^m$:

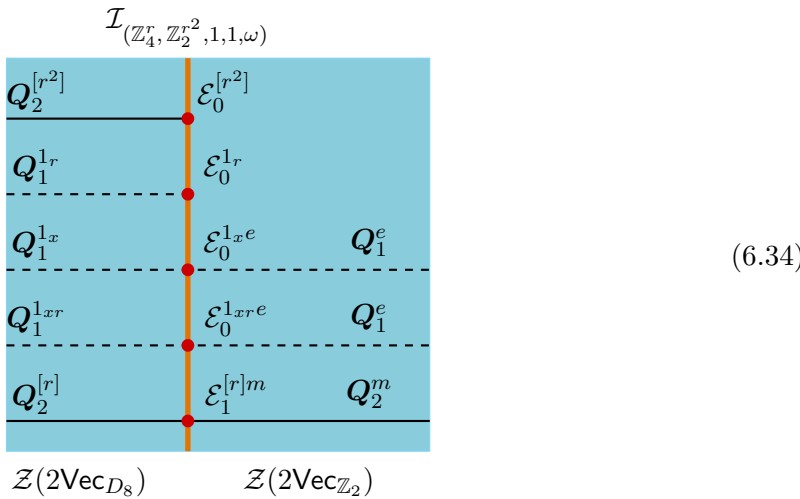

$$\tag{6.34}$$

There is also a choice of $\omega \in H^3(\mathbb{Z}_4, U(1)) = \mathbb{Z}_4$ that translates to the associators of the lines obtained from the bulk surfaces intersecting the interface.

$\boldsymbol{Q}_2^{[r]}, \boldsymbol{Q}_2^{[r^2]}$ **Condensed Interface.** By setting $H = D_8$, $N = \mathbb{Z}_4^r$ the surfaces $\boldsymbol{Q}_2^{[r]}$ and $\boldsymbol{Q}_2^{[r^2]}$ can end on the interface, while no (non-identity) topological lines can end since $H = D_8$. $\boldsymbol{Q}_1^E$ is confined since it braids non-trivially with the condensed $\boldsymbol{Q}_2^{r^2}$, and so are $\boldsymbol{Q}_1^{1x}$ and $\boldsymbol{Q}_1^{1xr}$ since they braid with $\boldsymbol{Q}_2^{[r]}$.

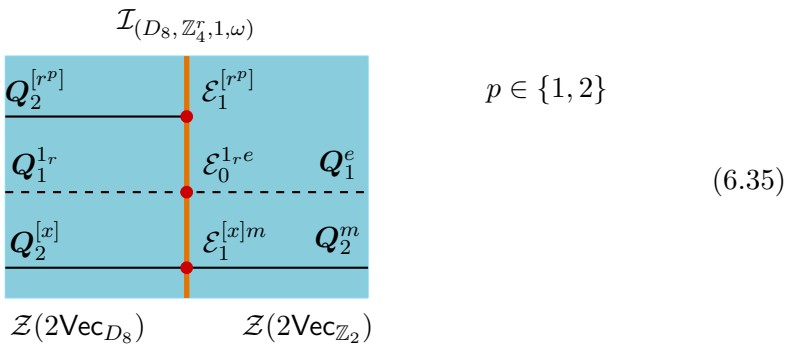

$$\tag{6.35}$$

There is also a choice of $\omega \in H^3(D_8, U(1))$ that translates to the associators of the lines obtained from the bulk surfaces intersecting the interface.

$\boldsymbol{Q}_2^{[x]}$,$\boldsymbol{Q}_2^{[r^2]}$ **Condensed Interface.** By setting $H = D_8$, $N = \mathbb{Z}_2^{r^2} \times \mathbb{Z}_2^x$ the surfaces $\boldsymbol{Q}_2^{[x]}$ and $\boldsymbol{Q}_2^{[r^2]}$ can end on the interface, while no (non-identity) topological lines can end since $H = D_8$. $\boldsymbol{Q}_1^E$ is confined since it braids non-trivially with the condensed $\boldsymbol{Q}_2^{r2}$, and so are $\boldsymbol{Q}_1^{1r}$ and $\boldsymbol{Q}_1^{1xr}$ since they braid with $\boldsymbol{Q}_2^{[x]}$.

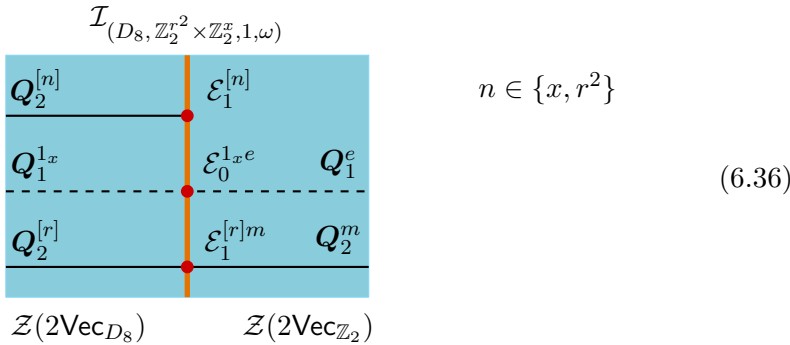

$$n \in \{x, r^2\}$$

$$\mathcal{Z}(2\mathsf{Vec}_{D_8}) \qquad\qquad \mathcal{Z}(2\mathsf{Vec}_{\mathbb{Z}_2})$$

(6.36)

There is also a choice of $\omega \in H^3(D_8, U(1))$ that translates to the associators of the lines obtained from the bulk surfaces intersecting the interface.

$\boldsymbol{Q}_2^{[xr]}$,$\boldsymbol{Q}_2^{[r^2]}$ **Condensed Interface.** Its properties are very similar to the previous case, and can be determined from it by using the automorphism relating the two presentations of $D_8$ in equation (6.25).

## 6.4 Gapless Phases

We can now consider the club sandwiches, obtained by inserting various $\mathfrak{B}_{\mathrm{sym}}$ on the club quiches from the last section, and choosing specific gapless phases (for a symmetry realized by the reduced TO) as $\mathfrak{B}_{\mathrm{phys}}$. We will highlight some interesting phases, such as igSPTs and igSSBs.

### 6.4.1 igSPT for $2\mathsf{Vec}_{D_8}$

By setting $\mathfrak{B}_{\mathrm{sym}} = \mathfrak{B}_{\mathrm{Dir}}$, we fix the symmetry to be $2\mathsf{Vec}_{D_8}$, i.e. $D_8$ 0-form symmetry, as reviewed in section 6.2.1.

If we fix $G = H = D_8$ and $N = \mathbb{Z}_2^{r^2}$, the reduced topological order $\mathcal{Z}(2\mathsf{Vec}_{\mathbb{Z}_2 \times \mathbb{Z}_2})$ can have a non-trivial 4-cocycle twist $\pi \in H^4(\mathbb{Z}_2 \times \mathbb{Z}_2, U(1)) = \mathbb{Z}_2 \times \mathbb{Z}_2$. To construct a representative of $\pi$ whose pullback to $D_8$ is trivial, we follow the approach described on page 26, by writing

$\pi = \eta \cup e$, for $\eta \in H^2(H/N, \widehat{N})$ and $e \in H^2(H/N, N)$. We choose $\eta \in H^2(\mathbb{Z}_2^{m_1} \times \mathbb{Z}_2^{m_2}, \mathbb{Z}_2^{\widehat{r}^2})$ to be:

$$\eta(m_1^{p_1} m_2^{p_2}, m_1^{q_1} m_2^{q_2} \mid \widehat{r}^2) = (-1)^{p_2 q_1}, \tag{6.37}$$

and $+1$ otherwise. For the section $s : \mathbb{Z}_2 \times \mathbb{Z}_2 \to D_8$, we pick:

$$s(m_1) = x, \quad s(m_2) = r, \quad s(m_1 m_2) = xr^3, \tag{6.38}$$

with representative $e \in H^2(\mathbb{Z}_2^{m_1} \times \mathbb{Z}_2^{m_2}, \mathbb{Z}_2^{r^2})$ of the extension class given by equation (2.33), whose non-trivial values are:

$$\begin{aligned} e(m_1, m_2) = r^2, & \qquad e(m_1, m_1 m_2) = r^2, \\ e(m_2, m_2) = r^2, & \qquad e(m_2, m_1 m_2) = r^2. \end{aligned} \tag{6.39}$$

Therefore $\pi = \eta \cup e \in H^4(\mathbb{Z}_2^{m_1} \times \mathbb{Z}_2^{m_2}, U(1))$ is given by:

$$\pi(m_1^{a_1} m_2^{a_2}, m_1^{b_1} m_2^{b_2}, m_1^{c_1} m_2^{c_2}, m_1^{d_1} m_2^{d_2}) = (-1)^{a_2 b_1 c_1 d_2 + a_2 b_1 c_2 d_2} \tag{6.40}$$

which we checked to be a representative of a non-trivial cohomology class in $H^4(\mathbb{Z}_2 \times \mathbb{Z}_2, U(1))$.

**Club Quiche from $Q_2^{[r^2]}$ Condensed Interface.** Let us consider the club quiche corresponding to the $Q_2^{[r^2]}$ condensed interface, see (6.26). This produces a boundary

$$\mathfrak{B}' = \mathfrak{B}_{e_1, e_2}, \tag{6.41}$$

with symmetry $\mathcal{S}' = 2\mathsf{Vec}_{\mathbb{Z}_2 \times \mathbb{Z}_2}^\pi$. We may input a $\mathbb{Z}_2 \times \mathbb{Z}_2$ 0-form symmetric theory with anomaly $\pi \in H^4(\mathbb{Z}_2 \times \mathbb{Z}_2)$, denoted by $\mathfrak{T}_{\mathbb{Z}_2 \times \mathbb{Z}_2}^\pi$, on the physical boundary. The $D_8$ symmetry is realized on this theory via the projection

$$\begin{aligned} 2\mathsf{Vec}_{D_8} &\longrightarrow 2\mathsf{Vec}_{\mathbb{Z}_2 \times \mathbb{Z}_2}^\pi, \\ (D_2^{r^{2p}}, D_2^{xr^{2p}}, D_2^{r^{2p+1}}, D_2^{xr^{2p+1}}) &\longmapsto (D_2^{\mathrm{id}}, D_2^{m_1}, D_2^{m_2}, D_2^{m_1 m_2}), \qquad p \in \{0, 1\} \end{aligned} \tag{6.42}$$

whose kernel is $2\mathsf{Vec}_{\mathbb{Z}_2^{r^2}}$ and the $D_8$ 0-form symmetry is realized as

$$\mathfrak{T}_{\mathbb{Z}_2 \times \mathbb{Z}_2}^\pi \,\circlearrowleft\, 2\mathsf{Vec}_{\mathbb{Z}_2 \times \mathbb{Z}_2}^\pi \longleftarrow 2\mathsf{Vec}_{D_8}. \tag{6.43}$$

The gapless phase resulting from $\mathcal{I}_{(D_8, \mathbb{Z}_2^{r^2}, \pi, \omega)}$ preserves the full $D_8$. For non-trivial $\pi$, equation (6.40), it is intrinsically gapless since if we gap the theory, $D_8$ must get spontaneously broken to a subgroup $H' < D_8$ such that $\pi|_{H'/\mathbb{Z}_2^{r^2}} = 1$. From table 8 we see that this holds for all proper subgroups $H' < D_8$ so this phase can be gapped to all $D_8$ gapped phases in which $Q_2^{[r^2]}$ is condensed (since it was already condensed on $\mathcal{I}_{(D_8, \mathbb{Z}_2^{r^2}, \pi, \omega)}$), except for the $D_8$ SPTs. This is shown in figure 7, where, for simplicity, we have set all 3-cocycles to the identity. It would be interesting to determine the input transitions for these igSPTs, which are $\mathbb{Z}_2 \times \mathbb{Z}_2$ symmetric gapless phases, with specific 4-cocycles.

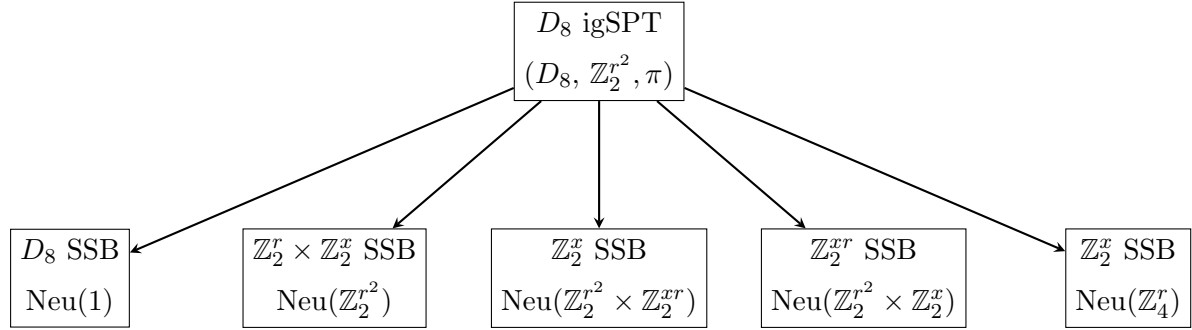

Figure 7: Possible deformations of the $D_8$ igSPT corresponding to $H = D_8$, $N = \mathbb{Z}_2^{r^2}$ and non-trivial $\pi \in H^4(\mathbb{Z}_2 \times \mathbb{Z}_2, U(1))$. For maximal algebras (bottom layer) we show the boundary condition $\text{Neu}(H')$, where, for simplicity, we set all 3-cocycles to be trivial. The igSPT can only be deformed to gapped SSB phases for (subgroups of) $D_8$.

### 6.4.2 $D_8$-symmetric Phase Transitions

These can be computed analogously to the $S_3$ example of section 5. Table 1 summarizes the results for transitions between two $D_8$ gapped phases: these are obtained from $D_8$ club quiches with reduced TO $\mathcal{Z}(2\text{Vec}_{\mathbb{Z}_2})$, by performing a KT transformation taking the input CFT to be 3d Ising. If the resulting gapless has a single copy of Ising it is a gSPT, otherwise it is gSSB for $D_8$ 0-form symmetry. Each gapless universe preserves a subgroup of $D_8$ ismororphic to $H$ while the universes are permuted by representatives of the other $H$-cosets of $D_8$. gSPTs thus correspond to phases for which $H = D_8$.

### 6.4.3 $2\text{Rep}(\mathbb{Z}_4^{(1)} \rtimes \mathbb{Z}_2^{(0)})$-symmetric Phase Transitions

By setting $\mathfrak{B}_{\text{sym}} = \mathfrak{B}_{\text{Neu}(\mathbb{Z}_2^x)}$, we fix the symmetry to be $2\text{Rep}(\mathbb{G})$ where $\mathbb{G} = \mathbb{Z}_4^{(1)} \rtimes \mathbb{Z}_2^{(0)}$, i.e. a 2-representation of 2-group symmetry that can be obtained from $D_8$ 0-form symmetry by gauging the non normal $\mathbb{Z}_2^x$, as reviewed in section 6.2.2.

We can thus determine the gapless phases corresponding to transitions between two $2\text{Rep}(\mathbb{G})$-symmetric gapped phases by gauging the $\mathbb{Z}_2^x$ symmetry of the $D_8$ CFTs of table 1. If $N = \mathbb{Z}_2^x$, after gauging we get a $\text{DW}(\mathbb{Z}_2)$ stacking: its charged line comes from the bulk $\boldsymbol{Q}_2^{[x]}$ ending on both $\mathfrak{B}_{\text{sym}}$ and $\mathfrak{B}_{\text{phys}}$. If $\mathbb{Z}_2^x$ is a subset of $H$ but not of $N$, we have to gauge the $\mathbb{Z}_2$ 0-form symmetry of 3d Ising: the resulting $\frac{\text{Ising}}{\mathbb{Z}_2^{(0)}}$ theory realizes a transition between $\mathbb{Z}_2^{(1)}$ trivial and $\mathbb{Z}_2^{(1)}$ SSB phases. If, finally, $\mathbb{Z}_2^x$ is not a subgroup of $H$, then we have to combine the gapless universes into $\mathbb{Z}_2^x$ orbits: upon gauging $\mathbb{Z}_2^x$ we obtain a single Ising from each pair that forms a $\mathbb{Z}_2^x$ orbit in the $D_8$ CFT. The results are summarized in table 2. We note that the transition between the $2\text{Rep}(\mathbb{G})/\mathbb{Z}_2^0$ SSB and the $2\text{Rep}(\mathbb{G})$ SSB phase is an igSSB for the 1-form sym-

metry: the number of $2\mathsf{Rep}(\mathbb{G})$ charged lines (obtained from bulk SymTFT surfaces ending on both $\mathfrak{B}_{\text{sym}}$ and $\mathfrak{B}_{\text{phys}}$) strictly increases when we gap the theory, as can be seen from the algebras in tables 7 and 8.

### 6.4.4  $2\mathsf{Rep}(D_8)$-symmetric Phase Transitions

By setting $\mathfrak{B}_{\text{sym}} = \mathfrak{B}_{\text{Neu}(D_8)}$, we fix the symmetry to be $2\mathsf{Rep}(D_8)$, i.e. $\mathsf{Rep}(D_8)$ 1-form symmetry, as reviewed in section 6.2.3. We can obtain the gapless phases corresponding to transitions between two $2\mathsf{Rep}(D_8)$-symmetric gapped phases by gauging the full $D_8$ 0-form symmetry of the $D_8$ CFTs of table 1. The SymTFT bulk surfaces labeled by $N \subset D_8$ give rise to genuine charged lines in a 3d $N$-Dijkgraaf Witten theory $\text{DW}(N)$ in each universe of the CFT: we then gauge the diagonal $\mathbb{Z}_2^{(0),\text{diag}}$ of the $\mathbb{Z}_2^x$ group automorphism in $\text{DW}(N)$ and the $\mathbb{Z}_2$ 0-form symmetry of 3d $\mathsf{Ising}$, similarly to the cases described in section 5. Note that for $H = \mathbb{Z}_2$ there are no group automorphism, so we only gauge the 0-form symmetry of $\mathsf{Ising}$. The results are summarized in table 3. We note that the bottom three phases, with gauged $\mathbb{Z}_2^{(0),\text{diag}}$, are igSSBs for the 1-form symmetry: the number of $2\mathsf{Rep}(D_8)$ charged lines (obtained from bulk SymTFT surfaces ending on both $\mathfrak{B}_{\text{sym}}$ and $\mathfrak{B}_{\text{phys}}$) strictly increases when we gap the theory, as can be seen from the algebras in tables 8 and 7.

### Acknowledgements

We thank Andrea Antinucci, David Hofmeier, Daniel Pajer, Du Pei, Andreas Stergiou, Devon Stockall and Jingxiang Wu for discussions. We thank Rui Wen for coordinating submission with his upcoming work [86]. SSN thanks the KITP at UCSB, for hospitality during the completion of this work. LB is funded as a Royal Society University Research Fellow through grant URF\R1\231467. The work of YG, S-JH, SSN and AW is supported by the UKRI Frontier Research Grant, underwriting the ERC Advanced Grant "Generalized Symmetries in Quantum Field Theory and Quantum Gravity". KI and SSN are supported in part by the EPSRC Open Fellowship (Schafer-Nameki) EP/X01276X/1. KI also acknowledges support through the Leverhulme-Peierls Fellowship. The work of AT is funded by Villum Fonden Grant no. VIL60714.

## A  Partial Ordering on Minimal Interfaces

This partial ordering can be seen from the corresponding condensable algebras [32], where the condensable algebra associated to the interface $\mathcal{I}_{(H,N,\pi,\omega)}$ is the $G$-crossed braided multifusion

category

$$\mathrm{Fun}_H(G, \mathsf{Vec}_N^{\omega|_N})\,, \tag{A.1}$$

with the crossed braided structure determined by $\omega$ and $\pi$ [25]. The desired partial orderings takes two steps

$$\mathcal{I}_{(H_1, N_1, \pi_1, \omega_1,)} < \mathcal{I}_{(H_2, N_1, \widetilde{\pi}, \omega_2)} < \mathcal{I}_{(H_2, N_2, \pi_2, \omega_2)}, \tag{A.3}$$

for suitable $\widetilde{\pi} \in H^4(H_2/N_1, U(1))$, which will be explicit in the following. For the partial ordering to hold true, the $G$-crossed braided structure on the condensable algebras should be compatible. In the first step, this is guaranteed by the requirement that $\omega_2 = \omega_1|_{H_2}$ and $\widetilde{\pi} = \pi_1|_{H_2/N_1}$[26], under which there is the inclusion

$$\mathrm{Fun}_{H_1}(G, \mathsf{Vec}_{N_1}^{\omega_1|_{N_1}}) \hookrightarrow \mathrm{Fun}_{H_2}(G, \mathsf{Vec}_{N_1}^{\omega_2|_{N_1}})$$
$$F \mapsto F\,. \tag{A.4}$$

Similarly, for the condensable algebra associated to $\mathcal{I}_{(H_2, N_1, \widetilde{\pi}, \omega_2)}$ to be a subalgebra of that associated to $\mathcal{I}_{(H_2, N_2, \pi_2, \omega_2)}$, the requirement $\widetilde{\pi} = p^* \pi_2$ guarantees the natural inclusion

$$\mathrm{Fun}_{H_2}(G, \mathsf{Vec}_{N_1}^{\omega_2|_{N_1}}) \hookrightarrow \mathrm{Fun}_{H_2}(G, \mathsf{Vec}_{N_2}^{\omega_2|_{N_2}})$$
$$F \mapsto F\,. \tag{A.5}$$

# B  Gapless Phases from Abelian DW Theories

In this section, we study phase transitions for symmetries that arise as gapped boundary conditions of DW theories for an abelian group $\mathbb{A}$ and trivial topological action in $H^4(\mathbb{A}, U(1))$. The minimal symmetries are generically 2-groups with mixed anomalies, and the non-minimal cases are obtained by stacking the Dirichlet BC with an $\mathbb{A}$-symmetric TQFT and gauging a subgroup $\mathbb{B} < \mathbb{A}$. The gapped BCs and gapped phases were completely classified in [1]. Here we will extend this analysis to gapless phases and phase-transitions in the presence of such symmetries.

## B.1  The SymTFT

Let $\mathbb{A}$ be a finite abelian group, and let us consider the SymTFT for the 0-form symmetry $\mathbb{A}$. Its topological defects are given in terms of the Drinfeld center $\mathcal{Z}(2\mathsf{Vec}_{\mathbb{A}})$, whose non-

---

[25]There is a technical subtlety that we suppressed in the main text, being the monoidal functor specifying the $H$ action $\rho : H \to \mathrm{Aut}^{\mathrm{br}}(\mathsf{Vec})$ on the trivially graded component of $\mathsf{Vec}_N^{\omega|_N}$, and technically we also require $H_1$ and $H_2$ to have compatible actions for the partial ordering to hold true, i.e., $\rho_i : H_i \to \mathrm{Aut}^{\mathrm{br}}(\mathsf{Vec})$ for $i = 1, 2$ are such that

$$\rho_2 = \rho_1 \circ \iota, \tag{A.2}$$

where $\iota : H_2 \hookrightarrow H_1$.

[26]The compatible $H_i$ action $\rho_2 = \rho_1 \circ \iota$ also plays a role here.

| BC | Symmetry | topological defects that end | SymTFT Quiche |
|---|---|---|---|
| $\mathfrak{B}_{\mathrm{Dir}}$ | $2\mathsf{Vec}_{\mathbb{A}}$ | $\begin{cases} \boldsymbol{Q}_2^{\mathrm{id}}, \\ \boldsymbol{Q}_1^{\widehat{a}}, & \widehat{a} \in \widehat{\mathbb{A}}. \end{cases}$ | $\mathfrak{B}_{\mathrm{Dir}}$    $\boldsymbol{Q}_1^{\widehat{a}\in\widehat{\mathbb{A}}}$    $\mathcal{E}_0^{\widehat{a}}$ |
| $\mathfrak{B}_{\mathrm{Neu},\nu}$ | $\mathsf{Rep}(\mathbb{A}) = \widehat{\mathbb{A}}$ | $\begin{cases} \boldsymbol{Q}_2^{a}, & a \in \mathbb{A} \\ \boldsymbol{Q}_1^{\mathrm{id}}. \end{cases}$ | $\mathfrak{B}_{\mathrm{Neu},\nu}$    $\boldsymbol{Q}_2^{a\in\mathbb{A}}$    $\mathcal{E}_1^{a}$ |
| $\mathfrak{B}_{\mathrm{Neu}(\mathbb{B}),\nu}$ | $2\mathsf{Vec}^{\beta}_{\mathbb{G}^{(2)}}$ | $\begin{cases} \boldsymbol{Q}_2^{b}, & b \in \mathbb{B} \\ \boldsymbol{Q}_1^{\widehat{a}}, & \widehat{a} \in \widehat{\mathbb{A}/\mathbb{B}}. \end{cases}$ | $\mathfrak{B}_{\mathrm{Neu}(\mathbb{B}),\nu}$    $\boldsymbol{Q}_2^{b\in\mathbb{B}}$    $\mathcal{E}_1^{b}$    $\boldsymbol{Q}_1^{\widehat{a}\in\widehat{\mathbb{A}/\mathbb{B}}}$    $\mathcal{E}_0^{\widehat{a}}$ |

Table 9: Summary of minimal gapped BC for $\mathcal{Z}(2\mathsf{Vec}_{\mathbb{A}})$. Here $\nu$ denotes the discrete torsion, $\mathbb{B} < \mathbb{A}$ is a subgroup, $\beta$ is defined by equation (B.15), $\widehat{a} \in \widehat{\mathbb{A}/\mathbb{B}}$ is identified with $\widehat{a} \in \widehat{\mathbb{A}}$ such that $\widehat{a}|_{\mathbb{B}} = 1$, and $\mathbb{G}^{(2)} = (\mathbb{A}/\mathbb{B})^{(0)} \times \mathbb{B}^{(1)}$ is a two-group.

condensation defects are

$$\{\boldsymbol{Q}_2^a, \quad a \in \mathbb{A}\} \quad \text{and} \quad \{\boldsymbol{Q}_1^{\widehat{a}}, \quad \widehat{a} \in \widehat{\mathbb{A}}\}. \tag{B.1}$$

where $\widehat{\mathbb{A}} = \mathrm{Hom}(\mathbb{A}, U(1))$ denotes the Pontryagin dual (or the group of multiplicative characters) of the abelian group $\mathbb{A}$. We have simplified our general notation The surface and line defects braid with

$$B(\boldsymbol{Q}_2^a, \boldsymbol{Q}_1^{\widehat{a}}) = \widehat{a}(a). \tag{B.2}$$

## B.2 Symmetry Boundaries

Both the minimal and the non-minimal gapped boundary conditions for abelian DW theories for a finite abelian group $\mathbb{A}$ were studied in detail in [2], which we briefly summarize here.

**Dirichlet** $\mathfrak{B}_{\mathbf{Dir}}$. The topological defects that can end on the Dirichlet boundary $\mathfrak{B}_{\mathrm{Dir}}$ are the identity surface $\boldsymbol{Q}_2^{\mathrm{id}}$, and all the topological lines $\boldsymbol{Q}_1^{\widehat{a}}$, labeled by $\widehat{a} \in \widehat{A}$. The Dirichlet BC realizes the symmetry $2\mathsf{Vec}_{\mathbb{A}}$, with symmetry generators $D_2^a$, $a \in \mathbb{A}$. Specifically, on $\mathfrak{B}_{\mathrm{Dir}}$, the

bulk surface operator $\boldsymbol{Q}_2^a$ ends on a line $\mathcal{E}_1^a$ attached to the $D_2^a$ symmetry defect. We denote this as

$$\boldsymbol{Q}_2^a\bigg|_{\mathrm{Dir}} = (D_2^a, \mathcal{E}_1^1)\,. \tag{B.3}$$

Similarly, the bulk line $\boldsymbol{Q}_1^{\widehat{a}}$ ends on an untwisted local operator

$$\boldsymbol{Q}_1^{\widehat{a}}\bigg|_{\mathrm{Dir}} = \mathcal{E}_0^{\widehat{a}}\,, \tag{B.4}$$

which are charged under the symmetry $2\mathsf{Vec}_{\mathbb{A}}$ due to the bulk braiding (B.2)

$$B(D_2^a, \mathcal{E}_0^{\widehat{a}}) = B(\boldsymbol{Q}_2^a, \boldsymbol{Q}_1^{\widehat{a}}) = \widehat{a}(a)\,. \tag{B.5}$$

**Neumann with $\nu$.** The Neumann boundary condition $\mathfrak{B}_{\mathrm{Neu},\nu}$ is obtained by gauging the $2\mathsf{Vec}_{\mathbb{A}}$ symmetry on $\mathfrak{B}_{\mathrm{Dir}}$ with the discrete torsion $\nu \in H^3(\mathbb{A}, U(1))$. The line $\mathcal{E}_1^a$, in the twisted sector of $D_2^a$ on $\mathfrak{B}_{\mathrm{Dir}}$ is gauged to the untwisted sector, hence all the defect surface $\boldsymbol{Q}_2^a$ for $a \in \mathbb{A}$, can end on non-genuine lines $\mathcal{E}_1^a$ on $\mathfrak{B}_{\mathrm{Neu},\nu}$,

$$\boldsymbol{Q}_2^a\bigg|_{\mathrm{Neu},\nu} = \mathcal{E}_1^a\,. \tag{B.6}$$

Note that these lines $\mathcal{E}_1^a$ are not unique, in fact there are $|\mathbb{A}|$ lines from fusing with the end of $\boldsymbol{Q}_2^{\mathrm{id}}$, which are genuine lines that form $\mathsf{Rep}(\mathbb{A})$. These non-genuine lines have $F$-symbols determined by the discrete torsion

$$F(\mathcal{E}_1^{a_1}, \mathcal{E}_1^{a_2}, \mathcal{E}_1^{a_3}) = \nu(a_1, a_2, a_3)\,. \tag{B.7}$$

On the other hand, the charged operators $\mathcal{E}_0^{\widehat{a}}$ on $\mathfrak{B}_{\mathrm{Dir}}$, $\widehat{a} \in \widehat{\mathbb{A}}$, are gauged into twisted sectors hence the lines $\boldsymbol{Q}_1^{\widehat{a}}$ do not end on $\mathfrak{B}_{\mathrm{Neu},\nu}$

$$\boldsymbol{Q}_1^{\widehat{a}}\bigg|_{\mathrm{Neu},\nu} = (D_1^{\widehat{a}}, \mathcal{E}_0^{\widehat{a}})\,, \tag{B.8}$$

to form the generators $D_1^{\widehat{a}}$ of the symmetry $\mathsf{Rep}(\mathbb{A})$ on $\mathfrak{B}_{\mathrm{Neu},\nu}$. The non-genuine lines are charged non-trivially under the symmetry $\mathsf{Rep}(\mathbb{A})$ due to the bulk braiding (B.2)

$$B(\mathcal{E}_1^a, D_1^{\widehat{a}}) = B(\boldsymbol{Q}_2^a, \boldsymbol{Q}_1^{\widehat{a}}) = \widehat{a}(a)\,. \tag{B.9}$$

**Partial Neumann** $(\mathbb{B}, \nu)$. The partial Neumann boundary condition $\mathfrak{B}_{\mathrm{Neu}(\mathbb{B}),\nu}$ is obtained by gauging a subgroup $\mathbb{B} < \mathbb{A}$ symmetry with a discrete torsion $\nu \in H^3(\mathbb{B}, U(1))$ on $\mathfrak{B}_{\mathrm{Dir}}$. After gauging $\mathbb{B} < \mathbb{A}$, a surface defect $\boldsymbol{Q}_2^b$ labeled by $b \in \mathbb{B}$ can end on a non-genuine line $\mathcal{E}_1^b$ on the boundary $\mathfrak{B}_{\mathrm{Neu}(\mathbb{B}),\nu}$

$$\boldsymbol{Q}_2^b \Big|_{\mathrm{Neu}(\mathbb{B}),\nu} = \mathcal{E}_1^b . \tag{B.10}$$

The $F$-symbols for these lines are determined by $\nu$

$$F(\mathcal{E}_1^{b_1}, \mathcal{E}_1^{b_2}, \mathcal{E}_1^{b_3}) = \nu(b_1, b_2, b_3) . \tag{B.11}$$

After this gauging, there is still a residual $\mathbb{A}/\mathbb{B}$ 0-form symmetry. Bulk lines $\boldsymbol{Q}_1^{\widehat{a}}$ which are uncharged under $\mathbb{B}$, i.e., those that satisfy

$$\widehat{a} \Big|_{\mathbb{B}} = 1 , \tag{B.12}$$

continue to end on local untwisted topological operators on $\mathfrak{B}_{\mathrm{Neu}(\mathbb{B}),\nu}$. The remaining lines generate a dual $\widehat{\mathbb{B}} \cong \mathbb{B}$ 1-form symmetry. Specifically the following two short exact sequences are useful

$$1 \longrightarrow \mathbb{B} \longrightarrow \mathbb{A} \stackrel{\phi}{\longrightarrow} \mathbb{A}/\mathbb{B} \longrightarrow 1 , $$
$$1 \longrightarrow \widehat{\mathbb{A}/\mathbb{B}} \longrightarrow \widehat{\mathbb{A}} \stackrel{\widehat{\phi}}{\longrightarrow} \widehat{\mathbb{B}} \longrightarrow 1 . \tag{B.13}$$

The boundary projections of the bulk SymTFT defects are given in terms of the projection maps $\phi$ and $\widehat{\phi}$ as

$$\boldsymbol{Q}_2^a \Big|_{\mathrm{Neu}(\mathbb{B}),\nu} = (D_2^{\phi(a)}, \mathcal{E}_1^a) , \qquad \boldsymbol{Q}_1^{\widehat{a}} \Big|_{\mathrm{Neu}(\mathbb{B}),\nu} = (D_1^{\widehat{\phi}(\widehat{a})}, \mathcal{E}_0^{\widehat{a}}) , \tag{B.14}$$

Let the extension class of the first short exact sequence in (B.13) be denoted as $\epsilon \in H^2(\mathbb{A}/\mathbb{B}, \mathbb{B})$. There is a possibly non-trivial mixed anomaly $\beta \in H^4\left((\mathbb{A}/\mathbb{B})^{(0)} \times \mathbb{B}^{(1)}, U(1)\right)$ between the 1-form $\mathbb{B}^{(1)} = \widehat{\mathbb{B}}$ and the 0-form $(\mathbb{A}/\mathbb{B})^{(0)}$ symmetry, described by the topological action [87]

$$\int_{M_4} B_2 \cup \epsilon(A_1) \in \mathbb{R}/2\pi\mathbb{Z} , \tag{B.15}$$

where $\epsilon \in H^2(\mathbb{A}/\mathbb{B}, \mathbb{B})$ denotes the extension class

$$1 \to \mathbb{B} \to \mathbb{A} \to \mathbb{A}/\mathbb{B} \to 1 , \tag{B.16}$$

$B_2 \in H^2(M_4, \widehat{\mathbb{B}})$ and $A_1 \in H^1(M_4, \mathbb{A}/\mathbb{B})$ are the background gauge fields for the 1-form and 0-form symmetries respectively. We denote the symmetry on $\mathfrak{B}_{\mathrm{Neu}(\mathbb{B}),\nu}$ by

$$2\mathsf{Vec}^\beta((\mathbb{A}/\mathbb{B})^{(0)} \times \mathbb{B}^{(1)}) . \tag{B.17}$$

## B.3  Minimal Interfaces

Following the general theory, the minimal interfaces

$$\mathcal{I}_{(H,N,\pi,\omega)} \,, \tag{B.18}$$

where $N < H < \mathbb{A}$ are subgroups, $\omega \in H^3(H, U(1))$ and $\pi \in H^4(H/N, U(1))$ such that $p^*\pi = 0$, where $p : H \twoheadrightarrow H/N$. The surfaces that end on the interface are $\boldsymbol{Q}_2^n$ for $n \in N$. Whereas the reduced TO, which depends on $\pi$, is

$$\mathcal{Z}(2\mathsf{Vec}_{H/N}^\pi) \,. \tag{B.19}$$

The lines that can end on the interface are given by

$$\widehat{a} \in \widehat{\mathbb{A}/H} \,. \tag{B.20}$$

The cocycle $\omega$ specifies the associator of the lines at the intersection where $\boldsymbol{Q}_2^h$ passes through $\mathcal{I}_{(H,N,\pi,\omega)}$ to become $\boldsymbol{Q}_2^{p(h)}$. We denote these lines as $\mathcal{E}_1^h$. In the special case that $h \in N < H$, $\boldsymbol{Q}_2^h$ ends on the interface.

**Interfaces from Folding.**  We examine the construction of such interfaces from the folding and gauging process. Consider the Drinfeld center of the folded theory

$$\overline{\mathcal{Z}(2\mathsf{Vec}_{\mathbb{A}})} \boxtimes \mathcal{Z}(2\mathsf{Vec}_{H/N}^\pi) \,, \tag{B.21}$$

and impose the Dirichlet boundary condition. By appropriately gauging and unfolding, one can recover all the possible interfaces between $\mathcal{Z}(2\mathsf{Vec}_{\mathbb{A}})$ and $\mathcal{Z}(2\mathsf{Vec}_{H/N}^\pi)$. In particular, gauging the diagonal subgroup $H^{\text{diag}}$ on the Dirichlet boundary defined as

$$H \cong H^{\text{diag}} = \{(h, p(h)) \mid h \in H\} \subset \mathbb{A} \times H/N, \tag{B.22}$$

(recall $p : H \twoheadrightarrow H/N$ denotes the canonical projection) with a choice of discrete torsion $\omega \in H^3(H^{\text{diag}}, U(1))$, gives rise (after unfolding) to the SymTFT interface $\mathcal{I}_{(H,N,\omega,\pi)}$ between $\mathcal{Z}(2\mathsf{Vec}_{\mathbb{A}})$ and $\mathcal{Z}(2\mathsf{Vec}_{H/N}^\pi)$. The map from defects of $\mathcal{Z}(2\mathsf{Vec}_{H/N}^\pi)$ to defects of $\mathcal{Z}(2\mathsf{Vec}_{\mathbb{A}})$ is

$$
\begin{aligned}
\mathcal{F} : \quad & \mathcal{Z}(2\mathsf{Vec}_{H/N}^\pi) \to \mathcal{Z}(2\mathsf{Vec}_{\mathbb{A}}) \\
& \boldsymbol{Q}_2^{hN} \mapsto \bigoplus_{h' \in hN} \boldsymbol{Q}_2^{h'} \,, \\
& \boldsymbol{Q}_1^{\widehat{a}'} \mapsto \bigoplus_{\widehat{a} \in \text{Ind}_H^{\mathbb{A}} p^* \widehat{a}'} \boldsymbol{Q}_1^{\widehat{a}} \,,
\end{aligned} \tag{B.23}
$$

where $hN \in H/N$ is a coset, $\widehat{a}' \in \widehat{H/N}$ and the direct sum over $\widehat{a} \in \text{Ind}_H^{\mathbb{A}} p^* \widehat{a}'$ is taken over all the representations of $\mathbb{A}$ that appear in the decomposition of the induced representation, computed via

$$\text{Ind}_H^{\mathbb{A}} p^* \widehat{a}' = \mathbb{C}[\mathbb{A}] \otimes_{\mathbb{C}[H]} p^* \widehat{a}', \tag{B.24}$$

which is isomorphic to the direct sum of irreducible representations $\widehat{a} \in \text{Rep}(\mathbb{A})$ such that $\kappa(\widehat{a}) \cong p^* \widehat{a}'$ where $\kappa : \text{Rep}(\mathbb{A}) \to \text{Rep}(H)$ is the forgetful functor. Here $\mathbb{C}[G]$ denotes the group algebra, being the regular representation of a finite group $G$, i.e., $\mathbb{C}[G] \cong \bigoplus_R \dim(R) R$, summed over all irreducible representations of $G$. Pictorially, we find

$$\tag{B.25}$$

where $\widehat{a}' \in \widehat{H/N}$, $\mathcal{E}_0^{\widehat{a},\widehat{a}'}$ denotes the intersection of $Q_1^{\widehat{a}}$ and $Q_1^{\widehat{a}'}$ on the interface, and similarly, $\mathcal{E}_1^{h,p(h)}$ is the line where $Q_2^h$ and $Q_2^{p(h)}$ meets on the interface.

## B.4 Minimal Club Quiches

Consider an interface $\mathcal{I}_{(H,N,\pi,\omega)}$ between $\mathcal{Z}(2\text{Vec}_{\mathbb{A}})$ and $\mathcal{Z}(2\text{Vec}_{H/N}^\pi)$. By choosing different symmetry boundaries $\mathfrak{B}_{\text{sym}}$, we study the different symmetries $\mathcal{S}'$ and boundary conditions $\mathfrak{B}'$ after compactifying the interval occupied by $\mathcal{Z}(2\text{Vec}_{\mathbb{A}})$.

### B.4.1 $\mathbb{A}^{(0)}$ 0-Form Symmetry

Starting with the $\mathcal{S} = 2\text{Vec}_{\mathbb{A}}$ 0-form symmetry corresponding to the the symmetry boundary $\mathfrak{B}_{\text{sym}} = \mathfrak{B}_{\text{Dir}}$. Recall the lines that can end on $\mathcal{I}_{(H,N,\pi,\omega)}$ form $\widehat{\mathbb{A}/H}$, and all lines can end on $\mathfrak{B}_{\text{Dir}}$. After compactifying the $\mathcal{Z}(2\text{Vec}_{\mathbb{A}})$ SymTFT, the lines $Q_1^{\widehat{a}}$ for $\widehat{a} \in \widehat{\mathbb{A}/H}$ become the local operators $\mathcal{O}^{\widehat{a}}$, with OPEs satisfying the group multiplication of $\widehat{\mathbb{A}/H}$:

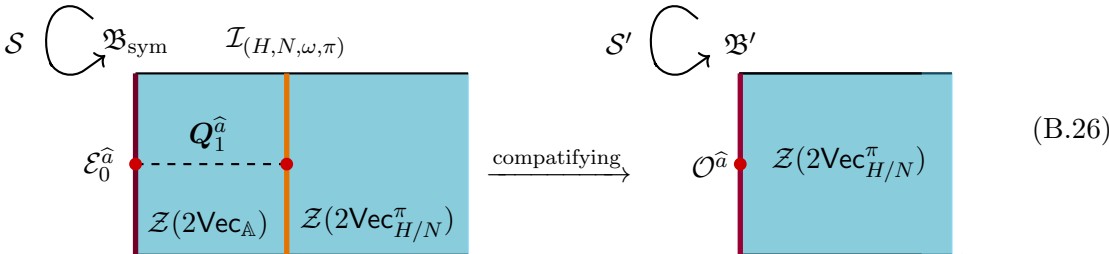

Note $\mathcal{O}^{\widehat{a}}$ is charged under $D_2^a$ if $a \notin H$, hence the symmetry $2\mathsf{Vec}_{\mathbb{A}}$ is broken to $2\mathsf{Vec}_H$. Fix a set $X$ of coset representatives for $\mathbb{A}/H$. From the above local operators, one can build $|\mathbb{A}/H|$ idempotents $\Pi_x$ labeled by $x \in X$ [27]

$$\Pi_x = \frac{1}{|\mathbb{A}/H|} \sum_{\widehat{a} \in \widehat{\mathbb{A}/H}} \widehat{a}(x) \mathcal{O}^{\widehat{a}}\,. \tag{B.27}$$

After taking the club quiche, the $\boldsymbol{Q}_2^n$ surface for $n \in N$ gives lines in twisted sectors

$$(D_2^n, \mathcal{L}^n) \tag{B.28}$$

that satisfy the $N$ fusion rule with associator $\omega|_N$. The boundary after compactifying the interval occupied by $\mathcal{Z}(2\mathsf{Vec}_{\mathbb{A}})$ is

$$\mathfrak{B}' = \bigoplus_{x \in X} (\mathfrak{B}_e)_x\,, \tag{B.29}$$

where $\mathfrak{B}_e$ is the Dirichlet boundary condition of the $\mathcal{Z}(2\mathsf{Vec}_{H/N}^\pi)$ SymTFT. The 2-category of topological defects on this boundary form

$$\mathcal{S}' = \mathrm{Mat}_{|X|}(2\mathsf{Vec}_{H/N}^\pi)\,, \tag{B.30}$$

which is a multifusion 2-category whose objects are $|X| \times |X|$ matrices with entries in $2\mathsf{Vec}_{H/N}^\pi$. We obtain the 2-functor between symmetry categories

$$\mathcal{F}: 2\mathsf{Vec}_{\mathbb{A}} \to \mathrm{Mat}_{|X|}(2\mathsf{Vec}_{H/N}^\pi)\,, \tag{B.31}$$

Under this functor, $D_2^a$ acts on the components of $\mathcal{S}'$ via the natural $\mathbb{A}$ action on $H$ cosets in $\mathbb{A}$. Specifically, $H < \mathbb{A}$ leaves the components fixed. In $H$, further, $N < H$ acts trivially upto a modification of associators, representative of an $N$-SPT labeled by $\omega \in H^3(N, U(1))$. Finally, the $H/N$ symmetry is realized as a $H/N$ symmetry on a given component $\mathfrak{B}_e$ of the reduced TO. Putting all this together, we obtain the functor

$$\mathcal{F}(D_2^a) = \bigoplus_{x \in X} \left( D_2^{p(h)} \right)_{y,x}\,, \tag{B.32}$$

---

[27] Again, recall we view $\widehat{a} \in \widehat{\mathbb{A}/H}$ as the element $\widehat{a} \in \widehat{\mathbb{A}}$ such that $\widehat{a}|_H = 1$ associated via the universal property of a quotient group.

where $a = yhx^{-1}$ for some unique $y \in X$ and $h \in H$ given $x \in X$,

$$\left(D_2^{p(h)}\right)_{y,x} = 1_{y,x} \otimes \left(D_2^{p(h)}\right)_{x,x} = \left(D_2^{p(h)}\right)_{y,y} \otimes 1_{y,x} . \tag{B.33}$$

and $1_{y,x}$ is the invertible surface that has the following linking action on the idempotents labeled by $x' \in X$

$$1_{y,x}\Pi_{x'} = \delta_{x,x'}\Pi_y . \tag{B.34}$$

### B.4.2 1-Form Symmetry

By gauging with discrete torsion $\nu \in H^3(\mathbb{A}, U(1))$ we get 1-form symmetry $\mathbb{A}^{(1)}$ (or $2\mathsf{Rep}(\mathbb{A})$). Lets consider the interfaces in this case: The lines that can end on the interface $\mathcal{I}(H, N, \omega)$ form $\widehat{\mathbb{A}/H} < \widehat{\mathbb{A}}$. None of the non-trivial lines can end on this boundary. The surfaces that can both end on $\mathfrak{B}_{\mathrm{Neu},\nu}$ and on the interface $\mathcal{I}_{(H,N,\omega)}$ are $\boldsymbol{Q}_2^n$ for $n \in N$. They form line operators $\mathcal{L}^n$ after compactifying the interval occupied by $\mathcal{Z}(2\mathsf{Vec}_{\mathbb{A}})$

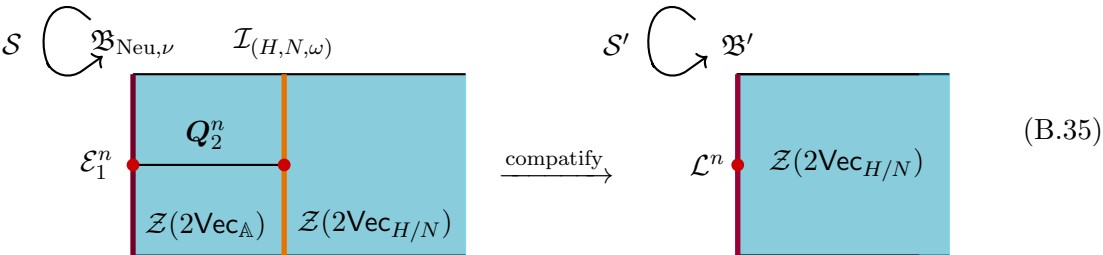

Therefore after compactifying the interval occupied by $\mathcal{Z}(2\mathsf{Vec}_{\mathbb{A}})$, these provide local operators attached to the generators of the 1-form symmetry subgroup $\widehat{\mathbb{A}/H}$, which is trivialized. The 1-form symmetry that acts faithfully within this gapless phase is given by the projection $\widehat{\mathbb{A}} \longrightarrow \widehat{H}$. Recall that the category of genuine and non-genuine lines on the interface form the MTC

$$\mathcal{Z}(\mathsf{Vec}_H^\omega) , \tag{B.36}$$

Among these there is $\mathsf{Rep}(H)$ subcategory which are the genuine lines. $\mathcal{E}_1^h$ are attached to $\boldsymbol{Q}_2^h$ in $\mathcal{Z}(2\mathsf{Vec}_{\mathbb{A}})$ and $Q_2^{p(h)}$ in $\mathcal{Z}(2\mathsf{Vec}_{H/N})$. Therefore after compactifying the interval occupied by $\mathcal{Z}(2\mathsf{Vec}_{\mathbb{A}})$, only the lines $\mathcal{E}_1^n$ get added to the 2-category of (genuine) defects on the boundary, meanwhile the remaining charged lines in $\mathcal{Z}(\mathsf{Vec}_H^\omega)$ are non-genuine lines attached to surfaces in the reduced TO. Physically it means that the $\widehat{N}$ is spontaneoulsy broken while the fate of the sub-symmetry $\widehat{H/N}$ is undecided in the gapless phase. In summary the boundary condition $\mathfrak{B}'$ is

$$\mathfrak{B}' = \frac{\mathfrak{B}_m \boxtimes \mathfrak{T}_{\mathrm{DW}(H),\omega}}{(H/N)^{(1),\mathrm{diag}}} , \tag{B.37}$$

The symmetry category on this boundary is

$$\mathcal{S}' = \frac{2\mathsf{Rep}(H/N) \boxtimes \mathfrak{T}_{\mathrm{DW}(H),\omega}}{(H/N)^{(1),\mathrm{diag}}} \, . \tag{B.38}$$

The functor

$$\mathcal{F} : 2\mathsf{Rep}(\mathbb{A}) \longrightarrow \mathcal{S}' \, , \tag{B.39}$$

maps

$$\mathcal{F}(D_1^{\widehat{a}}) = D_1^{\phi(\widehat{a})} \, , \tag{B.40}$$

where $\phi : \widehat{\mathbb{A}} \to \widehat{H}$ and $D_1^{\phi(\widehat{a})}$ is the topological Wilson line in $\mathfrak{T}_{\mathrm{DW}(H),\omega}$ that generates the $\widehat{H}$ 1-form symmetry.

In summary we have the following physical picture: The UV theory has $\mathbb{A}^{(1)} = \widehat{\mathbb{A}}$ symmetry. The subgroup $\widehat{\mathbb{A}/H}$ is trivial in this gapless phase, and the remaining $\widehat{H}$ acts faithfully. In $\widehat{H}$, $\widehat{N}$ is broken to the subgroup $\widehat{H/N}$, which survives in the reduced TO.

### B.4.3 $\quad 2\mathsf{Vec}^{\beta}_{(\mathbb{A}/\mathbb{B})^{(0)} \times \mathbb{B}^{(1)}}$ Symmetry

More generally, gauging a subgroup $\mathbb{B} < \mathbb{A}$ with discrete torsion $\nu \in H^3(\mathbb{B}, U(1))$ on $\mathfrak{B}_{\mathrm{Dir}}$, we obtain the symmetry $2\mathsf{Vec}^{\beta}_{(\mathbb{A}/\mathbb{B})^{(0)} \times \mathbb{B}^{(1)}}$, where $\beta$ is defined by (B.15), on the symmetry boundary $\mathfrak{B}_{\mathrm{Neu}(\mathbb{B}),\nu}$.

The lines that can end on both $\mathfrak{B}_{\mathrm{Neu}(\mathbb{B}),\nu}$ and $\mathcal{I}_{(H,N,\omega)}$ are $\boldsymbol{Q}_1^{\widehat{a}}$ for $\widehat{a} \in \widehat{\mathbb{A}/(\mathbb{B}H)}$ [28]. They give rise to local operators $\mathcal{O}^{\widehat{a}}$ after compactifying the $\mathcal{Z}(2\mathsf{Vec}_{\mathbb{A}})$ interval. The surfaces that can end on both $\mathfrak{B}_{\mathrm{Neu}(\mathbb{B}),\nu}$ and $\mathcal{I}_{(H,N,\omega)}$ are $\boldsymbol{Q}_2^n$ for $n \in \mathbb{B} \cap N$. They form line operators $\mathcal{L}^n$ after compactifying the $\mathcal{Z}(2\mathsf{Vec}_{\mathbb{A}})$ interval:

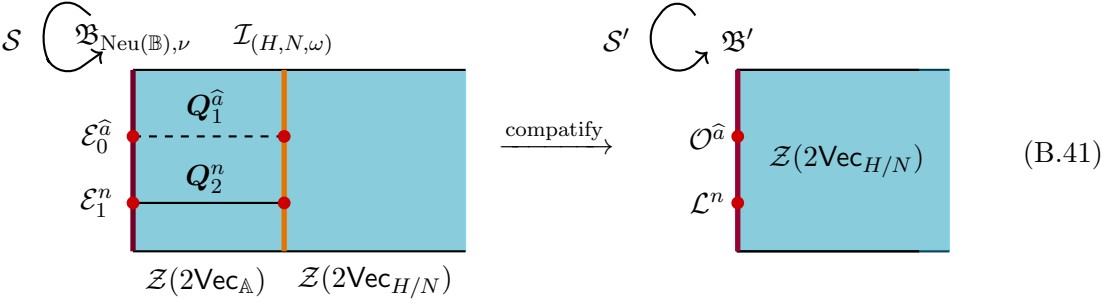

$$\tag{B.41}$$

$\boldsymbol{Q}_1^{\widehat{a}}$ with $\widehat{a}|_{\mathbb{B} \cap N} \neq 1$ braids non-trivially with $\boldsymbol{Q}_2^n$ hence these symmetries are broken, while $\boldsymbol{Q}_1^{\widehat{a}}$ with $\widehat{a}|_{\mathbb{B} \cap N} = 1$ braids trivially with $\boldsymbol{Q}_2^n$ hence these symmetries are preserved. The 1-form symmetry $\mathbb{B}^{(1)}$ is hence broken to $(\mathbb{B}/(\mathbb{B} \cap N))^{(1)}$. Similarly, $\mathcal{O}^{\widehat{a}}, \widehat{a} \in \widehat{\mathbb{A}/(\mathbb{B}H)}$ is charged non-trivially under $D_2^a$ for $a \notin \mathbb{B}H$, hence the 0-form symmetry $(\mathbb{A}/\mathbb{B})^{(0)}$ is broken to $((\mathbb{B}H)/\mathbb{B})^{(0)}$. Hence the original symmetry $2\mathsf{Vec}^{\beta}_{(\mathbb{A}/\mathbb{B})^{(0)} \times \mathbb{B}^{(1)}}$ is broken to $2\mathsf{Vec}^{\beta}_{(\mathbb{B}H/\mathbb{B})^{(0)} \times (\mathbb{B}/(\mathbb{B} \cap N))^{(1)}}$.

---

[28] $\widehat{a} \in \widehat{A}$ satisfies $\widehat{a}|_{\mathbb{B}} = 1$ and $\widehat{a}|_H = 1$, i.e., $\widehat{a}$ is trivial on the product group $\mathbb{B}H$

The genuine line operator $\mathcal{L}^n$ for $n \in \mathbb{B} \cap N$ has $\mathbb{B} \cap N$ fusion rule with non-trivial $F$-symbol specified by $\omega|_{\mathbb{B} \cap N}$. They are charged non-trivially under $D_1^{\widehat{a}}$ for $\widehat{a} \in \widehat{A}$ such that $\widehat{a}|_{\mathbb{B} \cap N} \neq 1$ via (B.2). The $\boldsymbol{Q}_2^{hN}$ surface in the reduced TO is connected to $\boldsymbol{Q}_2^h$ for $h \in hN$ across the interface and becomes the $\mathcal{E}_1^h$ lines in the twisted sector of $D_2^h$ (and is untwisted if $hN = N$ is the trivial coset). Fix a set $X$ of coset representatives for $\mathbb{A}/(\mathbb{B}H)$. From the above local operators, one can build $|\mathbb{A}/H|$ idempotents $\Pi_x$ labeled by $x \in X$ [29]

$$\Pi_x = \frac{1}{|\mathbb{A}/(\mathbb{B}H)|} \sum_{\widehat{a} \in \widehat{\mathbb{A}/(\mathbb{B}H)}} \widehat{a}(x) \mathcal{O}^{\widehat{a}} . \tag{B.42}$$

One can now determine the induced BC and symmetries on these. We will refrain from this as it has the potential to obscure matters (afterall we are discussing abelian group symmetries and their gauging). Nevertheless we will provide a summary of when in this instance there can be igSPT and igSSB phases later.

## B.5  Criteria for igSPT and igSSB

Let us focus on some interesting cases of gapless phases, where the symmetry protects the criticality: igSPTs and igSSBs. For abelian DW theories, we can provide criteria for the existence of such phases (for non-abelian groups these are more complicated and we provide examples in the main text).

A useful insight is the following: when $N \neq H$ and $\pi$ is non-trivial, the partial order (2.28) tells us that a phase, determined by the interface $\mathcal{I}_{(H,N,\pi,\omega)}$, can only flow to gapped phases labeled by $H' < H$ and $\omega|_{H'}$ such that the 4-cocycle trivializes on $H'/N$, i.e., $\pi|_{H'/N} = 1$. This will directly allow us to determine criteria for the igSPT and igSSB phases, i.e. when a given interface $\mathcal{I}_{(H,N,\pi,\omega)}$ and a given symmetry boundary can give rise to such a phase (once a suitable input phase transition is placed at the symmetry boundary).

$2\mathsf{Vec}_{\mathbb{A}}$ **Symmetry.**    Consider the 0-form symmetry $2\mathsf{Vec}_{\mathbb{A}}$. The number of vacua is determined by the number of bulk lines that can end on both $\mathfrak{B}_{\mathrm{Dir}}$ and the interface $\mathcal{I}_{(H,N,\pi,\omega)}$, which in turn is determined by $H$. Hence a phase (either an SPT or an SSB) is intrinsically gapless whenever

$$\pi \neq 1 . \tag{B.43}$$

This means that the reduced TO is a DW theory with a non-trivial twist. Furthermore, the gapless phase associated to the interface $\mathcal{I}_{(H,N,\pi,\omega)}$ is a gSPT if $H = \mathbb{A}$, otherwise it is a gSSB. An example of a $2\mathsf{Vec}_{\mathbb{A}}$ igSPT was provided in [43] for $\mathbb{A} = \mathbb{Z}_2 \times \mathbb{Z}_4$ and related to

---

[29] Again, recall we view $\widehat{a} \in \widehat{\mathbb{A}/(\mathbb{B}H)}$ as the element $\widehat{a} \in \widehat{\mathbb{A}}$ such that $\widehat{a}|_H = 1$ and $\widehat{a}|_{\mathbb{B}} = 1$.

the "Cheshire charge" discussed in [88]. Condition (B.43) for intrinsically gapless phases for 0-form symmetry holds also for non-abelian groups and we will provide an example of a $D_8$ igSPT in section 6.

**1-Form Symmetry $\mathbb{A}^{(1)}$.** Since $\mathcal{L}^n$, $n \in N$ are charged non-trivially under the generators $D_1^{\widehat{a}}$, $\widehat{a}|_N \neq 1$, for the $2\mathsf{Rep}(\mathbb{A})$ symmetry, the $2\mathsf{Rep}(\mathbb{A})$ symmetry is broken to $2\mathsf{Rep}(\mathbb{A}/N)$. Hence only the interface $\mathcal{I}_{(H,N,\pi,\omega)}$ with $N = 1$ gives rise to an SPT phase, and it is a gapped phase iff $H = N = 1$.

For $H \neq 1$ and $N = 1$, a gSPT phase determined by $\mathcal{I}_{(H,1,\omega,\pi)}$ can always flow to the SPT determined by $\mathcal{I}_{(1,1,1,1)}$. There are no igSPT phases for $2\mathsf{Rep}(\mathbb{A})$ symmetries. However, there can be instances of **igSSB** phases for $2\mathsf{Rep}(G)$ symmetry, where $G$ is non-abelian, for which we provide examples in sections 5 and 6.

Similarly, $\mathcal{I}_{(H,N,\omega,\pi)}$ determines a **gSSB** phase which can always flow to the SSB determined by $\mathcal{I}_{(N,N,\omega|_N,1)}$, with the same symmetry breaking pattern. Hence there are no igSSB phases for $2\mathsf{Rep}(\mathbb{A})$ symmetries either.

**$2\mathsf{Vec}^{\beta}_{(\mathbb{A}/\mathbb{B})^{(0)} \times \mathbb{B}^{(1)}}$ Symmetry.** For the 1-group symmetry we find the following: Recall there are lines $\mathcal{L}^n$, $n \in \mathbb{B} \cap N$ that break the 1-form symmetry $\mathbb{B}^{(1)}$ to $(\mathbb{B}/\mathbb{B} \cap N)^{(1)}$. Similarly, the presence of the local operators $\mathcal{O}^{\widehat{a}}$, $\widehat{a} \in \widehat{\mathbb{A}/\mathbb{B}H}$ breaks the 0-form symmetry $(\mathbb{A}/\mathbb{B})^{(0)}$ to $(\mathbb{B}H/\mathbb{B})^{(0)}$, i.e., the original symmetry $2\mathsf{Vec}^{\beta}_{(\mathbb{A}/\mathbb{B})^{(0)} \times \mathbb{B}^{(1)}}$ is broken to $2\mathsf{Vec}^{\beta}_{(\mathbb{B}H/\mathbb{B})^{(0)} \times (\mathbb{B}/\mathbb{B} \cap N)^{(1)}}$ in the phase determined by an interface $\mathcal{I}_{(H,N,\pi,\omega)}$.

Hence if both $\mathbb{B}H = \mathbb{A}$ and $\mathbb{B} \cap N = 1$, the corresponding phase is a gSPT. Otherwise if $\mathbb{B}H \neq \mathbb{A}$ or $\mathbb{B} \cap N \neq 1$, the phase is a gSSB. Furthermore, such a phase is intrinsically gapless, if there is a subgroup $H' < H$ such that $\pi|_{H'/N} = 1$, $\mathbb{B}H' = \mathbb{B}H$ and $\mathbb{B} \cap H' = \mathbb{B} \cap N$.

## B.6 KT Transformations and Gapless Phases

To obtain a symmetric gapless (or gapped) phase, we simply complete the club quiche into a club sandwich by providing a gapless physical boundary condition. Operationally, this amounts to replacing $\mathfrak{B}_e$ and $\mathfrak{B}_m$ in the $\mathcal{S}$-symmetric BCs $\mathfrak{B}'$ by $H/N$ 0-form symmetric theory (gapped or gapless) and $\widehat{H/N}$ 1-form symmetric theory. Concretely these theories take the following form:

- For $\mathcal{S} = 2\mathsf{Vec}_{\mathbb{A}}$, the phase that can be constructed from $\mathcal{I}_{(H,N,\pi,\omega)}$ is

$$\bigoplus_{x \in X} \left( \mathfrak{T}^{(0)}_{H/N} \right)_x, \tag{B.44}$$

where $\mathfrak{T}_{H/N}^{(0)}$ is an $H/N$ 0-form symmetric CFT, with the action of $\mathcal{S}$ same as on $\mathfrak{B}'$ in (B.29).

- For the 1-form symmetry $\mathbb{A}$, i.e. $\mathcal{S} = 2\mathsf{Rep}(\mathbb{A})$, the phase associated to the interface $\mathcal{I}_{(H,N,\pi,\omega)}$ is

$$\frac{\mathfrak{T}_{\widehat{H/N}}^{(1)} \boxtimes \mathfrak{T}_{\mathrm{DW}(H),\omega}}{(H/N)^{(1),\mathrm{diag}}} , \tag{B.45}$$

where $\mathfrak{T}_{\widehat{H/N}}^{(1)}$ is an $\widehat{H/N}$ 1-form symmetric CFT, with the action of $\mathcal{S}$ same as on $\mathfrak{B}'$ in (B.38).

We can now use this to KT transform phases, from the smaller $(H/N)^{(0)}$ and $(H/N)^{(1)}$ etc symmetries to the larger ones, and thereby construct e.g. phase transitions (inputting the phase transition for the smaller symmetry).

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
