# Peer review of "Gapless Phases in (2+1)d with Non-Invertible Symmetries"

_SciPost Physics_

## Round 1 · Referee Report · Anonymous (Referee 1) · 2025-9-30

Report

The paper generalizes the KT transformation and apply it to the study of gapless phases in (2+1) using SymTFT, and provide a way to construct 2nd order phase transition with a larger categorical symmetry from a given one with a smaller symmetry. The paper has a good balance between abstract approach and concrete examples. I believe it is beneficial for readers from both Math and Physics backgrounds. However, before I can recommend this paper to be published, I hope the authors could address the following questions:

  1. On page 14, given that $\mathcal{Z}(2\text{Vec}_{H/N}^\pi)$ will appear with a non-trivial $\pi \in H^4$. It is helpful if the authors to comment on the effect of non-trivial $\pi$ here.
  2. Around page 22, when discussing the interfaces $\mathcal{I}(H,N,\pi,\omega)$, for completeness, it will be nice if the authors could describe what happens to lines $\mathbf{Q}_1^R$ where $R \notin ker(\kappa)$ and surfaces $\mathbf{Q}_2^{[g]}$ where $g \notin H$.
  3. On page 31, below (3.1), can the authors explain what is physically meaning of assumption where $\mathcal{A}_e$ is fusion?
  4. At the end of page 45, for the case $H = G$ and $N = 1$, there could be non-trivial choice of $\omega \in H^3(H,U(1))$. Are the interfaces trivial for all choices of $\omega$ or it is only trivial when $\omega = 0$?
  5. In Section 4.2, it would be nice if the authors could describe the KT transformations in the case where $\omega$ in the $\mathcal{I}(H,N,\omega)$ is non-trivial.
  6. In Eq (5.17), will the RHS (currently being written as $2\text{Vec}(\mathbb{Z}_3^{(1)}\rtimes \mathbb{Z}_2^{(0)})$) depend the label for the twist $\omega \in H^3(\mathbb{Z}_3,U(1))$?
  7. In Eq (5.65), it would be helpful if the authors could move the definition of $\text{Mat}_n$ of some 2-categories from below (B.30) to here.

Besides the above questions, there are a few potential typos that the authors should check:

  1. Page 45, at the end of the first paragraph of Section 4, "." is missing.
  2. Page 49, "The $m^p$ surfaces becomes ..." should be "The $m^p$ surfaces become ..."; "Non-trival $\omega$ ..." should be "Non-trivial $\omega$".
  3. Page 51, Eq (4.17) and below, $\text{DW}(\mathbb{Z}{Z_4})\omega$ looks like a typo.
  4. Page 52, at the end of the first paragraph of Section 4.4, $D_1^{e20}$ looks like a typo.
  5. Page 57, Eq (4.35) $Z_2^{(1)}$ should be $\mathbb{Z}_2^{(1)}$.
  6. Page 74, below Eq (5.52), $\mathbb{Z}_3 \times \mathbb{Z}_3 \cong \in H^3(S_3,U(1))$ should be $\mathbb{Z}_3 \times \mathbb{Z}_2 \cong H^3(S_3,U(1))$.
  7. Page 77, below Eq (5.72), $\mathcal{Z}(A^{\text{id}})$ should be $\mathcal{Z}(\mathcal{A}^{id})$.
  8. Page 79, Eq (5.81), there shouldn't be a $j$ on the exponent of $\phi$? Below, $\mathcal{Z}(A^{\text{id}})$ should be $\mathcal{Z}(\mathcal{A}^{id})$.
  9. Page 84, below Eq (5.108), $\mathcal{Z}(\mathcal{A}^{\text{id}23}$ looks like a typo.
  10. Page 87, below Eq (5.132), $p' \in H^3(\mathbb{Z}_2,U(1)$ should be $p' \in H^3(\mathbb{Z}_3,U(1))$.

Recommendation

Ask for major revision

---

## Round 1 · Referee Report · Anonymous (Referee 2) · 2025-10-22

Report

The paper presents a systematic and ambitious framework for studying gapless phases (in 2+1 dimensions) in the presence of categorical (i.e., non-invertible) symmetries, building on the earlier work on gapped phases. The authors extend the concept of the Symmetry Topological Field Theory (SymTFT) to interfaces (“club sandwiches”) between topological orders, generalising the well-known Kennedy–Tasaki (KT) map, and apply the method to Dijkgraaf–Witten (DW) theories for finite groups . They also illustrate their approach by concrete examples (abelian and non-abelian). The construction yields new kinds of intrinsically gapless symmetry-protected phases (igSPTs) and spontaneous symmetry breaking phases (igSSBs) in (2+1)d. Overall, the work constitutes a major step toward a “categorical Landau paradigm” for gapless phases with non-invertible symmetries. Thus I recommend it published in Scipost.

Requested changes

  1. A brief discussion (even a paragraph) of potential physical realisations (condensed‐matter systems, lattice models) of the predicted gapless phases would be better (even if speculative).

Recommendation

Publish (meets expectations and criteria for this Journal)

---

## Editorial Decision

awaiting_resubmission